# TAPB: an interventional debiasing framework for alleviating target prior bias in drug-target interaction prediction

Gaoming Lin[1,7], Xin Zhang [2,3,7], Zhonghao Ren [4,5], Quan Zou [2], Prayag Tiwari [6] ✉, Changjun Zhou [1] ✉ & Yijie Ding[2] ✉

Drug Target Interaction (DTI) prediction is vital for drug repurposing. Previous DTI studies on BioSNAP and BindingDB datasets often attribute biased predictions to "drug bias," while our work reveals "target prior bias" as the predominant issue. This bias stems from the "prior tendency," characterized by the imbalanced label distribution of targets in the training data. From causal lens, target "prior tendency" is a confounder, causing models trained with $P(Y|\mathbf{D}, \mathbf{T})$ to learn spurious associations between targets and labels rather than genuine interaction mechanisms. In this study, we introduce alleviating **Ta**rget **P**rior **B**ias in Drug-Target Interaction Prediction (TAPB), a novel debiasing framework that employs amino acid randomization, confounder alignment module (CAM), and interventional training to compute $P(Y|\mathbf{D}, do(\mathbf{T}))$ via backdoor adjustment, thereby addressing this bias. TAPB achieves competitive performance over existing approaches, demonstrating enhanced generalization and providing interpretable insights into DTIs.

Drug Target Interaction (DTI) prediction is indispensable in the exploration of potential applications for existing drugs, significantly expediting their transition from experimental phases to clinical application[1]. Computational methods for drug discovery span distinct strategies. On one hand, virtual docking simulation[2] explores biomolecular interactions from a structural perspective. On the other hand, a multitude of data-driven methods have been developed for DTI prediction, including traditional machine learning[3,4] and deep learning methods, which have converged on two primary types: those leveraging graph data for recommendations[5] and those utilizing sequence data for predictions. Notably, sequence-based prediction methods offer token-level interpretability, providing valuable insights that can guide drug repurposing. These methods often employ a dual-tower architecture to accommodate diverse input formats, e.g. SMILES[6], amino acid sequences[7–9], fingerprints[10], or molecular graphs[11,12], etc.

These inputs are encoded into embeddings and then aggregated for binary classification tasks, estimating the interaction probability $P(Y|\mathbf{D}, \mathbf{T})$ between a drug $\mathbf{D}$ and a target $\mathbf{T}$. Public sequence datasets commonly used in DTI research, e.g. BioSNAP and BindingDB, are sourced from established databases including DrugBank[13] and the Binding Database[14], respectively, as shown in Fig. 1a. Recently, DrugBAN[12] has divided these datasets into in-domain and cross-domain splits, enabling a more systematic and rigorous evaluation of model performance under diverse conditions.

Despite significant advancements in model architecture and feature engineering, DTI models trained on sequence datasets, e.g. BioSNAP and BindingDB, exhibit a bias towards specific inputs for predictions, rather than capturing the true mechanisms of drug-target interactions. This limits models' generalization capability, posing a significant barrier to their application in drug repurposing. Previous

[1]School of Computer Science and Technology, Zhejiang Normal University, Jinhua, Zhejiang, China. [2]Yangtze Delta Region Institute (Quzhou), University of Electronic Science and Technology of China, Quzhou, Zhejiang, China. [3]The Quzhou Affiliated Hospital of Wenzhou Medical University, Quzhou People's Hospital, Quzhou, Zhejiang, China. [4]State Key Laboratory of Chemo and Biosensing, College of Computer Science and Electronic Engineering, Hunan University, Changsha, Hunan, China. [5]The Ministry of Education Key Laboratory of Fusion Computing of Supercomputing and Artificial Intelligence, Hunan University, Changsha, Hunan, China. [6]School of Information Technology, Halmstad University, Halmstad, Sweden. [7]These authors contributed equally: Gaoming Lin, Xin Zhang. ✉e-mail: prayag.tiwari@hh.se; zhouchangjun@zjnu.edu.cn; wuxi_dyj@csj.uestc.edu.cn

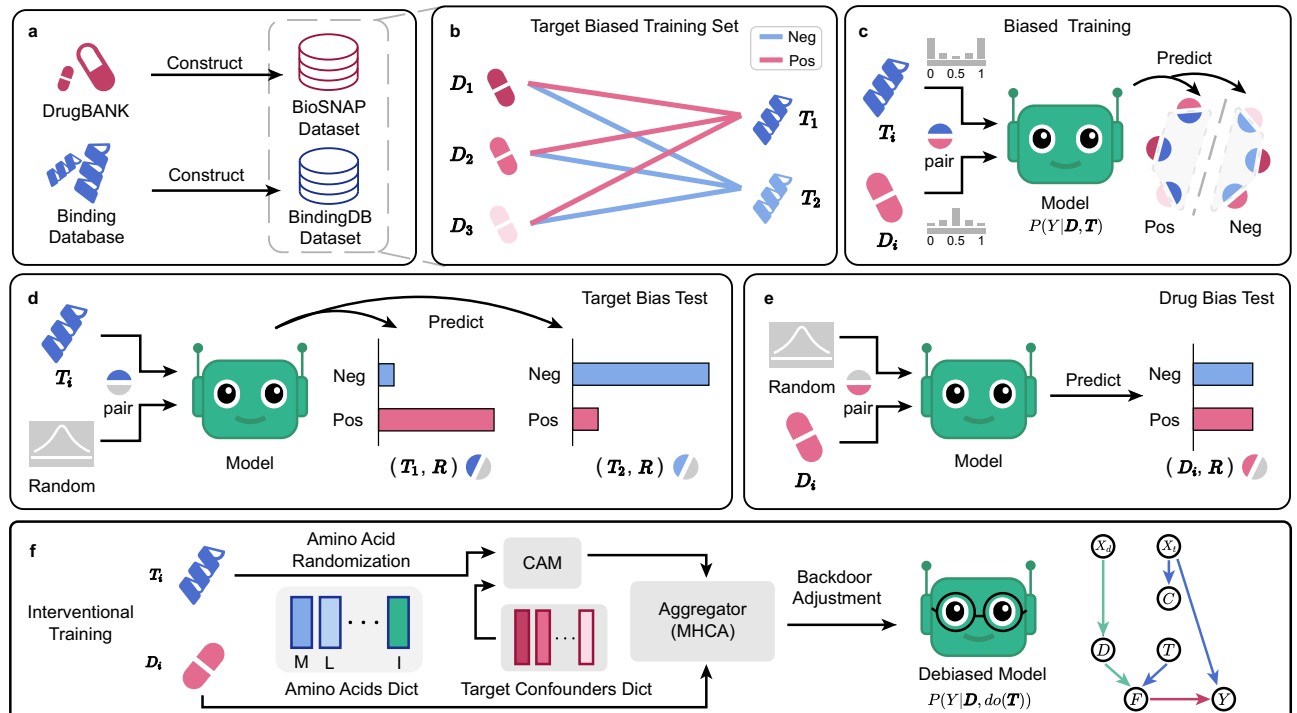

**Fig. 1 | Overview of DTI bias analysis and our framework. a** Dataset constructions of BioSNAP and BindingDB. **b** A Sketch of target-biased training sets, where certain targets (**T₁** and **T₂**) exhibit positive and negative "prior tendencies", respectively, while the bias in drugs **Dᵢ** is less pronounced. **c** A Sketch of the biased training process, where models learn from datasets with an inherent "target bias". **d** "Target bias" tests demonstrate that pairing a randomly generated feature **R** with target **T₁** yields higher positive scores compared to negative ones, with the opposite observed for **T₂**. **e** "Drug bias" tests show relatively balanced scores when a drug **D** is paired with a randomly generated feature **R**, indicating lesser influence on predictions. **f** Our TAPB interventional training, combining amino acid randomization, confounder alignment module (CAM), and multi-head cross-attention (MHCA) to compute $P(Y|\mathbf{D}, do(\mathbf{T}))$ under the assumption of our SCM.

research, e.g. TransformerCPI[15], DrugBAN[12] and UdanDTI[16], has attributed this issue to "hidden pattern bias", i.e. "drug bias". UdanDTI leverages an asymmetrical architecture and attentive aggregation to strengthen the target branch and downplay the drug branch, thus mitigating "drug bias." However, our findings indicate that "target prior bias", i.e. "target bias", plays a more substantial role in biased predictions across both in-domain and cross-domain splits of BioSNAP and BindingDB. This bias reflects the tendency of models to rely primarily on target-specific features when making predictions.

The cause of "target prior bias" is "prior tendency". Intuitively, "prior tendency" describes the imbalance of positive and negative interaction labels across individual drugs or targets in the DTI training set. Models can minimize loss by simply capturing the label tendencies of drugs or targets rather than the true interaction mechanisms. For example, in Fig. 1b, targets **T₁** and **T₂** in the training set have more positive and negative labels, respectively, while the drugs' label distribution is relatively averaged. This imbalance can cause models to memorize the observed targets' labels in the training set rather than genuine drug-target interactions, leading to biased predictions. Inspired by CF-VQA[17], we designed and performed our bias test for targets and drugs as shown in Fig. 1d, e, respectively. For the training set in Fig. 1b, when the input changes from DTI pairs (**D**, **T**) to pairs containing a randomly generated tensor **R**, i.e. (**D**, **R**) and (**T**, **R**), the model tends to make predictions based on the observed target label tendencies from the training set when given (**T**, **R**). In contrast, predictions from (**D**, **R**) remain close to average scores. This result underscores the significant influence of target "prior tendency." In Fig. 1c, we characterize this data-distorted training as biased training.

To verify that "prior tendency" causes biased predictions on sequence DTI datasets, e.g. BindingDB and BioSNAP, we constructed two counter-prior datasets: one with high drug "prior tendency" and

another with a balanced "prior tendency." The findings presented in Section "Prior tendency causes biased predictions" support our hypothesis. It is essential to note that the "target prior bias" identified originates from the BioSNAP and BindingDB datasets. In contrast, other DTI prediction methods with different datasets and model architectures, e.g. NRLMF[18] and[19], may display varying biases.

In this work, we introduce an interventional debiasing framework for alleviating **Ta**rget **P**rior **B**ias (TAPB) in drug-target interaction prediction. From causal lens, the target "prior tendency" of DTI sequence datasets is a confounder that opens up backdoor paths for targets and predictions, making it difficult for DTI models to make unbiased predictions through $P(Y|\mathbf{D}, do(\mathbf{T}))$. This issue has not been adequately addressed in previous studies. As shown in Fig. 1f, TAPB employs amino acid randomization and confounder alignment module to compute $P(Y|\mathbf{D}, do(\mathbf{T}))$ via theoretically exact backdoor adjustment, where $do(\cdot)$ denotes the intervention that sets the variable to a specific value, blocking all incoming paths to the variable. The backdoor adjustment computes $P(Y|\mathbf{D}, do(\mathbf{T}))$ via observed data without performing actual interventions. The contributions of this work are summarized as follows:

- In this study, we re-evaluate the BioSNAP and BindingDB datasets, both in-domain and cross-domain splits. We identify "target prior bias" as a key source of prediction bias in these datasets, a cause distinct from the previously recognized "drug bias." Our statistical analysis supports this conclusion, and counter-tendency experiments confirm that "prior tendency" underlies both "drug bias" and "target bias" in DTI models trained on sequence datasets.
- We reframe DTI prediction through causal lens and propose TAPB, an interventional debiasing framework that integrates: amino acid randomization to disrupt spurious correlations via residue deletion (70%) and mutation (20%), and backdoor

adjustment to compute $P(Y|\mathbf{D}, do(\mathbf{T}))$ through a confounder dictionary $C$ and confounder alignment module. Amino acid randomization not only diversifies input data and reduces memory usage but also enhances training efficiency.

- Extensive experiments on four public datasets demonstrate that TAPB establishes a new benchmark in DTI prediction. The framework's adaptability offers potential improvements for other DTI models, provided our assumptions are satisfied.

## Results

### DTI prediction formulation on sequence datasets

Due to the absence of token-level ground truth in DTI sequence datasets, previous studies, e.g. MolTrans[7], TransformerCPI[15], and DrugBAN[12], typically reformulate DTI prediction as a binary classification task. Let $X = \{X_d, X_t, y\}$ denote a set of DTI data points, where $X_d$ represents the Simplified Molecular Input Line Entry System (SMILES) of the small molecule, $X_t$ denotes the amino acid sequence of the target, and $y$ is a binary label indicating the presence or absence of an interaction between the drug and the target. The general approach for DTI prediction involves three main steps: 1) Feature encoding: segment or convert the input SMILES and target sequences separately, and employ various respective encoders $f_d(\cdot)$ and $f_t(\cdot)$ to encode features, e.g. CNN[20], ResNet[21], GCN[22], LSTM[23] and BERT[24], etc. Drug feature and target feature are denoted as $\mathbf{D}$ and $\mathbf{T}$, respectively; 2) Feature fusion: aggregate the features $\mathbf{D}$ and $\mathbf{T}$ using an aggregator $\mathcal{F}(\cdot)$, which could be feature concatenation[10], Bilinear Attention Network (BAN)[25], Transformer[26], or other aggregators; 3) Prediction: Using the pooling $\sigma(\cdot)$ and a classification head $g_y(\cdot)$ for binary classification, i.e. predicting through $P(Y|\mathbf{D}, \mathbf{T})$, which can be formulated as:

$$\mathbf{D} = f_d(X_d), \mathbf{T} = f_t(X_t), \mathbf{F} = \mathcal{F}(\mathbf{D}, \mathbf{T}), Y = g_y(\sigma(\mathbf{F})) \quad (1)$$

Building upon this framework, previous studies focused on enhancing model performance by refining feature encoders and aggregators, or incorporating additional features.

### Drugs bias vs target bias: which is more severe?

Previous studies, e.g. TransformerCPI[15], DrugBAN[12] and UdanDTI[16], trained on DTI sequence datasets have acknowledged the presence of biased predictions, with a common assumption that "drug bias" causes models to rely more on drug features. However, we question whether "drug bias" is the predominant factor influencing biased predictions in DTI sequence datasets such as BindingDB and BioSNAP. To test whether drug or target features are more influential, we employed t-SNE[27] visualizations of classification features. Specifically, for each drug-target pair $(\mathbf{D}, \mathbf{T})$ in the training set, we created two types of inputs: 1) the original drug feature $\mathbf{D}$ combined with a Gaussian-distributed random tensor $\mathbf{R}$ replacing the target feature $\mathbf{T}$; 2) the original target feature $\mathbf{T}$ paired with a Gaussian-distributed random tensor $\mathbf{R}$ replacing the drug feature $\mathbf{D}$. These inputs are denoted as $(\mathbf{D}, \mathbf{R})$ ("drug bias" test) and $(\mathbf{T}, \mathbf{R})$ ("target bias" test), respectively, when passed through the pre-trained models. The random feature $\mathbf{R} \in \mathbb{R}^{L \times d_m}$, where $L$ denotes the length of the input sequence, and $d_m$ denotes the dimension of the model.

We trained DrugBAN[12] and TransformerCPI[15] on the above datasets as our subject of study. Hyperparameter settings of DrugBAN and TransformerCPI are provided in Supplementary Tables 11 and 9, respectively. In DrugBAN, $f_d(\cdot)$ is a 3-layer GCN, $f_t(\cdot)$ is a 3-layer 1D CNN, and $\mathcal{F}(\cdot)$ is a Bilinear Attention Network (BAN)[25]. In TransformerCPI, $f_d(\cdot)$ is a 3-layer GCN, $f_t(\cdot)$ uses word2vec[28] embeddings followed by a 1D CNN with gate linear units, and $\mathcal{F}(\cdot)$ is a cross-attention transformer, where the query is the drug feature $\mathbf{D}$, and the key and value are the target feature $\mathbf{T}$. For $(\mathbf{D}, \mathbf{R})$, node and edge features were replaced with two random tensors $\mathbf{R}$, while for $(\mathbf{T}, \mathbf{R})$, target embeddings were replaced with $\mathbf{R}$. For the in-domain splits, the

training set was chosen as the visualization data, whereas for the cross-domain splits, the source training set was selected for visualization.

If predictions are unbiased, the t-SNE visualizations for both $(\mathbf{D}, \mathbf{R})$ and $(\mathbf{T}, \mathbf{R})$, which receive randomized meaningless inputs $\mathbf{R}$, should not exhibit any preference for positive class instances, i.e. positive class features should be randomly distributed in t-SNE visualizations. Conversely, if "hidden pattern bias" is predominant, $(\mathbf{D}, \mathbf{R})$ might exhibit a tendency toward positive class clustering, while $(\mathbf{T}, \mathbf{R})$ should show a random distribution.

However, the visualization results contradict these expectations. As shown in Fig. 2e, g, m, and o, under the BindingDB in-domain and cross-domain settings, no matter what drug encoder $f_d(\cdot)$ or target encoder $f_t(\cdot)$ or aggregators $\mathcal{F}(\cdot)$ were used, the positive classification features of $(\mathbf{D}, \mathbf{R})$ are nearly randomly distributed, while t-SNE visualizations of $(\mathbf{T}, \mathbf{R})$ in Fig. 2a, c, i, k exhibits significant positive class clustering. Similarly, under the BioSNAP in-domain and cross-domain settings, Fig. 2b, d, j, l shows more pronounced positive class clustering for $(\mathbf{T}, \mathbf{R})$ compared to Fig. 2f, h, n, p. These observations suggest that models trained on these datasets exhibit stronger "target bias" than "drug bias", prompting the question: why do models have inner patterns that rely more heavily on targets?

### Prior tendency causes biased predictions

We assume that the biased predictions are caused by "prior tendency", which we formally define as systematic label distribution biases inherent to individual drugs or targets in the DTI sequence dataset. Specifically, this refers to statistically significant deviations in the positive/negative sample ratios observed across different drugs (drug-level prior) or targets (target-level prior), which create spurious correlations that models can exploit to minimize loss without learning true interaction mechanisms. To quantify "prior tendency" across different DTI datasets, we devised the following label test:

$$z_i = \frac{\sum_j y_{ij}}{n_i} \quad (2)$$

$$Z = \sum_i |z_i - 0.5| + 0.5 \quad (3)$$

where $y_{ij}$ denotes the $j$-th label of the $i$-th sequence, $n_i$ denotes the occurrence count of the $i$-th sequence, $z_i$ denotes each sequence's "prior tendency" which is rounded to one decimal place for better visualization, and $Z$ denotes the overall "prior tendency" across all sequences in the dataset, ranging from 0.5 to 1.0.

Furthermore, beyond the heuristic score $Z$, we designed a rigorous permutation test grounded in a null model where interaction labels $Y$ are independent of target-specific effects. Under this null hypothesis, each drug-target pair's label follows a Bernoulli distribution parameterized by the global positive interaction proportion:

$$g = \frac{\sum Y}{N} \quad (4)$$

where $N$ denotes the total number of drug-target pairs, representing random label assignment without target-specific biases. To evaluate statistical significance, we employed a weighted sum of squared deviations as our test statistic:

$$T = \sum_{i=1}^{M} n_i(z_i - g)^2 \quad (5)$$

where $M$ is the total number of unique sequences, i.e. total number of unique drugs or targets in the dataset. The $n_i$ weighting ensures proportional contribution by sample size while maintaining sensitivity for

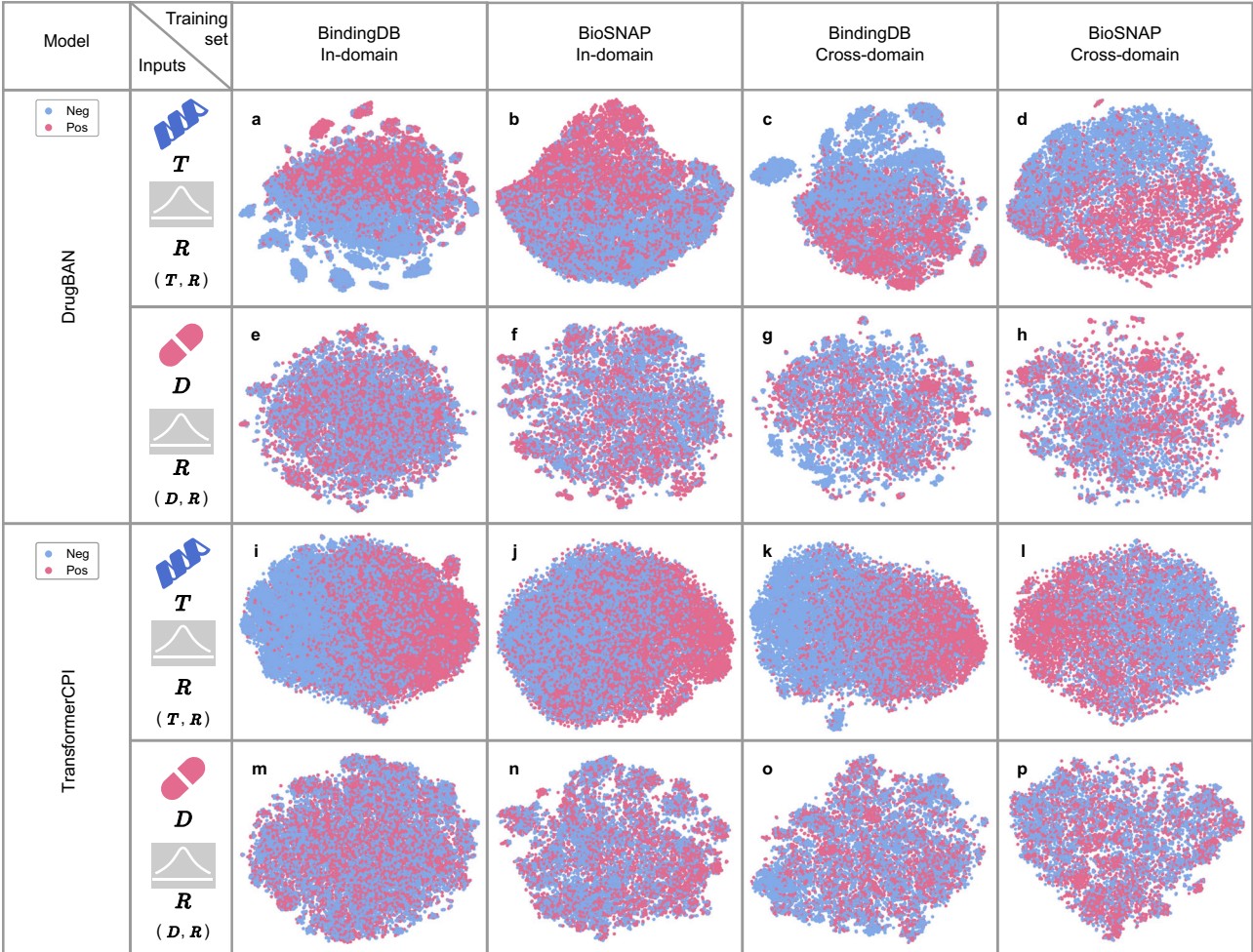

**Fig. 2 | The t-SNE visualization of classification features (D, R) and (T, R) of DrugBAN and TransformerCPI on the BindingDB and BioSNAP datasets.** **a** DrugBAN trained on the in-domain split of the BindingDB dataset with inputs (**T, R**). **b** DrugBAN trained on the in-domain split of the BioSNAP dataset with inputs (**T, R**). **c** DrugBAN trained on the cross-domain split of the BindingDB dataset with inputs (**T, R**). **d** DrugBAN trained on the cross-domain split of the BioSNAP dataset with inputs (**T, R**). **e** DrugBAN trained on the in-domain split of the BindingDB dataset with inputs (**D, R**). **f** DrugBAN trained on the in-domain split of the BioSNAP dataset with inputs (**D, R**). **g** DrugBAN trained on the cross-domain split of the BindingDB dataset with inputs (**D, R**). **h** DrugBAN trained on the cross-domain split of the BioSNAP dataset with inputs (**D, R**). **i** TransformerCPI trained on the in-domain split of the BindingDB dataset with inputs (**T, R**). **j** TransformerCPI trained on the in-domain split of the BioSNAP dataset with inputs (**T, R**). **k** TransformerCPI trained on the cross-domain split of the BindingDB dataset with inputs (**T, R**). **l** TransformerCPI trained on the cross-domain split of the BioSNAP dataset with inputs (**T, R**). **m** TransformerCPI trained on the in-domain split of the BindingDB dataset with inputs (**D, R**). **n** TransformerCPI trained on the in-domain split of the BioSNAP dataset with inputs (**D, R**). **o** TransformerCPI trained on the cross-domain split of the BindingDB dataset with inputs (**D, R**). **p** TransformerCPI trained on the cross-domain split of the BioSNAP dataset with inputs (**D, R**).

sparse targets. Our permutation procedure preserves drug-protein pair structures while randomly reshuffling labels across all pairs for $B = 1000$ iterations, with p-value computed as:

$$p\text{-value} = \frac{1 + \sum_{b=1}^{B} \mathbf{1}(T_b \geq T_{\text{obs}})}{1 + B} \qquad (6)$$

where $T_b$ is the permuted statistic and $T_{\text{obs}}$ the observed value. This non-parametric approach maintains DTI data structures through fixed pairings with permuted labels, ensures small-sample robustness by avoiding asymptotic assumptions, and naturally adapts to class imbalance.

We calculated the frequency of each "prior tendency", overall "prior tendency", and corresponding statistical significance, i.e. p-value for drugs as $P_d$ and p-value for targets $P_t$, across 4 datasets. As shown in Fig. 3a, in the in-domain split of the BindingDB dataset, target label frequencies exhibit extreme bimodal concentrations at 0 and 1 ($P_t = 0.000$), while drug label frequencies center near 0.5 with no

significant deviation ($P_d = 1.000$). Figure 3b reveals significant drug deviations ($P_d = 0.000$) alongside persistently extreme target imbalance ($P_t = 0.000$) in the cross-domain split of the BindingDB dataset. Figure 3c, d shows significant deviations for both entities ($P_t = 0.000$, $P_d = 0.000$) in the BioSNAP in-domain and cross-domain splits, with attenuated but still pronounced target imbalance. Figure 3e confirms targets consistently exhibit higher prior tendency $Z$ than drugs across all configurations.

However, the simultaneous occurrence of "prior tendency" and biased prediction in DTI models does not imply a causal relationship between them. To verify that it is indeed the "prior tendency" that causes the biased predictions, we re-split the BindingDB dataset as follows:

Drug biased training set: as shown in Fig. 4a, we formed a positive pairs set $S_p$ by selecting all positive sample pairs from the BindingDB in-domain training set, ensuring each drug or target appears only once. Next, negative pairs set $S_n$ were created by randomly assigning the rest drugs not in set $S_p$ to targets in set $S_p$ as negative samples. Finally, we

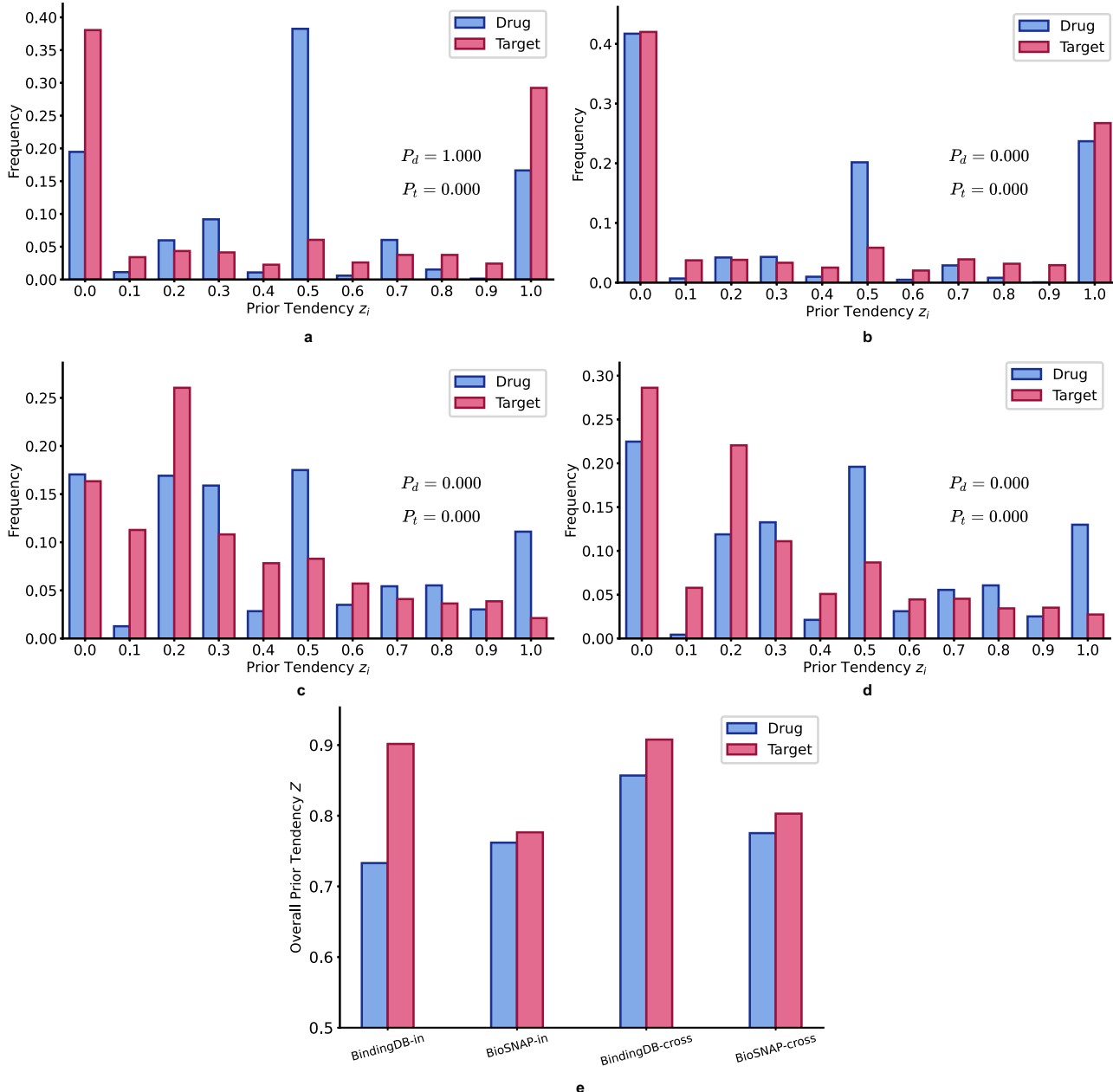

**Fig. 3 | Statistical visualization of "prior tendency" for drugs and targets in the BindingDB and BioSNAP datasets. a** "Prior tendency" frequency distribution of $z_i$ and p-value for drugs and targets in the BindingDB in-domain split training set. **b** "Prior tendency" frequency distribution of $z_i$ and p-value for drugs and targets in the BindingDB cross-domain split training set. **c** "Prior tendency" frequency distribution of $z_i$ and p-value for drugs and targets in the BioSNAP in-domain split training set. **d** Label frequency distribution of $Z$ for drugs and targets in the BioSNAP cross-domain split training set. **e** Quantification of the overall "prior tendency" for labels associated with drugs and targets across different datasets. *P*-values were derived from a one-sided permutation test with 1000 iterations ($P_d$ for drugs, $P_t$ for targets), with no adjustments for multiple comparisons. Source data are provided as a Source Data file.

merged sets $S_p$ and $S_n$, resulting in a training set with a 50% positive-negative split, where drugs have "prior tendency" while targets have a balanced label distribution.

"Balanced" training set: as shown in Fig. 4d, we formed a positive pairs set $S_p$, same as drug biased training set, and generated a negative pairs set $S_n$ by randomly shuffling drugs within $S_p$ to create negative samples. Finally, we combined sets $S_p$ and $S_n$ to produce a training set with a balanced distribution of positive and negative labels for every single drug and target.

We trained DrugBAN on the two counter-prior training sets using the same hyperparameters and conducted the bias test. Visualization of classification features for (**T**, **R**) and (**D**, **R**) in Fig. 4b, c

reveals that, unlike the pronounced clustering of (**T**, **R**) in Fig. 2, the positive pairs in Fig. 4c (**D**, **R**) show greater degree of clustering, whereas those in Fig. 4b (**T**, **R**) does not. In contrast, on the balanced training set, both (**T**, **R**) and (**D**, **R**) in Fig. 4e, f exhibits random distributions. This suggests that higher "prior tendency" in the training data leads to biased predictions. Consequently, merely adding more features[29] or altering encoders and aggregators can not solve the issue of biased prediction as long as "prior tendency" persists in the data.

Given that the publicly available DTI sequence datasets, BioSNAP and BindingDB, exhibit stronger "target prior bias," we specifically address it in our proposed method.

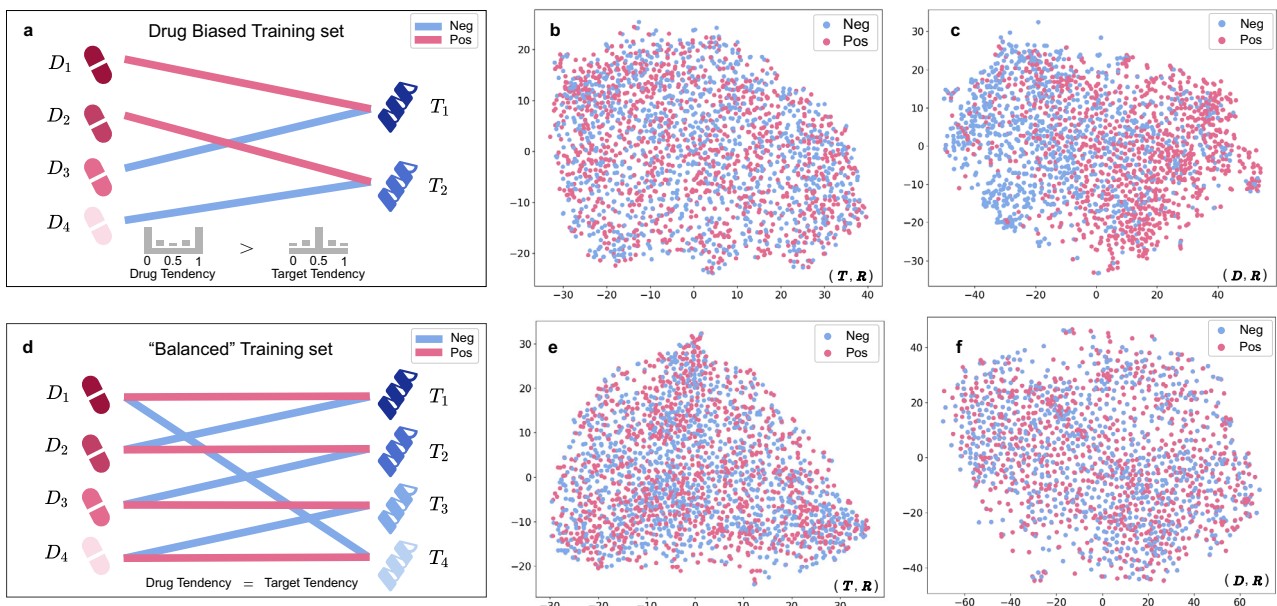

**Fig. 4 | Construction of drug-biased and "balanced" training sets, and t-SNE visualization of classification features (D, R) and (T, R) for DrugBAN. a** A sketch of the drug biased training set construction. **b** Test of "target bias" in the drug-biased training set using (**T**, **R**). **c** Test of "drug bias" in the drug-biased training set using (**D**, **R**). **d** A sketch of the "balanced" training set construction. **e** Test of "target bias" in the "balanced" training set using (**T**, **R**). **f** Test of "drug bias" in the "balanced" training set using (**D**, **R**).

## TAPB framework

In this paper, we introduce an interventional debiasing framework for alleviating target prior bias in drug-target interaction prediction (TAPB) as shown in Fig. 5.

The TAPB framework fundamentally differs from conventional DTI models through its integration of amino acid randomization, confounder alignment module, and interventional training to estimate $P(Y|\mathbf{D}, do(\mathbf{T}))$ via backdoor adjustment. As shown in Fig. 5a, the interventional training computes $P(Y|\mathbf{D}, do(\mathbf{T}))$ via backdoor adjustment by incorporating all target confounder clusters $c_i \in \mathbf{C}$. This requires the confounder dictionary $\mathbf{C}$ and the confounder alignment module $g_t(\cdot)$ as prerequisites.

The confounder dictionary $\mathbf{C}$ is constructed through K-Means[30] clustering on ESM-2 features from all training targets, as shown in Fig. 5b. The cluster centers constitute the dictionary $\mathbf{C}$, while the sample proportion within each cluster $c_i$ defines the adjustment weight $P(c_i)$. Since ESM-2 was pre-trained on datasets disjoint from DTI benchmarks, this eliminates the risk of label leakage.

The confounder alignment module $g_t(\cdot)$, illustrated in Fig. 5d, operates during interventional training. It processes each confounder cluster center $c_i$ to generate confounder-conditioned representations $\mathbf{T}_{c_i}$, and partitioned fused features $\mathbf{F}_{c_i}$. A shared classifier $g_y(\cdot)$ then computes $P(Y|\mathbf{D}, \mathbf{T}, c_i)$ for all $\mathbf{F}_{c_i}$, enabling the computation of $P(Y|\mathbf{D}, do(\mathbf{T}))$ via backdoor adjustment under our SCM.

Amino acid randomization in Fig. 5c regularizes input sequences. First, 70% of residues in ESM-2 features are randomly deleted to reduce computation and disrupt sequence patterns. Subsequently, each residue feature undergoes independent mutation with 20% probability by replacement via random sampling from the amino acid dictionary. This dual randomization prevents spurious correlation learning by disrupting label-specific motifs.

We did not employ unsupervised domain adaptation (UDA) techniques, e.g. CDAN[31], and achieved better results under the cross-domain settings, indicating the strong generalization of TAPB. Note that TAPB is a debiasing framework, and replacing encoders or aggregators can further enhance performance. The components of TAPB are generic, with the computation of backdoor adjustment

requiring the satisfaction of certain assumptions. The pseudocode of our method is provided in Supplementary Algorithm 1.

## Datasets and evaluation protocol

To ensure a rigorous and comprehensive assessment, we evaluated the model's classification performance across four publicly available datasets under six settings: in-domain and cross-domain splits of BindingDB and BioSNAP datasets, in-domain split of the Davis dataset, and cold split of the Human dataset. Supplementary Note 1 and Supplementary Table 1 provide an overview of the datasets. Additionally, Supplementary Note 5 and Supplementary Fig. 3 reveal "target prior bias" in the Davis dataset and "drug prior bias" in the Human dataset.

For the in-domain splits, datasets were randomly divided into training, validation, and test sets in a 7:1:2 ratio. Notably, in these in-domain scenarios, targets exhibit significantly higher overlap across training, validation, and test sets compared to drugs. In contrast, the cross-domain splits-constructed by DrugBAN-consist of a source domain training set, a target domain training set, and a target domain test set, with no overlap between source domain drugs/targets and the target domain data (CVS4).

For all datasets, we performed five independent runs with different random seeds and reported the area under the receiver operating characteristic curve (AUROC), the area under the precision-recall curve (AUPRC), accuracy, sensitivity, and specificity. The Youden Index was adopted to adjust the optimal threshold, offering a more effective balance between sensitivity and specificity. For the in-domain splits, we selected the model checkpoint with the highest validation AUROC and reported test set performance. Following DrugBAN's protocol for cross-domain datasets, we trained models without domain adaptation techniques on the source domain and evaluated them directly on the target domain test set, reporting the resulting metrics.

We compared TAPB with five baseline models-MolTrans[7], TransformerCPI[15], DrugBAN[12], PSICHIC[8], and MlanDTI[9]. Unlike these models, which rely on $P(Y|\mathbf{D}, \mathbf{T})$ for predictions, TAPB computes $P(Y|\mathbf{D}, do(\mathbf{T}))$ via backdoor adjustment for predictions. The hyperparameter settings for TransformerCPI, MolTrans, DrugBAN (on the in-domain splits of BindingDB, BioSNAP, and Davis and cold split of the

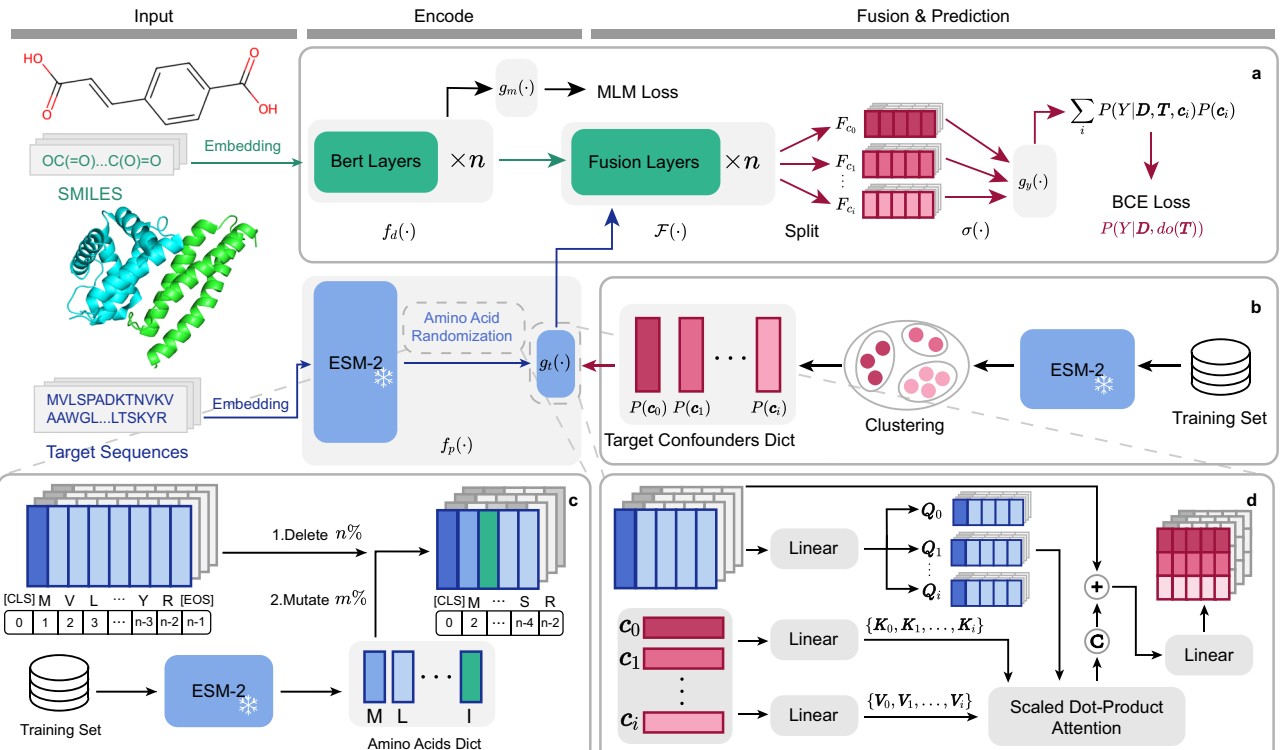

**Fig. 5 | Architecture of the TAPB framework. a** TAPB Interventional Training: The drug encoder BERT $f_d(\cdot)$ generates drug features **D** from SMILES. ESM-2 pre-extracted target features **E** undergo amino acid randomization and are then processed by the CAM $g_t$. All cluster centers $c_i \in \mathbf{C}$ act as keys/values in CAM $g_t$ with **E**. Fused features **F** are partitioned into $I$ segments $\mathbf{F}_{c_i}$, each globally pooled and fed to classifier $g'_y$ for estimating confounder-conditioned probabilities $P(Y|\mathbf{D}, \mathbf{T}, \mathbf{c_i})$. Finally, $P(Y|\mathbf{D}, \mathbf{do}(\mathbf{T}))$ is computed via backdoor adjustment. **b** Target Confounder

Dictionary **C**: Obtained via K-Means clustering on ESM-2 target features from training sets. **c** Amino Acid Randomization: 1. Random deletion of 70% residue features; 2. Mutation of remaining residues to random features from the amino acid dictionary. **d** Confounder Alignment Module (CAM, $g_t(\cdot)$): Attention-weighted summation fuses $\mathbf{C_i}$ with target features, followed by dimensionality reduction and residual connection, maintaining explicit path $X_t \to \mathbf{C} \to \mathbf{T}$ across training.

Human dataset), DrugBAN-da (on the cross-domain splits of BindingDB and BioSNAP datasets), TAPB, PSICHIC, and MlanDTI are detailed in Supplementary Note 4 and Supplementary Tables 9–15. For each model, hyperparameters remained consistent across all datasets unless otherwise specified. The key hyperparameters of TAPB-target confounder dictionary size, target random deletion ratio, and mutation rate-were tuned on the cross-domain split of the BioSNAP dataset, as shown in Supplementary Fig. 1. Accordingly, comparative conclusions were not drawn from this dataset. A summary of the per-seed AUROC and AUPRC values across all hyperparameter tuning experiments is provided in Supplementary Note 2 and Supplementary Table 2.

**In-domain comparison**

TAPB exhibits comprehensive dominance over all baselines on the in-domain split of the BioSNAP dataset, as evidenced by its larger polygon area across each evaluation metric in Fig. 6a. As shown in Supplementary Table 3, compared to the next best baseline PSICHIC, TAPB shows significant improvements: a 2.3% increase in AUROC; 2.2% in AUPRC; 2.7% in accuracy; 3.4% in sensitivity; and 1.9% in specificity. These statistically significant gains underscore TAPB's effectiveness in alleviating target prior bias.

Despite the severe "target bias" in BindingDB, where models can achieve high performance by merely memorizing the target-leading to strong results across all baselines-we still conducted a fair comparison. As illustrated in Fig. 6b, TAPB maintains strong competitiveness, narrowly trailing the top-performing method DrugBAN by only 0.2% in AUROC and 0.3% in AUPRC as shown in Supplementary Table 3, demonstrating competitive performance.

Notably, TAPB outperforms all baselines across every metric on the Davis dataset, as shown in Fig. 6c. Supplementary Table 3 confirms that it exceeds the strongest baseline, PSICHIC, by 2.1% in AUROC and 7.4% in AUPRC-the largest performance gap observed among all datasets. This underscores TAPB's exceptional ability to capture complex interaction patterns. In the challenging cold split scenario of the Human dataset, where "drug bias" is the dominant one, we also evaluated TAPB's performance under this opposite condition. As shown in Fig. 6f, TAPB maintains competitive performance, surpassing DrugBAN by 2.0% in AUROC and 2.8% in AUPRC, and outperforming three out of five baselines. This result demonstrates that TAPB, although designed to alleviate "target bias", also achieves strong performance on drug-biased datasets, highlighting its robustness and generalizability beyond its intended application context.

**Cross-domain comparison**

As illustrated in Fig. 6d, e, TAPB exhibits excellent cross-domain generalization capabilities. Comprehensive performance comparisons are provided in Supplementary Table 4. On the cross-domain split of the BindingDB dataset, TAPB maintains strong competitiveness, delivering notable results with an AUROC of 0.676, accuracy of 0.630, and specificity of 0.565, while surpassing DrugBAN-da (w/ CDAN) by 7.5% in AUROC, 5.8% in AUPRC, and 5.1% in accuracy. Notably, even without using domain adaptation, our method still outperforms DrugBAN-da and exhibits superior generalization, validating the effectiveness of our debiasing framework. The results on the cross-domain split of the BioSNAP dataset, as shown in Fig. 6d, are included for completeness, for the interested reader.

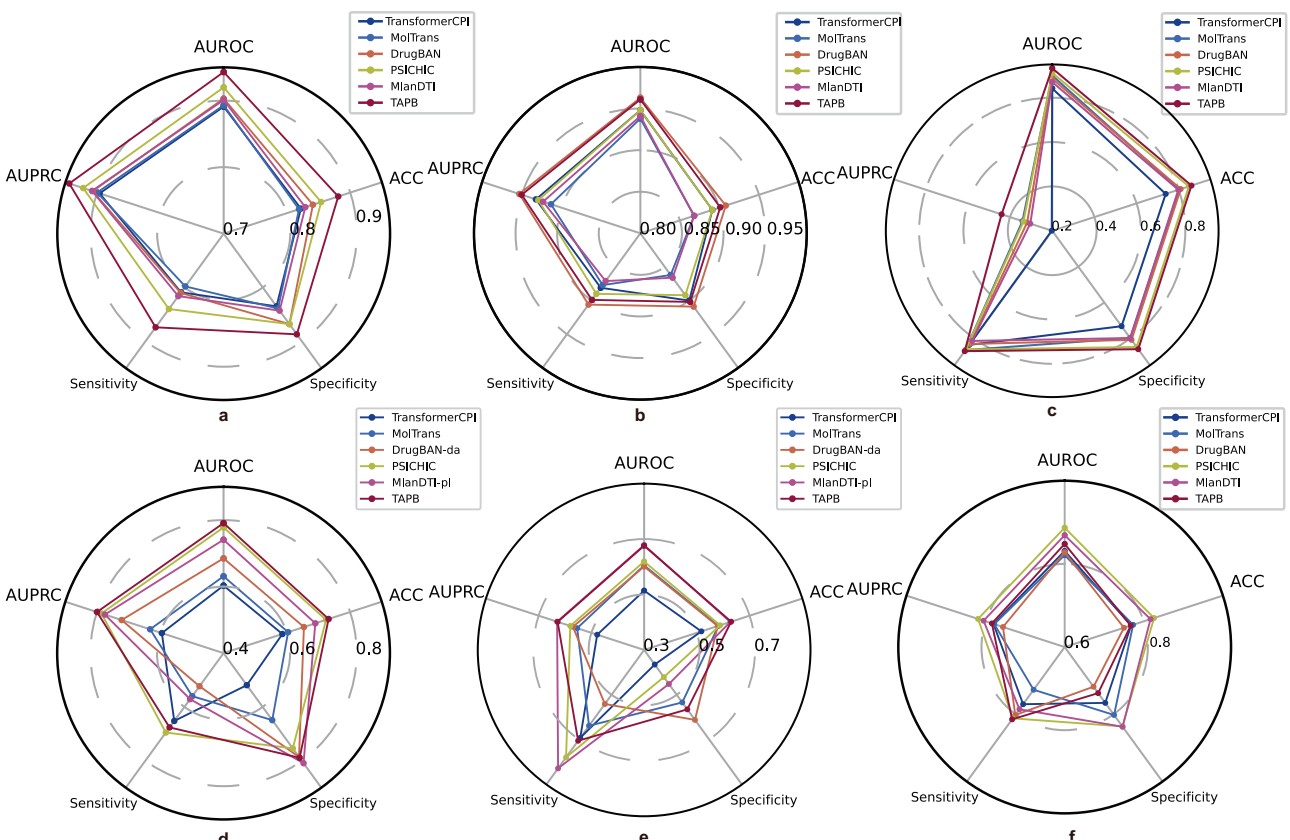

**Fig. 6 | Radar chart comparisons of performance on BioSNAP and BindingDB, Davis, and Human. a** In-domain evaluation on BioSNAP. **b** In-domain evaluation on BindingDB. **c** In-domain evaluation on Davis. **d** Cross-domain evaluation on BioSNAP. **e** Cross-domain evaluation on BindingDB. **f** Cold-split evaluation on Human. Experiments were performed with five different random seeds across all datasets. Source data are provided as a Source Data file.

TAPB's strong cross-domain generalizability stems from its core approach of mitigating target prior bias, thereby avoiding reliance on spurious target-label correlations. Conventional models relying on $P(Y|D, T)$ to predict drug-target interactions exhibit severe performance degradation when encountering out-of-distribution targets. In contrast, TAPB's interventional training paradigm, which incorporates amino acid randomization, disrupts these spurious correlations. Our method enables consistent generalization beyond training distributions, allowing TAPB to disentangle authentic DTI patterns from dataset-specific biases.

## Ablation study

We conducted ablation studies on three datasets: the in-domain split of the Davis dataset, and the cross-domain splits of the BioSNAP and BindingDB datasets. Using a total of seven TAPB variants, these studies were designed to comprehensively evaluate the impact of our key components: (1) TAPB-CNN: Replacing the ESM-2 encoder with an untrained CNN, (2) TAPB-Base: Baseline dual-tower architecture with ESM-2 encoders, binary classification loss, and average pooling (w/o interventional training), (3) TAPB-R: TAPB-Base augmented with amino acid randomization, (4) TAPB-RM: TAPB-R enhanced with masked language modeling (MLM) loss, (5) TAPB-RM-BA: TAPB-RM with backdoor adjustment (without CAM, omitting $X_t \rightarrow C \rightarrow T$), (6) TAPB-RM-CAM: TAPB-RM with CAM (w/o backdoor adjustment), and (7) TAPB-Full: Complete TAPB model integrating all proposed components (ESM-2, randomization, MLM, CAM, and backdoor adjustment). Unless specified, all experiments of TAPB were conducted with identical hyperparameters to those in Supplementary Table 13. Each experiment included five independent runs with different random seeds. Comprehensive ablation results

and residue random deletion generalizability are presented in Supplementary Tables 5–8.

ESM-2 encoder contribution: Given ESM-2's strong representation capacity, we ablated its usage in the TAPB-Base architecture by replacing it with a randomly initialized CNN encoder (denoted TAPB-CNN). As shown in Fig. 7a–c, TAPB-Base significantly outperformed the TAPB-CNN, demonstrating the advantage of pretrained protein encoders. Meanwhile, to confirm that the ESM-2 features do not cause "target bias", we conducted "target bias" and "drug bias" tests on the "balanced" dataset introduced in our previous section, as detailed in Supplementary Note 3. As shown in Supplementary Fig. 2, neither test exhibited clustering similar to that in Fig. 2, indicating that incorporating the ESM-2 encoder enhances target representation and does not cause "target bias", which is primarily triggered by the data.

Amino acid randomization and MLM loss: Amino acid randomization significantly enhances model performance and serves as the most direct approach to prevent model from memorizing the target, thereby avoiding insufficient learning of interaction patterns. As shown in Fig. 7a–c, TAPB-R consistently achieves higher AUROC and AUPRC scores than TAPB-Base across all three datasets, particularly on the Davis dataset, demonstrating the effectiveness of our randomization strategy and validating the rationale behind preventing target memorization. TAPB-RM marginally outperforms both TAPB-R and TAPB-Base on all three datasets. Although the performance improvement is less pronounced compared to amino acid randomization, the drug MLM loss effectively strengthens drug representation in target-biased datasets, thereby reducing the influence of the target.

Interventional training: According to our theoretical analysis, TAPB requires both CAM and backdoor adjustment to compute

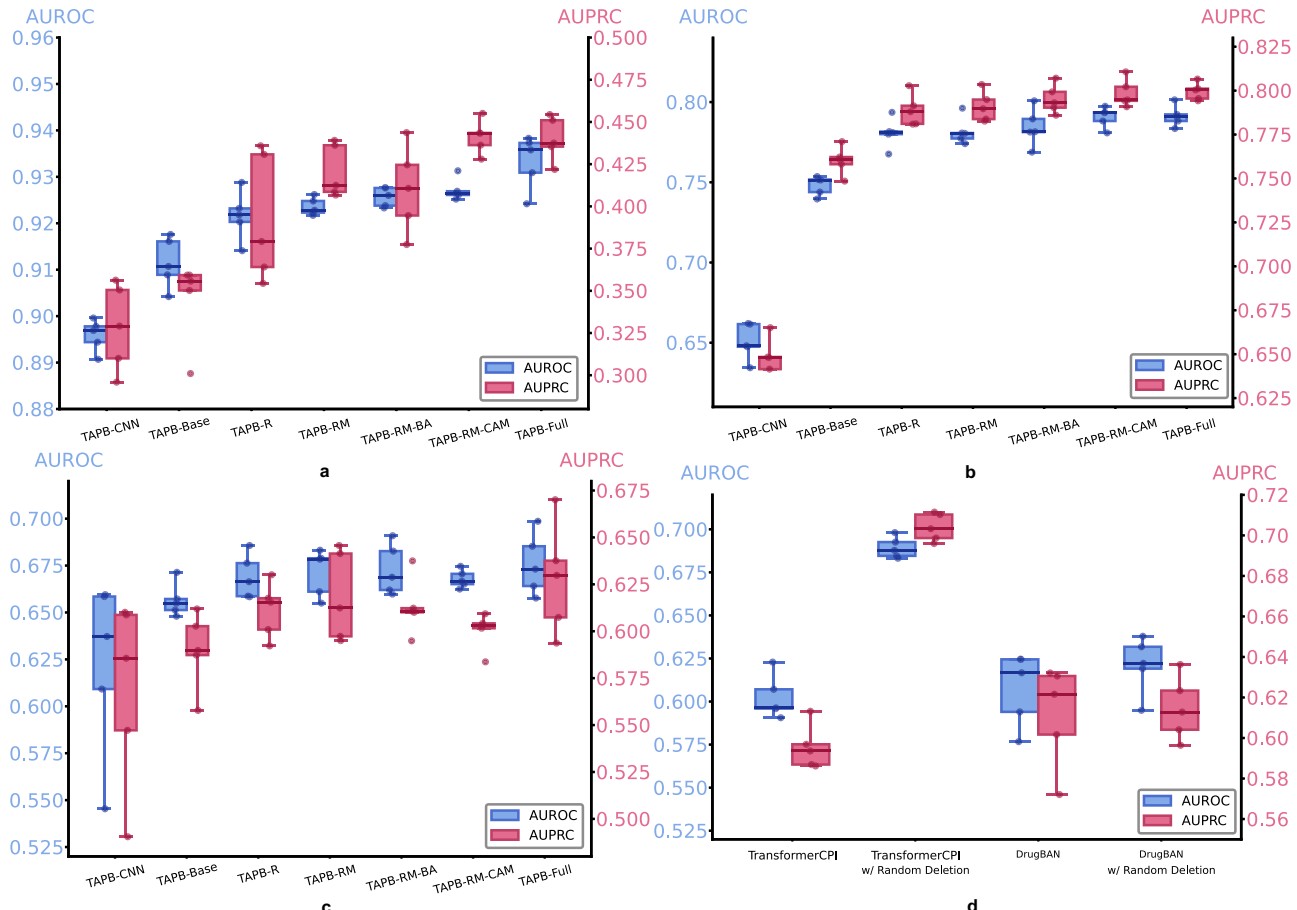

**Fig. 7 | Ablation studies results. a** Ablation study of TAPB key components on the Davis dataset. **b** Ablation study of TAPB key components on the cross-domain split of the BioSNAP dataset. **c** Ablation study of TAPB key components on the cross-domain split of the BindingDB dataset. **d** AUROC and AURPC of TransformerCPI, TransformerCPI w/ random deletion, DrugBAN, and DrugBAN w/ random deletion on the cross-domain split of the BioSNAP dataset. Ablation studies were performed with five different random seeds across all datasets. Box plots display the median (centre), 25–75th percentiles (box bounds), and minima-maxima within 3 times IQR (whiskers). Individual data points ($n = 5$) are overlaid. Source data are provided as a Source Data file.

$P(Y|\mathbf{D}, do(\mathbf{T}))$. Solely employing CAM violates the assumptions of our SCM, while the backdoor adjustment is specifically designed for our SCM and is theoretically invalid without CAM. To validate this, we designed ablation variants-TAPB-RM-BA, TAPB-RM-CAM, and TAPB-Full. As shown in Fig. 7a–c, when operating with only one module, TAPB-RM-BA and TAPB-RM-CAM exhibit comparable performance, while significant performance gains are exclusively observed in TAPB-Full. This pattern is particularly pronounced on the Davis dataset and consistently evident across the BioSNAP and BindingDB datasets. The comparative analysis of these three variants empirically validates that our theoretically grounded design aligns with the expected theoretical outcomes.

Generalizability of residue random deletion: To test the generalizability of our approach, we integrated residue random deletion into DrugBAN (Non_DA) and TransformerCPI, using the hyperparameters specified in Supplementary Tables 16 and 9, respectively. Only residue random deletion was selected because our residue mutation and interventional training require both a pretrained encoder and MHCA-based aggregation. Results on the cross-domain split of the BioSNAP dataset are as shown in Fig. 7d, TransformerCPI with random deletion exhibited substantial gains of nearly 10% in both AUROC and AUPRC, while DrugBAN with random deletion showed a 1% AUROC improvement over the baseline. This discrepancy may arise from TransformerCPI's aggregator architecture being more suitable for modeling DTI under this modification. These

results confirm our residual random deletion as a general, model-agnostic design.

## Interpretability of TAPB

TAPB provides insights at both molecular and amino acid levels, offering useful information for drug repurposing. The model uses eight attention heads in the final layer of the aggregator, each capturing distinct interaction patterns. These attention maps are visualized to interpret the model's focus. To highlight potential binding sites, we aggregate the multi-head attention maps by averaging over the attention heads, yielding separate attention scores for drugs and targets. These scores are compared with ground truth ligand–protein interaction maps obtained from X-ray crystallography, with interactions visualized contacts within 5 Å radius of ligand. For targets, key regions around binding sites are highlighted based on attention maps, with important amino acids distinctly colored. The model-predicted interactions matching the ground truth are marked with a red box. Additionally, the top five atoms in the drug attention maps, which indicate their predicted contributions to binding, are visualized using RDKit[32].

Docking calculations were performed using AutoDock Vina (v1.2.5)[33]. 2D ligand–protein interaction diagrams were generated with the Ligand Interaction Diagram module in Maestro (v13.5, Schrödinger LLC), and 3D interaction diagrams were prepared using PyMOL (v2.5,

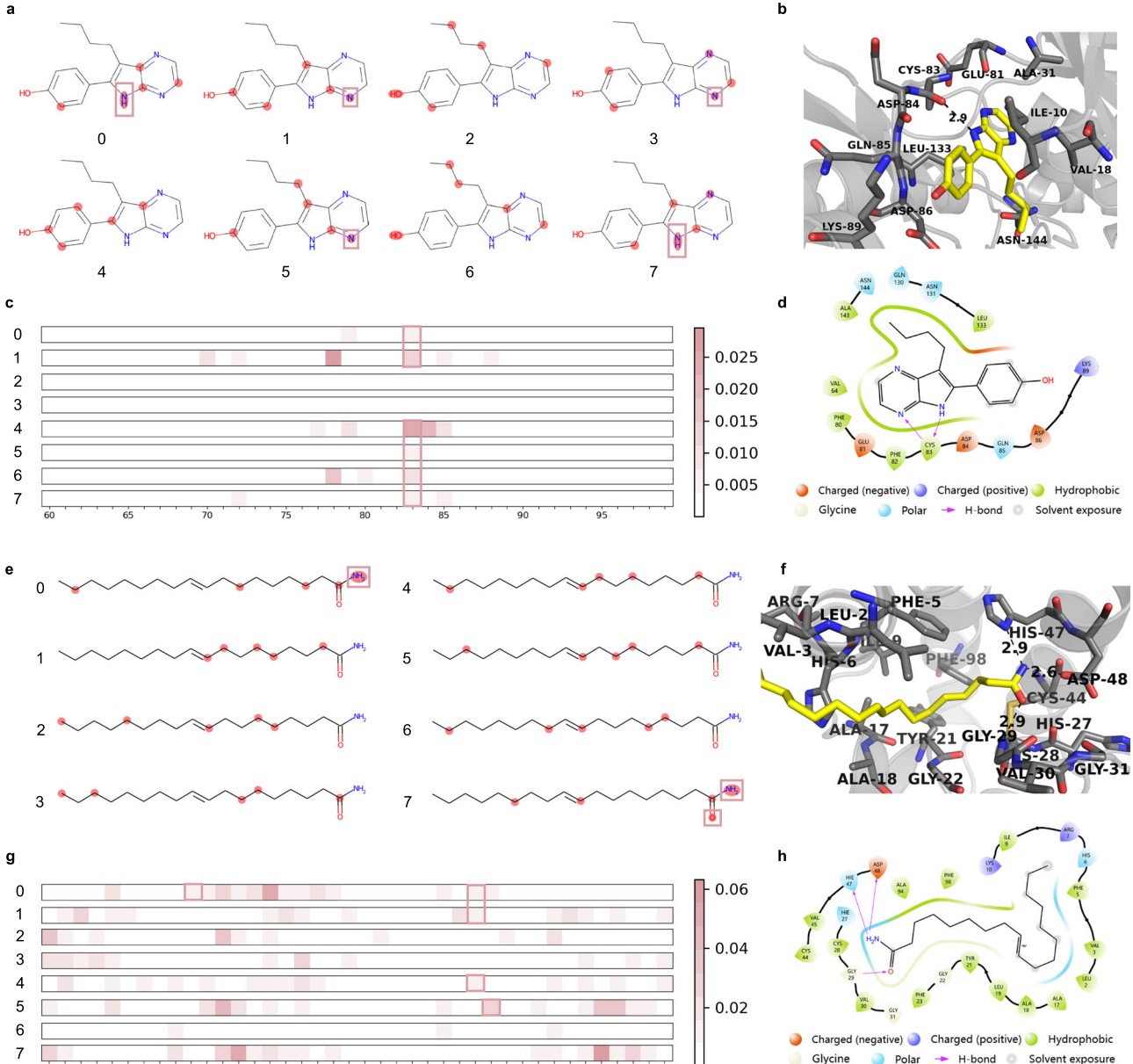

**Fig. 8 | Visualization of TAPB's attention maps and comparison with actual ligand-protein binding sites. a** Drug Aloisine A 2D structure with top 5 highlighted atoms based on attention maps. **b** The interactions and real binding sites of Aloisine A in the ligand-protein complex structure (PDB ID: 1UNG). **c** Target attention maps for 1UNG, highlighting amino acids in the protein structure. **d** The interactions and real binding sites of Aloisine A in ligand-protein complex structure (PDB ID: 1UNG).

**e** Drug Elaidamide 2D structure with top 5 highlighted atoms based on attention maps. **f** The interactions and real binding sites of Elaidamide in the ligand-protein complex structure (PDB ID: 1KQU). **g** Target attention maps for 1KQU, highlighting amino acids in the protein structure. **h** The interactions and real binding sites of Elaidamide in the ligand-protein complex structure (PDB ID: 1KQU). The dashed lines in the 3D interaction diagrams represent H bound.

Schrödinger LLC; https://pymol.org), with other residues, secondary structure elements, and surface maps shown in gray.

Two positive co-crystallized structures sourced from the Protein Data Bank (PDB)[34] were chosen from the BioSNAP in-domain test set: Aloisine A (PDB ID: 1UNG)[35] and Elaidamide (PDB ID: 1KQU)[36].

For PDB ID: 1UNG, Aloisine A (RP107) is a potent cyclin-dependent kinase (CDK) inhibitor. TAPB's drug attention identifies these hydrogen bonds and interaction sites in both 2D and 3D docking diagrams, as indicated in Fig. 8a0, a1, a3, a5 and a7, b, d. The model captures a nitrogen atom acting as a hydrogen bond acceptor, interacting with the main chain of CYS83, while another nitrogen atom acts as a hydrogen bond donor interacting with the same residue, as displayed in Fig. 8a0, a7. Furthermore, Fig. 8c0, c1, c4, c5, c6, c7 emphasizes the

role of CYS83 in ligand-protein binding, further validating TAPB's precise detection of the true binding sites.

For PDB ID: 1KQU(Human phospholipase A2 complexed with a substrate analog), Elaidamide is a fatty acid amide that has been found in the cerebrospinal fluid of sleep-deprived cats and inhibits human synovial phospholipase A2 (PLA2). TAPB accurately identifies these interaction sites: the hydroxyl group acting as a hydrogen bond donor, interacting with the main chain of GLY29, and the amino group functioning as a hydrogen bond donor, interacting with HIS47 and ASP48, as depicted in Fig. 8e0, e7, g0, g1, g4, g5. The target attention map correctly highlights the significance of GLY29, HIS47, and ASP48 in ligand-protein binding, as illustrated in Fig. 8 g0, g1, g4, g5. We again present 3D and 2D docking diagrams,

showing two hydrogen bonds within 5 Å, as depicted in Fig. 8f and h, respectively.

Although DTI models from previous studies provided interpretability and could reveal hidden interactions, they were trained on biased data, making them susceptible to "target prior bias" and potentially genuine interactions. TAPB effectively identifies and mitigates this bias, markedly improving the accuracy of interaction detection. Consequently, TAPB provides more reliable predictions for downstream computational screening and experimental validation.

## Discussion

Our study successfully identifies and mitigates "target prior bias," a phenomenon that has been underappreciated in previous studies. Through a series of experiments, we confirmed that "prior tendency," characterized by the imbalanced label distribution of targets, is a confounder that leads to a spurious correlation between targets and predictions in DTI prediction. Our proposed framework, TAPB, effectively addresses this bias by employing amino acid randomization, CAM and interventional training. These methods not only improve generalization but also yield stronger predictive capability, ultimately producing a more robust and reliable model.

The concept of "prior tendency" in this study extends beyond merely the distribution of labels in the dataset. It encompasses a broader spectrum of potential biases that can arise from various sources, including specific functional groups in drugs, subsequences in targets, or even other non-sequence features[37]. This bias can lead models to capture spurious correlations rather than genuine drug-target interactions, thereby impairing their generalization. The mitigation of bias is not limited to backdoor adjustment alone. Various approaches, including contrastive learning in multimodal framework, e.g. CLIP[38] and ConPLex[39], can effectively address this bias. However, it is crucial to recognize that eliminating prior bias does not guarantee complete bias removal, as other forms of bias[40] may persist. Achieving truly accurate and reliable DTI prediction remains an ongoing challenge that requires sustained research efforts and methodological innovations. Future DTI models should be trained on datasets that are as free as possible from such biases and should be evaluated based on biological metrics rather than merely algorithmic performance. These biological metrics could potentially be distinct from the labels present in the training data, thereby compelling the model to uncover more authentic interactions.

Although TAPB accurately predicts the binding sites for both Aloisine A and 1UNG, and Elaidamide and 1KQU, it also generates a significant amount of noise. For instance, only a few attention heads in Fig. 8a, e explore the true binding sites, and there is low consistency across heads in predicting these sites. Similarly, for the target, as shown in Fig. 8c, g, the attention weights for each amino acid are relatively small, likely due to the long sequence and the Softmax normalization. Additionally, the attention heads that focus on drug and target interactions are different, suggesting that the model may not fully synchronize the relevant attention mechanisms for both. This inconsistency implies that TAPB's predictions are not entirely stable and could be influenced by latent biases, similar to "target prior bias."

UDA techniques, e.g. CDAN[31] used in DrugBAN[12] and MCD[41] used in UdanDTI[16], require access to both source and target domain data for model adaptation, which typically leads to improved cross-domain generalization performance. In contrast, we aim to explore a more universal and convenient zero-shot prediction paradigm, where TAPB utilizes only the source domain training set. This approach avoids the computational burden and application complexity associated with repeatedly constructing target domain sets and retraining for novel drugs or targets. Our comparisons with UdanDTI are provided in Supplementary Note 6, Supplementary Tables 17, 18.

There have been numerous efforts to construct unbiased datasets, yet creating a completely unbiased DTI dataset remains challenging. In this paper, we provide new insights to address bias from causal perspective. The implications of our findings extend beyond DTI prediction, as the "prior tendency" phenomenon could be prevalent in other domains. Future research could explore the application of TAPB in other areas, e.g. DTA[11], multi-view fusion[42], or VQA[43], where similar biases may occur. Additionally, further investigation into the mechanisms underlying "prior tendency" from causal lens could lead to the development of more robust models that are less susceptible to spurious correlations. As DTI prediction continues to evolve, the integration of causal inference techniques will be crucial in ensuring that models capture genuine interactions and generalize effectively to new data.

Since the confounder is unobservable, we attempted to implement the proxy variables based confounder adjustment method from ref. 44. However, significant computational challenges emerged when integrating this into our deep learning pipeline, particularly regarding the reliable estimation of the distribution and numerical instability during matrix inversion.

Notably, current biological dataset constraints limit proxy variables to sequence-derived features, which may not sufficiently capture the full spectrum of confounding biological mechanisms. Future incorporation of multimodal data, e.g. structural or functional annotations, could enhance proxy quality by providing orthogonal information sources that better approximate latent confounders.

While causal inference theory provides principled solutions for unobserved confounders, e.g. refs. 44–46, adaptation to deep learning frameworks remains nontrivial. We explicitly acknowledge these limitations in our discussion and will prioritize bridging this methodological gap in future work, with particular focus on multimodal proxy refinement.

While our adjustment for **C** satisfies the backdoor criterion and is theoretically exact for causal effect identification in the SCM of Fig. 9b, where amino acid randomization effectively disrupts target patterns, different valid adjustment sets may vary significantly in their finite-sample performance. As demonstrated by Runge[47], in SCM with hidden variables, multiple adjustment sets can be theoretically equivalent for causal identification but exhibit different asymptotic variances. For observable adjustment sets, there exist optimal minimal adjustment sets that yield the smallest asymptotic variance among all minimal valid sets[48]. Our choice of adjusting for **C** balances statistical robustness and computational efficiency, acknowledging that while alternative valid adjustment sets might offer improved statistical efficiency in certain scenarios, they may entail higher computational costs or data requirements. Future work could explore optimal adjustment set selection specifically for DTI predictions.

## Methods

### Analysis DTI through causal inference

We construct a structural causal model (SCM)[49] to demonstrate the causal relationships within the DTI model. As shown in Fig. 9a, there are 6 nodes and 6 edges in the conventional DTI model's SCM, and TAPB introduces an additional node **C** and establishes the path $X_t \rightarrow \mathbf{C} \rightarrow \mathbf{T}$. $X_d$ represents SMILES in training set, $X_t$ represents target sequences in training set, **D** represents drug feature extracted by drug encoder $f_d(\cdot)$, **T** represents target feature extracted by target encoder $f_t(\cdot)$, **C** represents our target confounder dictionary obtained via K-Means clustering, **F** represents the fusion feature, and $Y$ represents the prediction. "Target prior bias" hides in $X_t$.

$X_d \rightarrow \mathbf{D}$: This path indicate that the drug feature **D** is extracted from the SMILES $X_d$ in the training set.

$X_t \rightarrow \mathbf{C} \rightarrow \mathbf{T}$: This path indicates that the target feature **T** is extracted from the sequence $X_t$ via the clustering **C**.

$\mathbf{D} \rightarrow \mathbf{F}$ and $\mathbf{T} \rightarrow \mathbf{F}$: These two paths indicate the generation of fusion feature **F** by aggregator $\mathcal{F}(\cdot)$.

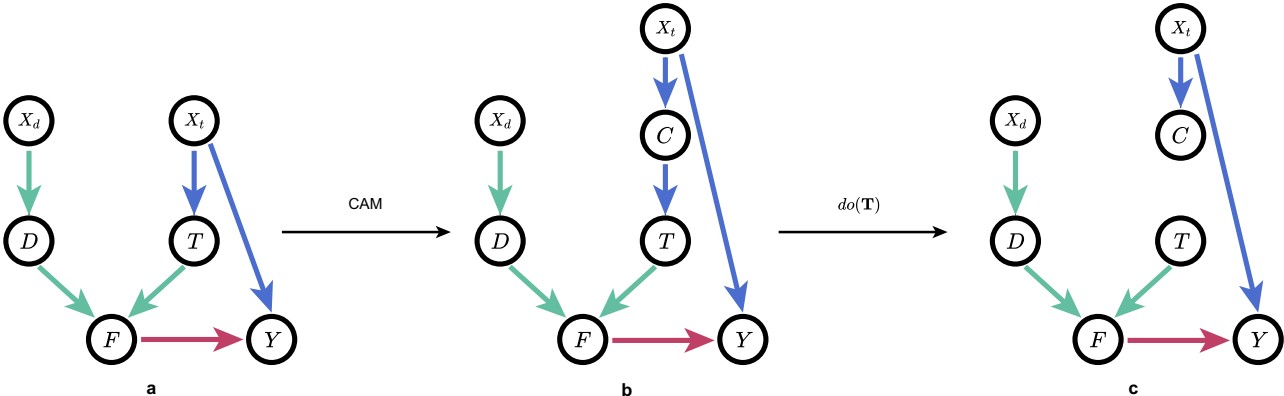

**Fig. 9 | SCM of conventional DTI models and TAPB. a** SCM of the conventional DTI models exhibiting "target prior bias." **b** SCM of biased training without amino acid randomization, adding **C** into $X_t \to$ **T**. **c** SCM for $P(Y|\mathbf{D}, do(\mathbf{T}))$ where $do(\mathbf{T})$ blocks the path $X_t \to \mathbf{C} \to \mathbf{T}$. We compute $P(Y|\mathbf{D}, do(\mathbf{T}))$ via backdoor adjustment without actual intervention.

**F** → *Y*: This path indicates that the prediction *Y* is based on fusion feature **F**.

$X_t$ → *Y*: This path indicates that the prediction *Y* is affected by $X_t$, i.e., the "target prior bias" causes biased prediction.

Figure 9a shows that SCM exists backdoor paths **T** ← $X_t$ → *Y*. From causal lens, the prior of $X_t$ confounds **T** and *Y*, leading to spurious correlations. To suppress this bias, a more effective mechanism is needed to handle the actual causal relationship between drug, target, and predictions.

**TAPB encoders.** Drug encoder: We employ BERT with rotational position encoding (RoPE)[50] as the drug encoder. The input SMILES sequences $X_t$ in one training batch are tokenized using Molformer[51] dictionary and tokenizer, and then embedded into a high-dimensional representations $E_d \in \mathbb{R}^{B \times L_d \times D_m}$:

$$\mathbf{E_d} = \text{Embedding}(X_d) \tag{7}$$

where *B* denotes the batch size, $L_d$ denotes the length of $X_t$ and $D_m$ denotes the dimension of the model. Next, TAPB stacks *n* layers (*n* = 3) of BERT layers with RoPE to construct more complex and abstract contextual representations. The output $\mathbf{D} \in \mathbb{R}^{B \times L_d \times D_m}$ of the entire drug encoder is a context-sensitive depth representation of the input drug SMILES:

$$\mathbf{D} = f_d(\mathbf{E_d}) = \text{BERT}(\mathbf{E_d}) \tag{8}$$

Target encoder: We employ ESM-2[52] as the target encoder. ESM-2 is employed as the ESMFold protein feature encoder, replacing multiple sequence alignment (MSA) and structural template parts, with positional embeddings modified to RoPE, and supports longer amino acid sequence encoding. Given that ESMFold is trained on significantly larger datasets and demonstrates competitive performance compared to AlphaFold[53], ESM-2 exhibits exceptional capability in extracting 3D structural information from protein sequences, making it highly suitable for drug-target interaction (DTI) prediction tasks. The input target amino acid sequences $X_d$ in one training batch are tokenized using the ESM-2 dictionary and tokenizer, and then embedded into high-dimensional representations $\mathbf{E_t} \in \mathbb{R}^{B \times L_t \times D_e}$:

$$\mathbf{E_t} = \text{Embedding}(X_t) \tag{9}$$

where $L_t$ denotes the length of $X_t$ and $D_e$ denotes the dimension of the ESM-2 encoded feature. The ESM-2 is then used to extract ESM-2 encoded features $\mathbf{E} \in \mathbb{R}^{B \times L_t \times D_e}$:

$$\mathbf{E} = f_t(\mathbf{E_t}) = \text{ESM-2}(\mathbf{E_t}) \tag{10}$$

In practical training, ESM-2 encoded features are pre-extracted and saved, significantly reducing memory burden and accelerating the training process.

**Aggregator**

Following TransformerCPI, TAPB adopts the same aggregator $\mathcal{F}(\cdot)$ with Multi-head Cross Attention (MHCA), which is essential for our estimation of confounder-conditioned probabilities $P(Y|\mathbf{D}, \mathbf{T}, \mathbf{c_i})$. First, each fusion layer of the aggregator takes the output of the previous layer $\mathbf{F} \in \mathbb{R}^{B \times L_d \times D_m}$ into the self-attention layer, followed by residual connection and layer normalization:

$$\mathbf{F} = \text{LayerNorm}(\mathbf{F} + \text{Self Attention}(\mathbf{F}, \mathbf{F}, \mathbf{F})) \tag{11}$$

The output **F** is then transformed through a linear layer to obtain $\mathbf{Q} \in \mathbb{R}^{B \times L_d \times D_m}$, while target features **T** are projected via two separate linear layers into $\mathbf{K} \in \mathbb{R}^{B \times L_t \times D_m}$ and $\mathbf{V} \in \mathbb{R}^{B \times L_t \times D_m}$, which are fed into the cross-attention layer followed by residual connection and layer normalization:

$$\mathbf{F} = \text{LayerNorm}(\mathbf{F} + \text{MHCA}(\mathbf{Q}, \mathbf{K}, \mathbf{V})) \tag{12}$$

The MHCA is formally defined as:

$$\text{MHCA}(\mathbf{Q}, \mathbf{K}, \mathbf{V}) = g_a(\text{Concat}(\mathbf{h}_1, \ldots, \mathbf{h}_i)) \tag{13}$$

$$\mathbf{h}_i = \text{Softmax}\left(\frac{\mathbf{Q_i}\mathbf{K_i}^\top}{\sqrt{d_k}}\right)\mathbf{V_i} \tag{14}$$

where $\mathbf{Q_i} \in \mathbb{R}^{B \times L_d \times D_k}$ corresponds to the *i*-th head split from **Q**, while $\mathbf{K_i}, \mathbf{V_i} \in \mathbb{R}^{B \times L_t \times D_k}$ represent the *i*-th head split from **K** and **V** respectively, $g_a(\cdot)$ is a dimension-preserving linear layer, the MHCA in the aggregator contains *H* heads, and $D_k = D_m/H$. A feed-forward network (FFN) with residual connection and layer normalization is then applied:

$$\mathbf{F} = \text{LayerNorm}(\mathbf{F} + \text{FFN}(\mathbf{F})) \tag{15}$$

Three identical aggregator layers are stacked, completing the aggregator architecture for TAPB. Next, we introduce how to suppress "target prior bias" and estimate $P(Y|\mathbf{D}, do(\mathbf{T}))$.

## Amino acid randomization

Amino acid randomization includes 2 parts: residue random deletion and residue feature mutation.

Residue random deletion: As shown in Fig. 5c, for each ESM-2 encoded feature, 70% of their residues are randomly deleted, except the [cls] token, akin to the approach used by Masked Autoencoders (MAE)[54] where 75% of image patches are masked.

Residue feature mutation: After residue random deletion, for each remaining residue feature except special tokens in batch, has a 80% probability of remaining unchanged, and a 20% probability of being replaced by a random residue feature in the amino acids dict. The amino acid dict is obtained by average pooling every kind of amino acid feature (i.e., the last hidden state) extracted by ESM-2 in the training set.

By randomly deleting and independently mutating residues, we create a scenario akin to a randomized experiment that helps to disrupt the backdoor path $T \leftarrow X_t \rightarrow Y$. Intuitively, amino acid randomization can prevent models from memorizing the spurious correlations between targets and labels. Furthermore, the residue random deletion, reducing the sequence length, lowers computational costs, thereby accelerating training and allowing for the exploration of larger models. This is essential for deepening our understanding of the extensive and complex space of drug-target interactions.

## TAPB interventional training

To adjust confounders in Fig. 9a, the backdoor adjustment for SCM in Fig. 9a is formulated as:

$$P(Y|\mathbf{D}, do(\mathbf{T})) = \sum_{x_t} P(x_t) P(Y|\mathbf{D}, \mathbf{T}, X_t = x_t) \qquad (16)$$

Regrettably, this is infeasible. Unlike previous causal debiasing vision models, e.g. IFSL[55], VCRCNN[56] or IBMIL[57], whose tasks involve specific objects and observable confounders-the learned preferences in DTI models (potentially corresponding to protein families, sub-sequence lengths, or other latent factors) constitute unobservable confounders.

Furthermore, computing $P(Y|\mathbf{D}, \mathbf{T}, X_t)$ for every $X_t$ presents implementation challenges in deep learning frameworks: Since target sequences $X_t$ remain static in non-augmented datasets, each target feature $T$ corresponds to a single sequence and confounder category. Thus, for each DTI pair ($\mathbf{D}$, $\mathbf{T}$), the model can only predict one $P(Y|\mathbf{D}, \mathbf{T}, X_t)$ per forward. While inserting $x_t$-corresponding sub-sequences (if observable) during data augmentation could theoretically satisfy exact backdoor adjustment, this approach would increase computational costs-requiring additional forward/backward per augmented ($\mathbf{D}$, $\mathbf{T}$) pair-incurring prohibitive resource overhead and architectural inefficiency. Since $X_t$ is unobservable, direct estimation of $P(Y|\mathbf{D}, \mathbf{T}, X_t)$ is infeasible. However, under the causal assumptions of Fig. 9b, the confounder dictionary $\mathbf{C}$ serves as a valid adjustment set that is theoretically equivalent to adjusting for $X_t$.

Unlike previous deep learning debiasing methods, e.g. VCRCNN[56] and IBMIL[57], that employ the Normalized Weighted Geometric Mean (NWGM)[58] to approximate the backdoor adjustment, we implement theoretically exact backdoor adjustment formula by estimating $P(Y|\mathbf{D}, \mathbf{T}, \mathbf{c_i})$ for all $\mathbf{c_i} \in \mathbf{C}$ to compute $P(Y|\mathbf{D}, do(\mathbf{T}))$, while maintaining computational efficiency in deep learning frameworks. Our SCM yields the backdoor adjustment:

$$P(Y|\mathbf{D}, do(\mathbf{T})) = \sum_i P(Y|\mathbf{D}, \mathbf{T}, \mathbf{c_i}) P(\mathbf{c_i}) \qquad (17)$$

where $\mathbf{c_i}$ denotes cluster centers. As shown in Fig. 5b, confounder dictionary $\mathbf{C}$ and $P(\mathbf{c_i})$ are derived as follows: We cluster ESM-2-encoded features $\mathbf{E}$ (preceding $\mathbf{T}$ generation) via K-Means[30] across the

training set, and use the resulting cluster centers to construct a confounder dictionary $\mathbf{C} \in \mathbb{R}^{I \times D_e}$. Here, $I$ is the dictionary size (equivalent to the number of heads $H$ in the aggregator's MHCA). Since ESM-2 was pre-trained on disjoint datasets, DTI label leakage risks are eliminated. The sample proportion per cluster serves as the adjustment weight $P(\mathbf{c_i})$.

The path $X_t \rightarrow \mathbf{C} \rightarrow \mathbf{T}$ is established via our confounder alignment module (CAM) $g_t(\cdot)$. As shown in Fig. 5e, CAM fuse cluster centers $\mathbf{c_i} \in \mathbf{C}$ serves as the key $\mathbf{K_i}$ and value $\mathbf{V_i}$ for a distinct attention head within $g_t(\cdot)$, where they interact with the ESM-2 features $\mathbf{E}$:

$$\mathbf{T_i} = \text{Softmax}\left(\frac{\mathbf{Q_i}\mathbf{K_i}^\top}{\sqrt{d_k}}\right)\mathbf{V_i} \qquad (18)$$

$$\mathbf{T} = g(\text{Concat}(\mathbf{T_0}, \mathbf{T_1}, \ldots, \mathbf{T_i}) + \mathbf{E}) \qquad (19)$$

where $\mathbf{Q_i}$ represents the $i$-th head of linear projected $\mathbf{E}$, while $\mathbf{K_i}$ and $\mathbf{V_i}$ correspond to the $i$-th cluster center $\mathbf{c_i}$ projected through separate linear layers. Here $D_k$ denotes the dimension of $i$-th head $\mathbf{Q_i}$, and $g(\cdot)$ is a linear layer $\mathbb{R}^{B \times L_d \times D_e} \rightarrow \mathbb{R}^{B \times L_d \times D_m}$. CAM incorporates confounder features $\mathbf{c_i}$ into the $\mathbf{E}$ via multi-head attention. Since amino acid randomization disrupts the original confounding information and pattern within the target features, this enables confounder-conditioned features to be integrated into $\mathbf{T}$, establishing the path $X \rightarrow \mathbf{C} \rightarrow \mathbf{T}$. Note that computational costs remain minimal since $I \ll \text{length}(X_t)$.

To approximately estimate all confounder-conditioned probabilities $P(Y|\mathbf{D}, \mathbf{T}, \mathbf{c_i})$ within per forward, we leverage the independent interaction mechanism of MHCA in the aggregator. Eq. (14) shows that MHCA partitions features along the embedding dimension into $h$ independent heads. Since $\mathbf{K}$ and $\mathbf{V}$ remain invariant across layers, each $\mathbf{Q_i}$ can individually extract $\mathbf{c_i}$-relevant information. Due to this independent interaction mechanism, decomposing MHCA output into $I$ heads yields distinct $\mathbf{F_{c_i}} \in \mathbb{R}^{B \times L_d \times D_k}$ approximations. This enables simultaneous estimation of all $P(Y|\mathbf{D}, \mathbf{T}, \mathbf{c_i})$ in one forward. Note that $H$ must equal the confounder dictionary size $I$ and satisfy that $D_m$ is divisible by $H$.

The resulting output feature $\mathbf{F} \in \mathbb{R}^{B \times L \times D_m}$, where $D_m = I \times D_k$, is then split along its feature dimension into $I$ segments, each corresponding to one confounder cluster:

$$\mathbf{F} = \left[\mathbf{F_{c_0}}, \mathbf{F_{c_1}}, \cdots, \mathbf{F_{c_i}}\right] \qquad (20)$$

Here, $\mathbf{F_{c_i}}$ represents the feature segment associated with the $i$-th confounder cluster. Finally, after applying average pooling to each $\mathbf{F_{c_i}}$, a classification head $g_y(\cdot)$ is used to estimate all $P(Y|\mathbf{D}, \mathbf{T}, \mathbf{c_i})$:

$$P(Y|\mathbf{D}, \mathbf{T}, \mathbf{c_i}) = \text{Softmax}(g_y(\mathbf{F_{c_i}})) \qquad (21)$$

then, we can parameterize Eq. (17) via TAPB in the following form:

$$P(Y|\mathbf{D}, do(\mathbf{T})) = \sum_i P(\mathbf{c_i})\text{Softmax}(g_y(\mathbf{F_{c_i}})) \qquad (22)$$

Therefore, $P(Y|\mathbf{D}, do(\mathbf{T}))$ can be computed via Eq. (22) and integrated into deep learning training to adjust for confounders. Under our SCM in Fig. 9c, this implementation provides a theoretically exact estimation of the causal effect, while maintaining computational efficiency in deep learning frameworks. The binary classification loss $\mathcal{L}_b$ for TAPB can be denoted as follows:

$$\mathcal{L}_b = -\sum_{i=1} y_i \log(\hat{y}_i) \qquad (23)$$

where $y_i$ is the label, and $\hat{y}_i$ is the predicted probability (i.e., from *Softmax*) for class *i*. Furthermore, we follow the masked language modeling in BERT to enhance the semantic features extracted by the drug encoder $f_d(\cdot)$. Specifically, 15% of all tokens in each sequence are randomly selected, with an 80% probability of being replaced by a [mask] token, a 10% probability of remaining unchanged, and a 10% probability of being replaced by a random token. The masked tokens are then predicted using $f_d(\cdot)$ and $g_m(\cdot)$, and the loss $\mathcal{L}_{\text{mlm}}$ is calculated by:

$$\mathcal{L}_{\text{mlm}} = -\sum_{i=1}^{N}\sum_{j=1}^{L_d} m_{ij} \log P(w_{ij}|\mathbf{H}_i) \qquad (24)$$

where *N* is the number of samples, $m_{ij}$ is a binary mask (1 position *j* is masked, 0 otherwise), and $P(w_{ij}|\mathbf{H}_i)$ denotes the predicted probability for the token $w_{ij}$ at position *j* of the *i*-th sample, with $\mathbf{H}_i \in \mathbb{R}^{D_m}$ representing the contextual representations generated by the $f_d(\cdot)$. The total loss $\mathcal{L}$ for the TAPB can be denoted as follow:

$$\mathcal{L} = \mathcal{L}_b + \mathcal{L}_{mlm} \qquad (25)$$

The integration of amino acid randomization with our TAPB interventional training framework establishes a generalizable methodology for other DTI models, requiring only that the dataset exhibits "target prior bias" while utilizing both MHCA and a pre-trained target encoder.

**Reporting summary**

Further information on research design is available in the Nature Portfolio Reporting Summary linked to this article.

## Data availability

The BioSNAP, BindingDB, and Human datasets are publicly available at DrugBAN[12] GitHub repository (https://github.com/peizhenbai/DrugBAN). Davis dataset is available at ConPLex[39] GitHub repository (https://github.com/samsledje/ConPLex_dev). All datasets are also available at our GitHub repository (https://github.com/GaomingL1n/TAPB). Source data are provided with this paper.

## Code availability

The source code, visualization details, and implementation details of this study are freely available at our GitHub repository (https://github.com/GaomingL1n/TAPB) with a DOI[59] of https://doi.org/10.5281/zenodo.17350833.

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

## Acknowledgements

This work is supported in part by the National Natural Science Foundation of China (NSFC 62425107 to Q.Z., 62272418 to C.Z., 62172076 to Y.D.), the Zhejiang Provincial Natural Science Foundation of China (Grant No. LY23F020003 to Y.D.), and the Municipal Government of Quzhou (Grant No. 2024D002 to Y.D., 2023D018 to X.Z.).

## Author contributions

Y.D., C.Z., P.T., and Q.Z. supervised the research. G.L. contributed to the overall design and experiments. G.L. and X.Z. contributed to writing and editing the original manuscript. G.L., X.Z., Z.R., and Y.D. contributed to refining and optimizing figures. G.L., X.Z., Z.R., Q.Z., P.T., C.Z., and Y.D. contributed to the manuscript preparation and revision.

## Funding

## Competing interests

The authors declare no competing interests.
