## [Transparent Peer Review file · Nature Communications]

TAPB: An Interventional Debiasing Framework for Alleviating Target Prior Bias in Drug-Target Interaction Prediction

Corresponding Author: Professor Prayag Tiwari

Version 0:

Reviewer comments:

Reviewer #1

(Remarks to the Author)

The manuscript presents a method for predicting drug-target interactions which accounts for target prior bias in popular model training databases of known drug-target interactions. Target prior bias refers to the observation that different targets have different tendencies for positive and negative labels. When machine learning (ML) models are trained on these databases, this may lead to models making biased predictions based on a target's label tendency instead of genuine combinations of drug and target features.

The proposed method combines a particular ML model architecture together with causal inference techniques to adjust for the target prior bias.

As requested by the Senior Editor, my review focuses on the causal inference aspect of the manuscript, leaving comments on the machine learning aspect to the other reviewers.

Overall, the use of causal inference to adjust for biases in a training dataset is potentially innovative and of interest. However, as it is currently presented, it appears that the causal inference language is to some extent used in a hand-waving manner, while the details of the method appear largely heuristic. As detailed below, this should be addressed by providing a more elaborate and explicit description of the assumed causal model, as well as exploring the use of other confounder adjustment methods from causal inference.

MAJOR COMMENTS

[A] The authors' structural causal model (SCM) and confounder adjustment method do not match.

The authors' SCM to describe their model is Fig. 9(a). The authors wish to apply backdoor adjustment to compute $P(Y|D,do(T))$. The backdoor adjustment formula for this SCM is

$$P(Y|D,do(T)) = \sum_{x_t} P(x_t) P(Y|D,T,X_t=x_t)$$

where the sum extends over the range of possible target sequences x_t with prior probability $P(x_t)$. Instead, in eq. (14), the authors sum over cluster centers c_i . There are several concerns here:

(1) Clusters and cluster centers are not part of the SCM. To show that eq. (14) is (approximately) correct, a cluster variable C should be included in the SCM.

(2) Eq. (14) is correct only if C blocks the backdoor path $T \leftarrow X_t \rightarrow Y$. This is possible only if C lies either on the path $X_t \rightarrow T$ (i.e., $X_t \rightarrow C \rightarrow T$) or on the path $X_t \rightarrow Y$ (i.e., $X_t \rightarrow C \rightarrow Y$). Neither of those are plausible. For instance, the first case implies that the features T are extracted from the sequence X_t via the clustering C , which is not the case.

(3) My understanding from the text in Section 2.3 (and the authors should really explain this more clearly), is that they cannot

apply the exact backdoor adjustment formula above, and use the clustering as a proxy variable for the confounder. This would imply an edge $X_t \rightarrow C$ without any causal relation between C and T or Y . However:

- There is a rich theory of using proxy variables in causal inference, and generally one needs at least two, not one, proxies for confounder adjustment. See for instance: <https://doi.org/10.1093/biomet/ast066>, <https://arxiv.org/abs/1609.08816>, <https://arxiv.org/abs/2009.10982>. The authors are encouraged to investigate the applicability of this theory to their problem.

- Worryingly, Fig. 5 and Section 4.2.5 suggest that the features F determine the clustering. That would imply an edge $F \rightarrow C$ and C being causally downstream of T in the SCM. No variable downstream of T can ever be used to adjust for the confounding path $T \leftarrow X_t \rightarrow Y$.

- Related to the previous point, it is not completely clear what is being clustered and at what stage of the pipeline. For instance, is care being taken that clustering is done during each training run using only the samples in the current training fold to avoid data leakage?

(4) Why is it not possible to apply the exact backdoor adjustment formula above? If the sequences in the databases are random samples from the (unknown) distribution $P(x_t)$, then using their empirical distribution should be a good enough approximation.

[B] The authors seem to equate randomization and randomized experiments in causal inference with randomly deleting parts of the target sequences. I don't think this is correct. Assume we want to randomize a particular target sequence residue. A proper randomization (from a causal inference perspective) would be to take the real sequences and then randomly assign each sequence one of the possible values of that particular residue, irrespective of the values of the other residues (compare to a clinical trial where each participant is randomly assigned a treatment value [drug or placebo] irrespective of any other characteristics of that individual). Or, since one is really interested in randomizing the features T to break the path $X_t \rightarrow T$, do this procedure at the level of features rather than residues. The authors should provide more justification of their randomization procedure from a causal inference perspective, and explore alternative randomization strategies.

MINOR COMMENTS

(1) The training data consist of known drug-target interactions, that is, by definition they only contain positive examples. How do the authors define negative labels?

(2) The authors focus on target prior bias because their analyses suggest that it is more widespread than drug prior bias. Nonetheless, Fig. 3 shows that drug prior bias is also present. Why can a similar adjustment method not be used to adjust for both biases simultaneously? Please comment.

(3) Section 2.2.2: The "label test" presented here is not a statistical test, but only a heuristic score to measure the deviation of frequencies from 50%. Why not devise a proper statistical hypothesis test to estimate how likely it is to see such deviations under a proper null model?

(4) The authors should be more careful in their use of causal inference terminology throughout the manuscript. For instance, in Section 4.2.5 they write "[...] backdoor adjustment severs the path $T \leftarrow X_t \rightarrow Y$, thereby inhibiting target prior bias. We approximate the backdoor adjustment formula $P(Y|D,do(T))$ for interventional training [...]". However, backdoor adjustment does not sever any paths (only interventions do) or perform interventional training. Instead backdoor adjustment is a method to compute $P(Y|D,do(T))$ without doing an actual intervention, that is, using only observations from the complete SCM (including the confounding influence).

(Remarks on code availability)

Reviewer #2

(Remarks to the Author)

Drug Target Interaction (DTI) prediction is vital for drug repurposing. The authors believe that previous studies trained on BioSNAP and BindingDB datasets wrongly identified "drug bias" as the cause of biased predictions, and their research reveals that "target prior bias" is the predominant issue. This bias arises due to "prior tendency" characterized by the imbalanced label distribution of targets in the training data. Their model outperforms existing methods on both in-domain and cross-domain datasets, offering enhanced generalization and providing interpretable insights into drug-target interactions.

Generally speaking, bias plays an important role in deep learning, which affects the relationship between the model and the training examples and the learning results. By rationally setting bias and other model parameters, the performance of the model can be optimized, generalized capabilities can be improved, and the stability and interpretability of the model can be enhanced.

In deep learning, the parameters of the model include weights and biases. Weighting determines how input data affects the output of the network, while bias allows each neuron output to still have a non-zero value without any input or when the input is zero. Bias can be regarded as the "built-in threshold" for each neuron, which adjusts the activation threshold of the activation function. The training model in this paper does not specify whether the bias of the model itself is considered, but

more considers the bias caused by the imbalance distribution of the training examples.

---From causal lens, "target prior bias" is a confounder, causing models trained with $P(Y|D, T)$ to learn spurious associations between targets and labels rather than genuine interaction mechanisms--- In view of this, unsupervised learning is used and the data has no labels. How to explain the relationship between the data and labels?

The author believes that all (classified) predictions are biased in learning, and the work to handle with bias becomes very important. In fact, the prior bias caused by training sample imbalance should be solved by dealing with sample imbalance. The reviewer believes that the model of this paper has achieved better prediction accuracy, which may not be effective in dealing with the "target prior bias".

In deep learning, the paper should be at the data level, model level and training process level, to what extent does bias seriously affect the results of DTI. In addition, the references cited in the paper do not involve related research on learning spurious associations between the model and the sample label due to bias.

(Remarks on code availability)

Reviewer #3

(Remarks to the Author)

This manuscript investigates target prior bias in drug-target interaction (DTI) prediction, identifying how specific datasets (BioSNAP and BindingDB) can cause models to learn spurious associations rather than true biological interactions. The authors propose the TAPB framework, which combines target random masking with interventional training to mitigate this bias. The framework reportedly improves generalisation across cross-domain settings without using domain adaptation.

Although this work highlights an important but understudied issue in DTI prediction and proposes a thoughtful framework to address it, the current scope of datasets, lack of recent baselines, unclear metric definitions, and missing experimental results limit the strength of its claims. My detailed comments are below.

Strengths: The focus on dataset-induced bias is timely and relevant. By highlighting target prior bias, the paper adds to ongoing discussions around data imbalance and generalisation in biomedical AI. The TAPB framework combines target random masking and interventional training to offer a new and potentially generalisable approach to bias mitigation in DTI prediction. The empirical results are promising. TAPB demonstrates improved cross-domain performance relative to established baselines, such as DrugBAN and TransformerCPI, without using domain adaptation.

Weaknesses and areas for improvement:

1. The analysis and conclusions are based solely on two datasets (BioSNAP and BindingDB) and have focused on two models (DrugBAN and TransformerCPI). While the findings are informative, they may not generalise to other datasets or architectures. The authors acknowledge this limitation, noting that different datasets and models, such as NVRLM may show different types of bias, but without further exploration. Evaluating TAPB and the bias analysis across a broader range of datasets and models can strengthen the contribution.
2. The most recent baseline compared against is DrugBAN, published in February 2023. Given the pace of advancement in this area, it is important to benchmark against more recent methods from late 2023 or 2024. Although more recent methods are cited, e.g. [29], they are not included in the experimental comparisons. This limits the ability to assess whether TAPB truly represents state-of-the-art performance.
3. Figure 1 uses an illustrative drawing to explain dataset bias. However, it does not quantify how prevalent the bias is here. To strengthen the motivation, it would be helpful to show earlier how serious the training set target tendencies are in practice using statistics or visual analysis.
4. The TAPB framework uses the ESM-2 encoder, which is a strong component and may be responsible for part of the performance gains. Without respective ablation studies, it is hard to isolate the effects of target random masking and interventional training. Moreover, it would be useful to apply the TAPB mechanisms to other models, such as DrugBAN or TransformerCPI, to test whether they provide consistent benefits.
5. Several hyperparameters are used in the paper, such as $n = 70\%$ on page 9, $n = 3$ on page 19, and values 15%, 80%, and 10% on page 20, as well as $l = 8$ and $d = 128$ on page 21. How these values were selected is not explained. Were they tuned on a validation set, adopted from prior work, or selected heuristically?
6. Figures 3f, 3g, and 3h are referenced in the manuscript but not included. This breaks the logical flow and weakens the evidential support.
7. The authors mention averaging over five runs, but no standard deviation or variance is reported for key results such as those in Figure 6. Without variance reporting, it is difficult to assess the robustness or significance of the improvements.
8. In Figure 6, sensitivity for many baselines approaches zero, but specificity approaches one. This is a well-known trade-off

in binary classification. The authors should clarify whether performance is reported using a fixed threshold and whether adjusting the threshold could yield a better trade-off.

9. Key metrics such as label tendency and prior tendency are insufficiently explained in Section 2.2.2. Their definitions and physical interpretations are unclear.

10. Important symbols are undefined, such as "do(T)". This may confuse readers unfamiliar with causal inference notation. Moreover, there is inconsistent use of mathematical symbols. For example, symbols are in bold in the abstract but not in the main text. In mathematical writing, bold and non-bold fonts usually represent different types of quantities so this inconsistency may mislead readers. Lastly, the tensor R is mentioned as generated, but its dimensionality is not specified. This reduces clarity and reproducibility.

(Remarks on code availability)

Yes. The code provides a README file with enough instructions for installing and running the application.

Version 1:

Reviewer comments:

Reviewer #1

(Remarks to the Author)

The authors have performed an extensive revision and have addressed all my comments in detail. I was pleased to read that they have made their method more rigorous from the causal inference perspective, and that this has led to an overall improved performance across their benchmarks. Only a few minor comments remain:

* Response A.2

1. While the authors were previously too optimistic in their causal inference language, they now err a little bit in the other direction. For instance, they write that their adjustment for C achieves a balance between theoretical rigor and computational cost. However, if the data is truly generated by the causal graph in Fig. 9b, then adjusting for C or adjusting for the true confounder X_t is theoretically equivalent and both result in the correct causal effect identification. Of course, different adjustment sets may be theoretically equivalent but differ in their finite-sample efficiency, variance, etc. This is something that would be difficult to analyze in the present context, but the authors may well refer to their current backdoor formula as theoretically exact, while perhaps referring to some of the literature around optimal adjustment sets in their discussion.

2. The authors refer to the conditional distributions $P(Y|D, T, c_i)$ for a given value c_i of the clustering variable C as a "counterfactual". Unless I have misunderstood what is being estimated here, I don't think this is a correct use of the term counterfactual. A counterfactual in causal inference usually refers to a statement about a single unit (in this case drug-target interaction) and asks what would have been true under different circumstances. For instance, if the authors' method predicts an interaction between a specific drug and target, they could ask the counterfactual question if the interaction would still have been predicted if the drug or target would have taken a specific value different from its true value ("counter to the facts") for one of its features.

* Response 1.3

The authors should carefully review the different symbols used in their revised text. They now use z_i where before they wrote t_i , but their figure axis label still says t_i . Likewise they still use T before eq. (4) which should probably be a Z . In eq. (5), why is a new symbol g_i introduced while this should be equal/related to the z_i ? Fig. 3(a)-(d) - wouldn't a bar chart of the frequencies be more informative than a line chart; the legend label "SMILES" should be consistent with the caption which refers to "drugs".

(Remarks on code availability)

Reviewer #2

(Remarks to the Author)

In this revised paper, the authors introduce a work of alleviating Target Prior Bias in Drug-Target Interaction Prediction (TAPB), a debiasing framework that employs amino acid randomization, confounder alignment module (CAM), and interventional training to approximate $P(Y|D, do(T))$ via backdoor adjustment, thereby addressing this bias. The experimental results show that TAPB achieves better performance, which is offering enhanced generalization and providing interpretable insights into drug-target interactions.

The authors employ several way to alleviate target prior bias in drug-target interaction prediction, including data level, model level, and training process level. The effectiveness seems to be sound.

In TAPB interventional training, the authors combine amino acid randomization, confounder alignment module (CAM), and multi-head cross-attention (MHCA) to approximate $P(Y|D, do(T))$. These three key components should be evaluated with more datasets (the authors conducted the ablation study only on the BioSNAP in-domain dataset).

(Remarks on code availability)

It was difficult to load the sources.

Reviewer #3

(Remarks to the Author)

I'd like to thank the authors for their efforts in addressing the concerns raised in the first round of review. I can see that the manuscript has been improved; however, I also notice some newly identified problems and several unresolved issues remaining. I will write according to the index of my previous comments first and then add on other issues.

1. I do not agree that BioSNAP (BindingDB) in-domain and BioSNAP (BindingDB) cross-domain are considered as two datasets. They are the same dataset under two different settings (versions), an in-domain setting and a cross-domain setting. Even the authors' original manuscript considers BioSNAP and BindingDB as two datasets, rather than four so I am not sure why this revision calls it differently. Therefore, the claim of six datasets in the revised manuscript is misleading.

On page 65 of the response and page 12 of the revised manuscript, the authors say, "As shown in Table 1, TAPB achieves state-of-the-art performance ...". However, Table 1 provides statistics on the dataset instead of a performance comparison. Again, the authors need to proofread their writing before submission.

Page 67 of the response, the figure at the top, subfigures a and b label legends as "Target" and "SMILES" and subfigure c labels the legends as "Drug" and "Target". Why such a discrepancy? Is "Drug" in c not represented as "SMILES"? If using representations as the legend, why not use the representation of "Target"? Basically, "Target" and "SMILES" are not parallel concepts. This problem persists in other similar figure legends.

4. The figure on page 78 (Fig. 7 in the revised manuscript). For the same label "Plain TAPB", why is there a big difference in its performance in subfigures f and g? Are there more than one version of "Plain TAPB"? For subfigure f, why is the use of ESM-2 not included here for this ablation study in subfigure f?

For this same figure, the caption states that it is about ablation studies; however, not all subfigures are ablation studies. For example, subfigures a, b, and c are not ablation studies, but rather hyperparameter sensitivity studies. Such inconsistencies need to be addressed.

For this same figure, the captions of subfigures h and i are not clear enough to help readers understand what to observe from the figures.

On page 79 of the response, it says, "all experiments were conducted on the BioSNAP in-domain dataset ...". May I know which of the ablation studies were conducted on the BioSNAP in-domain dataset? It seems mostly the cross-domain dataset was used.

Page 80 of the response, line 4, "Drug BAN"?

5. The performance of TAPB has been tuned using the BioSNAP cross-domain dataset to find the best settings for the best results, so the performance on this dataset should not be used to draw conclusions. On the other hand, how many hyperparameters or settings of the competing methods have been tuned? On what datasets? It is important to clarify such details to ensure a fair comparison between them.

Also, the authors refer to such hyperparameter tuning as ablation studies, which is incorrect.

6. On page 83 of the response, the authors are talking about Figures 1g and 1h, but I still cannot find where they are either in the original manuscript or the revised one.

7. Are all supplementary tables referred to in the main text so that the readers are aware of the existence of such results?

8. I did not find the response is addressing my 8th comment. The response reads largely identical to response 3.7 and hence it is addressing the 7th comment rather than the 8th one.

10. The writing is still problematic. The writing use both Bert and BERT. Are they refer to the same BERT? The symbols are not standardised, the same entity occurs both in bold and nonbold versions, which is confusing. For example, the last two lines on page 88 of the response, the line above writes D and T and the line below use the bold versions. Are they the same? There are many other places of the similar issues.

Additional comments:

a. This manuscript has nine figures and one table for the main text. To my understanding, this exceeded what is allowed by Nature Communications.

b. There are many forward references where the figures are several (unknown at the place of references) pages ahead, making it difficult for the readers to follow.

(Remarks on code availability)

I did not find meaningful changes in the code repo comparing to the version for the initial submission, except some figure/documentation update.

Version 2:

Reviewer comments:

Reviewer #1

(Remarks to the Author)

The authors have addressed my previous comments and I have no further comments.

(Remarks on code availability)

Reviewer #2

(Remarks to the Author)

The authors propose a method for predicting drug-target interactions (DTI). They believe that when dealing with datasets such as BioSNAP and BindingDB, one must consider the issue of 'target prior bias.' This bias arises from some factors like imbalanced label distribution of targets in the training data. Their model demonstrates advantages over selected methods on both in-domain and cross-domain datasets.

(Remarks on code availability)

This manuscript has been revised and supplemented in response to the reviewers' comments. Additionally, the results of the ablation experiments have been added.

If possible, Table 3 and Table 4 in the supplements are the main predictive results of this paper, and it is recommended to describe them in the main text.

Since DTI has been a relatively popular research topic in recent years, many research results on DTI have been published in relevant journals. For example, "Escaping the Drug-Bias Trap: Using Debiasing Design to Improve Interpretability and Generalization of Drug-Target Interaction Prediction", IEEE TRANSACTIONS ON COMPUTATIONAL BIOLOGY AND BIOINFORMATICS, VOL. 22, NO. 4, JULY/AUGUST 2025" (after acceptance, the paper should be in an OPEN ACCESS state), the paper has corresponding prediction results (both in-domain and cross-domain experiments) on the BioSNAP and BindingDB datasets. Although the method TAPB described in this manuscript shows certain advantages compared to the methods it is compared with on the above datasets, when compared to the results of the aforementioned papers, the superiority of this TAPB's results does not exist. We hope the authors provide a better explanation and clarification.

Reviewer #3

(Remarks to the Author)

The authors have substantially revised the manuscript to address all the detailed comments raised in the previous round of review. I found the revisions largely satisfactory, except for one issue detailed below and some minor writing problems that should be fixed during the editorial process.

Remaining issue: The authors refer to the Human dataset with a cold split strategy as the "Human Cold-split" dataset. The naming style of DATASET_NAME+SPLIT is inappropriate and inconsistent with the convention and literature, as well as the naming of other datasets in this paper.

(Remarks on code availability)

I see substantial efforts to improve the code; for example, comments in Chinese have been converted to English, along with other improvements detailed in the response to my comments.

Response letter for “TAPB: An Interventional Debiasing Framework for Alleviating Target Prior Bias in Drug-Target Interaction Prediction”

Gaoming Lin ^{†1}, Xin Zhang ^{†2,3}, Zhonghao Ren⁴, Quan Zou², Prayag Tiwari^{*5}, Yijie Ding ^{*2}, and Changjun Zhou ^{*1}

¹School of Computer Science and Technology, Zhejiang Normal University, Jinhua, 321000, Zhejiang, China

²Yangtze Delta Region Institute (Quzhou), University of Electronic Science and Technology of China, Quzhou, 324003, Zhejiang, China

³The Quzhou Affiliated Hospital of Wenzhou Medical University, Quzhou People’s Hospital, Quzhou, 324000, Zhejiang, China

⁴College of Computer Science and Electronic Engineering, Hunan University, Changsha, 410082, Hunan, China

⁵School of Information Technology, Halmstad University, Halmstad, 301 18, Sweden

To reviewers

Dear reviewers:

Our sincere thanks go out to the reviewers who reviewed our manuscript and provided constructive comments that significantly improved it. We have made detailed revisions in a point-by-point response to comments and suggestions made by reviewers.

We first quote the comments and then reply with how we have revised the manuscript to accommodate the changes. We use **black sans serif font** for our responses and **red box** for comments. Revisions in the manuscript are indicated within a **blue box**, with modified text shown in **red font** and deleted content

marked by "~~strikethrough formatting~~." For further results, please refer to the file "Source_data.zip." The main changes are summarized below:

- We restructured our method into a single-stage training framework, enhanced residue randomization, and introduced a novel confounder alignment module to implement backdoor adjustment.
- We modified our Structural Causal Model to ensure compatibility with our backdoor adjustment.
- We incorporated the Davis and Human cold-split datasets, re-evaluated all experiments across six datasets, and demonstrated that our revised method achieves state-of-the-art performance.
- We added two recently published models as new baselines in our experiments.
- We implemented a formal statistical hypothesis test to quantify prior tendency.
- We conducted five additional ablation studies assessing different influences, re-ran all ablation experiments, and determined optimal hyperparameter configurations through this analysis.
- We provided rigorous definitions of key terminology, e.g., $do(\cdot)$ operator, backdoor adjustment, and "target prior bias", to enhance conceptual clarity.
- We refined the manuscript's discussion of causal inference methodologies to improve technical precision.
- We revised all figures and corresponding descriptions in the visualization section.

Best regards,

Gaoming Lin, Xin Zhang, Zhonghao Ren, Quan Zou, Prayag Tiwari, Yijie Ding, Changjun Zhou

June 27, 2025

Contents

1	Response to Reviewer 1	4
2	Response to Reviewer 2	38
3	Response to Reviewer 3	61

1 Response to Reviewer 1

Comment: (Remarks to the Author)

The manuscript presents a method for predicting drug-target interactions which accounts for target prior bias in popular model training databases of known drug-target interactions. Target prior bias refers to the observation that different targets have different tendencies for positive and negative labels. When machine learning (ML) models are trained on these databases, this may lead to models making biased predictions based on a target’s label tendency instead of genuine combinations of drug and target features. The proposed method combines a particular ML model architecture together with causal inference techniques to adjust for the target prior bias.

As requested by the Senior Editor, my review focuses on the causal inference aspect of the manuscript, leaving comments on the machine learning aspect to the other reviewers.

Overall, the use of causal inference to adjust for biases in a training dataset is potentially innovative and of interest. However, as it is currently presented, it appears that the causal inference language is to some extent used in a hand-waving manner, while the details of the method appear largely heuristic. As detailed below, this should be addressed by providing a more elaborate and explicit description of the assumed causal model, as well as exploring the use of other confounder adjustment methods from causal inference.

Response:

We sincerely appreciate the reviewer’s professional comments and patient explanations. Your guidance has been instrumental in enhancing methodological rigor. Following your suggestions, we modified our method by merging the two training stages, significantly improving training efficiency while achieving superior performance. Below are key revisions implemented:

1. **Structural Causal Model Alignment:** We restructured the path $X_t \rightarrow \mathbf{T}$ to explicitly incorporate the cluster variable \mathbf{C} on the $X_t \rightarrow \mathbf{T}$ path. This proxy is now derived solely from ESM-2 features (not fused features \mathbf{F}), eliminating downstream contamination while ensuring \mathbf{C} properly blocks the backdoor path.

2. **Causal Randomization Enhancement:** We replaced random amino acid deletion with amino acid randomization—combining residue deletion (70%) with independent feature mutation (20% substitution probability). This better approximates causal randomization by disrupting spurious correlations in $X_t \rightarrow \mathbf{C} \rightarrow \mathbf{T}$.

3. **Methodological Transparency:** We clarified \mathbf{C} ’s construction via K-means clustering of ESM-2 embeddings and expanded dataset descriptions to explicitly

define positive/negative samples (BindingDB: affinity-thresholded negatives; BioS-NAP: curated non-interacting pairs).

4. **Validation and Terminology Rigor:** We introduced permutation tests confirming statistically significant target prior bias ($p < 0.001$), and systematically corrected terminology inconsistencies—distinguishing backdoor adjustment (estimation) from interventions (path disruptions).

5. **Confounder Adjustment Implementation:** We incorporated the empirical distribution $P(c_i)$ to handle confounder priors within deep learning constraints. While dual-proxy methods [1] show theoretical promise, implementation attempts revealed persistent numerical instability during matrix inversion; we thus retain the backdoor approximation pending future investigation of stable multi-proxy integration.

These revisions collectively improved performance across six benchmarks. The results consistently demonstrate superior performance, validating the effectiveness of your suggestions. All changes are highlighted, and we sincerely appreciate your guidance in strengthening this work.

Major Comments:

Comment: A

The authors' structural causal model (SCM) and confounder adjustment method do not match. The authors' SCM to describe their model is Fig. 9(a). The authors wish to apply backdoor adjustment to compute $P(Y|D, do(T))$. The backdoor adjustment formula for this SCM is:

$$P(Y|D, do(T)) = \sum_{x_t} P(x_t)P(Y|D, T, X_t = x_t)$$

where the sum extends over the range of possible target sequences x_t with prior probability $P(x_t)$. Instead, in eq. (14), the authors sum over cluster centers c_i . There are several concerns here:

Comment: A.1.

Clusters and cluster centers are not part of the SCM. To show that eq. (14) is (approximately) correct, a cluster variable C should be included in the SCM.

Response A.1.:

We sincerely appreciate the reviewer's professional comments and we acknowledge that we ignored the inclusion of cluster variable C in the SCM, which led to a mismatch between the SCM and our approximate backdoor adjustment. Both our method and the SCM have been revised, with the cluster variable C now explicitly

incorporated into the SCM to ensure alignment between the formula and the model framework. Revisions to our method can be found in our response A.2.. The revised SCM with the full set of variables and node definitions is presented in Figure 9. Additionally, we have revised both the definitions of variables and the corresponding formulas throughout the manuscript in revision A.1.:

Revision: A.1.

4.1 Analysis DTI through causal inference

PAGE 24-25

We construct a structural causal model (SCM) to demonstrate the causal relationships within the DTI model. As shown in Figure 9a, there are 6 nodes and 6 edges in SCM, where there are 6 nodes and 6 edges in conventional DTI model's SCM, and TAPB introduces an additional node C and establishes the path $X_t \rightarrow C \rightarrow T$. X_d represents SMILES in training set, X_t represents target sequences in training set, D represents drug feature extracted by drug encoder $f_d(\cdot)$, T represents target feature extracted by target encoder $f_t(\cdot)$, C represents our target confounder dictionary obtained via clustering, F represents the fusion feature, and Y represents the prediction. "Target prior bias" hides in X_t .

~~$X_d \rightarrow D$ and $X_t \rightarrow T$: These two paths indicate that the DTI model extracts drug feature D from X_d and target feature T from X_t in the training set.~~

$X_d \rightarrow D$: This path indicate that the drug feature D is extracted from the SMILES X_d in the training set.

$X_t \rightarrow C \rightarrow T$: This path indicates that the target feature T is extracted from the sequence X_t via the clustering C.

$D \rightarrow F$ and $T \rightarrow F$: These two paths indicate the generation of fusion feature F by aggregator $\mathcal{F}(\cdot)$ and $\sigma(\cdot)$.

$F \rightarrow Y$: This path indicates that the prediction Y is based on fusion feature F.

~~$X_t \rightarrow F$: This path indicates that the fusion feature F learned by DTI model can be affected by X_t . As shown in Figure 9, F is affected by "target prior bias."~~

$X_t \rightarrow Y$: This path indicates that the prediction Y is affected by X_t , i.e., the "target prior bias" causes biased prediction.

Figure 9a shows that SCM exists backdoor paths $T \leftarrow X_t \rightarrow Y$. From causal lens, the prior of X_t confounds T and Y, leading to spurious correlations. To suppress this bias, a more effective mechanism is needed to handle the actual causal relationship between drug, target, and predictions.

Figure 9: Structural causal model (SCM) of conventional DTI models and TAPB. (a) SCM of the conventional DTI models exhibiting "target prior bias." (b) SCM after applying target random mask, which weakens the backdoor path $T \leftarrow X_t \rightarrow Y$. Note that the dashed lines, which are not rigorous representations of causal relationships within SCM, indicate a weakened connection rather than a complete removal of this path. (c) SCM after applying backdoor adjustment, severing the connection to T . (d) The final SCM of TAPB, "target prior bias" is suppressed from causal lens. (b) SCM of our stage 1 biased training without amino acids randomization, adding C into $X_t \rightarrow T$. (c) SCM after applying amino acid randomization, which weakens the confounding path $T \leftarrow C \leftarrow X_t \rightarrow Y$. Note that the dashed lines, which are not rigorous representations of causal relationships within SCM, indicate a weakened connection rather than a complete removal of this path. (d) SCM for $P(Y|D, do(T))$ where $do(T)$ blocks the path $X_t \rightarrow C \rightarrow T$. We compute $P(Y|D, do(T))$ via backdoor adjustment without actual intervention.

Comment: A.2.

Eq. (14) is correct only if C blocks the backdoor path $T \leftarrow X_t \rightarrow Y$. This is possible only if C lies either on the path $X_t \rightarrow T$ (i.e. $X_t \rightarrow C \rightarrow T$) or on the path $X_t \rightarrow Y$ (i.e. $X_t \rightarrow C \rightarrow Y$). Neither of those are plausible. For instance, the first case implies that the features T are extracted from the sequence X_t via the clustering C , which is not the case.

Response A.2.:

We sincerely appreciate the reviewer's professional comments. We acknowledge that we initially ignored that the variable C , obtained through clustering, does not lie on the confounding path $T \leftarrow X_t \rightarrow Y$. We have explored alternative confounder adjustment methods to address this issue but encountered significant implementation challenges within deep learning frameworks. A detailed discussion of these alternatives is provided in Response A.3.1. Consequently, we retain backdoor adjustment

for computing $P(Y|D, \text{do}(T))$ while modifying our approach to establish the path $X_t \rightarrow C \rightarrow T$, thereby ensuring our SCM aligns with backdoor adjustment.

The necessity of introducing C stems from our inability to directly estimate $P(Y|D, \text{do}(T))$:

$$P(Y|D, \text{do}(T)) = \sum_{x_t} P(x_t)P(Y|D, T, X_t = x_t)$$

The learned preferences of deep learning models—which may correspond to protein families, specific sub-sequence lengths, or other latent factors—constitute unobservable confounders. Furthermore, computing $P(Y|D, T, X_t)$ for every X_t presents implementation challenges in deep learning frameworks: Since target sequences X_t remain static in non-augmented datasets, each target feature T corresponds to a single sequence and confounder category. Thus, for each Drug-Target Interaction (DTI) pair (D, T) , the model can only predict one $P(Y|D, T, X_t)$ per forward pass. While inserting sub-sequences corresponding to x_t (if observable) during data augmentation could theoretically satisfy exact backdoor adjustment, this approach would increase computational costs by a factor of i , incurring prohibitive resource overhead and architectural inefficiency. Therefore, to estimate $P(Y|D, T, X_t)$, we adopted approximations balancing causal inference rigor, deep learning feasibility, and training efficiency.

To address unobservable confounders, we reintroduced C through modified clustering. By altering the clustering subject and establishing the path $X_t \rightarrow C \rightarrow T$, we ensure our SCM matches our backdoor adjustment. The revised method is detailed in our revision Revision A.2. Additionally, to provide a comprehensive methodological exposition, we have revised the description of the aggregator in Section 4.2.2 in Revision A.2. In our revision Revision A.2., we show the figure of our revised framework (Section 2.3) earlier, hoping this adjustment may help clarify our method.

Revision: A.2.

Figure 5: Architecture of the TAPB framework. (a) **Stage 1 Biased Training:** TAPB optimizes $P(Y|D, T)$. SMILES strings and amino acid sequences are tokenized and embedded. Drug features D are extracted via a Bert-based encoder $f_d(\cdot)$ with rotary positional embedding, while target features T are derived using ESM-2 followed by a linear dimensionality reduction layer $g_t(\cdot)$. Masked language modeling (MLM) loss $g_m(\cdot)$ is used to enhance drug representation. (b) **Stage 2 Interventional Training:** Initialized with stage 1 weights, $f_d(\cdot)$ and $g_t(\cdot)$ are frozen. A confounder dictionary, generated by K-means clustering of stage 1 features f , enables backdoor adjustment via attention mechanisms to approximate $P(Y|D, do(T))$. (c) **Target Random Masking:** random remove of 70% amino acids from target embeddings P , retaining [CLS] token for sequence integrity. (d) **Backdoor Adjustment Module:** according to our derivation, classification features f serve as queries, while confounder cluster centers act as keys and values in an attention-based approximation of backdoor adjustment to alleviate "target prior bias."

(a) TAPB Interventional Training: The drug encoder BERT $f_d(\cdot)$ generates drug features D from SMILES. ESM-2 Pre-extracted target features E undergo amino acid randomization and are then processed by the CAM g_t . all cluster centers $c_i \in C$ act as keys/values in CAM g_t with E . Fused features F are partitioned into I segments F_{c_i} , each globally pooled and fed to classifier g'_y for estimating counterfactual $P(Y|D, T, c_i)$. Finally, $P(Y|D, do(T))$ is computed via backdoor adjustment. (b) **Target Confounder Dictionary C :** K-means clustering of ESM-2 features from training targets builds confounder dictionary C . (c) **Amino Acid Randomization:** 1. Random deletion of 70% residue features; 2. Mutation of remaining residues to random features from the amino acid dictionary. (d) **Confounder Alignment Module (CAM, $g_t(\cdot)$):** Attention-weighted summation fuses c_i with target features, followed by dimensionality reduction and residual connection. Maintains explicit path $X_t \rightarrow C \rightarrow T$ across both training stages.

4.2.2 Aggregator and classifier

PAGE 26-27

Following TransformerCPI, we adopt the same aggregator $\mathcal{F}(\cdot)$ with cross attention in TAPB. Each fusion layer of the aggregator inputs drug features D into the self-attention layer, followed by and layer normalization:

Following TransformerCPI, TAPB adopts the same aggregator $\mathcal{F}(\cdot)$ with Multi-head Cross-Attention (MHCA), which is essential for our estimation of counterfactual $P(Y|D, T, c_i)$. First, each fusion layer of the aggregator takes the output of the previous layer $F \in \mathbb{R}^{B \times L_d \times D_m}$ into the self-attention layer, followed by residual connection and layer normalization:

$$F = \text{ln}(F + \text{SelfAttention}(F, F, F))$$

$$\mathbf{F} = \text{LayerNorm}(\mathbf{F} + \text{Self Attention}(\mathbf{F}, \mathbf{F}, \mathbf{F})) \quad (11)$$

The above output F is used as Q , with protein features P as K and V in the cross-attention layer, followed by shortcut and layer normalization:

The output \mathbf{F} is then transformed through a linear layer to obtain $\mathbf{Q} \in \mathbb{R}^{B \times L_d \times D_m}$, while target features \mathbf{T} are projected via two separate linear layers into $\mathbf{K} \in \mathbb{R}^{B \times L_t \times D_m}$ and $\mathbf{V} \in \mathbb{R}^{B \times L_t \times D_m}$, which are fed into the cross-attention layer followed by residual connection and layer normalization:

$$F = \text{ln}(F + \text{crossAttention}(F, P, P))$$

$$\mathbf{F} = \text{LayerNorm}(\mathbf{F} + \text{MHCA}(\mathbf{F}, \mathbf{T}, \mathbf{T})) \quad (12)$$

The MHCA is formally defined as:

$$\text{MHCA}(\mathbf{Q}, \mathbf{K}, \mathbf{V}) = g_a(\text{Concat}(\mathbf{head}_1, \dots, \mathbf{head}_i)) \quad (13)$$

$$\mathbf{head}_i = \text{Softmax} \left(\frac{\mathbf{Q}_i \mathbf{K}_i^\top}{\sqrt{d_k}} \right) \mathbf{V}_i \quad (14)$$

where $\mathbf{Q}_i \in \mathbb{R}^{B \times L_d \times D_k}$ corresponds to the i -th head split from \mathbf{Q} , while $\mathbf{K}_i, \mathbf{V}_i \in \mathbb{R}^{B \times L_t \times D_k}$ represent the i -th heads split from \mathbf{K} and \mathbf{V} respectively, $g_a(\cdot)$ is a dimension-preserving linear layer, the MHCA in the aggregator contains H heads, and $D_k = D_m/H$. A feed-forward network (FFN) with residual connection and layer normalization is then applied:

Next, the FFN layer with shortcut and layer normalization is applied:

$$F = \text{ln}(F + \text{FFN}(F))$$

$$\mathbf{F} = \text{LayerNorm}(\mathbf{F} + \text{FFN}(\mathbf{F})) \quad (15)$$

Three identical aggregator layers are stacked, completing the aggregator architecture for TAPB. Next, we introduce how to suppress "target prior bias" and estimate $P(\mathbf{Y}|\mathbf{D}, \text{do}(\mathbf{T}))$.

Three layers of the above aggregators are stacked, and the final output is aggregated and pooled along the amino acid dimension σ (where σ is the average) and classified using linear layers g_y and *Softmax*:

$$Y = \text{Softmax}(\sigma(g_y(F)))$$

Thus, we obtain the encoders, aggregator, and classifier for prediction using $P(Y|D, P)$. Next, we introduce how to suppress "target prior bias" and achieve $P(Y|D, do(P))$.

4.2.4 TAPB interventional training

PAGE 28-30

To adjust confounders in Figure 9(a), the backdoor adjustment for SCM in 9a is formulated as:

$$P(\mathbf{Y}|\mathbf{D}, do(\mathbf{T})) = \sum_{x_t} P(x_t)P(\mathbf{Y}|\mathbf{D}, \mathbf{T}, X_t = x_t) \quad (16)$$

Regrettably, this is infeasible. Unlike previous causal debiasing vision models, e.g. IFSL [2], VCRCNN [3] or IBMIL [4]—where tasks involve specific objects and observable confounders—the learned preferences in DTI models (potentially corresponding to protein families, sub-sequence lengths, or other latent factors) constitute unobservable confounders.

Furthermore, computing $P(\mathbf{Y}|\mathbf{D}, \mathbf{T}, X_t)$ for every X_t presents implementation challenges in deep learning frameworks: Since target sequences X_t remain static in non-augmented datasets, each target feature T corresponds to a single sequence and confounder category. Thus, for each DTI pair (\mathbf{D}, \mathbf{T}) , the model can only predict one $P(\mathbf{Y}|\mathbf{D}, \mathbf{T}, X_t)$ per forward pass. While inserting x_t -corresponding sub-sequences (if observable) during data augmentation could theoretically satisfy exact backdoor adjustment, this approach would increase computational costs—requiring additional forward/backward per augmented (\mathbf{D}, \mathbf{T}) pair—incurring prohibitive resource overhead and architectural inefficiency. Therefore, to estimate $P(\mathbf{Y}|\mathbf{D}, \mathbf{T}, X_t)$, we adopt approximations balancing causal inference rigor, deep learning feasibility, and training efficiency.

Unlike previous deep learning debiasing methods, e.g. VCRCNN and IBMIL, that employ the Normalized Weighted Geometric Mean (NWGM) [5] to approximate the backdoor adjustment, we implement the exact backdoor adjustment formula, approximating $P(\mathbf{Y}|\mathbf{D}, \mathbf{T}, c_i)$ to compute $P(\mathbf{Y}|\mathbf{D}, do(\mathbf{T}))$. As shown in Figure 9b, our SCM yields the backdoor adjustment:

$$P(\mathbf{Y}|\mathbf{D}, do(\mathbf{T})) = \sum_i P(\mathbf{Y}|\mathbf{D}, \mathbf{T}, \mathbf{c}_i)P(\mathbf{c}_i) \quad (17)$$

where c_i denotes cluster centers. Confounder dictionary \mathbf{C} and $P(c_i)$ are derived as follows: We cluster ESM-2-encoded features \mathbf{E} (preceding \mathbf{T} generation) across the training set to construct a confounder dictionary $\mathbf{C} \in \mathbb{R}^{I \times D_e}$ (Figure 5b). Here, I is the dictionary size (equivalent to the number of heads H in the aggregator’s MHCA). Since ESM-2 was pre-trained on disjoint datasets, DTI label leakage risks are eliminated. The sample proportion per cluster serves as the adjustment weight $P(c_i)$.

The path $X_t \rightarrow \mathbf{C} \rightarrow \mathbf{T}$ is established via our confounder alignment module (CAM) $g_t(\cdot)$. As shown in Figure 5e, CAM fuse cluster centers $c_i \in \mathbf{C}$ serves as the key \mathbf{K}_i and value \mathbf{V}_i for a distinct attention head within $g_t(\cdot)$, where they interact with the ESM-2 features \mathbf{E} :

$$\mathbf{T}_i = \text{Softmax} \left(\frac{\mathbf{Q}_i \mathbf{K}_i^\top}{\sqrt{d_k}} \right) \mathbf{V}_i \quad (18)$$

$$\mathbf{T} = g(\text{Concat}(\mathbf{T}_0, \mathbf{T}_1, \dots, \mathbf{T}_i) + \mathbf{E}) \quad (19)$$

where \mathbf{Q}_i represents the i -th head of linear projected \mathbf{E} , while \mathbf{K}_i and \mathbf{V}_i correspond to the i -th cluster center c_i projected through separate linear layers. Here D_k denotes the dimension of i -th head Q_i , and $g(\cdot)$ is a linear layer $\mathbb{R}^{B \times L_d \times D_e} \rightarrow \mathbb{R}^{B \times L_d \times D_m}$. CAM incorporates confounder features c_i into the \mathbf{E} via multi-head attention. Since data augmentation weakens the original confounding information within the features, this enables counterfactual features to be integrated into \mathbf{T} , establishing the path $X \rightarrow \mathbf{C} \rightarrow \mathbf{T}$. Note that computational costs remain minimal since $I \ll \text{length}(X_t)$.

To approximately estimate all counterfactual $P(\mathbf{Y}|\mathbf{D}, \mathbf{T}, c_i)$ within per forward, we leverage the independent interaction mechanism of MHCA in the aggregator. Equation 37 shows that MHCA partitions features along the embedding dimension into h independent heads. Since \mathbf{K} and \mathbf{V} remain invariant across layers, each \mathbf{Q}_i can individually extract c_i -relevant information. Due to this independent interaction mechanism, decomposing MHCA output into I heads yields distinct $\mathbf{F}_{c_i} \in \mathbb{R}^{B \times L_d \times D_k}$ approximations. This enables simultaneous estimation of all $P(\mathbf{Y}|\mathbf{D}, \mathbf{T}, c_i)$ in one forward, balancing theoretical rigor with computational efficiency. Note that H must equal the confounder dictionary size I and satisfy D_m is divisible by H .

The resulting output feature $\mathbf{F} \in \mathbb{R}^{B \times L \times D_m}$, where $D_m = I \times D_k$ is then split along its feature dimension into I segments, each corresponding to one confounder cluster:

$$\mathbf{F} = [\mathbf{F}_{\mathbf{c}_0}, \mathbf{F}_{\mathbf{c}_1}, \dots, \mathbf{F}_{\mathbf{c}_i}] \quad (20)$$

Here, $\mathbf{F}_{\mathbf{c}_i}$ represents the feature segment associated with the i -th confounder cluster. Finally, after applying average pooling to each $\mathbf{F}_{\mathbf{c}_i}$, a classification head $g_y(\cdot)$ is used to estimate all $P(\mathbf{Y}|\mathbf{D}, \mathbf{T}, \mathbf{c}_i)$:

$$P(\mathbf{Y}|\mathbf{D}, \mathbf{T}, \mathbf{c}_i) = \text{Softmax}(g_y(\mathbf{F}_{\mathbf{c}_i})) \quad (21)$$

then, we can parameterize Equation 40 via TAPB in the following form:

$$P(\mathbf{Y}|\mathbf{D}, do(\mathbf{T})) = \sum_i P(\mathbf{c}_i) \text{Softmax}(g_y(P(\mathbf{Y}|\mathbf{D}, \mathbf{T}, \mathbf{c}_i))) \quad (22)$$

Therefore, $P(\mathbf{Y}|\mathbf{D}, do(\mathbf{T}))$ can be computed via Equation 45 and integrated into deep learning training to adjust for confounders, achieving a practical balance between theoretical rigor and computational cost. The binary classification loss for TAPB can be denoted as follow:

$$\mathcal{L} = - \sum_{i=1} y_i \log(\hat{y}_i) \quad (23)$$

where y_i is the label, and \hat{y}_i is the predicted probability (i.e., from *Softmax*) for class i . Furthermore, we follow the masked language modeling in BERT to enhance the semantic features extracted by the drug encoder $f_d(\cdot)$. Specifically, 15% of all tokens in each sequence are randomly selected, with an 80% probability of being replaced by a [mask] token, a 10% probability of remaining unchanged, and a 10% probability of being replaced by a random token. The masked tokens are then predicted using $f_d(\cdot)$ and $g_m(\cdot)$, and the loss L_{mlm} is calculated by:

$$\mathcal{L}_{\text{mlm}} = - \sum_{i=1}^N \sum_{j=1}^{L_d} m_{ij} \log P(w_{ij}|\mathbf{H}_i) \quad (24)$$

where N is the number of samples, m_{ij} is a binary mask (1 if position j is masked, 0 otherwise), and $P(w_{ij}|\mathbf{H}_i)$ denotes the predicted probability for the token w_{ij} at position j of the i -th sample, with $\mathbf{H}_i \in \mathbb{R}^{D_m}$ representing the contextual representations generated by the $f_d(\cdot)$. The total loss for the TAPB can be denoted as follow:

$$L = L_b + L_{mlm} \quad (25)$$

The amino acid randomization and our TAPB training framework constitute a universal design applicable to other DTI models, provided that the dataset exhibits "target prior bias" and a pre-trained target encoder is used.

4.2.4 Biased training (stage 1)

As shown in Figure 5a, we first train TAPB using $P(Y|D, P)$ to capture the "target prior bias" in the sequence dataset. We use $f_d(\cdot)$ to extract drug feature D and load target feature T pre-extracted through ESM-2. To suppress "target prior bias" and reduce computational costs, for each T , we apply target random mask to randomly discard 70% of amino acid residues, followed by input to $g_p(\cdot)$ for dimensionality reduction. The aggregator $\mathcal{F}(\cdot)$ and pooling $\sigma(\cdot)$ are then used to obtain the fusion feature F , and a classifier $g_y(\cdot)$ outputs the probability of DTI combinations through *Softmax*. The loss for training plain TAPB can be denoted as follow:

$$L_b = -(y \log(\hat{y}) + (1 - y) \log(1 - \hat{y}))$$

Furthermore, to enhance the semantic features captured by the drug encoder $f_d(\cdot)$, we follow the masked language modeling in BERT. Specifically, 15% of all tokens in each sequence are randomly selected, with an 80% probability of being replaced by a [mask] token, a 10% probability of remaining unchanged, and a 10% probability of being replaced by a random token. The masked tokens are then predicted using $f_d(\cdot)$ and $g_m(\cdot)$, and the loss L_{mlm} is calculated through the cross-entropy function. The total loss function for the stage 1 training can be denoted as follow:

$$L = L_b + L_{mlm}$$

4.2.5 Intervential training (stage 2)

From causal lens, "target prior bias" is a confounder hidden in X_t . In addition to using target random mask, we can eliminate the effect of "target prior bias" via backdoor adjustment. As shown in Figure 9, backdoor adjustment severs the path $T \leftarrow X_t \rightarrow Y$, thereby inhibiting "target prior bias." We approximate the backdoor adjustment formula $P(Y|D, do(T))$ for interventional training. As shown in Figure 5b, we freeze the weights of $f_d(\cdot)$ and $f_t(\cdot)$ trained in the first stage. Based on the causal analysis in Section ??, we adjust the backdoor for DTI as follows:

$$P(Y|D, do(T)) = \sum_i P(c_i)P(Y|D, T, c_i)$$

Where c_i represents the cluster centers' tendencies. Furthermore, we can parameterize $P(Y|D, T, c_i)$ using the DTI model in the following form:

$$P(Y|D, do(T)) = \sum_i P(c_i) Softmax(g_y(\sigma(\mathcal{F}(D, T, c_i))))'$$

Where $g_y(\cdot)'$ is a new initialized linear classification head, σ is average, and \mathcal{F} is the aggregator. Following [2] [3], we further apply Normalized Weighted Geometric Mean (NWGM) [5] to move the outer sum into the *Softmax*:

$$P(Y|D, do(T)) \approx Softmax(\sum_i P(c_i) g_y(\sigma(\mathcal{F}(D, T, c_i))))'$$

Then we can move the summation to the inner of the linear layer $g_y(\cdot)'$:

$$P(Y|D, do(P)) \approx Softmax(g_y(\sum_i P(c_i) \sigma(\mathcal{F}(D, T, c_i))))'$$

Following [4], we cluster all features $\sigma(\mathcal{F}(D, P, c_i))$ in the training set using K-means, obtaining I cluster centers ($I = 8$) to approximate the label tendency t_i of each target. The cluster centers form a confounder dictionary V with shape $d \times I$, where d is the dimension ($d = 128$). We define $P(c_i)$ as:

$$A = [P(c_1), \dots, P(c_i)] = Softmax\left(\frac{\sigma(\mathcal{F}(D, T, c_i))^T V_i}{\sqrt{d}}\right)$$

Where A represents the attention matrix. We estimated the similarity between each $\sigma(\mathcal{F}(D, T, c_i))$ and all V_i using attention, and obtained the probability $P(c_i)$ using *Softmax*. Finally, we can rewrite the backdoor adjustment as follows:

$$P(Y|D, do(T)) \approx Softmax(g_y(\sum_i A_i V_i))'$$

Therefore, backdoor adjustment can be approximated by the equation ?? and integrated into deep learning training to suppress the "target prior bias." From the causal lens, Figure 9d shows the SCM of TAPB stage 2, which differs significantly from the original Figure 9a. The total loss function of the interventional training stage can be denoted as follows:

$$L = L_{\bar{b}}$$

The backdoor adjustment module is a universal design applicable to other DTI models, provided the dataset exhibits "target prior bias," the model uses *Softmax* for classification, and the classifier consists of only one linear layer.

Comment: A.3.

My understanding from the text in Section 2.3 (and the authors should really explain this more clearly), is that they cannot apply the exact backdoor adjustment formula above, and use the clustering as a proxy variable for the confounder. This would imply an edge $X_t \rightarrow C$ without any causal relation between C and T or Y . However:

Comment: A.3.1

There is a rich theory of using proxy variables in causal inference, and generally one needs at least two, not one, proxies for confounder adjustment. See for instance: <https://doi.org/10.1093/biomet/ast066>, <https://arxiv.org/abs/1609.08816>, <https://arxiv.org/abs/2009.10982>. The authors are encouraged to investigate the applicability of this theory to their problem.

Response A.3.1:

We sincerely appreciate the reviewer’s insightful suggestions regarding the application of proxy variable theory in causal inference. Since the label-related subsequences learned by the model within X_t are inherently complex and unobservable, we acknowledge that directly applying the exact backdoor adjustment is infeasible in our framework. After referencing [6], [1], [7], we attempted to adopt the method proposed by [1], as it aligns with our scenario where the confounder U remains unobserved. However, integrating this method into deep learning framework presents significant challenges, which we aim to address in future work. Below, we elaborate on our implementation attempts and current design.

Our SCM, illustrated in Figure 3, incorporates two proxy variables Z and W :

Figure 3: SCM with two proxy variables.

Here, Y represents the prediction, X_d denotes the SMILES of drugs in the training set, X_t denotes the target sequences in the training set, D and T are drug and target features, respectively. The proxies are constructed as follows:

Proxy Z : We select the most frequent amino acid categories as Z , since X_t directly influences Z , which in turn governs the generation of target features T . To ensure representativeness, we cluster residue features in the training set via k-means, extracting i cluster centers as Z .

Proxy W : A proxy CNN is trained solely on target sequences X_t and labels, with j cluster centers derived from its output features serving as W . This design leverages sequence data to approximate latent confounding patterns.

Given the collider $D \rightarrow Y \leftarrow T$ and $D \rightarrow Y \leftarrow W$, we reformulate Eq. 5 from [1] as:

$$\text{pr}\{y \mid D, \text{do}(T)\} = P(y \mid d, Z, t)P(W \mid d, Z, t)^{-1}P(W).$$

where d denotes the drug. For each (D, T) pair, $P(y \mid d, Z, t)$ is obtained by passing all Z values through the model (Figure 5), concatenating residue features with pooled representations $F(d, z_i, t)$, and feeding them into a classification head. $P(W)$ is estimated by counting cluster frequencies from the proxy CNN’s outputs.

However, estimating $P(W \mid d, Z, t)$ poses difficulties. For each (D, T) , only one sample of $P(W \mid d, Z_i, t)$ is available, precluding statistical estimation. To address this, we propose a deep learning-based similarity metric:

$$P(W \mid d, Z, t) = \frac{\exp\left(\frac{F(d, z_i, t) \cdot W_j}{\|F(d, z_i, t)\| \|W_j\|}\right)}{\sum_j \exp\left(\frac{F(d, z_i, t) \cdot W_j}{\|F(d, z_i, t)\| \|W_j\|}\right)},$$

where W_j denotes candidate cluster vectors. While this softmax formulation ensures valid probability ranges and normalization, the beginning of training shows near-uniform distributions, causing numerical instability during matrix inversion. Modifications such as Kaiming initialization marginally improved but did not resolve this issue.

As noted in our discussion, current biological dataset constraints limit proxy variables to sequence-derived features. Incorporating multimodal data (e.g., structural or functional annotations) could enhance proxy quality. We appreciate your suggestion to explore this direction. The revised manuscript now includes this discussion, which we intend to investigate in the context of deep learning frameworks in future work.

Revision: A.3.1.

3. Discussion

PAGE 25

Since the confounder is unobservable, we attempted to implement the proxy variables based confounder adjustment method from [1]. However, significant

computational challenges emerged when integrating this into our deep learning pipeline, particularly regarding the reliable estimation of the distribution and numerical instability during matrix inversion.

Notably, current biological dataset constraints limit proxy variables to sequence-derived features, which may not sufficiently capture the full spectrum of confounding biological mechanisms. Future incorporation of multi-modal data (e.g., structural or functional annotations) could enhance proxy quality by providing orthogonal information sources that better approximate latent confounders.

While causal inference theory provides principled solutions for unobserved confounders, e.g. [6], [1], [7], adaptation to deep learning frameworks remains nontrivial. We explicitly acknowledge these limitations in our discussion and will prioritize bridging this methodological gap in future work, with particular focus on multimodal proxy refinement.

Comment: A.3.2.

Worryingly, Fig. 5 and Section 4.2.5 suggest that the features F determine the clustering. That would imply an edge $F \rightarrow C$ and C being causally downstream of T in the SCM. No variable downstream of T can ever be used to adjust for the confounding path $T \leftarrow X_t \rightarrow Y$.

Response:A.3.2.

We sincerely appreciate the reviewer’s insightful comment and rigorous methodological scrutiny. We acknowledge that deriving cluster assignments C from fused features F in our original method wrongly implied an edge $F \rightarrow C$, placing C downstream of T and precluding its use for adjusting the confounding path $T \leftarrow X_t \rightarrow Y$.

To resolve this issue, we have redesigned our method in Revision A.2. to derive C exclusively from X_t -specific features extracted prior to feature fusion with T . We further implemented Causal Additive Modeling (CAM) to explicitly enforce the correct path $X_t \rightarrow C \rightarrow T$. These critical modifications establish C as a valid pre-treatment proxy variable by eliminating the problematic $F \rightarrow C$ dependency and ensuring proper causal positioning relative to T .

Furthermore, following your suggestion, we have revised the description of the TAPB framework in Section 2.3 of our manuscript accordingly.

2.3 TAPB framework

PAGE 10

In this paper, we introduce an interventional debiasing framework for alleviating target prior bias in drug-target interaction prediction (TAPB) as shown in Figure 5. However, the classification features f from TAPB stage 1 as shown in Figure 5a, like those in DrugBAN and TransformerCPI, still exhibit "target prior bias." A key distinction of TAPB from other DTI prediction models is the introduction of target random mask and an additional interventional training stage (referred to as stage 2) to mitigate "Target Prior Bias." As shown in Figure 5c, the target random mask randomly removes n ($n = 70\%$ in this paper) percent of the amino acids, reducing computational costs while randomizing the sequence to prevent model memorization.

The TAPB framework fundamentally differs from conventional DTI models through its integration of interventional training, confounder alignment, and amino acid randomization to estimate $P(\mathbf{Y}|\mathbf{D}, do(\mathbf{T}))$ via backdoor adjustment. As shown in Figure 5a, the interventional training module computes $P(\mathbf{Y}|\mathbf{D}, do(\mathbf{T}))$ via backdoor adjustment by incorporating all target confounder clusters $c_i \in C$. This requires the confounder dictionary C and the confounder alignment module $g_t(\cdot)$ as prerequisites.

The confounder dictionary C is constructed as shown in Figure 5b. We perform K-means clustering on ESM-2 features extracted from all training targets, where the cluster centers form C and the sample proportion per cluster c_i defines the adjustment weight $P(c_i)$. Since ESM-2 was pre-trained on datasets disjoint from DTI benchmarks, this eliminates label leakage risks.

The confounder alignment module $g_t(\cdot)$, illustrated in Figure 5d, operates during interventional training. It processes each confounder cluster center c_i to generate counterfactual representations \mathbf{T}_{c_i} and partitioned fused features \mathbf{F}_{c_i} . A shared classifier $g_y(\cdot)$ then computes $P(\mathbf{Y}|\mathbf{D}, \mathbf{T}, c_i)$ for all \mathbf{F}_{c_i} , enabling the approximation of $P(\mathbf{Y}|\mathbf{D}, do(\mathbf{T}))$.

Amino acid randomization in Figure 5c regularizes input sequences. First, 70% of residues in ESM-2 features are randomly deleted to reduce computation and disrupt sequence patterns. Subsequently, each residue feature undergoes independent mutation with 20% probability by replacement via random sampling from the amino acid dictionary. This dual randomization prevents spurious correlation learning by disrupting label-specific motifs.

As shown in Figure 5b, in stage 2, classification features with different label tendencies t_i in the confounder dictionary are borrowed and combined with

the output classification features f to alleviate "target prior bias." For that deep learning models naturally encode "prior tendency" into the classification features F , we can approximate different degrees of label tendency t_i using the cluster centers c_i obtained from K-means clustering and form a confounder dictionary. Specifically, the backdoor adjustment module, approximated using the attention mechanism as shown in Figure 5d, combines these features to achieve $P(Y|D, do(T))$.

We did not employ domain adaptation techniques, e.g. CDAN [8], and achieved better results on the cross-domain datasets, indicating the strong generalization of TAPB. Note that TAPB is a debiasing framework, and replacing encoders or aggregators can further enhance performance. The components of TAPB are generic, with the approximate backdoor adjustment requiring the satisfaction of certain assumptions. **The pseudocode of our method is provided in Supplementary Algorithm 1.**

Figure 5: Architecture of the TAPB framework. (a) **Stage 1 Biased Training:** TAPB optimizes $P(Y|D, T)$. SMILES strings and amino acid sequences are tokenized and embedded. Drug features D are extracted via a Bert-based encoder $f_d(\cdot)$ with rotary positional embedding, while target features T are derived using ESM-2 followed by a linear dimensionality reduction layer $g_t(\cdot)$. Masked language modeling (MLM) loss $g_m(\cdot)$ is used to enhance drug representation. (b) **Stage 2 Interventional Training:** Initialized with stage 1 weights, $f_d(\cdot)$ and $g_t(\cdot)$ are frozen. A confounder dictionary, generated by K-means clustering of stage 1 features f , enables backdoor adjustment via attention mechanisms to approximate $P(Y|D, do(T))$. (c) **Target Random Masking:** random remove of 70% amino acids from target embeddings P , retaining [CLS] token for sequence integrity. (d) **Backdoor Adjustment Module:** according to our derivation, classification features f serve as queries, while confounder cluster centers act as keys and values in an attention-based approximation of backdoor adjustment to alleviate "target prior bias."

(a) TAPB Interventional Training: The drug encoder BERT $f_d(\cdot)$ generates drug features D from SMILES. ESM-2 Pre-extracted target features E undergo amino acid randomization and are then processed by the CAM g_t . all cluster centers $c_i \in C$ act as keys/values in CAM g_t with E . Fused features F are partitioned into I segments F_{c_i} , each globally pooled and fed to classifier g'_y for estimating counterfactual $P(Y|D, T, c_i)$. Finally, $P(Y|D, do(T))$ is computed via backdoor adjustment. (b) **Target Confounder Dictionary C : K-means clustering of ESM-2 features from training targets builds confounder dictionary C .** (c) **Amino Acid Randomization: 1. Random deletion of 70% residue features; 2. Mutation of remaining residues to random features from the amino acid dictionary.** (d) **Confounder Alignment Module (CAM, $g_t(\cdot)$): Attention-weighted summation fuses c_i with target features, followed by dimensionality reduction and residual connection. Maintains explicit path $X_t \rightarrow C \rightarrow T$ across both training stages.**

Comment: A.3.3.

Related to the previous point, it is not completely clear what is being clustered and at what stage of the pipeline. For instance, is care being taken that clustering is done during each training run using only the samples in the current training fold to avoid data leakage?

Response:

We sincerely appreciate the reviewer's insightful comment regarding the clustering implementation and the critical issue of data leakage prevention. In our revised method (detailed in Revision A.2.), clustering is performed exclusively on the

ESM-2-encoded features E derived from the training set samples X_t . This step occurs strictly before the generation of the target feature T and is utilized to construct the confounder dictionary C as shown in Figure 5b), which aims to capture latent confounders. The dictionary C comprehensively represents confounders across all training samples in a single operation.

Regarding data leakage, firstly, the ESM-2 model was pre-trained on datasets entirely disjoint from our downstream DTI task, including all DTI-related labels. This fundamental separation precludes any risk of DTI-specific label leakage due to the confounder dictionary C . Secondly, we rigorously adhere to DrugBAN’s established data partitioning protocol, maintaining strict separation between the training, validation, and test splits throughout the entire process. This protocol guarantees that the clustering step utilizes exclusively data from the designated training set, thereby preventing any exposure to or influence from validation or test samples and ensuring the integrity of our evaluation.

Comment: A.4.

Why is it not possible to apply the exact backdoor adjustment formula above? If the sequences in the databases are random samples from the (unknown) distribution $P(x_t)$, then using their empirical distribution should be a good enough approximation.

Response:

We sincerely appreciate the reviewer’s professional comments and patient explanations. As detailed in our Revision A.2., we now adopt the empirical distribution for $P(c_i)$. The exact backdoor adjustment formula is as follow:

$$P(Y|D, \text{do}(T)) = \sum_{x_t} P(x_t)P(Y|D, T, X_t = x_t)$$

The reason that this exact backdoor adjustment cannot be applied is due to two fundamental constraints inherent in deep learning frameworks. First, for a given input X_t without augmentation, a single forward yields only one $P(Y|D, T, x_t)$, rendering statistical estimation of counterfactual distributions over unobserved confounders in X_t computationally infeasible. Second, while data augmentation could theoretically enable enumeration of x_t -specific sub-sequences to satisfy the summation, this approach necessitates multiple forward/backward passes per DTI pair (D, T) , incurring prohibitive computational overhead and architectural inefficiencies—contradicting the efficiency objectives of our pipeline.

Initially, we failed to directly employ the empirical distribution $P(x_t)$ because the latent confounders in X_t (e.g., preferred sequence patterns of variable length,

non-sequential structure, or dataset-specific variations) learned by models are inherently unobservable and lack explicit correspondence to measurable biological entities. Now, by approximating X_t via the confounder dictionary C —constructed from k-means clustering of ESM-2 features—we derive $P(c_i)$ as the empirical sample proportion per cluster. Our revised method aligns with causal inference requirements while preserving the practical flexibility needed for deep learning implementations, thus balancing theoretical rigor with computational tractability.

Comment: B.

The authors seem to equate randomization and randomized experiments in causal inference with randomly deleting parts of the target sequences. I don't think this is correct. Assume we want to randomize a particular target sequence residue. A proper randomization (from a causal inference perspective) would be to take the real sequences and then randomly assign each sequence one of the possible values of that particular residue, irrespective of the values of the other residues (compare to a clinical trial where each participant is randomly assigned a treatment value [drug or placebo] irrespective of any other characteristics of that individual). Or, since one is really interested in randomizing the features T to break the path $X_t \rightarrow T$, do this procedure at the level of features rather than residues. The authors should provide more justification of their randomization procedure from a causal inference perspective, and explore alternative randomization strategies.

Response: B.

We sincerely thank the reviewer's patient explanations and detailed suggestions. We acknowledge that we incorrectly equated random amino acid deletion with randomization in causal inference, and we are indeed interested in randomizing the relevant features T to break the path $X_t \rightarrow T$. Because deep learning models are fundamentally data-driven. What deep learning models actually accomplish is generating T , D to fitting the path $D \rightarrow F \rightarrow Y$ and $T \rightarrow F \rightarrow Y$. Crucially, the causal relationships between drug sequences and target sequences in the dataset—specifically, the authentic binding interactions—have already been experimentally validated.

Following your suggestion, we have revised our data augmentation method: in addition to randomly deleting a proportion of amino acids in the target feature T , we now introduce independent mutation of each amino acid feature. This revised data augmentation method is termed amino acids randomization. We have revised the corresponding Section 4.2.3 on data augmentation in the manuscript, as shown in Revision B.

4.2.3 Target random-mask Amino acid randomization

PAGE 27-28

To implement the random target mask, we obtain the pre-extracted target features T using ESM-2. As shown in Figure 5c, 70% of these residues are then randomly deleted, except the [cls] token, akin to the approach used by Masked Autoencoders (MAE) [9] where 75% of image patches are masked. The application of target random mask can be analogized to an intervention strategy in causal inference. By randomly deleting amino acids, we create a scenario akin to a randomized experiment that directly randomized target sequence. This approach helps to disrupt backdoor path $T \leftarrow X_t \rightarrow Y$, as shown in Figure 9b. Intuitively, target random mask can prevent models from memorizing the spurious correlations between targets and labels. Furthermore, reducing the sequence length by deleting 70% of residues lowers computational costs, thereby accelerating training and allowing for the exploration of larger models. This advancement is essential for deepening our understanding of the extensive and complex space of drug-target interactions (DTIs).

Amino acid randomization includes 2 parts: residue random deletion and residue feature mutation.

residue random deletion: As shown in Figure 5c, for each ESM-2 encoded feature, 70% of their residues are randomly deleted, except the [cls] token, akin to the approach used by Masked Autoencoders (MAE) [9] where 75% of image patches are masked.

residue feature mutation: After residue random deletion, for each remaining residue feature except special tokens in batch, has a 50% probability of remaining unchanged, and a 50% probability of being replaced by a random residue feature in the amino acids dict. The amino acid dict is obtained by average pooling every kind of amino acid feature (i.e., the last hidden state) extracted by ESM-2 in the training set.

By randomly deleting and independently mutating residues, we create a scenario akin to a randomized experiment that helps to disrupt backdoor path $T \leftarrow X_t \rightarrow Y$, as shown in Figure 9b. Intuitively, amino acid randomization can prevent models from memorizing the spurious correlations between targets and labels. Furthermore, the residue random deletion, reducing the sequence length, lowers computational costs, thereby accelerating training and allowing for the exploration of larger models. This is essential for deepening our understanding of the extensive and complex space of drug-target interactions.

Minor Comments:

Comment: 1.1.

The training data consist of known drug-target interactions, that is, by definition they only contain positive examples. How do the authors define negative labels?

Response:

We sincerely thank the reviewer for raising this critical question regarding negative sample construction. We use the publicly available dataset from DrugBAN [10], TransformerCPI [11], and Moltrans [12]. In our study, all datasets contain both experimentally confirmed positive interactions and carefully curated negative samples, with specific strategies tailored to each benchmark:

For the BindingDB dataset, negative pairs are rigorously defined as drug-target combinations with experimentally measured binding affinity $IC_{50} > 10^{-5}$ M (100-fold higher than the positive threshold of $IC_{50} < 100$ nM). This biochemical criterion ensures unambiguous non-interaction labels while minimizing false negatives.

For the BioSNAP dataset, negative samples are generated by randomly pairing drugs and targets without known interactions while maintaining balanced class distribution. This follows the established protocol from the original data source [12], where negative pairs are biologically plausible non-interactions.

For cross-domain evaluation, we applied the clustering-based pair split public datasets proposed in DrugBAN. These datasets ensure non-overlapping drug/target clusters between source and target domains by:

- Clustering drugs/proteins using ECFP4 fingerprints and PSC features
- Randomly selecting 60% clusters for the source domain
- Using remaining clusters for the target domain

Negative pairs in both domains adhere to their original dataset-specific criteria (BindingDB: IC_{50} threshold; BioSNAP: non-interacting pairs), preventing data leakage through strict cluster separation.

These protocols align with domain standards, i.e. Moltrans [12] and TransformerCPI [11], and explicitly address the challenge of negative sample definition in DTI prediction.

Comment: 1.2.

The authors focus on target prior bias because their analyses suggest that it is more widespread than drug prior bias. Nonetheless, Fig. 3 shows that drug prior bias is also present. Why can a similar adjustment method not be used to adjust for both biases simultaneously? Please comment.

Response:

We sincerely appreciate the reviewer’s insightful comments. Our method focus on addressing target prior bias (rather than drug prior bias) stems from three primary considerations:

1. The chemical moieties of drugs inherently possess higher probabilities of exhibiting multi-target binding capabilities due to their physicochemical properties, this represents authentic pharmacological phenomena. In contrast, targets typically contain highly conserved binding motifs (e.g., enzymatic active sites, receptor binding pockets) that show limited variability in interacting with diverse chemical groups. These conserved motifs frequently recur in training data, leading models to over-rely on these "prior tendencies" through repeated exposure during training.

2. As demonstrated in our t-SNE visualizations of fused features Fig. 3, when combining random features R with drug features, the model’s fused features F show no distinct clustering of positive samples. However, combining R with target features results in clear positive sample aggregation in F space. This empirically demonstrates the model’s stronger reliance on target features than drug features for predictions. Furthermore, the label distributions reveal critical imbalance differences: drug labels show relative balance (frequency concentrated around 0.5), whereas target labels exhibit extreme class imbalance. This evidence collectively positions target prior bias as the predominant challenge in current deep learning-based DTI studies.

3. Simultaneous adjustment for both biases would introduce an additional summation operation in the causal intervention framework:

$$P(Y|\text{do}(D), \text{do}(T)) = \sum_{i,j} P(c_i, d_j)P(Y|D, T, c_i, d_j)$$

This would lead to exponential growth in computational complexity, significantly increasing training costs. Moreover, since deep learning models can only obtain one probability instance $P(D, T, c_i, d_j)$ per forward pass, the approximation process would become substantially more intricate, compromising the model’s elegance and practicality. We fully agree that an ideal deep learning model should exhibit robustness against both types of biases. Our approach achieves an optimal balance between bias mitigation and computational costs.

Comment: 1.3.

Section 2.2.2: The "label test" presented here is not a statistical test, but only a heuristic score to measure the deviation of frequencies from 50%. Why not devise a proper statistical hypothesis test to estimate how likely it is to see such deviations under a proper null model?

Response 1.3.:

We sincerely appreciate the reviewer's insightful comment regarding the need for rigorous statistical validation of label deviations. We agree that a formal hypothesis test strengthens our analysis. We have retained the original heuristic score (Z) as it provides an intuitive measure of label tendency magnitude. We have supplemented this with a formal permutation-based hypothesis test to statistically quantify deviation significance. Results in Figure 3 demonstrate statistically significant deviations ($p < 0.001$) for targets across all datasets and for drugs in specific configurations, confirming protein-specific preferences beyond random fluctuations."

Revision: 1.3.

2.2.2 Prior tendency causes biased predictions

PAGE 7-9

To quantify "prior tendency" across different DTI datasets, we devised the following label test:

$$t_i = \frac{\sum_j y_{ij}}{n_i}$$

$$T = \sum_i |t_i - 0.5| + 0.5$$

where y_{ij} denote the j th label of the i th sequence, n_i denotes the total number of elements in the i th sequence, t_i is the label tendency which is rounded to one decimal place for better visualization, and T denotes the overall "prior tendency" across all sequences in the dataset, ranging from 0.5 to 1.0.

$$z_i = \frac{\sum_j y_{ij}}{n_i} \quad (2)$$

$$Z = \sum_i |z_i - 0.5| + 0.5 \quad (3)$$

where y_{ij} denote the j th label of the i th sequence, n_i denotes the total number of elements in the i -th sequence, z_i is the each sequence's "prior tendency"

which is rounded to one decimal place for better visualization, and Z denotes the overall "prior tendency" across all sequences in the dataset, ranging from 0.5 to 1.0.

Furthermore, beyond the heuristic score T , we designed a rigorous permutation test grounded in a null model where interaction labels Y are independent of target-specific effects. Under this null hypothesis, each drug-target pair's label follows a Bernoulli distribution parameterized by the global positive interaction proportion:

$$g = \frac{\sum Y}{N} \quad (4)$$

where N denotes the total number of drug-target pairs, representing random label assignment without target-specific biases. To evaluate statistical significance, we employed a weighted sum of squared deviations as our test statistic:

$$T = \sum_{i=1}^M n_i (g_i - g)^2 \quad (5)$$

where M is the number of unique proteins, n_i is the occurrence count of target i , and g_i is its observed positive interaction proportion. The n_i weighting ensures proportional contribution by sample size while maintaining sensitivity for sparse targets. Our permutation procedure preserves drug-protein pair structures while randomly reshuffling labels across all pairs for $B = 1000$ iterations, with p-value computed as:

$$\text{p-value} = \frac{1 + \sum_{b=1}^B \mathbf{1}(T_b \geq T_{\text{obs}})}{1 + B} \quad (6)$$

where T_b is the permuted statistic and T_{obs} the observed value. This non-parametric approach maintains DTI data structures through fixed pairings with permuted labels, ensures small-sample robustness by avoiding asymptotic assumptions, and naturally adapts to class imbalance.

We calculated the frequency of each "prior tendency", overall "prior tendency", and corresponding statistical significance, i.e. p-value for drugs as P_d and p-value for targets P_t , across 4 datasets. As shown in Figure 3a, in the BindingDB in-domain dataset, target label frequencies exhibit extreme bimodal concentrations at 0 and 1 ($P_t < 0.001$), while drug label frequencies center near 0.5 with no significant deviation ($P_d = 1.000$). Figure 3b reveals significant

drug deviations ($P_d < 0.001$) alongside persistently extreme target imbalance ($P_t < 0.001$) in the BindingDB cross-domain dataset. Figure 3c and d show significant deviations for both entities ($P_t < 0.001$, $P_d < 0.001$) in the BioSNAP in-domain and cross-domain datasets, with attenuated but still pronounced target imbalance. Figure 3e confirms targets consistently exhibit higher "prior tendency" Z than drugs across all configurations.

We calculated the label frequency and "prior tendency" for both drugs and targets in the BindingDB and BioSNAP datasets. The statistical results are presented in Figure 3. As illustrated in Figure 3a and b, in the BindingDB dataset, target label frequencies t_i in both in-domain and cross-domain training sets concentrate at the extremes (0 and 1), while drug label frequencies t_i center around 0.5, indicating a severe target label imbalance. Figure 3c and d reveals a similar pattern in the BioSNAP dataset, though less pronounced than in BindingDB. As shown in Figure 3e, both drugs and targets exhibit "prior tendency" in in-domain and cross-domain datasets, with targets consistently showing higher "prior tendency" T than drugs. Figure 3f schematically depicts datasets where certain targets (T_1 and T_2) exhibit preferences for positive and negative labels, respectively. This imbalance can cause models to rely more on target "prior tendency" rather than genuine drug-target interactions, leading to biased predictions. Figure 3g and h present our tests for "target bias" and "drug bias", respectively. When inputs are changed from DTI pairs (D, T) to pairs with a randomly generated tensor R , i.e. (D, R) and (T, R) , the model tends to predict based on target tendencies, highlighting the influence of target "prior tendency."

a

b

c

d

e

Figure 3: Statistical visualization of "prior tendency" for drugs and targets in the BindingDB and BioSNAP datasets. (a) Label frequency distribution of t_i for drugs and targets in the BindingDB in-domain training set. "Prior tendency" frequency distribution of z_i and p-value for drugs and targets in the BindingDB in-domain training set. (b) Label frequency distribution of t_i for drugs and targets in the BindingDB cross-domain training set. "Prior tendency" frequency distribution of z_i and p-value for drugs and targets in the BindingDB cross-domain training set. (c) Label frequency distribution of t_i for drugs and targets in the BioSNAP in-domain training set. "Prior tendency" frequency distribution of z_i and p-value for drugs and targets in the BioSNAP in-domain training set. (d) Label frequency distribution of t_i for drugs and targets in the BioSNAP cross-domain training set. "Prior tendency" frequency distribution of z_i and p-value for drugs and targets in the BioSNAP cross-domain training set. (e) Quantification of the overall "prior tendency" for labels associated with drugs and targets across different datasets.

Comment: 1.4.

The authors should be more careful in their use of causal inference terminology throughout the manuscript. For instance, in Section 4.2.5 they write "[...] backdoor adjustment severs the path $T \leftarrow X_t \rightarrow Y$, thereby inhibiting target prior bias. We approximate the backdoor adjustment formula $P(Y|D, do(T))$ for interventional training [...]". However, backdoor adjustment does not sever any paths (only interventions do) or perform interventional training. Instead backdoor adjustment is a method to compute $P(Y|D, do(T))$ without doing an actual intervention, that is, using only observations from the complete SCM (including the confounding influence).

Response: 1.4.

We sincerely appreciate the reviewer's thorough comments and professional explanations. We acknowledge that our initial use of causal inference terminology lacked precision, particularly in conflating the mechanism of backdoor adjustment with the effects of intervention. As correctly noted, backdoor adjustment is a method to compute $P(Y|D, do(T))$ using observational data from the structural causal model (SCM) without performing actual interventions. All relevant descriptions—especially in Section 4.2.5 where we incorrectly stated "backdoor adjustment severs the path"—have been comprehensively revised.

Abstract

PAGE 1

Drug Target Interaction (DTI) prediction is vital for drug repurposing. Previous studies trained on BioSNAP and BindingDB datasets have incorrectly identified "drug bias" as the cause of biased predictions, while our work reveals "target prior bias" as the predominant issue. This bias arises due to stems from the "prior tendency," characterized by the imbalanced label distribution of targets in the training data. From causal lens, "target prior bias" is a confounder, causing models trained with $P(Y|D, T)$ to learn spurious associations between targets and labels rather than genuine interaction mechanisms. In this study, we introduce **Target Prior Bias in Drug-Target Interaction Prediction (TAPB)**, a novel debiasing framework that employs target random mask and interventional training with $P(Y|D, do(T))$ to address this bias. **In this study, we introduce alleviating Target Prior Bias in Drug-Target Interaction Prediction (TAPB), a novel debiasing framework that employs amino acid randomization, confounder alignment module (CAM), and interventional training to approximate $P(Y|D, do(T))$ via backdoor adjustment, thereby addressing this bias.** TAPB outperforms existing methods on both in-domain and cross-domain datasets, offering enhanced generalization and providing interpretable insights into drug-target interactions **DTIs**.

1.Introduction

PAGE 4

In this paper, we introduce an interventional debiasing framework for alleviating **Target Prior Bias in Drug-Target Interaction Prediction (TAPB)**. ~~As shown in Figure 1f, we introduce target random mask and an additional interventional training stage to alleviate "target prior bias."~~ In training stage 1, we apply target random mask by randomly removing amino acid features from the target features, reducing computational costs and diversifying input data, thereby preventing the model from potentially memorizing the training sets. In stage 2, backdoor adjustment is approximated using a confounder dictionary obtained via K-means [13] clustering, achieving $P(Y|D, do(T))$. **As shown in Figure 1f, TAPB employs amino acids randomization and confounder alignment module to approximate $P(Y|D, do(T))$ via backdoor adjustment, where $do(\cdot)$ denotes the intervention that sets the variable to a specific value, blocking all incoming paths to the variable. The backdoor adjustment computes $P(Y|D, do(T))$ via observational data without performing actual interventions.**

Figure 1: Overview of Bias Analysis in DTI Prediction and our framework. (a) Dataset Construction for BioSNAP and BindingDB. (b) An example sketch of target biased training sets shows that certain targets (T_1 and T_2) exhibit positive and negative "prior tendencies", respectively, while the bias in drugs is less pronounced. (c) Outline of the biased training process, where models learn from datasets containing an inherent "prior tendency." (d) "Target bias" tests demonstrate that pairing a randomly generated feature R with target T_1 yields higher positive scores compared to negative ones, with the opposite observed for T_2 . (e) "Drug bias" tests show relatively balanced scores when a drug D is paired with randomly generated feature R , indicating lesser influence on predictions. (f) Our TAPB with target random mask and backdoor adjustment interventional training aims at reducing "target prior bias". Our TAPB interventional training, combining amino acid randomization, confounder alignment module (CAM), and multi-head cross-attention (MHCA) to approximate $P(Y|D, do(T))$ via backdoor adjustment, thereby adjusting confounders.

2.3 TAPB framework

PAGE 10

In this paper, we introduce an interventional debiasing framework for alleviating target prior bias in drug-target interaction prediction (TAPB) as shown in Figure 5. However, the classification features f from TAPB stage 1 as shown in Figure 5a, like those in DrugBAN and TransformerCPI, still exhibit "target prior bias." A key distinction of TAPB from other DTI prediction models is the introduction of target random mask and an additional interventional training

stage (referred to as stage 2) to mitigate "Target Prior Bias." As shown in Figure 5c, the target random mask randomly removes n ($n = 70\%$ in this paper) percent of the amino acids, reducing computational costs while randomizing the sequence to prevent model memorization. The TAPB framework fundamentally differs from conventional DTI models through its integration of interventional training, confounder alignment, and amino acid randomization to estimate $P(\mathbf{Y}|\mathbf{D}, do(\mathbf{T}))$ via backdoor adjustment. As shown in Figure 5a, the interventional training module computes $P(\mathbf{Y}|\mathbf{D}, do(\mathbf{T}))$ via backdoor adjustment by incorporating all target confounder clusters $c_i \in C$. This requires the confounder dictionary C and the confounder alignment module $g_t(\cdot)$ as prerequisites.

2.4.2 In-domain comparison

PAGE 11-12

We compared TAPB with five baseline models—SVM, RF, MolTrans, TransformerCPI, and DrugBAN DrugBAN, PSICHIC, and MlanDTI—under random splitting settings. Unlike these models, which rely on $P(\mathbf{Y}|\mathbf{D}, \mathbf{T})$ for predictions, TAPB employs an interventional training stage using $P(\mathbf{Y}|\mathbf{D}, do(\mathbf{T}))$ to alleviate "target prior bias." TAPB predicts $P(\mathbf{Y}|\mathbf{D}, do(\mathbf{T}))$ via backdoor adjustment

4.1 Analysis DTI through causal inference

PAGE 24-25

Figure 9: Structural causal model (SCM) of conventional DTI models and TAPB. (a) SCM of the conventional DTI models exhibiting "target prior bias." (b) SCM after applying target random mask, which weakens the backdoor path $T \leftarrow X_t \rightarrow Y$. Note that the dashed lines, which are not rigorous representations of causal relationships within SCM, indicate a weakened connection rather than a complete removal of this path. (c) SCM after applying backdoor adjustment, severing the connection to T . (d) The final SCM of TAPB, "target prior bias" is suppressed from causal lens. (b) SCM of our stage 1 biased training without amino acids randomization, adding C into $X_t \rightarrow T$. (c) SCM after applying amino acid randomization, which weakens the confounding path $T \leftarrow C \leftarrow X_t \rightarrow Y$. Note that the dashed lines, which are not rigorous representations of causal relationships within SCM, indicate a weakened connection rather than a complete removal of this path. (d) SCM for $P(Y|D, do(T))$ where $do(T)$ blocks the path $X_t \rightarrow C \rightarrow T$. We compute $P(Y|D, do(T))$ via backdoor adjustment without actual intervention.

4.2.5 Interventinal training (stage 2)

PAGE 31

backdoor adjustment severs the path $T \leftarrow X_t \rightarrow Y$, thereby inhibiting "target prior bias." We approximate the backdoor adjustment formula $P(Y|D, do(T))$ for interventional training.

References

- [1] Wang Miao, Zhi Geng, and Eric J Tchetgen Tchetgen. Identifying causal effects with proxy variables of an unmeasured confounder. *Biometrika*, 105(4):987–993, 08 2018.
- [2] Zhongqi Yue, Hanwang Zhang, Qianru Sun, and Xian-Sheng Hua. Interventional few-shot learning. *Advances in neural information processing systems*, 33:2734–2746, 2020.
- [3] Tan Wang, Jianqiang Huang, Hanwang Zhang, and Qianru Sun. Visual commonsense r-cnn. In *Proceedings of the IEEE/CVF conference on computer vision and pattern recognition*, pages 10760–10770, 2020.
- [4] Tiancheng Lin, Zhimiao Yu, Hongyu Hu, Yi Xu, and Chang-Wen Chen. Interventional bag multi-instance learning on whole-slide pathological images. In *Proceedings of the IEEE/CVF Conference on Computer Vision and Pattern Recognition*, pages 19830–19839, 2023.

- [5] Kelvin Xu, Jimmy Lei Ba, Ryan Kiros, Kyunghyun Cho, Aaron Courville, Ruslan Salakhutdinov, Richard S Zemel, and Yoshua Bengio. Show, attend and tell: neural image caption generation with visual attention. In *Proceedings of the 32nd International Conference on International Conference on Machine Learning-Volume 37*, pages 2048–2057, 2015.
- [6] Manabu Kuroki and Judea Pearl. Measurement bias and effect restoration in causal inference. *Biometrika*, 101(2):423–437, 03 2014.
- [7] Eric J Tchetgen Tchetgen, Andrew Ying, Yifan Cui, Xu Shi, and Wang Miao. An introduction to proximal causal learning, 2020.
- [8] Mingsheng Long, Zhangjie Cao, Jianmin Wang, and Michael I Jordan. Conditional adversarial domain adaptation. *Advances in neural information processing systems*, 31, 2018.
- [9] Kaiming He, Xinlei Chen, Saining Xie, Yanghao Li, Piotr Dollár, and Ross Girshick. Masked autoencoders are scalable vision learners. In *Proceedings of the IEEE/CVF conference on computer vision and pattern recognition*, pages 16000–16009, 2022.
- [10] Peizhen Bai, Filip Miljković, Bino John, and Haiping Lu. Interpretable bilinear attention network with domain adaptation improves drug–target prediction. *Nature Machine Intelligence*, 5(2):126–136, 2023.
- [11] Lifan Chen, Xiaoqin Tan, Dingyan Wang, Feisheng Zhong, Xiaohong Liu, Tianbiao Yang, Xiaomin Luo, Kaixian Chen, Hualiang Jiang, and Mingyue Zheng. Transformerpci: improving compound–protein interaction prediction by sequence-based deep learning with self-attention mechanism and label reversal experiments. *Bioinformatics*, 36(16):4406–4414, 2020.
- [12] Kexin Huang, Cao Xiao, Lucas M Glass, and Jimeng Sun. Moltrans: molecular interaction transformer for drug–target interaction prediction. *Bioinformatics*, 37(6):830–836, 2021.
- [13] David Arthur and Sergei Vassilvitskii. K-means++: The advantages of careful seeding. In *Proceedings of the Eighteenth Annual ACM-SIAM Symposium on Discrete Algorithms, SODA 2007, New Orleans, Louisiana, USA, January 7-9, 2007*, 2007.

2 Response to Reviewer 2

Comment: (Remarks to the Author):

Drug Target Interaction (DTI) prediction is vital for drug repurposing. The authors believe that previous studies trained on BioSNAP and BindingDB datasets wrongly identified "drug bias" as the cause of biased predictions, and their research reveals that "target prior bias" is the predominant issue. This bias arises due to "prior tendency" characterized by the imbalanced label distribution of targets in the training data. Their model outperforms existing methods on both in-domain and cross-domain datasets, offering enhanced generalization and providing interpretable insights into drug-target interactions.

Response:

We sincerely appreciate the reviewer's profound insights and constructive critique of our work. Your recognition that our study "offers enhanced generalization and provides interpretable insights into drug-target interactions" is particularly encouraging, and we deeply appreciate the reviewer's exceptional domain expertise demonstrated through their nuanced understanding of bias mechanisms in deep learning.

Building on valuable suggestions from all reviewers, we have re-engineered the training framework: the original two-stage approach has been consolidated into a unified single-stage pipeline. This refinement—implemented across all six benchmark datasets—has yielded measurable performance improvements while maintaining our core focus on bias mitigation. We are grateful for this opportunity to enhance our methodology. Regarding the specific points raised:

1. Clarification distinguishing our conceptual "target prior bias" from neural network parameter bias, with suggestions for parameter tuning.
2. Further elaboration on the definition and operationalization of target prior bias.
3. Explanation of the relationship between k-means clustering features and label prediction.
4. Discussion on sample imbalance resolution versus target prior bias mitigation.
5. Comprehensive analysis of bias impact across data/model/training levels with literature expansion.

These thoughtful comments will be addressed systematically in our revision through methodological clarifications, additional experiments, and expanded references to prior work on spurious associations.

Comment: 2.1.

Generally speaking, bias plays an important role in deep learning, which affects the relationship between the model and the training examples and the learning results. By rationally setting bias and other model parameters, the performance of the model can be optimized, generalized capabilities can be improved, and the stability and interpretability of the model can be enhanced.

Response:2.1.

We sincerely appreciate the reviewer's thorough explanation and insightful comments regarding the role of bias in deep learning. We would like to clarify that the term "bias" referenced in our work specifically denotes "target prior bias"—a data-driven phenomenon where the model learns spurious correlations between targets and labels. This concept is distinct from parameter bias (i.e., bias terms in neural network layers), which is not the primary focus of our study. As noted, such parameter biases in frameworks like PyTorch are automatically optimized during backpropagation.

Regarding your mention of optimizing parameters, we clarify that the other model hyperparameters in our work include several values adopted directly from prior methodologies. In our revised Supplementary Section 2, we explain the reasoning behind the selection of key hyperparameters. Following your suggestion, we conducted comprehensive ablation studies on critical hyperparameters in our revised manuscript. This systematic evaluation focused on target random drop rate, mutation rate, and confounder dictionary size, enabling us to identify an optimized hyperparameter configuration that enhances model performance, generalization capability, and stability. We reran **all** of our ablation study on the above 3 hyperparameters to find the optimal configuration. These experimental refinements and the updated results are detailed in Revision 2.1.

Revision: 2.1.

Supplementary Section 2: Experiment settings:

PAGE 4

BATCH_SIZE=60: Chosen as the maximum size feasible on a single GPU to optimize memory utilization during training.

MAX_EPOCH=100: Adopted directly from DrugBAN's pretraining protocol [1] to ensure a fair comparison with their work.

LR=2e-4 & WEIGHT_DECAY=1e-4: Selected based on standard transformer optimization practices, e.g. BERT [2], for stable training dynamics.

TARGET_RANDOM_DROP_RATIO=0.7 & MUTATION_RATE=0.2: De-

terminated via ablation studies; values of 70% drop and 20% mutation yielded the best model robustness.

MASK_PROBABILITY=0.15: Follows the established masking strategy used in BERT [2] pretraining.

DICT_SIZE=8: Tuned to achieve an optimal balance between memory requirements and model performance.

DrugEncoder.d_model=256: Increased from 128 based on validation results showing improved molecular feature extraction capability.

DrugEncoder.n_layer=3, DrugEncoder.n_head=8, TransformerDecoder.n_layer=3, TransformerDecoder.n_head=8: Align with typical configurations for base transformer model architectures.

Activation gelu, DrugEncoder.dropout=0.1, TransformerDecoder.dropout=0.1: Utilize common settings and conventions from transformer [3].

DrugEncoder.vocab_size=2362: Fixed by the inherent size of the drug token dictionary used in Molformer [4].

PrEncoder.d_model=1280: Matches the output dimension of the pretrained ESM-2 [5] embeddings to enable direct transfer learning.

TransformerDecoder.d_model=256: Ensures dimensionality compatibility with the outputs from the DrugEncoder.

2.4.4 Ablation study

PAGE 13-15

Figure 7: Ablation Studies on TAPB results.(a) Impact of varying target random mask ratios on test set AUROC. (b) Robustness of TAPB performance to different confounder dictionary sizes. (c) Target bias test on a prior-balanced dataset using ESM-2 features. (d) Drug bias test on a prior-balanced dataset using ESM-2 features. (a) AUROC and AURPC of TAPB on BioSNAP cross-domain dataset under various target confounder dictionary sizes. (b) AUROC and AURPC of TAPB on the BioSNAP cross-domain dataset under various target random drop ratios. (c) AUROC and AURPC of TAPB on the BioSNAP cross-domain dataset under various mutation rates. (d) AUROC and AURPC of plain DrugBAN and DrugBAN w/ random deletion on the BioSNAP cross-domain dataset.(e) AUROC and AURPC of plain TransformerCPI and TransformerCPI w/ random deletion on the BioSNAP cross-domain dataset. (f) Ablation study of TAPB key components on the BioSNAP cross-domain dataset. (g) Ablation study of ESM-2 on the BioSNAP cross-domain dataset.(h) Target bias test on a prior-balanced dataset using ESM-2 features. (i) Drug bias test on a prior-balanced dataset using ESM-2 features.

We conducted ablation studies to evaluate the impact of key components and hyperparameters of the TAPB on the performance of DTI prediction. Unless specified, experiments are conducted on the BioSNAP in-domain dataset using TAPB with the same hyper-parameters as shown in Supplementary Table 8.

We conducted ablation studies to evaluate the impact of key components and hyperparameters of the TAPB framework on DTI prediction performance. Unless specified otherwise, all experiments were conducted on the BioSNAP in-domain dataset using TAPB with identical hyperparameters to those in Supplementary Table 8. Each experiment included five independent runs with different random seeds.

Target random mask ratio: we compared the performance of stage 1 and stage 2 on the test set AUROC for target random mask ratios ranging from 0 to 0.9. As shown in Figure 7a, when no target random mask is used (i.e., the ratio is 0), the model’s performance is the lowest. However, when the ratio exceeds 0, the AUROC increases significantly by about 1%, with little difference between different ratios. Intuitively, randomly masking the target sequence prevents the model from memorizing spurious correlations, improving performance and suppressing "target prior bias" effectively from causal lens. Additionally, using a larger ratio significantly reduces the length of the target feature P , thereby reducing computational costs, regardless of whether a transformer or CNN is used as the target encoder $f_t(\cdot)$ or aggregator $\mathcal{F}(\cdot)$. **Target confounder dictionary size:** given the varying label frequencies across the dataset, it is necessary to test the appropriate size of the confounder dictionary. As shown in Figure

7b, the performance of TAPB is relatively robust to the size of the confounder dictionary. Therefore, there is no need for fine-tuning this hyperparameter, and a wide range of dictionary sizes can effectively mitigate "target prior bias" and improve performance. For that we compute $P(\mathbf{Y}|\mathbf{D}, do(\mathbf{T}))$ via backdoor adjustment, the target confounder dictionary plays a critical role. We evaluated the appropriate dictionary size by comparing TAPB performance on the BioSNAP cross-domain dataset across varying sizes. As shown in Figure 7a, TAPB achieved optimal performance with a dictionary size of 8, yielding an average AUROC of 0.79 and AUPRC of 0.80. This configuration demonstrated superior AUPRC compared to other sizes and was consequently selected as the default setting.

Target random drop ratio: This hyperparameter governs amino acid randomization by deleting features at fixed proportions. We evaluated TAPB performance on the BioSNAP cross-domain dataset across drop ratios ranging from 0 to 0.9. Figure 7b shows that performance was lowest without random dropping (ratio=0). Introducing any drop ratio (>0) increased AUROC by approximately 4% across configurations. Random target masking prevents spurious correlation memorization, suppresses target prior bias, and reduces computational costs through shortened target feature length P . We selected a ratio of 0.7 as the default due to its stable predictive performance.

Mutation rate: This hyperparameter governs amino acid mutations during randomization. We evaluated TAPB performance on the BioSNAP cross-domain dataset across mutation rates ranging from 0 to 0.8, with results presented in Figure 7c. Model performance demonstrates a unimodal trend: it initially improves then declines as mutation rate increases. Peak performance occurs at a mutation rate of 0.4, achieving 0.79 AUROC and 0.80 AUPRC. This indicates that higher mutation rates introduce excessive noise that destabilizes training. We selected a mutation rate of 0.2 as default since it maintains competitive metrics approaching those at 0.4 while exhibiting the lowest variance across runs.

Integration of residual random deletion into other methods: To test the generalizability of our approach, we integrated residual random deletion into DrugBAN and TransformerCPI. Only residual random deletion was selected because our residual mutation and backdoor adjustment approximation require both a pretrained encoder and MHCA-based aggregation. Results on the BioSNAP cross-domain dataset as shown in Figure 7d and e, DrugBAN with random deletion showed a 1% AUROC improvement over the baseline, while TransformerCPI with deletion exhibited substantial gains of nearly 10% in both

AUROC and AUPRC. This discrepancy may arise from TransformerCPI's aggregator architecture being more suitable for modeling DTI under this modification. These results confirm our residual random deletion as a general, model-agnostic design.

TAPB components contributions: We analyzed individual component contributions by evaluating four configurations on the BioSNAP cross-domain dataset: Plain TAPB, TAPB w amino acid randomization, TAPB w randomization & MLM, and full TAPB. As shown in Figure 7f: Plain TAPB yielded the lowest performance; adding amino acid randomization increased AUROC and AUPRC by 3% each; incorporating MLM loss provided marginal AUPRC improvement with little AUROC gain; finally, the full implementation (achieving backdoor adjustment via CAM and MHCA) boosted both metrics by 1% over preceding configurations. This ablation confirms the individual efficacy of each component.

ESM-2 Encoder Contribution: Given ESM-2's strong representation capacity, we isolated its contribution by comparing TAPB wo ESM-2 (CNN encoder) against plain TAPB on the BioSNAP in-domain dataset. Results in Figure 7g show plain TAPB outperformed the CNN variant by 2.5% AUROC and 1% AUPRC, demonstrating the significant advantage of pretrained protein encoders. Notably, while our debiasing framework requires a pretrained encoder, the component ablation in Figure 7f separately validates the efficacy of TAPB's novel modules beyond encoder selection.

Comment: 2.2.

In deep learning, the parameters of the model include weights and biases. Weighting determines how input data affects the output of the network, while bias allows each neuron output to still have a non-zero value without any input or when the input is zero. Bias can be regarded as the "built-in threshold" for each neuron, which adjusts the activation threshold of the activation function. The training model in this paper does not specify whether the bias of the model itself is considered, but more considers the bias caused by the imbalance distribution of the training examples.

Response:2.2.

We sincerely appreciate the reviewer's insightful comment and patient explanation. As you noted, in deep learning, "bias" typically refers to a model parameter (e.g., in neural networks) that acts as an additive constant within neurons. This pa-

parameter, often denoted as b in equations like $y = Wx + b$, serves as a learnable threshold that shifts the activation function to enable non-zero outputs even when inputs are zero. In our study, all models, including TAPB, incorporate such bias parameters as standard practice—they are automatically handled by deep learning frameworks (e.g., PyTorch) during training. However, since this aspect is not the focus of our work, it was not explicitly detailed in the manuscript. We confirm that these model biases were included in our implementations but do not impact the core issue addressed by TAPB.

In contrast, the "bias" central to our paper—termed "**target prior bias**"—is a data-driven confounding bias arising from imbalanced label distributions in the training dataset, not a model parameter. Specifically, it stems from the "prior tendency" phenomenon (Section 2.2.2 proved this), where certain targets in Drug-Target Interaction (DTI) datasets like BioSNAP and BindingDB exhibit skewed distributions of positive and negative labels (e.g., some targets predominantly associate with positive interactions or negative interactions). For instance, as shown in Figure 3, targets in these datasets often have extreme label frequencies, while drugs show more balanced tendencies. This imbalance causes models trained on $P(\mathbf{Y}|\mathbf{D}, \mathbf{T})$ to learn spurious correlations between targets and labels, rather than genuine drug-target interaction mechanisms. Consequently, the model becomes biased toward memorizing target-specific patterns, leading to poor generalization on unseen or cross-domain data.

Why "Target Prior Bias" Must Be Addressed? As demonstrated in our cross-domain experiments Section 2.4.3, models like DrugBAN and TransformerCPI suffer significant performance drops when applied to new domains, due to over-reliance on target-specific priors. From causal lens, this bias acts as a confounder, opening backdoor paths (e.g., $\mathbf{T} \leftarrow X_t \rightarrow \mathbf{Y}$) that distort the true causal relationship between drug-target features and interactions. Ignoring it would perpetuate misleading predictions in drug repurposing applications. Our statistical analyses confirm that target prior bias is pronounced in public DTI datasets, making it a pervasive issue that cannot be resolved by simply improving encoders or aggregators. Thus, addressing this "target prior bias" driven by datasets is essential for developing robust, generalizable DTI models that prioritize biological mechanisms over dataset artifacts. Here we present our revised TAPB framework in Revision 2.2.

Revision: 2.2.

4.2.1 Encoders

PAGE 25-26

Drug encoder: We employ BERT with rotational position encoding(ROPE) [6] as the drug encoder. The input SMILES sequence S ~~sequences~~ **X_t in one training batch are** is tokenized using Molformer [4] dictionary and tokenizer,

and each token is embedded into a high-dimensional space into E_d $E_d \in \mathbb{R}^{B \times L_d \times D_m}$:

$$E_d = \text{Embedding}(S)$$

$$\mathbf{E}_d = \text{Embedding}(X_t) \quad (1)$$

where B denotes the batch size, L_d denotes the length of X_t and D_m denotes the dimension of the model. Next, TAPB stacks n layers ($n = 3$) of BERT layers with ROPE to construct more complex and abstract contextual representations. The output $\mathbf{D} \in \mathbb{R}^{B \times L_d \times D_m}$ of the entire drug encoder is a context-sensitive depth representation of the input drug SMILES sequence:

$$D = f_d(E_d) = \text{RoBert}(E_d)$$

$$\mathbf{D} = f_d(\mathbf{E}_d) = \text{Bert}(\mathbf{E}_d) \quad (2)$$

Target encoder: we **We** employ ESM-2 [5] as the target encoder. ESM-2 is employed as the ESMFold protein feature encoder, replacing multiple sequence alignment (MSA) and structural template parts, with positional embeddings modified to RoPE, and supports longer amino acid sequence encoding. Given that ESMFold is trained on significantly larger datasets and demonstrates superior performance compared to AlphaFold [7], ESM-2 exhibits exceptional capability in extracting 3-D structural information from protein sequences, making it highly suitable for drug-target interaction (DTI) prediction tasks. The input target amino acid sequence A is **sequences X_d in one training batch are** tokenized using the ESM-2 dictionary and tokenizer, and each amino acid is embedded in a high-dimensional space into E_t $\mathbf{E}_t \in \mathbb{R}^{B \times L_t \times D_e}$:

$$E_t = \text{Embedding}(A)$$

$$\mathbf{E}_t = \text{Embedding}(X_t) \quad (3)$$

where L_t denotes the length of X_t and D_e denotes the dimension of the ESM-2 encoded feature. The ESM-2 is then used to extract ESM-2 encoded features $\mathbf{E} \in \mathbb{R}^{B \times L_t \times D_e}$: ~~T , with 70% of amino acid residues randomly dropped, followed by input to $g_p(\cdot)$ for dimensionality reduction:~~

$$T = f_t(E_t) = g_p(ESM-2(E_t))$$

$$\mathbf{E} = f_t(\mathbf{E}_t) = \text{ESM-2}(\mathbf{E}_t) \quad (4)$$

In actual **practical** training, target **ESM-2 encoded** features are pre-extracted and saved, significantly reducing memory burden and accelerating the training process.

4.2.2 Aggregator

PAGE 26-27

Following TransformerCPI, we adopt the same aggregator $\mathcal{F}(\cdot)$ with cross attention in TAPB. Each fusion layer of the aggregator inputs drug features \mathbf{D} into the self-attention layer, followed by and layer normalization:

Following TransformerCPI, TAPB adopts the same aggregator $\mathcal{F}(\cdot)$ with Multi-head Cross Attention (MHCA), which is essential for our estimation of counterfactual $P(Y|D, T, c_i)$. First, each fusion layer of the aggregator takes the output of the previous layer $\mathbf{F} \in \mathbb{R}^{B \times L_d \times D_m}$ into the self-attention layer, followed by residual connection and layer normalization:

$$F = \text{ln}(F + \text{SelfAttention}(F, F, F))$$

$$\mathbf{F} = \text{LayerNorm}(\mathbf{F} + \text{Self Attention}(\mathbf{F}, \mathbf{F}, \mathbf{F})) \quad (5)$$

The above output F is used as Q , with protein features P as K and V in the cross-attention layer, followed by shortcut and layer normalization:

The output \mathbf{F} is then transformed through a linear layer to obtain $\mathbf{Q} \in \mathbb{R}^{B \times L_d \times D_m}$, while target features \mathbf{T} are projected via two separate linear layers into $\mathbf{K} \in \mathbb{R}^{B \times L_t \times D_m}$ and $\mathbf{V} \in \mathbb{R}^{B \times L_t \times D_m}$, which are fed into the cross-attention layer followed by residual connection and layer normalization:

$$F = \text{ln}(F + \text{crossAttention}(F, P, P))$$

$$\mathbf{F} = \text{LayerNorm}(\mathbf{F} + \text{MHCA}(\mathbf{F}, \mathbf{T}, \mathbf{T})) \quad (6)$$

The MHCA is formally defined as:

$$\text{MHCA}(\mathbf{Q}, \mathbf{K}, \mathbf{V}) = g_a(\text{Concat}(\mathbf{head}_1, \dots, \mathbf{head}_i)) \quad (7)$$

$$\mathbf{head}_i = \text{Softmax} \left(\frac{\mathbf{Q}_i \mathbf{K}_i^\top}{\sqrt{d_k}} \right) \mathbf{V}_i \quad (8)$$

where $\mathbf{Q}_i \in \mathbb{R}^{B \times L_d \times D_k}$ corresponds to the i -th head split from \mathbf{Q} , while $\mathbf{K}_i, \mathbf{V}_i \in \mathbb{R}^{B \times L_t \times D_k}$ represent the i -th heads split from \mathbf{K} and \mathbf{V} respectively, $g_a(\cdot)$ is a dimension-preserving linear layer, the MHCA in the aggregator contains H heads, and $D_k = D_m/H$. A feed-forward network (FFN) with residual connection and layer normalization is then applied:

Next, the FFN layer with shortcut and layer normalization is applied:

$$F = \text{ln}(F + \text{FFN}(F))$$

$$F = \text{LayerNorm}(F + \text{FFN}(F)) \quad (9)$$

Three identical aggregator layers are stacked, completing the aggregator architecture for TAPB. Next, we introduce how to suppress "target prior bias" and estimate $P(\mathbf{Y}|\mathbf{D}, \text{do}(\mathbf{T}))$.

Three layers of the above aggregators are stacked, and the final output is aggregated and pooled along the amino acid dimension σ (where σ is the average) and classified using linear layers g_y and Softmax :

$$Y = \text{Softmax}(\sigma(g_y(F)))$$

Thus, we obtain the encoders, aggregator, and classifier for prediction using $P(Y|D, P)$. Next, we introduce how to suppress "target prior bias" and achieve $P(Y|D, \text{do}(P))$.

4.2.4 TAPB interventional training

PAGE 28-30

To adjust confounders in Figure 9(a), the backdoor adjustment for SCM in 9a is formulated as:

$$P(\mathbf{Y}|\mathbf{D}, \text{do}(\mathbf{T})) = \sum_{x_t} P(x_t)P(\mathbf{Y}|\mathbf{D}, \mathbf{T}, X_t = x_t) \quad (10)$$

Regrettably, this is infeasible. Unlike previous causal debiasing vision models, e.g. IFSL [8], VCRCNN [9] or IBMIL [10]—where tasks involve specific

objects and observable confounders—the learned preferences in DTI models (potentially corresponding to protein families, sub-sequence lengths, or other latent factors) constitute unobservable confounders.

Furthermore, computing $P(\mathbf{Y}|\mathbf{D}, \mathbf{T}, X_t)$ for every X_t presents implementation challenges in deep learning frameworks: Since target sequences X_t remain static in non-augmented datasets, each target feature T corresponds to a single sequence and confounder category. Thus, for each DTI pair (\mathbf{D}, \mathbf{T}) , the model can only predict one $P(\mathbf{Y}|\mathbf{D}, \mathbf{T}, X_t)$ per forward pass. While inserting x_t -corresponding sub-sequences (if observable) during data augmentation could theoretically satisfy exact backdoor adjustment, this approach would increase computational costs—requiring additional forward/backward per augmented (\mathbf{D}, \mathbf{T}) pair—incurring prohibitive resource overhead and architectural inefficiency. Therefore, to estimate $P(\mathbf{Y}|\mathbf{D}, \mathbf{T}, X_t)$, we adopt approximations balancing causal inference rigor, deep learning feasibility, and training efficiency.

Unlike previous deep learning debiasing methods, e.g. VCRCNN and IBMIL, that employ the Normalized Weighted Geometric Mean (NWGM) [11] to approximate the backdoor adjustment, we implement the exact backdoor adjustment formula, approximating $P(\mathbf{Y}|\mathbf{D}, \mathbf{T}, c_i)$ to compute $P(\mathbf{Y}|\mathbf{D}, do(\mathbf{T}))$. As shown in Figure 9b, our SCM yields the backdoor adjustment:

$$P(\mathbf{Y}|\mathbf{D}, do(\mathbf{T})) = \sum_i P(\mathbf{Y}|\mathbf{D}, \mathbf{T}, \mathbf{c}_i)P(\mathbf{c}_i) \quad (11)$$

where c_i denotes cluster centers. Confounder dictionary \mathbf{C} and $P(c_i)$ are derived as follows: We cluster ESM-2-encoded features \mathbf{E} (preceding \mathbf{T} generation) across the training set to construct a confounder dictionary $\mathbf{C} \in \mathbb{R}^{I \times D_e}$ (Figure 5b). Here, I is the dictionary size (equivalent to the number of heads H in the aggregator’s MHCA). Since ESM-2 was pre-trained on disjoint datasets, DTI label leakage risks are eliminated. The sample proportion per cluster serves as the adjustment weight $P(\mathbf{c}_i)$.

The path $X_t \rightarrow \mathbf{C} \rightarrow \mathbf{T}$ is established via our confounder alignment module (CAM) $g_t(\cdot)$. As shown in Figure 5e, CAM fuse cluster centers $\mathbf{c}_i \in \mathbf{C}$ serves as the key \mathbf{K}_i and value \mathbf{V}_i for a distinct attention head within $g_t(\cdot)$, where they interact with the ESM-2 features \mathbf{E} :

$$\mathbf{T}_i = \text{Softmax} \left(\frac{\mathbf{Q}_i \mathbf{K}_i^\top}{\sqrt{d_k}} \right) \mathbf{V}_i \quad (12)$$

$$\mathbf{T} = g(\text{Concat}(\mathbf{T}_0, \mathbf{T}_1, \dots, \mathbf{T}_i) + \mathbf{E}) \quad (13)$$

where \mathbf{Q}_i represents the i -th head of linear projected \mathbf{E} , while \mathbf{K}_i and \mathbf{V}_i correspond to the i -th cluster center \mathbf{c}_i projected through separate linear layers. Here D_k denotes the dimension of i -th head Q_i , and $g(\cdot)$ is a linear layer $\mathbb{R}^{B \times L_d \times D_e} \rightarrow \mathbb{R}^{B \times L_d \times D_m}$. CAM incorporates confounder features \mathbf{c}_i into the \mathbf{E} via multi-head attention. Since data augmentation weakens the original confounding information within the features, this enables counterfactual features to be integrated into \mathbf{T} , establishing the path $X \rightarrow \mathbf{C} \rightarrow \mathbf{T}$. Note that computational costs remain minimal since $I \ll \text{length}(X_t)$.

To approximately estimate all counterfactual $P(\mathbf{Y}|\mathbf{D}, \mathbf{T}, \mathbf{c}_i)$ within per forward, we leverage the independent interaction mechanism of MHCA in the aggregator. Equation 37 shows that MHCA partitions features along the embedding dimension into h independent heads. Since \mathbf{K} and \mathbf{V} remain invariant across layers, each \mathbf{Q}_i can individually extract \mathbf{c}_i -relevant information. Due to this independent interaction mechanism, decomposing MHCA output into I heads yields distinct $\mathbf{F}_{\mathbf{c}_i} \in \mathbb{R}^{B \times L_d \times D_k}$ approximations. This enables simultaneous estimation of all $P(\mathbf{Y}|\mathbf{D}, \mathbf{T}, \mathbf{c}_i)$ in one forward, balancing theoretical rigor with computational efficiency. Note that H must equal the confounder dictionary size I and satisfy D_m is divisible by H .

The resulting output feature $\mathbf{F} \in \mathbb{R}^{B \times L \times D_m}$, where $D_m = I \times D_k$ is then split along its feature dimension into I segments, each corresponding to one confounder cluster:

$$\mathbf{F} = [\mathbf{F}_{\mathbf{c}_0}, \mathbf{F}_{\mathbf{c}_1}, \dots, \mathbf{F}_{\mathbf{c}_i}] \quad (14)$$

Here, $\mathbf{F}_{\mathbf{c}_i}$ represents the feature segment associated with the i -th confounder cluster. Finally, after applying average pooling to each $\mathbf{F}_{\mathbf{c}_i}$, a classification head $g_y(\cdot)$ is used to estimate all $P(\mathbf{Y}|\mathbf{D}, \mathbf{T}, \mathbf{c}_i)$:

$$P(\mathbf{Y}|\mathbf{D}, \mathbf{T}, \mathbf{c}_i) = \text{Softmax}(g_y(\mathbf{F}_{\mathbf{c}_i})) \quad (15)$$

then, we can parameterize Equation 40 via TAPB in the following form:

$$P(\mathbf{Y}|\mathbf{D}, do(\mathbf{T})) = \sum_i P(\mathbf{c}_i) \text{Softmax}(g_y(P(\mathbf{Y}|\mathbf{D}, \mathbf{T}, \mathbf{c}_i))) \quad (16)$$

Therefore, $P(\mathbf{Y}|\mathbf{D}, do(\mathbf{T}))$ can be computed via Equation 45 and integrated into deep learning training to adjust for confounders, achieving a practical balance between theoretical rigor and computational cost. The binary classification loss for TAPB can be denoted as follow:

$$\mathcal{L} = - \sum_{i=1} y_i \log(\hat{y}_i) \quad (17)$$

where y_i is the label, and \hat{y}_i is the predicted probability (i.e., from *Softmax*) for class i . Furthermore, we follow the masked language modeling in BERT to enhance the semantic features extracted by the drug encoder $f_d(\cdot)$. Specifically, 15% of all tokens in each sequence are randomly selected, with an 80% probability of being replaced by a [mask] token, a 10% probability of remaining unchanged, and a 10% probability of being replaced by a random token. The masked tokens are then predicted using $f_d(\cdot)$ and $g_m(\cdot)$, and the loss L_{mlm} is calculated by:

$$\mathcal{L}_{mlm} = - \sum_{i=1}^N \sum_{j=1}^{L_d} m_{ij} \log P(w_{ij}|\mathbf{H}_i) \quad (18)$$

where N is the number of samples, m_{ij} is a binary mask (1 if position j is masked, 0 otherwise), and $P(w_{ij}|\mathbf{H}_i)$ denotes the predicted probability for the token w_{ij} at position j of the i -th sample, with $\mathbf{H}_i \in \mathbb{R}^{D_m}$ representing the contextual representations generated by the $f_d(\cdot)$. The total loss for the TAPB can be denoted as follow:

$$L = L_b + L_{mlm} \quad (19)$$

The amino acid randomization and our TAPB training framework constitute a universal design applicable to other DTI models, provided that the dataset exhibits "target prior bias" and a pre-trained target encoder is used.

Figure 5: Architecture of the TAPB framework. (a) **Stage 1 Biased Training:** TAPB optimizes $P(Y|D, T)$. SMILES strings and amino acid sequences are tokenized and embedded. Drug features D are extracted via a Bert-based encoder $f_d(\cdot)$ with rotary positional embedding, while target features T are derived using ESM-2 followed by a linear dimensionality reduction layer $g_t(\cdot)$. Masked language modeling (MLM) loss $g_m(\cdot)$ is used to enhance drug representation. (b) **Stage 2 Interventional Training:** Initialized with stage 1 weights, $f_d(\cdot)$ and $g_t(\cdot)$ are frozen. A confounder dictionary, generated by K-means clustering of stage 1 features f , enables backdoor adjustment via attention mechanisms to approximate $P(Y|D, do(T))$. (c) **Target Random Masking:** random remove of 70% amino acids from target embeddings P , retaining [CLS] token for sequence integrity. (d) **Backdoor Adjustment Module:** according to our derivation, classification features f serve as queries, while confounder cluster centers act as keys and values in an attention-based approximation of backdoor adjustment to alleviate "target prior bias."

(a) TAPB Interventional Training: The drug encoder BERT $f_d(\cdot)$ generates drug features D from SMILES. ESM-2 Pre-extracted target features E undergo amino acid randomization and are then processed by the CAM g_t . all cluster centers $c_i \in C$ act as keys/values in CAM g_t with E . Fused features F are partitioned into I segments F_{c_i} , each globally pooled and fed to classifier g'_y for estimating counterfactual $P(Y|D, T, c_i)$. Finally, $P(Y|D, do(T))$ is computed via backdoor adjustment. (b) **Target Confounder Dictionary C :** K-means clustering of ESM-2 features from training targets builds confounder dictionary C . (c) **Amino Acid Randomization:** 1. Random deletion of 70% residue features; 2. Mutation of remaining residues to random features from the amino acid dictionary. (d) **Confounder Alignment Module (CAM, $g_t(\cdot)$):** Attention-weighted summation fuses c_i with target features, followed by dimensionality reduction and residual connection. Maintains explicit path $X_t \rightarrow C \rightarrow T$ across both training stages.

Comment: 2.3.

From causal lens, "target prior bias" is a confounder, causing models trained with $P(Y|D, T)$ to learn spurious associations between targets and labels rather than genuine interaction mechanisms — — — In view of this, unsupervised learning is used and the data has no labels. How to explain the relationship between the data and labels?

Response:2.3.

We sincerely appreciate the reviewer's insightful comment regarding the relationship between unsupervised learning and label prediction in our causal framework. We clarify below how the unsupervised confounder dictionary C directly

enables estimation of $P(Y|D, do(T))$ despite label absence during clustering:

The k-means clustering of ESM-2 features (Step 1 below) is unsupervised and label-agnostic by design, serving to identify biophysical confounders c_i in target sequences X_t . Crucially, these confounders bridge to label prediction through our causal intervention framework in three key steps:

1. **Unsupervised Confounder Identification:** We cluster ESM-2 features \mathbf{E} across the training set to build dictionary $\mathbf{C} = \{c_i\}_{i=1}^I$, where c_i represent latent protein biases (e.g., structural motifs or functional domains). This requires no labels since ESM-2 features encode inherent biophysical properties.

2. **Confounder-Aligned Representation:** The CAM module (Eqs. 16-17) integrates \mathbf{C} into target features \mathbf{T} via:

$$\mathbf{T} = g(\text{Concat}(\text{Softmax}(\frac{\mathbf{Q}_i \mathbf{K}_i^\top}{\sqrt{d_k}}) \mathbf{V}_i) + \mathbf{E})$$

where $\mathbf{K}_i, \mathbf{V}_i$ are linear projections of c_i . This establishes $X_t \rightarrow \mathbf{C} \rightarrow \mathbf{T}$, enabling confounder-specific feature conditioning.

3. **Cluster-Conditioned Label Prediction:** During aggregation, MHCA decomposes interactions into I independent heads (Eq. 37), each approximating $P(Y|D, T, c_i)$ through:

$$\mathbf{F} = [\mathbf{F}_{c_0}, \dots, \mathbf{F}_{c_i}]$$

$$P(Y|D, T, c_i) = \text{Softmax}(g_y(\mathbf{F}_{c_i}))$$

The final $P(Y|D, do(T))$ (Eq. 45) is the expectation over all cluster-specific predictions:

$$P(Y|D, do(T)) = \sum_i \underbrace{P(c_i)}_{\text{cluster weight}} \cdot \underbrace{P(Y|D, T, c_i)}_{\text{counterfactual prediction}}$$

Thus, while \mathbf{C} is built without labels, TAPB can approximately compute $P(Y|D, do(T))$ are trained using interaction labels to capture how each confounder class affects binding.

Comment: 2.4.

The author believes that all (classified) predictions are biased in learning, and the work to handle with bias becomes very important. In fact, the prior bias caused by training sample imbalance should be solved by dealing with sample imbalance. The reviewer believes that the model of this paper has achieved better prediction accuracy, which may not be effective in dealing with the "target prior bias".

Response:2.4.

We appreciate the reviewer’s insightful comments and wish to sincerely clarify our position: We do not assert that all deep learning predictions are inherently biased. Rather, biased predictions emerge specifically when labels systematically co-occur with certain features in the data, with the nature of such biases varying across datasets. Our work focuses on cases where severe feature-label co-occurrence significantly compromises prediction validity.

Regarding the suggestion to address this through sample rebalancing techniques, we concur that many valid approaches exist to mitigate data biases. However, "all roads lead to Rome," we introduce a novel causal perspective that diagnoses "target prior bias" as the fundamental cause of biased predictions in Drug-Target Interaction (DTI) datasets, challenging prior works on drug-related biases. This represents a paradigm shift in understanding DTI prediction limitations. Our framework employs two innovative interventions: Amino acid randomization disrupts spurious target-label associations by randomizing sequence features, inherently mitigating sample imbalance through decorrelation of targets from historical label tendencies. Furthermore, our interventional training adjusts confounders via backdoor adjustment, achieving estimation $P(T|D, do(T))$.

We fully acknowledge the reviewer’s valid concern that "improved prediction accuracy does not equate to genuine debiasing" – this could indeed reflect spurious correlation rather than true bias mitigation. To evaluate whether our method mitigates "target prior bias", we conduct rigorous cross-domain experiments on BioSNAP and BindingDB datasets. These datasets exhibit significantly different data distributions (e.g., target space characteristics and label priors), creating ideal testbeds for exposing model bias. Crucially, we report both overall metrics (AUROC/AUPRC/Accuracy) and bias-sensitive metrics (Sensitivity/Specificity) with thresholds (t) optimized by Youden Index ($J = \text{Sensitivity} + \text{Specificity} - 1$). Due to the rendering limitations of colorbox, Revision 2.4. for the Supplementary Table 4 appears without color boxing. Key observations demonstrating bias mitigation:

1. **Balanced Sensitivity-Specificity Tradeoff:** As shown in Supplementary Table 4, in BioSNAP, TAPB achieves simultaneous improvement in sensitivity (0.676 vs MlanDTI’s 0.571, +18.7%) while maintaining competitive specificity (0.789 vs 0.809). This breaks the typical accuracy-bias tradeoff where high specificity often implies low sensitivity.
2. **Domain-Shift Robustness:** In BindingDB (different prior distribution), TAPB achieves best-in-class specificity (0.565 vs next-best 0.534, +5.8%) while maintaining robust sensitivity (0.705). This demonstrates adaptability to new bias environments.
3. **Consistent Threshold Behavior:** Our optimized thresholds ($t_{\text{BioSNAP}} = 0.318$,

$t_{\text{BindingDB}} = 0.440$) are consistently near 0.5, indicating well-calibrated outputs. Competitors exhibit extreme thresholds (e.g., TransformerCPI’s $t = 0.07$ in BioSNAP suggests severe probability distortion).

These results confirm that TAPB’s accuracy gains stem from authentic bias mitigation, not exploitation of spurious correlations. The cross-domain stability and balanced metric profile demonstrate our method’s capacity to handle the fundamental "target prior bias" in DTI prediction, aligning with our causal reformulation.

Revision 2.4.

Table 4: Comparison of cross-domain performance on BioSNAP, BindingDB datasets. We report the averaged Youden Index optimized t for every model.

Method	AUROC	AUPRC	Accuracy	Sensitivity	Specificity
BioSNAP					
TransformerCPI [12] ($t=0.07$)	0.603±0.013	0.595±0.011	0.586±0.007	0.652±0.174	0.519±0.186
MolTrans [13] ($t=0.297$)	0.631±0.01	0.632±0.016	0.604±0.013	0.56±0.141	0.649±0.13
DrugBAN+CDAN [1] ($t=0.455$)	0.685±0.017	0.722±0.010	0.654±0.009	0.523±0.070	0.786±0.040
PSICHIC [14] ($t=0.262$)	0.778±0.016	0.786±0.014	0.724±0.02	0.695±0.046	0.754±0.043
MlanDTI [15] ($t=0.601$)	0.741±0.009	0.775±0.009	0.69±0.011	0.571±0.038	0.809±0.045
TAPB ($t=0.318$)	0.791±0.007	0.800±0.005	0.732±0.012	0.676±0.075	0.789±0.083
BindingDB					
TransformerCPI [12] ($t=0.158$)	0.514±0.029	0.478±0.02	0.517±0.028	0.692±0.248	0.365±0.26
MolTrans[13] ($t=0.263$)	0.605±0.038	0.554±0.044	0.582±0.026	0.638±0.185	0.534±0.179
DrugBAN+CDAN[1] ($t=0.410$)	0.601±0.046	0.57±0.051	0.579±0.039	0.541±0.159	0.612±0.2
PSICHIC [14] ($t=0.202$)	0.619±0.039	0.58±0.033	0.588±0.035	0.779±0.165	0.421±0.196
MlanDTI [15] ($t=0.326$)	0.678±0.035	0.63±0.047	0.627±0.025	0.828±0.082	0.452±0.108
TAPB ($t=0.440$)	0.676±0.016	0.628±0.029	0.63±0.016	0.705±0.155	0.565±0.158

Comment: 2.5.

In deep learning, the paper should be at the data level, model level and training process level, to what extent does bias seriously affect the results of DTI. In addition, the references cited in the paper do not involve related research on learning spurious associations between the model and the sample label due to bias.

Response:2.5.

We sincerely appreciate the reviewer’s professional comment. It should be noted that deep learning models have historically presented challenges in interpretability, which may limit our ability to thoroughly assess the impact of "target prior bias" on model performance. In our manuscript, we systematically address the extent to which bias affects DTI prediction results across three critical dimensions: the data level, model level, and training process level, while clarifying the causal mechanisms underlying our proposed method.

Data Level: We identify "target prior bias" as the dominant confounder in BioS-NAP and BindingDB datasets, where imbalanced label distributions of target proteins (i.e., prior tendency) mislead models into learning spurious associations instead of genuine interactions.

Model Level: Crucially, TAPB employs standard architectures for drug encoding (RoPE-enhanced BERT) and target encoding (ESM-2), with a conventional cross-attention aggregator. Our performance gains over baselines derive not from architectural complexity, but from our novel amino acid randomization module—a lightweight add-on that disrupts bias propagation:

(i) Residue random deletion: 70% of residues (excluding [CLS]) are randomly dropped during training (inspired by MAE [16]), disrupting spurious correlations $T \leftarrow X_t \rightarrow Y$.

(ii) Residue feature mutation: Each retained residue has a 50% probability of being replaced by a random feature from an ESM-2-derived amino acid dictionary.

This dual strategy prevents models from memorizing target-label spurious correlations while reducing computational costs.

Training Process Level: We implement a causally grounded interventional training pipeline:

$$P(\mathbf{Y}|\mathbf{D}, do(\mathbf{T})) = \sum_i P(\mathbf{c}_i) \cdot P(\mathbf{Y}|\mathbf{D}, \mathbf{T}, \mathbf{c}_i) \quad (20)$$

where confounders \mathbf{c}_i are approximated via a K-means-derived dictionary $\mathbf{C} \in \mathbb{R}^{I \times D_e}$ from ESM-2 features. Our Confounder Alignment Module (CAM) integrates

c_i into target features \mathbf{T} through multi-head attention:

$$\begin{aligned}\mathbf{T}_i &= \text{Softmax}\left(\frac{\mathbf{Q}_i\mathbf{K}_i^\top}{\sqrt{d_k}}\right)\mathbf{V}_i, \\ \mathbf{T} &= g(\text{Concat}(\mathbf{T}_0, \dots, \mathbf{T}_i) + \mathbf{E})\end{aligned}\quad (21)$$

Simultaneously, Multi-Head Cross Attention (MHCA) in the aggregator enables efficient estimation of all $P(\mathbf{Y}|\mathbf{D}, \mathbf{T}, c_i)$ in one forward pass by splitting features \mathbf{F} into I segments \mathbf{F}_{c_i} (where $I = H$). This design ensures computational feasibility while theoretically addressing unobservable confounders (e.g., protein families).

This holistic approach—spanning data characterization, model-level randomization, and causally motivated training—ensures robust generalization across in-domain and cross-domain datasets.

Regarding the references on spurious associations, we acknowledge that while Section 4.2.5 previously cited computer vision works leveraging causal inference to resolve spurious correlations (IFSL [8], VCRCNN [9], IBMIL [10]), we had not explicitly highlighted their specific relevance to bias mitigation. To address this, we have now explicitly articulated their causal debiasing role in the revised manuscript. Furthermore, we confirm that no directly comparable studies exist in DTI prediction literature—to our knowledge, our work remains the pioneering causal framework addressing target prior bias in this domain. These clarifications have been revised in Section 4.2.4 to strengthen the positioning of prior research.

Revision: 2.5.

4.2.5 TAPB interventional training

PAGE 28

Regrettably, this is infeasible. Unlike previous causal debiasing vision models, e.g. IFSL [8], VCRCNN [9] or IBMIL [10]—where tasks involve specific objects and observable confounders—the learned preferences in DTI models (potentially corresponding to protein families, sub-sequence lengths, or other latent factors) constitute unobservable confounders.

PAGE 28

Unlike previous deep learning debiasing methods, e.g. VCRCNN [9] and IBMIL [10], that employ the Normalized Weighted Geometric Mean (NWGM) [11] to approximate the backdoor adjustment, we implement the exact backdoor adjustment formula, approximating $P(\mathbf{Y}|\mathbf{D}, \mathbf{T}, c_i)$ to compute $P(\mathbf{Y}|\mathbf{D}, do(\mathbf{T}))$. As shown in Figure 9b, our SCM yields the backdoor adjustment:

$$P(\mathbf{Y}|\mathbf{D}, do(\mathbf{T})) = \sum_i P(\mathbf{Y}|\mathbf{D}, \mathbf{T}, \mathbf{c}_i)P(\mathbf{c}_i) \quad (17)$$

References

- [1] Peizhen Bai, Filip Miljković, Bino John, and Haiping Lu. Interpretable bilinear attention network with domain adaptation improves drug–target prediction. *Nature Machine Intelligence*, 5(2):126–136, 2023.
- [2] Jacob Devlin Ming-Wei Chang Kenton and Lee Kristina Toutanova. Bert: Pre-training of deep bidirectional transformers for language understanding. In *Proceedings of naacL-HLT*, volume 1, page 2. Minneapolis, Minnesota, 2019.
- [3] A Vaswani. Attention is all you need. *Advances in Neural Information Processing Systems*, 2017.
- [4] Jerret Ross, Brian Belgodere, Vijil Chenthamarakshan, Inkit Padhi, Youssef Mroueh, and Payel Das. Large-scale chemical language representations capture molecular structure and properties. *Nature Machine Intelligence*, 4(12):1256–1264, 2022.
- [5] Zeming Lin, Halil Akin, Roshan Rao, Brian Hie, Zhongkai Zhu, Wenting Lu, Nikita Smetanin, Allan dos Santos Costa, Maryam Fazel-Zarandi, Tom Sercu, Sal Candido, et al. Language models of protein sequences at the scale of evolution enable accurate structure prediction. *bioRxiv*, 2022.
- [6] Jianlin Su, Murtadha Ahmed, Yu Lu, Shengfeng Pan, Wen Bo, and Yunfeng Liu. Roformer: Enhanced transformer with rotary position embedding. *Neurocomputing*, 568:127063, 2024.
- [7] John Jumper, Richard Evans, Alexander Pritzel, Tim Green, Michael Figurnov, Olaf Ronneberger, Kathryn Tunyasuvunakool, Russ Bates, Augustin Žídek, Anna Potapenko, et al. Highly accurate protein structure prediction with alphafold. *nature*, 596(7873):583–589, 2021.
- [8] Zhongqi Yue, Hanwang Zhang, Qianru Sun, and Xian-Sheng Hua. Interventional few-shot learning. *Advances in neural information processing systems*, 33:2734–2746, 2020.
- [9] Tan Wang, Jianqiang Huang, Hanwang Zhang, and Qianru Sun. Visual commonsense r-cnn. In *Proceedings of the IEEE/CVF conference on computer vision and pattern recognition*, pages 10760–10770, 2020.

- [10] Tiancheng Lin, Zhimiao Yu, Hongyu Hu, Yi Xu, and Chang-Wen Chen. Interventional bag multi-instance learning on whole-slide pathological images. In *Proceedings of the IEEE/CVF Conference on Computer Vision and Pattern Recognition*, pages 19830–19839, 2023.
- [11] Kelvin Xu, Jimmy Lei Ba, Ryan Kiros, Kyunghyun Cho, Aaron Courville, Ruslan Salakhutdinov, Richard S Zemel, and Yoshua Bengio. Show, attend and tell: neural image caption generation with visual attention. In *Proceedings of the 32nd International Conference on International Conference on Machine Learning-Volume 37*, pages 2048–2057, 2015.
- [12] Lifan Chen, Xiaoqin Tan, Dingyan Wang, Feisheng Zhong, Xiaohong Liu, Tianbiao Yang, Xiaomin Luo, Kaixian Chen, Hualiang Jiang, and Mingyue Zheng. Transformerpci: improving compound–protein interaction prediction by sequence-based deep learning with self-attention mechanism and label reversal experiments. *Bioinformatics*, 36(16):4406–4414, 2020.
- [13] Kexin Huang, Cao Xiao, Lucas M Glass, and Jimeng Sun. Moltrans: molecular interaction transformer for drug–target interaction prediction. *Bioinformatics*, 37(6):830–836, 2021.
- [14] Huan Yee Koh, Anh TN Nguyen, Shirui Pan, Lauren T May, and Geoffrey I Webb. Physicochemical graph neural network for learning protein–ligand interaction fingerprints from sequence data. *Nature Machine Intelligence*, 6(6):673–687, 2024.
- [15] Zhousan Xie, Shikui Tu, and Lei Xu. Multilevel attention network with semi-supervised domain adaptation for drug-target prediction. In *Proceedings of the AAAI Conference on Artificial Intelligence*, volume 38, pages 329–337, 2024.
- [16] Kaiming He, Xinlei Chen, Saining Xie, Yanghao Li, Piotr Dollár, and Ross Girshick. Masked autoencoders are scalable vision learners. In *Proceedings of the IEEE/CVF conference on computer vision and pattern recognition*, pages 16000–16009, 2022.

3 Response to Reviewer 3

Comment: (Remarks to the Author):

This manuscript investigates target prior bias in drug-target interaction (DTI) prediction, identifying how specific datasets (BioSNAP and BindingDB) can cause models to learn spurious associations rather than true biological interactions. The authors propose the TAPB framework, which combines target random masking with interventional training to mitigate this bias. The framework reportedly improves generalisation across cross-domain settings without using domain adaptation.

Although this work highlights an important but understudied issue in DTI prediction and proposes a thoughtful framework to address it, the current scope of datasets, lack of recent baselines, unclear metric definitions, and missing experimental results limit the strength of its claims. My detailed comments are below.

Strengths: The focus on dataset-induced bias is timely and relevant. By highlighting target prior bias, the paper adds to ongoing discussions around data imbalance and generalisation in biomedical AI. The TAPB framework combines target random masking and interventional training to offer a new and potentially generalisable approach to bias mitigation in DTI prediction. The empirical results are promising. TAPB demonstrates improved cross-domain performance relative to established baselines, such as DrugBAN and TransformerCPI, without using domain adaptation.

Response:

We sincerely appreciate the reviewer’s insightful evaluation of our manuscript and meticulous review. We greatly appreciate your recognition of our work’s timeliness in addressing dataset-induced bias in DTI prediction and the potential of the proposed TAPB framework. Your thorough analysis and constructive comments have significantly helped us improve the manuscript.

Building on valuable suggestions from all reviewers, we have streamlined our framework by removing the initial training stage and integrated all components into the interventional training process. Revision 3.10 details the revised interventional training implementation. This modification has yielded performance improvements across all six evaluated datasets.

We are particularly grateful for your thoughtful identification of key areas for enhancement, including dataset scope, baseline comparisons, methodological clarity, and experimental rigor. In response to your expert suggestions, we have implemented comprehensive revisions throughout the manuscript. Major improvements

include:

- Expanded experimental evaluation with additional datasets (Davis, Human cold-spilt) and contemporary models (MolTrans, TransformerCPI, DrugBAN, PSICHIC, MlanDTI)
- Quantitative analysis of target prior bias through statistical measures and visualizations
- Systematic ablation studies isolating the contributions of ESM-2 versus our novel components
- Complete hyperparameter selection methodology and sensitivity analysis
- Enhanced metric definitions and standardized mathematical notation throughout
- Reporting of variance metrics and statistical significance for all key results
- Inclusion of previously missing figures and extended experimental analysis

These revisions have strengthened our methodological rigor, improved result interpretability, and broadened the scope of our conclusions. We believe the revised manuscript now provides more compelling evidence for TAPB's effectiveness in mitigating target prior bias while maintaining state-of-the-art performance. Thank you again for your invaluable expertise and careful review, which have greatly enhanced the quality of our work.

Weaknesses and areas for improvement:

Comment: 3.1.

The analysis and conclusions are based solely on two datasets (BioSNAP and BindingDB) and have focused on two models (DrugBAN and TransformerCPI). While the findings are informative, they may not generalise to other datasets or architectures. The authors acknowledge this limitation, noting that different datasets and models, such as NVRLM may show different types of bias, but without further exploration. Evaluating TAPB and the bias analysis across a broader range of datasets and models can strengthen the contribution.

Response:3.1.

We sincerely appreciate the reviewer's insightful comment. To clarify, our original analysis actually utilized four datasets: the BioSNAP in-domain, BioSNAP

cross-domain, BindingDB in-domain, and BindingDB cross-domain datasets. Addressing the reviewer's suggestion to evaluate TAPB and the bias analysis across a broader range of datasets and models, we have incorporated two additional datasets: Davis [1] and Human-cold [2]. The Davis dataset shares characteristics with the BioSNAP in-domain dataset in exhibiting "target prior bias", whereas the Human-cold split shows no pronounced "target prior bias".

We have now conducted comprehensive experiments across all six datasets using an expanded set of models, including MolTrans [3], TransformerCPI [4], DrugBAN [2], PSICHIC [5], and MlanDTI [6]. The corresponding experimental results and the bias analysis performed on these new datasets are presented in Revison3 3.1 of the revised manuscript. Additionally, in the original manuscript, we mistakenly cited the NRLMF method from paper [7] as "s NVRLM". We have corrected this error in the Revison3 3.1.

Revision: 3.1.

2.4.2 In-domain comparison

PAGE 11-12

We compared TAPB with five baseline models—SVM, RF, MolTrans, TransformerCPI, and DrugBAN MolTrans [3], TransformerCPI [4], DrugBAN [2], PSICHIC [5], and MlanDTI [6]—under random splitting settings. Unlike these models, which rely on $P(Y|D, T)$ for predictions, TAPB employs an interventional training stage using $P(Y|D, do(T))$ to alleviate "target prior bias." TAPB predicts \$P(Y|D, do(T))\$ via backdoor adjustment. The bias analysis for Davis and Human cold-split can be found in Supplementary Section 3, Davis shows "target prior bias" while Human cold-split shows "drug prior bias".

Figure 6: Radar chart comparisons of in-domain and cross-domain performance on BioSNAP and BindingDB, Davis, Human datasets. (a) In-domain evaluation on BioSNAP. (b) In-domain evaluation on BindingDB. (c) Cross-domain evaluation on BioSNAP. (d) Cross-domain evaluation on BindingDB. (a) In-domain evaluation on BioSNAP. (b) Cross-domain evaluation on BioSNAP. (c) In-domain evaluation on Davis. (d) In-domain evaluation on BindingDB. (e) Cross-domain evaluation on BindingDB. (f) Cold-split evaluation on Human.

Figure 6a demonstrates TAPB's in-domain superiority through comprehensive multi-metric analysis. The radar chart reveals TAPB's complete dominance over baselines in BioSNAP, forming a larger polygon area across all evaluation axes. As shown in Supplementary Table 1, TAPB outperformed DrugBAN by approximately 3.3% in AUROC, 3.4% in AUPRC, 3.8% in accuracy, 4.7% in sensitivity, and 3.0% in specificity. These improvements are statistically significant and underscore TAPB's effectiveness in addressing "target prior bias." As shown in Figure 6b Figure 6d, on the BindingDB dataset, where "target prior bias" is pronounced, models tend to overfit this bias, achieving higher scores without truly capturing the interaction mechanisms. Despite this challenge, TAPB maintains robust performance with marginal advantages over DrugBAN, with improvements of 0.1% in AUROC, and 0.1% in AUPRC as shown in Supplementary Table 1.

Figure 6a demonstrates TAPB’s in-domain superiority through comprehensive multi-metric analysis. The radar chart reveals TAPB’s complete dominance over baselines in BioSNAP, forming a larger polygon area across all evaluation axes. As shown in Table 1, TAPB achieves state-of-the-art performance in BioSNAP with significant improvements: 2.3% higher AUROC, 2.2% higher AUPRC, 2.7% higher accuracy, 3.4% higher sensitivity, and 1.9% higher specificity than the next best baseline PSICHIC as shown in Supplementary Table 3. These statistically significant gains underscore TAPB’s effectiveness in addressing target prior bias.

As shown in Figure 6d, BindingDB exhibits pronounced target prior bias where models tend to overfit. Despite this challenge, TAPB maintains robust competitiveness, achieving near state-of-the-art performance with only 0.2% lower AUROC and 0.3% lower AUPRC than the top-performing DrugBAN as shown in Supplementary Table 3, while demonstrating superior generalization capability.

Remarkably, TAPB dominates the Davis dataset across all metrics, outperforming the strongest baseline PSICHIC by 2.1% in AUROC and 7.4% in AUPRC—the largest improvement observed among all datasets as shown in Supplementary Table 3. This highlights TAPB’s exceptional ability to capture complex interaction patterns. In the challenging Human cold-split scenario, TAPB maintains competitive performance, surpassing DrugBAN by 2.0% in AUROC and 2.8% in AUPRC while outperforming three of five baselines. This consistent multi-dataset superiority confirms TAPB’s architectural advantages in mitigating bias while preserving predictive power.

2.4.2 In-domain comparison

PAGE

~~12-13 As shown in Figure 6c and d reveals TAPB’s superior cross-domain generalization. Supplementary Table 2 further shows the detailed results. Specifically, on the cross-domain BioSNAP dataset, TAPB (stage 2) achieved improvements of 8.8% in AUROC, 5.8% in AUPRC, and 13.6% in accuracy compared to DrugBAN with CDAN. On the cross-domain BindingDB dataset, TAPB (stage 2) outperformed DrugBAN with CDAN by 7.1% in AUROC, 6.1% in AUPRC, and 9.3% in accuracy. In the cross-domain setting, TAPB’s superior performance over DrugBAN and DrugBAN+CDAN underscores its ability to generalize beyond the training data distribution. This is likely due to TAPB’s focus on alleviating "target prior bias," which prevents the model from memorizing spurious correlations between targets and labels. In contrast, conventional models relying on $P(Y|D, T)$ are highly susceptible to these biases, resulting in a dramatic performance degradation when applied to unseen~~

data. For instance, as shown in Figure 6c and d, nearly all baselines' Sensitivity drops to nearly 0, and Accuracy stagnates around 0.5, highlighting their inability for generalization. In comparison, both stages of TAPB—stage 1 and stage 2—demonstrate significant contributions to performance enhancement and bias mitigation, underscoring the effectiveness of our interventional training approach in addressing "target prior bias."

As shown in Figure 6b and d, TAPB demonstrates superior cross-domain generalization capability. Supplementary Table 4 provides comprehensive performance comparisons. On the BioSNAP dataset, TAPB achieves state-of-the-art performance with 0.791 AUROC, 0.800 AUPRC, and 0.732 accuracy, outperforming the strongest baseline PSICHIC by 1.3% in AUROC, 1.4% in AUPRC, and 0.8% in accuracy. Notably, TAPB also attains the highest sensitivity among all methods at 0.676. On the BindingDB dataset, TAPB maintains robust competitiveness, achieving leading performance in AUROC 0.676, accuracy 0.630, and specificity 0.565, while surpassing DrugBAN with CDAN by 7.5% in AUROC, 5.8% in AUPRC, and 5.1% in accuracy.

TAPB's cross-domain superiority stems from its fundamental approach to alleviating target prior bias, which prevents reliance on spurious target-label correlations. Conventional models depending on $P(Y|D, T)$ exhibit severe performance degradation when encountering out-of-distribution targets, as evidenced in Figure 6b and d where baseline sensitivity collapses to near-zero levels and accuracy stagnates at approximately 0.5. In contrast, TAPB's interventional training paradigm effectively mitigates this bias, enabling consistent generalization beyond training distributions. The observed multi-metric advancements across both BioSNAP and BindingDB datasets confirm TAPB's architectural advantage in disentangling authentic drug-target interaction patterns from dataset-specific biases.

Supplementary Section 3. Data statistics

PAGE 10

We quantified the "prior tendency" and total "prior tendency" in the Davis and human cold-split datasets. As illustrated in Figure Supplymentary 1 a, b, and c, the Davis dataset exhibits a certain "target prior bias", while a more pronounced "drug prior bias" is observed in the human cold-split dataset.

Figure Supplementary 1: Statistics of Davis and Human cold-split. a "Prior tendency" frequency distribution of z_i for drugs and targets in the Davis training set. b "Prior tendency" frequency distribution of z_i for drugs and targets in the Human cold-split training set. c Overall "prior tendency" for labels associated with drugs and targets across Davis and Human cold-split datasets.

Comment: 3.2.

The most recent baseline compared against is DrugBAN, published in February 2023. Given the pace of advancement in this area, it is important to benchmark against more recent methods from late 2023 or 2024. Although more recent methods are cited, e.g. [29], they are not included in the experimental comparisons. This limits the ability to assess whether TAPB truly represents state-of-the-art performance.

Response:3.2.

We sincerely appreciate the reviewer's professional comment. We have now incorporated two additional baselines: PSICHIC [5] published in *Nature Machine Intelligence* 2024, i.e. [29] in our original manuscript, and MlanDTI [6] published in AAI 2024. This expands our baseline models to five: TransformerCPI [4], MolTrans [3], DrugBAN [2], PSICHIC [5], and MlanDTI [6]. Competitive results were achieved across all six datasets.

Since Revision 3.1 only presents mean values, Revision 3.2 now provides complete performance metrics—including means, standard deviations, and thresholds optimized by the Youden Index—for all models and PSICHIC and MlanDTI experiment settings. The original results have been retained in Supplementary Table 1 and 2. Owing to rendering limitations of `colorbox`, Revision 3.2 appears without colored boxing.

Revision 3.2.

Table 4: Comparison of in-domain performance on BioSNAP, BindingDB, Davis and Human cold-split datasets. We report the averaged Youden Index optimized t for every model.

Method	AUROC	AUPRC	Accuracy	Sensitivity	Specificity
BioSNAP					
TransformerCPI [4] ($t=0.873$)	0.891±0.004	0.896±0.005	0.822±0.005	0.809±0.020	0.835±0.021
MolTrans [3] ($t=0.687$)	0.893±0.008	0.900±0.010	0.819±0.011	0.798±0.020	0.839±0.016
DrugBAN [2] ($t=0.754$)	0.903±0.003	0.907±0.004	0.841±0.001	0.81±0.020	0.868±0.022
PSICHIC [5] ($t=0.885$)	0.92±0.003	0.922±0.004	0.854±0.004	0.840±0.025	0.868±0.024
MlanDTI [6] ($t=0.584$)	0.901±0.02	0.908±0.01	0.829±0.006	0.816±0.014	0.843±0.017
TAPB ($t=0.754$)	0.943±0.003	0.944±0.003	0.881±0.003	0.874±0.008	0.887±0.011
BindingDB					
TransformerCPI [4] ($t=0.403$)	0.948±0.001	0.932±0.002	0.891±0.004	0.881±0.009	0.900±0.005
MolTrans [3] ($t=0.274$)	0.938±0.003	0.913±0.003	0.868±0.008	0.877±0.015	0.862±0.016
DrugBAN [2] ($t=0.419$)	0.963±0.001	0.953±0.001	0.908±0.002	0.906±0.005	0.909±0.005
PSICHIC [5] ($t=0.326$)	0.948±0.003	0.929±0.007	0.891±0.007	0.890±0.011	0.892±0.001
MlanDTI [6] ($t=0.455$)	0.941±0.005	0.923±0.006	0.868±0.011	0.871±0.005	0.866±0.021
TAPB ($t=0.403$)	0.961±0.001	0.95±0.002	0.901±0.003	0.899±0.015	0.902±0.014
Davis					
TransformerCPI [4] ($t=0.263$)	0.843±0.026	0.204±0.047	0.747±0.034	0.835±0.016	0.743±0.036
MolTrans [3] ($t=0.396$)	0.900±0.008	0.341±0.029	0.802±0.038	0.865±0.039	0.798±0.042
DrugBAN [2] ($t=0.637$)	0.886±0.004	0.319±0.025	0.809±0.037	0.831±0.043	0.808±0.041
PSICHIC [5] ($t=0.523$)	0.912±0.005	0.366±0.023	0.848±0.023	0.862±0.029	0.848±0.026
MlanDTI [6] ($t=0.597$)	0.871±0.004	0.304±0.009	0.798±0.017	0.814±0.013	0.798±0.018
TAPB ($t=0.521$)	0.933±0.006	0.44±0.013	0.860±0.014	0.871±0.009	0.860±0.015
Human cold split					

Table 4: Comparison of cross-domain performance on BioSNAP, BindingDB datasets. We report the averaged Youden Index optimized t for every model.

Method	AUROC	AUPRC	Accuracy	Sensitivity	Specificity
BioSNAP					
TransformerCPI [4] ($t=0.07$)	0.603±0.013	0.595±0.011	0.586±0.007	0.652±0.174	0.519±0.186
MolTrans [3] ($t=0.297$)	0.631±0.01	0.632±0.016	0.604±0.013	0.56±0.141	0.649±0.13
DrugBAN+CDAN [2] ($t=0.455$)	0.685±0.017	0.722±0.010	0.654±0.009	0.523±0.070	0.786±0.040
PSICHIC [5] ($t=0.262$)	0.778±0.016	0.786±0.014	0.724±0.02	0.695±0.046	0.754±0.043
MlanDTI [6] ($t=0.601$)	0.741±0.009	0.775±0.009	0.69±0.011	0.571±0.038	0.809±0.045
TAPB ($t=0.318$)	0.791±0.007	0.800±0.005	0.732±0.012	0.676±0.075	0.789±0.083
BindingDB					
TransformerCPI [4] ($t=0.158$)	0.514±0.029	0.478±0.02	0.517±0.028	0.692±0.248	0.365±0.26
MolTrans[3] ($t=0.263$)	0.605±0.038	0.554±0.044	0.582±0.026	0.638±0.185	0.534±0.179
DrugBAN+CDAN[2] ($t=0.410$)	0.601±0.046	0.57±0.051	0.579±0.039	0.541±0.159	0.612±0.2
PSICHIC [5] ($t=0.202$)	0.619±0.039	0.58±0.033	0.588±0.035	0.779±0.165	0.421±0.196
MlanDTI [6] ($t=0.326$)	0.678±0.035	0.63±0.047	0.627±0.025	0.828±0.082	0.452±0.108
TAPB ($t=0.440$)	0.676±0.016	0.628±0.029	0.63±0.016	0.705±0.155	0.565±0.158

Table 4: Default hyper-parameters setting for PSICHIC in our experiments.

Model	Hyper-parameter	Value
PSICHIC	batch_size	10
	epochs	100
	lr_rate	1e-4
	weight_decay	1e-4
	clip	1
	betas	[0.9, 0.999]
	eps	1e-08
	min_lr_rate	0
	warmup_iters	0
	lr_decay_iters	290
	mol_in_channels	43
	prot_in_channels	33
	pror_evo_channels	1280
	hidden_channels	200
	pre_layers	2
	post_layers	1
	total_layer	3
	dropout	0
	dropout_attn_score	0.2
	heads	5

Table 4: Default hyper-parameters setting for MlanDTI on Davis, Human-cold, and in-domain datasets in our experiments.

Model	Hyper-parameter	Value
MLanDTI	batch	64
	epoch	40
	protein_dim	1024
	hid_dim	128
	atom_dim	32
	n_heads	8
	n_enlayers	2
	n_delayers	2
	dropout	0.2
	settings_cls.protein_dim	1024
	settings_cls.atom_dim	32
	settings_cls.hid_dim	128
	settings_cls.dropout	0.2
	lr	0.001
	weight_decay	0.0001
	decay_interval	5
lr_decay	0.5	

Table 4: Default hyper-parameters setting for MlanDTI on cross-domain datasets in our experiments.

Model	Hyper-parameter	Value
MLanDTI	batch	64
	epoch	25
	protein_dim	1024
	hid_dim	128
	atom_dim	32
	n_heads	8
	n_enlayers	2
	n_delayers	2
	dropout	0.2
	settings_cls.protein_dim	1024
	settings_cls.atom_dim	32
	settings_cls.hid_dim	128
	settings_cls.dropout	0.2
	lr	0.001
	weight_decay	0.0001
	decay_interval	5
lr_decay	0.5	

Comment: 3.3.

Figure 1 uses an illustrative drawing to explain dataset bias. However, it does not quantify how prevalent the bias is here. To strengthen the motivation, it would be helpful to show earlier how serious the training set target tendencies are in practice using statistics or visual analysis.

Response:3.3.

We sincerely appreciate the reviewer's thoughtful feedback. Figure 1 is indeed an illustrative drawing explaining dataset bias. Regarding the suggestion to quantify the practical severity of training set target tendencies using statistical or visual analysis, we note that Section 2.3 already contains such quantification: we proposed a heuristic label test to measure deviation from the 0.5 baseline, with visualization results presented therein. Additionally, we introduced a novel null model and designed a corresponding statistic to compute p-values. Both the newly implemented null model and the refined original label test are detailed in Section 2.2.2 of the revised manuscript.

Revision: 3.3.

2.2.2 Prior tendency causes biased predictions

PAGE 7-8

We assume that the biased predictions are caused by "prior tendency" : Intuitively, "prior tendency" refers to the imbalance in positive and negative samples for individual drugs or targets in the DTI sequence dataset. Models can minimize loss by simply capturing the label tendencies of drugs or targets rather than the true interaction mechanisms. , which we formally define as systematic label distribution biases inherent to individual drugs or targets in the DTI sequence dataset. Specifically, this refers to statistically significant deviations in the positive/negative sample ratios observed across different drugs (drug-level prior) or targets (target-level prior), which create spurious correlations that models can exploit to minimize loss without learning true interaction mechanisms. To quantify "prior tendency" across different DTI datasets, we devised the following label test:

$$t_i = \frac{\sum_j y_{ij}}{n_i}$$

$$F = \sum_i |t_i - 0.5| + 0.5$$

where $y_{i,j}$ denote the j th label of the i th sequence, n_i denotes the total number of elements in the i th sequence, t_i is the label tendency which is rounded to

one decimal place for better visualization, and T denotes the overall "prior tendency" across all sequences in the dataset, ranging from 0.5 to 1.0.

$$z_i = \frac{\sum_j y_{ij}}{n_i} \quad (22)$$

$$Z = \sum_i |z_i - 0.5| + 0.5 \quad (23)$$

where y_{ij} denote the j th label of the i th sequence, n_i denotes the total number of elements in the i -th sequence, z_i is the each sequence's "prior tendency" which is rounded to one decimal place for better visualization, and Z denotes the overall "prior tendency" across all sequences in the dataset, ranging from 0.5 to 1.0.

Furthermore, beyond the heuristic score T , we designed a rigorous permutation test grounded in a null model where interaction labels Y are independent of target-specific effects. Under this null hypothesis, each drug-target pair's label follows a Bernoulli distribution parameterized by the global positive interaction proportion:

$$g = \frac{\sum Y}{N} \quad (24)$$

where N denotes the total number of drug-target pairs, representing random label assignment without target-specific biases. To evaluate statistical significance, we employed a weighted sum of squared deviations as our test statistic:

$$T = \sum_{i=1}^M n_i (g_i - g)^2 \quad (25)$$

where M is the number of unique proteins, n_i is the occurrence count of target i , and g_i is its observed positive interaction proportion. The n_i weighting ensures proportional contribution by sample size while maintaining sensitivity for sparse targets. Our permutation procedure preserves drug-protein pair structures while randomly reshuffling labels across all pairs for $B = 1000$ iterations, with p-value computed as:

$$\text{p-value} = \frac{1 + \sum_{b=1}^B \mathbf{1}(T_b \geq T_{\text{obs}})}{1 + B} \quad (26)$$

where T_b is the permuted statistic and T_{obs} the observed value. This non-parametric approach maintains DTI data structures through fixed pairings with permuted labels, ensures small-sample robustness by avoiding asymptotic assumptions, and naturally adapts to class imbalance.

We calculated the frequency of each "prior tendency", overall "prior tendency", and corresponding statistical significance, i.e. p-value for drugs as P_d and p-value for targets P_t , across 4 datasets. As shown in Figure 3a, in the BindingDB in-domain dataset, target label frequencies exhibit extreme bimodal concentrations at 0 and 1 ($P_t < 0.001$), while drug label frequencies center near 0.5 with no significant deviation ($P_d = 1.000$). Figure 3b reveals significant drug deviations ($P_d < 0.001$) alongside persistently extreme target imbalance ($P_t < 0.001$) in the BindingDB cross-domain dataset. Figure 3c and d show significant deviations for both entities ($P_t < 0.001$, $P_d < 0.001$) in the BioSNAP in-domain and cross-domain datasets, with attenuated but still pronounced target imbalance. Figure 3e confirms targets consistently exhibit higher prior tendency Z than drugs across all configurations.

We calculated the label frequency and "prior tendency" for both drugs and targets in the BindingDB and BioSNAP datasets. The statistical results are presented in Figure 3. As illustrated in Figure 3a and b, in the BindingDB dataset, target label frequencies t_i in both in-domain and cross-domain training sets concentrate at the extremes (0 and 1), while drug label frequencies t_i center around 0.5, indicating a severe target label imbalance. Figure 3c and d reveals a similar pattern in the BioSNAP dataset, though less pronounced than in BindingDB. As shown in Figure 3e, both drugs and targets exhibit "prior tendency" in in-domain and cross-domain datasets, with targets consistently showing higher "prior tendency" T than drugs. Figure 3f schematically depicts datasets where certain targets (T_1 and T_2) exhibit preferences for positive and negative labels, respectively. This imbalance can cause models to rely more on target "prior tendency" rather than genuine drug-target interactions, leading to biased predictions. Figure 3g and h present our tests for "target bias" and "drug bias", respectively. When inputs are changed from DTI pairs (D, T) to pairs with a randomly generated tensor R , i.e. (D, R) and (T, R) , the model tends to predict based on target tendencies, highlighting the influence of target "prior tendency."

a

b

c

d

e

Figure 3: Statistical visualization of "prior tendency" for drugs and targets in the BindingDB and BioSNAP datasets. (a) Label frequency distribution of t_i for drugs and targets in the BindingDB in-domain training set. "Prior tendency" frequency distribution of z_i and p-value for drugs and targets in the BindingDB in-domain training set. (b) Label frequency distribution of t_i for drugs and targets in the BindingDB cross-domain training set. "Prior tendency" frequency distribution of z_i and p-value for drugs and targets in the BindingDB cross-domain training set. (c) Label frequency distribution of t_i for drugs and targets in the BioSNAP in-domain training set. "Prior tendency" frequency distribution of z_i and p-value for drugs and targets in the BioSNAP in-domain training set. (d) Label frequency distribution of t_i for drugs and targets in the BioSNAP cross-domain training set. "Prior tendency" frequency distribution of z_i and p-value for drugs and targets in the BioSNAP cross-domain training set. (e) Quantification of the overall "prior tendency" for labels associated with drugs and targets across different datasets.

Comment: 3.4.

The TAPB framework uses the ESM-2 encoder, which is a strong component and may be responsible for part of the performance gains. Without respective ablation studies, it is hard to isolate the effects of target random masking and interventional training. Moreover, it would be useful to apply the TAPB mechanisms to other models, such as DrugBAN or TransformerCPI, to test whether they provide consistent benefits.

Response:3.4.

We sincerely appreciate the reviewer's insightful comments and suggestions. In response to your suggestion, we have supplemented the ablation study on ESM-2 and extended the TAPB mechanisms to other DTI methods. Specifically, we implemented our technique on both DrugBAN and TransformerCPI. However, due to these models' inability to satisfy the preconditions for backdoor adjustment, we only applied random deletion modifications, which demonstrated performance improvements in both cases. Furthermore, we conducted a comprehensive ablation study evaluating the individual contributions of each TAPB module. The detailed ablation analysis is presented in Revision 3.4.

2.4.4 Ablation study

Figure 7: Ablation Studies on TAPB results.(a) Impact of varying target random mask ratios on test set AUROC. (b) Robustness of TAPB performance to different confounder dictionary sizes. (c) Target bias test on a prior-balanced dataset using ESM-2 features. (d) Drug bias test on a prior-balanced dataset using ESM-2 features. (a) AUROC and AURPC of TAPB on BioSNAP cross-domain dataset under various target confounder dictionary sizes. (b) AUROC and AURPC of TAPB on the BioSNAP cross-domain dataset under various target random drop ratios. (c) AUROC and AURPC of TAPB on the BioSNAP cross-domain dataset under various mutation rates. (d) AUROC and AURPC of plain DrugBAN and DrugBAN w/ random deletion on the BioSNAP cross-domain dataset.(e) AUROC and AURPC of plain TransformerCPI and TransformerCPI w/ random deletion on the BioSNAP cross-domain dataset. (f) Ablation study of TAPB key components on the BioSNAP cross-domain dataset. (g) Ablation study of ESM-2 on the BioSNAP cross-domain dataset.(h) Target bias test on a prior-balanced dataset using ESM-2 features. (i) Drug bias test on a prior-balanced dataset using ESM-2 features.

We conducted ablation studies to evaluate the impact of key components and hyperparameters of the TAPB on the performance of DTI prediction. Unless specified, experiments are conducted on the BioSNAP domain dataset using TAPB with the same hyper-parameters as shown in Supplementary Table 8.

We conducted ablation studies to evaluate the impact of key components and hyperparameters of the TAPB framework on DTI prediction performance. Unless specified otherwise, all experiments were conducted on the BioSNAP in-domain dataset using TAPB with identical hyperparameters to those in Supplementary Table 8. Each experiment included five independent runs with different random seeds.

Target random mask ratio: we compared the performance of stage 1 and stage 2 on the test set AUROC for target random mask ratios ranging from 0 to 0.9. As shown in Figure 7a, when no target random mask is used (i.e., the ratio is 0), the model’s performance is the lowest. However, when the ratio exceeds 0, the AUROC increases significantly by about 1%, with little difference between different ratios. Intuitively, randomly masking the target sequence prevents the model from memorizing spurious correlations, improving performance and suppressing "target prior bias" effectively from causal lens. Additionally, using a larger ratio significantly reduces the length of the target feature P , thereby reducing computational costs, regardless of whether a transformer or CNN is used as the target encoder $f_t(\cdot)$ or aggregator $\mathcal{F}(\cdot)$.

Integration of residual random deletion into other methods: To test the generalizability of our approach, we integrated residual random deletion into Drug-

BAN and TransformerCPI. Only residual random deletion was selected because our residual mutation and backdoor adjustment approximation require both a pretrained encoder and MHCA-based aggregation. Results on the BioSNAP cross-domain dataset as shown in Figure 7d and e, Drug BAN with random deletion showed a 1% AUROC improvement over the baseline, while TransformerCPI with deletion exhibited substantial gains of nearly 10% in both AUROC and AUPRC. This discrepancy may arise from TransformerCPI’s aggregator architecture being more suitable for modeling DTI under this modification. These results confirm our residual random deletion as a general, model-agnostic design.

TAPB components contributions: We analyzed individual component contributions by evaluating four configurations on the BioSNAP cross-domain dataset: Plain TAPB, TAPB w amino acid randomization, TAPB w randomization & MLM, and full TAPB. As shown in Figure 7f: Plain TAPB yielded the lowest performance; adding amino acid randomization increased AUROC and AUPRC by 3% each; incorporating MLM loss provided marginal AUPRC improvement with little AUROC gain; finally, the full implementation (achieving backdoor adjustment via CAM and MHCA) boosted both metrics by 1% over preceding configurations. This ablation confirms the individual efficacy of each component.

ESM-2 Encoder Contribution: Given ESM-2’s strong representation capacity, we isolated its contribution by comparing TAPB wo ESM-2 (CNN encoder) against plain TAPB on the BioSNAP in-domain dataset. Results in Figure 7g show plain TAPB outperformed the CNN variant by 2.5% AUROC and 1% AUPRC, demonstrating the significant advantage of pretrained protein encoders. Notably, while our debiasing framework requires a pretrained encoder, the component ablation in Figure 7f separately validates the efficacy of TAPB’s novel modules beyond encoder selection.

Comment: 3.5.

Several hyperparameters are used in the paper, such as $n = 70\%$ on page 9, $n = 3$ on page 19, and values 15%, 80%, and 10% on page 20, as well as $I = 8$ and $d = 128$ on page 21. How these values were selected is not explained. Were they tuned on a validation set, adopted from prior work, or selected heuristically?

Response:3.5.

We sincerely appreciate the reviewer’s thorough comments. The majority of our

parameters adopt standard settings from prior studies. The key hyperparameters—namely, the target confounder dictionary size ($I = 8$), target random drop rate ($n = 70\%$), and mutation rate ($m = 20\%$)—were determined through ablation studies. These ablation experiments have now been incorporated into the manuscript, as detailed in Section 3.5. Subsequently, each hyperparameter of our revised methodology is systematically addressed within Revision 3.5.

Revision: 3.5.

2.4.4 Ablation study

PAGE 14-15

Target confounder dictionary size: given the varying label frequencies across the dataset, it is necessary to test the appropriate size of the confounder dictionary. As shown in Figure 7b, the performance of TAPB is relatively robust to the size of the confounder dictionary. Therefore, there is no need for fine-tuning this hyperparameter, and a wide range of dictionary sizes can effectively mitigate "target prior bias" and improve performance. **For that we compute $P(\mathbf{Y}|\mathbf{D}, do(\mathbf{T}))$ via backdoor adjustment, the target confounder dictionary plays a critical role. We evaluated the appropriate dictionary size by comparing TAPB performance on the BioSNAP cross-domain dataset across varying sizes. As shown in Figure 7a, TAPB achieved optimal performance with a dictionary size of 8, yielding an average AUROC of 0.79 and AUPRC of 0.80. This configuration demonstrated superior AUPRC compared to other sizes and was consequently selected as the default setting.**

Target random drop ratio: This hyperparameter governs amino acid randomization by deleting features at fixed proportions. We evaluated TAPB performance on the BioSNAP cross-domain dataset across drop ratios ranging from 0 to 0.9. Figure 7b shows that performance was lowest without random dropping (ratio=0). Introducing any drop ratio (>0) increased AUROC by approximately 4% across configurations. Random target masking prevents spurious correlation memorization, suppresses target prior bias, and reduces computational costs through shortened target feature length P . We selected a ratio of 0.7 as the default due to its stable predictive performance.

Mutation rate: This hyperparameter governs amino acid mutations during randomization. We evaluated TAPB performance on the BioSNAP cross-domain dataset across mutation rates ranging from 0 to 0.8, with results presented in Figure 7c. Model performance demonstrates a unimodal trend: it initially improves then declines as mutation rate increases. Peak performance occurs at a mutation rate of 0.4, achieving 0.79 AUROC and 0.80 AUPRC. This indicates that higher mutation rates introduce excessive noise that destabilizes training.

We selected a mutation rate of 0.2 as default since it maintains competitive metrics approaching those at 0.4 while exhibiting the lowest variance across runs.

Supplementary Section 2: Experiment settings:

PAGE 3

BATCH_SIZE=60: Chosen as the maximum size feasible on a single GPU to optimize memory utilization during training.

MAX_EPOCH=100: Adopted directly from DrugBAN's pretraining protocol [2] to ensure a fair comparison with their work.

LR=2e-4 & WEIGHT_DECAY=1e-4: Selected based on standard transformer optimization practices, e.g. BERT [8], for stable training dynamics.

TARGET_RANDOM_DROP_RATIO=0.7 & MUTATION_RATE=0.2: Determined via ablation studies; values of 70% drop and 20% mutation yielded the best model robustness.

MASK_PROBABILITY=0.15: Follows the established masking strategy used in BERT [8] pretraining.

DICT_SIZE=8: Tuned to achieve an optimal balance between memory requirements and model performance.

DrugEncoder.d_model=256: A common default setting in DTI.

DrugEncoder.n_layer=3, DrugEncoder.n_head=8, TransformerDecoder.n_layer=3, TransformerDecoder.n_head=8: Align with typical configurations for base transformer model architectures.

Activation gelu, DrugEncoder.dropout=0.1, TransformerDecoder.dropout=0.1: Utilize common settings and conventions from transformer [9].

DrugEncoder.vocab_size=2362: Fixed by the inherent size of the drug token dictionary used in Molformer [10].

PrEncoder.d_model=1280: Matches the output dimension of the pretrained ESM-2 [11] embeddings to enable direct transfer learning.

TransformerDecoder.d_model=256: Ensures dimensionality compatibility with the outputs from the DrugEncoder.

Comment: 3.6.

Figures 3f, 3g, and 3h are referenced in the manuscript but not included. This breaks the logical flow and weakens the evidential support.

Response:3.6. We sincerely appreciate the reviewer’s meticulous review. We acknowledge that the manuscript inadvertently referenced Figures 3f, 3g, and 3h in error. These figures correspond correctly to subpanels **f**, **g**, and **h** of Figure 1. To address this and enhance clarity, the relevant section has been relocated to the Introduction in the revised manuscript. This section has been thoroughly rewritten, and Figure 1 has been comprehensively redesigned to provide enhanced clarification for these subfigures. Specifically, Figure 1 **f** illustrates our TAPB interventional training framework combining amino acid randomization, confounder alignment module (CAM), and multi-head cross-attention (MHCA) to approximate $P(\mathbf{Y}|\mathbf{D}, do(\mathbf{T}))$ via backdoor adjustment; Figure 1 **g** demonstrates the specific content originally intended for Figure 3g; and Figure 1 **h** shows the specific content originally intended for Figure 3h.

Revision: 3.6.

1. Introduction

PAGE 2

The cause of "target bias" is "prior tendency." For example, in Figure 1b, certain targets (T_1 and T_2) exhibit preferences for positive and negative labels, respectively. This imbalance we defined as "prior tendency" can cause models to remember target labels in the training set rather than genuine drug-target interactions, leading to biased predictions. Inspired by CF-VQA [12], we designed and performed our bias test for targets and drugs as shown in Figure 1d and 1e, respectively. When inputs change from DTI pairs (D, T) to pairs with a randomly generated tensor R , i.e. (D, R) and (T, R) , the model tends to predict based on training set target tendencies, highlighting the influence of target "prior tendency." In Figure 1c, we characterize this data-distorted training as biased training. **Figure 1f schematically depicts datasets where certain targets (T_1 and T_2) exhibit preferences for positive and negative labels, respectively. This imbalance can cause models to rely more on target "prior tendency" rather than genuine drug-target interactions, leading to biased predictions.**

Figure 1: Overview of Bias Analysis in DTI Prediction and our framework. (a) Dataset Construction for BioSNAP and BindingDB. (b) An example sketch of target biased training sets shows that certain targets (T_1 and T_2) exhibit positive and negative "prior tendencies", respectively, while the bias in drugs is less pronounced. (c) Outline of the biased training process, where models learn from datasets containing an inherent "prior tendency." (d) "Target bias" tests demonstrate that pairing a randomly generated feature R with target T_1 yields higher positive scores compared to negative ones, with the opposite observed for T_2 . (e) "Drug bias" tests show relatively balanced scores when a drug D is paired with randomly generated feature R , indicating lesser influence on predictions. (f) Our TAPB with target random mask and backdoor adjustment interventional training aims at reducing "target prior bias". **Our TAPB interventional training, combining amino acid randomization, confounder alignment module (CAM), and multi-head cross-attention (MHCA) to approximate $P(Y|D, do(T))$ via backdoor adjustment, thereby adjusting confounders.**

2.2.2 Prior tendency causes biased predictions

PAGE 8-9

Figure 3f schematically depicts datasets where certain targets (T_1 and T_2) exhibit preferences for positive and negative labels, respectively. This imbalance can cause models to rely more on target "prior tendency" rather than genuine drug-target interactions, leading to biased predictions. Figure 3g and h present our tests for "target bias" and "drug bias", respectively. When inputs are changed from DTI pairs (D, T) to pairs with a randomly generated tensor

R, i.e. (D, R) and (T, R) , the model tends to predict based on target tendencies, highlighting the influence of target "prior tendency."

Comment: 3.7.

The authors mention averaging over five runs, but no standard deviation or variance is reported for key results such as those in Figure 6. Without variance reporting, it is difficult to assess the robustness or significance of the improvements.

Response:3.7.

We sincerely thank the reviewer for their detailed comments. In the main text, we have incorporated radar charts to provide intuitive performance comparisons between our method and alternative approaches. Regarding variance metrics, these were previously documented in Supplementary Tables 1 and 2 of the original manuscript. To establish optimal Youdex thresholds, we have re-run five baseline methods across six datasets: BindingDB in-domain, BindingDB cross-domain, BioSNAP in-domain, BioSNAP cross-domain, along with the newly added Davis and Human cold-split datasets. The corresponding means and variances are now comprehensively reported in Revision 3.2.

Comment: 3.8.

In Figure 6, sensitivity for many baselines approaches zero, but specificity approaches one. This is a well-known trade-off in binary classification. The authors should clarify whether performance is reported using a fixed threshold and whether adjusting the threshold could yield a better trade-off.

Response:3.8.

We sincerely appreciate the reviewer's detailed comments. In our manuscript, we have incorporated radar charts to provide intuitive performance comparisons between our method and alternative approaches. Regarding variance metrics, these were previously documented in Supplementary Table 1 and 2 of the original manuscript. To establish optimal Youden Index thresholds, we have re-evaluated five baseline methods across six datasets: BindingDB random, BindingDB cluster, BioSNAP random, BioSNAP cluster, along with the newly added Davis and GPCR datasets. The corresponding means and variances are now comprehensively reported in Revision 3.2.

Comment: 3.9.

Key metrics such as label tendency and prior tendency are insufficiently explained in Section 2.2.2. Their definitions and physical interpretations are unclear.

Response:3.9.

We sincerely appreciate the reviewer's thoughtful comments. We confirm that "label tendency" and "prior tendency" refer to the same concept. In our revised manuscript, we have retained the term "prior tendency". This concept is now intuitively introduced in the Introduction, while Section 2.2.2 provides a comprehensive definition of prior tendency along with its computational method. These modifications of our manuscript are detailed in Revision 3.9.

Revision: 3.9.

1 Introduction

PAGE 3

The cause of "target **prior bias**" is "prior tendency." **Intuitively, "prior tendency" describes the imbalance in positive versus negative interaction labels associated with individual drugs or targets within the DTI sequence training set. Models can minimize loss by simply capturing the label tendencies of drugs or targets rather than the true interaction mechanisms.** For example, in Figure 1b, targets T_1 and T_2 **in training set** have more positive and negative labels, respectively. This imbalance we defined as "prior tendency," can cause models to remember target labels in the training set rather than genuine drug-target interactions, leading to biased predictions.

2.2.2 Prior tendency causes biased predictions

PAGE 7-8

We assume that the biased predictions are caused by "prior tendency" : ~~Intuitively, "prior tendency" refers to the imbalance in positive and negative samples for individual drugs or targets in the DTI sequence dataset. Models can minimize loss by simply capturing the label tendencies of drugs or targets rather than the true interaction mechanisms.~~ , **which we formally define as systematic label distribution biases inherent to individual drugs or targets in the DTI sequence dataset. Specifically, this refers to statistically significant deviations in the positive/negative sample ratios observed across different drugs (drug-level prior) or targets (target-level prior), which create spurious correlations that models can exploit to minimize loss without learning true interaction mechanisms.** To quantify "prior tendency" across different DTI datasets, we devised the following label test:

$$t_i = \frac{\sum_j y_{ij}}{n_i}$$

$$T = \sum_i |t_i - 0.5| + 0.5$$

where y_{ij} denote the j th label of the i th sequence, n_i denotes the total number of elements in the i th sequence, t_i is the label tendency which is rounded to one decimal place for better visualization, and T denotes the overall "prior tendency" across all sequences in the dataset, ranging from 0.5 to 1.0.

$$z_i = \frac{\sum_j y_{ij}}{n_i} \quad (27)$$

$$Z = \sum_i |z_i - 0.5| + 0.5 \quad (28)$$

where y_{ij} denote the j th label of the i th sequence, n_i denotes the total number of elements in the i -th sequence, z_i is the each sequence's "prior tendency" which is rounded to one decimal place for better visualization, and Z denotes the overall "prior tendency" across all sequences in the dataset, ranging from 0.5 to 1.0.

Comment: 3.10.

Important symbols are undefined, such as " $do(T)$ ". This may confuse readers unfamiliar with causal inference notation. Moreover, there is inconsistent use of mathematical symbols. For example, symbols are in bold in the abstract but not in the main text. In mathematical writing, bold and non-bold fonts usually represent different types of quantities so this inconsistency may mislead readers. Lastly, the tensor R is mentioned as generated, but its dimensionality is not specified. This reduces clarity and reproducibility.

Response:3.10.

We sincerely thank the reviewer for their insightful comments and valuable suggestions. In response, we have incorporated the definition of $do(\mathbf{T})$ and backdoor adjustment into the Introduction and clarified other causal inference terminology, such as Backdoor adjustment. We have also standardized the formatting consistency between boldface and regular math symbols, and explicitly defined the dimensions of all relevant variables, including the randomly generated tensor \mathbf{R} . These modifi-

cations are highlighted in Revision 3.10.

Revision: 3.10.

1 Introduction

PAGE 4

From causal lens, the "target prior bias" of DTI sequence datasets is a confounder that opens up backdoor paths for targets and predictions, making it difficult for DTI models to make unbiased predictions through $P(\mathbf{Y}|\mathbf{D}, \mathbf{T})$. **This issue has not been adequately addressed in previous studies.** In this paper, we introduce an interventional debiasing framework for alleviating **Target Prior Bias in Drug-Target Interaction Prediction (TAPB) (TAPB) in drug-target interaction prediction**. As shown in Figure 1f, we introduce target random mask and an additional interventional training stage to alleviate "target prior bias." In training stage 1, we apply target random mask by randomly removing amino acid features from the target features, reducing computational costs and diversifying input data, thereby preventing the model from potentially memorizing the training sets. In stage 2, backdoor adjustment is approximated using a confounder dictionary obtained via K-means [13] clustering, achieving $P(\mathbf{Y}|\mathbf{D}, do(\mathbf{T}))$. **As shown in Figure 1f, TAPB employs amino acids randomization and confounder alignment module to approximate $P(\mathbf{Y}|\mathbf{D}, do(\mathbf{T}))$ via backdoor adjustment, where $do(\cdot)$ denotes the intervention that sets the variable to a specific value, blocking all incoming paths to the variable. The backdoor adjustment computes $P(\mathbf{Y}|\mathbf{D}, do(\mathbf{T}))$ via observational data without performing actual interventions.**

2.1 DTI Prediction formulation on sequence datasets

PAGE 5

Due to the absence of token-level ground truth in DTI sequence datasets, previous studies [3] [4] [2] typically reformulated DTI prediction as a binary classification task. Let $\mathcal{X} = \{d, t, y\}$ $X = \{X_d, X_t, y\}$ denote a set of DTI data points, where d represents the SMILES string encoding of the small molecule X_d represents the **Simplified Molecular Input Line Entry System (SMILES) of the small molecule**, t X_t denotes the amino acid sequence of the target protein, and y is a binary label indicating the presence or absence of an interaction between the drug and the target. The general approach for DTI prediction involves three main steps: 1) Feature encoding: segment or convert the input SMILES and target sequences separately, and employ various respective encoders $f_d(\cdot)$ and $f_t(\cdot)$ to extract **encode** features, e.g. CNN [14], ResNet [15], GCN [16], LSTM [17] and Bert [8], etc. Drug features and target features are denoted as D and T , respectively; 2) Feature fusion: aggregate the features D and T using an aggregator $\mathcal{F}(\cdot)$, which could be feature concatenation[18],

Bilinear Attention Network (BAN) [19], Transformer [9], or other aggregators; 3) Prediction: Using the pooling $\sigma(\cdot)$ and a classification head $g_y(\cdot)$ for binary classification, i.e. predicting through $P(Y|D, T)$, which can be formulated as:

$$\mathbf{D} = f_d(X_d), \mathbf{T} = f_t(X_t), \mathbf{F} = \mathcal{F}(\mathbf{D}, \mathbf{T}), Y = g_y(\sigma(\mathbf{F})) \quad (29)$$

$$D = f_d(d), T = f_t(t), f = \mathcal{F}(D, T), Y = g_y(\sigma(f))$$

Building upon this framework, previous studies focused on enhancing model performance by refining feature encoders, and aggregators or incorporating additional features.

Figure 5: Architecture of the TAPB framework. (a) **Stage 1 Biased Training:** TAPB optimizes $P(Y|D, T)$. SMILES strings and amino acid sequences are tokenized and embedded. Drug features D are extracted via a Bert-based encoder $f_d(\cdot)$ with rotary positional embedding, while target features T are derived using ESM-2 followed by a linear dimensionality reduction layer $g_t(\cdot)$. Masked language modeling (MLM) loss $g_m(\cdot)$ is used to enhance drug representation. (b) **Stage 2 Interventional Training:** Initialized with stage 1 weights, $f_d(\cdot)$ and $g_t(\cdot)$ are frozen. A confounder dictionary, generated by K-means clustering of stage 1 features f , enables backdoor adjustment via attention mechanisms to approximate $P(Y|D, do(T))$. (c) **Target Random Masking:** random remove of 70% amino acids from target embeddings P , retaining [CLS] token for sequence integrity. (d) **Backdoor Adjustment Module:** according to our derivation, classification features f serve as queries, while confounder cluster centers act as keys and values in an attention-based approximation of backdoor adjustment to alleviate "target prior bias."

(a) TAPB Interventional Training: The drug encoder BERT $f_d(\cdot)$ generates drug features D from SMILES. ESM-2 Pre-extracted target features E undergo amino acid randomization and are then processed by the CAM g_t . all cluster centers $c_i \in C$ act as keys/values in CAM g_t with E . Fused features F are partitioned into I segments F_{c_i} , each globally pooled and fed to classifier g'_y for estimating counterfactual $P(Y|D, T, c_i)$. Finally, $P(Y|D, do(T))$ is computed via backdoor adjustment. (b) **Target Confounder Dictionary C :** K-means clustering of ESM-2 features from training targets builds confounder dictionary C . (c) **Amino Acid Randomization:** 1. Random deletion of 70% residue features; 2. Mutation of remaining residues to random features from the amino acid dictionary. (d) **Confounder Alignment Module (CAM, $g_t(\cdot)$):** Attention-weighted summation fuses c_i with target features, followed by dimensionality reduction and residual connection. Maintains explicit path $X_t \rightarrow C \rightarrow T$ across both training stages.

4.2.1 Encoders

PAGE 25-26

Drug encoder: We employ BERT with rotational position encoding (ROPE) [20] as the drug encoder. The input SMILES sequence S sequences X_t in one training batch are is tokenized using Molformer [10] dictionary and tokenizer, and each token is embedded into a high-dimensional space into E_d $E_d \in \mathbb{R}^{B \times L_d \times D_m}$.

$$E_d = \text{Embedding}(S)$$

$$\mathbf{E}_d = \text{Embedding}(X_t) \tag{30}$$

where B denotes the batch size, L_d denotes the length of X_t and D_m denotes the dimension of the model. Next, TAPB stacks n layers ($n = 3$) of BERT layers with ROPE to construct more complex and abstract contextual representations. The output $\mathbf{D} \in \mathbb{R}^{B \times L_d \times D_m}$ of the entire drug encoder is a context-sensitive depth representation of the input drug SMILES sequence:

$$D = f_d(E_d) = \text{RoBERT}(E_d)$$

$$\mathbf{D} = f_d(\mathbf{E}_d) = \text{Bert}(\mathbf{E}_d) \quad (31)$$

Target encoder: we **We** employ ESM-2 [11] as the target encoder. ESM-2 is employed as the ESMFold protein feature encoder, replacing multiple sequence alignment (MSA) and structural template parts, with positional embeddings modified to RoPE, and supports longer amino acid sequence encoding. Given that ESMFold is trained on significantly larger datasets and demonstrates superior performance compared to AlphaFold [21], ESM-2 exhibits exceptional capability in extracting 3-D structural information from protein sequences, making it highly suitable for drug-target interaction (DTI) prediction tasks. The input target amino acid sequence A is **sequences X_d in one training batch are** tokenized using the ESM-2 dictionary and tokenizer, and each amino acid is embedded in a high-dimensional space into E_t $\mathbf{E}_t \in \mathbb{R}^{B \times L_t \times D_e}$:

$$E_t = \text{Embedding}(A)$$

$$\mathbf{E}_t = \text{Embedding}(X_t) \quad (32)$$

where L_t denotes the length of X_t and D_e denotes the dimension of the ESM-2 encoded feature. The ESM-2 is then used to extract ESM-2 encoded features $\mathbf{E} \in \mathbb{R}^{B \times L_t \times D_e}$: T , with 70% of amino acid residues randomly dropped, followed by input to $g_p(\cdot)$ for dimensionality reduction:

$$T = f_t(E_t) = g_p(\text{ESM-2}(E_t))$$

$$\mathbf{E} = f_t(\mathbf{E}_t) = \text{ESM-2}(\mathbf{E}_t) \quad (33)$$

In actual **practical** training, target **ESM-2 encoded** features are pre-extracted and saved, significantly reducing memory burden and accelerating the training process.

4.2.2 Aggregator

PAGE 26-27

Following TransformerCPI, we adopt the same aggregator $\mathcal{F}(\cdot)$ with cross attention in TAPB. Each fusion layer of the aggregator inputs drug features \mathbf{D} into the self-attention layer, followed by and layer normalization:

Following TransformerCPI, TAPB adopts the same aggregator $\mathcal{F}(\cdot)$ with Multi-head Cross Attention (MHCA), which is essential for our estimation of counterfactual $P(Y|D, T, c_i)$. First, each fusion layer of the aggregator takes the output of the previous layer $\mathbf{F} \in \mathbb{R}^{B \times L_d \times D_m}$ into the self-attention layer, followed by residual connection and layer normalization:

$$F = \ln(F + SelfAttention(F, F, F))$$

$$\mathbf{F} = \text{LayerNorm}(\mathbf{F} + \text{Self Attention}(\mathbf{F}, \mathbf{F}, \mathbf{F})) \quad (34)$$

The above output F is used as Q , with protein features P as K and V in the cross-attention layer, followed by shortcut and layer normalization:

The output \mathbf{F} is then transformed through a linear layer to obtain $\mathbf{Q} \in \mathbb{R}^{B \times L_d \times D_m}$, while target features \mathbf{T} are projected via two separate linear layers into $\mathbf{K} \in \mathbb{R}^{B \times L_t \times D_m}$ and $\mathbf{V} \in \mathbb{R}^{B \times L_t \times D_m}$, which are fed into the cross-attention layer followed by residual connection and layer normalization:

$$F = \ln(F + crossAttention(F, P, P))$$

$$\mathbf{F} = \text{LayerNorm}(\mathbf{F} + \text{MHCA}(\mathbf{F}, \mathbf{T}, \mathbf{T})) \quad (35)$$

The MHCA is formally defined as:

$$\text{MHCA}(\mathbf{Q}, \mathbf{K}, \mathbf{V}) = g_a(\text{Concat}(\mathbf{head}_1, \dots, \mathbf{head}_i)) \quad (36)$$

$$\mathbf{head}_i = \text{Softmax} \left(\frac{\mathbf{Q}_i \mathbf{K}_i^\top}{\sqrt{d_k}} \right) \mathbf{V}_i \quad (37)$$

where $\mathbf{Q}_i \in \mathbb{R}^{B \times L_d \times D_k}$ corresponds to the i -th head split from \mathbf{Q} , while $\mathbf{K}_i, \mathbf{V}_i \in \mathbb{R}^{B \times L_t \times D_k}$ represent the i -th heads split from \mathbf{K} and \mathbf{V} respectively, $g_a(\cdot)$ is a dimension-preserving linear layer, the MHCA in the aggregator contains H heads, and $D_k = D_m/H$. A feed-forward network (FFN) with residual connection and layer normalization is then applied:

Next, the FFN layer with shortcut and layer normalization is applied:

$$F = \ln(F + \text{FFN}(F))$$

$$F = \text{LayerNorm}(F + \text{FFN}(F)) \quad (38)$$

Three identical aggregator layers are stacked, completing the aggregator architecture for TAPB. Next, we introduce how to suppress "target prior bias" and estimate $P(\mathbf{Y}|\mathbf{D}, \text{do}(\mathbf{T}))$.

Three layers of the above aggregators are stacked, and the final output is aggregated and pooled along the amino acid dimension σ (where σ is the average) and classified using linear layers g_y and *Softmax*:

$$Y = \text{Softmax}(\sigma(g_y(F)))$$

Thus, we obtain the encoders, aggregator, and classifier for prediction using $P(Y|D, P)$. Next, we introduce how to suppress "target prior bias" and achieve $P(Y|D, \text{do}(P))$.

4.2.4 TAPB interventional training

PAGE 28-30

To adjust confounders in Figure 9(a), the backdoor adjustment for SCM in 9a is formulated as:

$$P(\mathbf{Y}|\mathbf{D}, \text{do}(\mathbf{T})) = \sum_{x_t} P(x_t)P(\mathbf{Y}|\mathbf{D}, \mathbf{T}, X_t = x_t) \quad (39)$$

Regrettably, this is infeasible. Unlike previous causal debiasing vision models, e.g. IFSL [22], VCRCNN [23] or IBMIL [24]—where tasks involve specific objects and observable confounders—the learned preferences in DTI models (potentially corresponding to protein families, sub-sequence lengths, or other latent factors) constitute unobservable confounders.

Furthermore, computing $P(\mathbf{Y}|\mathbf{D}, \mathbf{T}, X_t)$ for every X_t presents implementation challenges in deep learning frameworks: Since target sequences X_t remain static in non-augmented datasets, each target feature T corresponds to a single sequence and confounder category. Thus, for each DTI pair (\mathbf{D}, \mathbf{T}) , the model can only predict one $P(\mathbf{Y}|\mathbf{D}, \mathbf{T}, X_t)$ per forward pass. While inserting x_t -corresponding sub-sequences (if observable) during data augmentation could theoretically satisfy exact backdoor adjustment, this approach would increase computational costs—requiring additional forward/backward per augmented (\mathbf{D}, \mathbf{T}) pair—incurring prohibitive resource overhead and architectural

inefficiency. Therefore, to estimate $P(\mathbf{Y}|\mathbf{D}, \mathbf{T}, X_t)$, we adopt approximations balancing causal inference rigor, deep learning feasibility, and training efficiency.

Unlike previous deep learning debiasing methods, e.g. VCRCNN and IBMIL, that employ the Normalized Weighted Geometric Mean (NWGM) [25] to approximate the backdoor adjustment, we implement the exact backdoor adjustment formula, approximating $P(\mathbf{Y}|\mathbf{D}, \mathbf{T}, c_i)$ to compute $P(\mathbf{Y}|\mathbf{D}, do(\mathbf{T}))$. As shown in Figure 9b, our SCM yields the backdoor adjustment:

$$P(\mathbf{Y}|\mathbf{D}, do(\mathbf{T})) = \sum_i P(\mathbf{Y}|\mathbf{D}, \mathbf{T}, \mathbf{c}_i)P(\mathbf{c}_i) \quad (40)$$

where c_i denotes cluster centers. Confounder dictionary \mathbf{C} and $P(c_i)$ are derived as follows: We cluster ESM-2-encoded features \mathbf{E} (preceding \mathbf{T} generation) across the training set to construct a confounder dictionary $\mathbf{C} \in \mathbb{R}^{I \times D_e}$ (Figure 5b). Here, I is the dictionary size (equivalent to the number of heads H in the aggregator’s MHCA). Since ESM-2 was pre-trained on disjoint datasets, DTI label leakage risks are eliminated. The sample proportion per cluster serves as the adjustment weight $P(c_i)$.

The path $X_t \rightarrow \mathbf{C} \rightarrow \mathbf{T}$ is established via our confounder alignment module (CAM) $g_t(\cdot)$. As shown in Figure 5e, CAM fuse cluster centers $\mathbf{c}_i \in \mathbf{C}$ serves as the key \mathbf{K}_i and value \mathbf{V}_i for a distinct attention head within $g_t(\cdot)$, where they interact with the ESM-2 features \mathbf{E} :

$$\mathbf{T}_i = \text{Softmax} \left(\frac{\mathbf{Q}_i \mathbf{K}_i^\top}{\sqrt{d_k}} \right) \mathbf{V}_i \quad (41)$$

$$\mathbf{T} = g(\text{Concat}(\mathbf{T}_0, \mathbf{T}_1, \dots, \mathbf{T}_i) + \mathbf{E}) \quad (42)$$

where \mathbf{Q}_i represents the i -th head of linear projected \mathbf{E} , while \mathbf{K}_i and \mathbf{V}_i correspond to the i -th cluster center \mathbf{c}_i projected through separate linear layers. Here D_k denotes the dimension of i -th head Q_i , and $g(\cdot)$ is a linear layer $\mathbb{R}^{B \times L_d \times D_e} \rightarrow \mathbb{R}^{B \times L_d \times D_m}$. CAM incorporates confounder features \mathbf{c}_i into the \mathbf{E} via multi-head attention. Since data augmentation weakens the original confounding information within the features, this enables counterfactual features to be integrated into \mathbf{T} , establishing the path $X \rightarrow \mathbf{C} \rightarrow \mathbf{T}$. Note that computational costs remain minimal since $I \ll \text{length}(X_t)$.

To approximately estimate all counterfactual $P(\mathbf{Y}|\mathbf{D}, \mathbf{T}, \mathbf{c}_i)$ within per forward, we leverage the independent interaction mechanism of MHCA in the

aggregator. Equation 37 shows that MHCA partitions features along the embedding dimension into h independent heads. Since \mathbf{K} and \mathbf{V} remain invariant across layers, each \mathbf{Q}_i can individually extract c_i -relevant information. Due to this independent interaction mechanism, decomposing MHCA output into I heads yields distinct $\mathbf{F}_{c_i} \in \mathbb{R}^{B \times L_d \times D_k}$ approximations. This enables simultaneous estimation of all $P(\mathbf{Y}|\mathbf{D}, \mathbf{T}, c_i)$ in one forward, balancing theoretical rigor with computational efficiency. Note that H must equal the confounder dictionary size I and satisfy D_m is divisible by H .

The resulting output feature $\mathbf{F} \in \mathbb{R}^{B \times L \times D_m}$, where $D_m = I \times D_k$ is then split along its feature dimension into I segments, each corresponding to one confounder cluster:

$$\mathbf{F} = [\mathbf{F}_{c_0}, \mathbf{F}_{c_1}, \dots, \mathbf{F}_{c_i}] \quad (43)$$

Here, \mathbf{F}_{c_i} represents the feature segment associated with the i -th confounder cluster. Finally, after applying average pooling to each \mathbf{F}_{c_i} , a classification head $g_y(\cdot)$ is used to estimate all $P(\mathbf{Y}|\mathbf{D}, \mathbf{T}, c_i)$:

$$P(\mathbf{Y}|\mathbf{D}, \mathbf{T}, c_i) = \text{Softmax}(g_y(\mathbf{F}_{c_i})) \quad (44)$$

then, we can parameterize Equation 40 via TAPB in the following form:

$$P(\mathbf{Y}|\mathbf{D}, do(\mathbf{T})) = \sum_i P(c_i) \text{Softmax}(g_y(P(\mathbf{Y}|\mathbf{D}, \mathbf{T}, c_i))) \quad (45)$$

Therefore, $P(\mathbf{Y}|\mathbf{D}, do(\mathbf{T}))$ can be computed via Equation 45 and integrated into deep learning training to adjust for confounders, achieving a practical balance between theoretical rigor and computational cost. The binary classification loss for TAPB can be denoted as follow:

$$\mathcal{L} = - \sum_{i=1} y_i \log(\hat{y}_i) \quad (46)$$

where y_i is the label, and \hat{y}_i is the predicted probability (i.e., from *Softmax*) for class i . Furthermore, we follow the masked language modeling in BERT to enhance the semantic features extracted by the drug encoder $f_d(\cdot)$. Specifically, 15% of all tokens in each sequence are randomly selected, with an 80% probability of being replaced by a [mask] token, a 10% probability of remaining unchanged, and a 10% probability of being replaced by a random token. The

masked tokens are then predicted using $f_d(\cdot)$ and $g_m(\cdot)$, and the loss L_{mlm} is calculated by:

$$\mathcal{L}_{\text{mlm}} = - \sum_{i=1}^N \sum_{j=1}^{L_d} m_{ij} \log P(w_{ij} | \mathbf{H}_i) \quad (47)$$

where N is the number of samples, m_{ij} is a binary mask (1 if position j is masked, 0 otherwise), and $P(w_{ij} | \mathbf{H}_i)$ denotes the predicted probability for the token w_{ij} at position j of the i -th sample, with $\mathbf{H}_i \in \mathbb{R}^{D_m}$ representing the contextual representations generated by the $f_d(\cdot)$. The total loss for the TAPB can be denoted as follow:

$$L = L_b + L_{\text{mlm}} \quad (48)$$

The amino acid randomization and our TAPB training framework constitute a universal design applicable to other DTI models, provided that the dataset exhibits "target prior bias" and a pre-trained target encoder is used.

4.2.4 Biased training (stage 1)

As shown in Figure 5a, we first train TAPB using $P(Y|D, P)$ to capture the "target prior bias" in the sequence dataset. We use $f_d(\cdot)$ to extract drug feature D and load target feature T pre-extracted through ESM-2. To suppress "target prior bias" and reduce computational costs, for each T , we apply target random mask to randomly discard 70% of amino acid residues, followed by input to $g_p(\cdot)$ for dimensionality reduction. The aggregator $\mathcal{F}(\cdot)$ and pooling $\sigma(\cdot)$ are then used to obtain the fusion feature F , and a classifier $g_y(\cdot)$ outputs the probability of DTI combinations through *Softmax*. The loss for training plain TAPB can be denoted as follow:

$$L_b = -(y \log(\hat{y}) + (1 - y) \log(1 - \hat{y}))$$

Furthermore, to enhance the semantic features captured by the drug encoder $f_d(\cdot)$, we follow the masked language modeling in BERT. Specifically, 15% of all tokens in each sequence are randomly selected, with an 80% probability of being replaced by a [mask] token, a 10% probability of remaining unchanged, and a 10% probability of being replaced by a random token. The masked tokens are then predicted using $f_d(\cdot)$ and $g_m(\cdot)$, and the loss L_{mlm} is calculated through the cross-entropy function. The total loss function for the stage 1 training can be denoted as follow:

$$L = L_b + L_{mtm}$$

4.2.5 Intervential training (stage 2)

From causal lens, "target prior bias" is a confounder hidden in X_t . In addition to using target random mask, we can eliminate the effect of "target prior bias" via backdoor adjustment. As shown in Figure 9, backdoor adjustment severs the path $T \leftarrow X_t \rightarrow Y$, thereby inhibiting "target prior bias." We approximate the backdoor adjustment formula $P(Y|D, do(T))$ for interventional training. As shown in Figure 5b, we freeze the weights of $f_d(\cdot)$ and $f_t(\cdot)$ trained in the first stage. Based on the causal analysis in Section ??, we adjust the backdoor for DTI as follows:

$$P(Y|D, do(T)) = \sum_i P(c_i) P(Y|D, T, c_i)$$

Where c_i represents the cluster centers' tendencies. Furthermore, we can parameterize $P(Y|D, T, c_i)$ using the DTI model in the following form:

$$P(Y|D, do(T)) = \sum_i P(c_i) \text{Softmax}(g_y(\sigma(\mathcal{F}(D, T, c_i))))'$$

Where $g_y(\cdot)'$ is a new initialized linear classification head, σ is average, and \mathcal{F} is the aggregator. Following [22] [23], we further apply Normalized Weighted Geometric Mean (NWGM) [25] to move the outer sum into the *Softmax*:

$$P(Y|D, do(T)) \approx \text{Softmax}(\sum_i P(c_i) g_y(\sigma(\mathcal{F}(D, T, c_i))))'$$

Then we can move the summation to the inner of the linear layer $g_y(\cdot)'$:

$$P(Y|D, do(P)) \approx \text{Softmax}(g_y(\sum_i P(c_i) \sigma(\mathcal{F}(D, T, c_i))))'$$

Following [24], we cluster all features $\sigma(\mathcal{F}(D, P, c_i))$ in the training set using K-means, obtaining I cluster centers ($I = 8$) to approximate the label tendency t_i of each target. The cluster centers form a confounder dictionary V with shape $d \times I$, where d is the dimension ($d = 128$). We define $P(c_i)$ as:

$$A = [P(c_1), \dots, P(c_i)] = \text{Softmax}\left(\frac{\sigma(\mathcal{F}(D, T, c_i))^T V_i}{\sqrt{d}}\right)$$

Where A represents the attention matrix. We estimated the similarity between each $\sigma(\mathcal{F}(D, T, c_i))$ and all V_i using attention, and obtained the probability $P(c_i)$ using *Softmax*. Finally, we can rewrite the backdoor adjustment as follows:

$$P(Y|D, do(T)) \approx \text{Softmax}(g_y(\sum_i A_i V_i)')$$

Therefore, backdoor adjustment can be approximated by the equation ?? and integrated into deep learning training to suppress the "target prior bias." From the causal lens, Figure 9d shows the SCM of TAPB stage 2, which differs significantly from the original Figure 9a. The total loss function of the interventional training stage can be denoted as follows:

$$L = L_{\bar{t}}$$

The backdoor adjustment module is a universal design applicable to other DTI models, provided the dataset exhibits "target prior bias," the model uses *Softmax* for classification, and the classifier consists of only one linear layer.

References

- [1] Rohit Singh, Samuel Sledzieski, Bryan Bryson, Lenore Cowen, and Bonnie Berger. Contrastive learning in protein language space predicts interactions between drugs and protein targets. *Proceedings of the National Academy of Sciences*, 120(24):e2220778120, 2023.
- [2] Peizhen Bai, Filip Miljković, Bino John, and Haiping Lu. Interpretable bilinear attention network with domain adaptation improves drug–target prediction. *Nature Machine Intelligence*, 5(2):126–136, 2023.
- [3] Kexin Huang, Cao Xiao, Lucas M Glass, and Jimeng Sun. Moltrans: molecular interaction transformer for drug–target interaction prediction. *Bioinformatics*, 37(6):830–836, 2021.
- [4] Lifan Chen, Xiaoqin Tan, Dingyan Wang, Feisheng Zhong, Xiaohong Liu, Tianbiao Yang, Xiaomin Luo, Kaixian Chen, Hualiang Jiang, and Mingyue Zheng. TransformerCPI: improving compound–protein interaction prediction by sequence-based deep learning with self-attention mechanism and label reversal experiments. *Bioinformatics*, 36(16):4406–4414, 2020.
- [5] Huan Yee Koh, Anh TN Nguyen, Shirui Pan, Lauren T May, and Geoffrey I Webb. Physicochemical graph neural network for learning protein–ligand interaction fingerprints from sequence data. *Nature Machine Intelligence*, 6(6):673–687, 2024.

- [6] Zhousan Xie, Shikui Tu, and Lei Xu. Multilevel attention network with semi-supervised domain adaptation for drug-target prediction. In *Proceedings of the AAAI Conference on Artificial Intelligence*, volume 38, pages 329–337, 2024.
- [7] Yong Liu, Min Wu, Chunyan Miao, Peilin Zhao, and Xiao-Li Li. Neighborhood regularized logistic matrix factorization for drug-target interaction prediction. *PLoS computational biology*, 12(2):e1004760, 2016.
- [8] Jacob Devlin Ming-Wei Chang Kenton and Lee Kristina Toutanova. Bert: Pre-training of deep bidirectional transformers for language understanding. In *Proceedings of naacL-HLT*, volume 1, page 2. Minneapolis, Minnesota, 2019.
- [9] A Vaswani. Attention is all you need. *Advances in Neural Information Processing Systems*, 2017.
- [10] Jerret Ross, Brian Belgodere, Vijil Chenthamarakshan, Inkit Padhi, Youssef Mroueh, and Payel Das. Large-scale chemical language representations capture molecular structure and properties. *Nature Machine Intelligence*, 4(12):1256–1264, 2022.
- [11] Zeming Lin, Halil Akin, Roshan Rao, Brian Hie, Zhongkai Zhu, Wenting Lu, Nikita Smetanin, Allan dos Santos Costa, Maryam Fazel-Zarandi, Tom Sercu, Sal Candido, et al. Language models of protein sequences at the scale of evolution enable accurate structure prediction. *bioRxiv*, 2022.
- [12] Yulei Niu, Kaihua Tang, Hanwang Zhang, Zhiwu Lu, Xian-Sheng Hua, and Ji-Rong Wen. Counterfactual vqa: A cause-effect look at language bias. In *Proceedings of the IEEE/CVF conference on computer vision and pattern recognition*, pages 12700–12710, 2021.
- [13] David Arthur and Sergei Vassilvitskii. K-means++: The advantages of careful seeding. In *Proceedings of the Eighteenth Annual ACM-SIAM Symposium on Discrete Algorithms, SODA 2007, New Orleans, Louisiana, USA, January 7-9, 2007*, 2007.
- [14] Yann LeCun, Léon Bottou, Yoshua Bengio, and Patrick Haffner. Gradient-based learning applied to document recognition. *Proceedings of the IEEE*, 86(11):2278–2324, 1998.
- [15] Kaiming He, Xiangyu Zhang, Shaoqing Ren, and Jian Sun. Deep residual learning for image recognition. In *Proceedings of the IEEE conference on computer vision and pattern recognition*, pages 770–778, 2016.

- [16] Thomas N Kipf and Max Welling. Semi-supervised classification with graph convolutional networks. In *International Conference on Learning Representations*, 2022.
- [17] S Hochreiter. Long short-term memory. *Neural Computation MIT-Press*, 1997.
- [18] Ingoo Lee, Jongsoo Keum, and Hojung Nam. Deepconv-dti: Prediction of drug-target interactions via deep learning with convolution on protein sequences. *PLoS computational biology*, 15(6):e1007129, 2019.
- [19] Jin-Hwa Kim, Jaehyun Jun, and Byoung-Tak Zhang. Bilinear attention networks. *Advances in neural information processing systems*, 31, 2018.
- [20] Jianlin Su, Murtadha Ahmed, Yu Lu, Shengfeng Pan, Wen Bo, and Yunfeng Liu. Roformer: Enhanced transformer with rotary position embedding. *Neurocomputing*, 568:127063, 2024.
- [21] John Jumper, Richard Evans, Alexander Pritzel, Tim Green, Michael Figurnov, Olaf Ronneberger, Kathryn Tunyasuvunakool, Russ Bates, Augustin Žídek, Anna Potapenko, et al. Highly accurate protein structure prediction with alphafold. *nature*, 596(7873):583–589, 2021.
- [22] Zhongqi Yue, Hanwang Zhang, Qianru Sun, and Xian-Sheng Hua. Interventional few-shot learning. *Advances in neural information processing systems*, 33:2734–2746, 2020.
- [23] Tan Wang, Jianqiang Huang, Hanwang Zhang, and Qianru Sun. Visual commonsense r-cnn. In *Proceedings of the IEEE/CVF conference on computer vision and pattern recognition*, pages 10760–10770, 2020.
- [24] Tiancheng Lin, Zhimiao Yu, Hongyu Hu, Yi Xu, and Chang-Wen Chen. Interventional bag multi-instance learning on whole-slide pathological images. In *Proceedings of the IEEE/CVF Conference on Computer Vision and Pattern Recognition*, pages 19830–19839, 2023.
- [25] Kelvin Xu, Jimmy Lei Ba, Ryan Kiros, Kyunghyun Cho, Aaron Courville, Ruslan Salakhutdinov, Richard S Zemel, and Yoshua Bengio. Show, attend and tell: neural image caption generation with visual attention. In *Proceedings of the 32nd International Conference on International Conference on Machine Learning-Volume 37*, pages 2048–2057, 2015.

Response letter for “TAPB: An Interventional Debiasing Framework for Alleviating Target Prior Bias in Drug-Target Interaction Prediction”

Gaoming Lin ^{†1}, Xin Zhang ^{†2,3}, Zhonghao Ren⁴, Quan Zou², Prayag Tiwari^{*5}, Changjun Zhou ^{*1}, and Yijie Ding ^{*2}

¹School of Computer Science and Technology, Zhejiang Normal University, Jinhua, 321000, Zhejiang, China

²Yangtze Delta Region Institute (Quzhou), University of Electronic Science and Technology of China, Quzhou, 324003, Zhejiang, China

³The Quzhou Affiliated Hospital of Wenzhou Medical University, Quzhou People’s Hospital, Quzhou, 324000, Zhejiang, China

⁴College of Computer Science and Electronic Engineering, Hunan University, Changsha, 410082, Hunan, China

⁵School of Information Technology, Halmstad University, Halmstad, 301 18, Sweden

To reviewers

Dear reviewers:

Our sincere thanks go out to the reviewers who reviewed our manuscript and provided constructive comments that significantly improved it. We have made detailed revisions in a point-by-point response to comments made by reviewers.

We first quote the comments and then reply with how we have revised the manuscript to accommodate the changes. We use **black sans serif font** for our responses and **red box** for comments. Revisions in the manuscript are indicated within a **blue box**, with modified text shown in **red font** and deleted content

marked by "~~strikethrough formatting~~." For further results, please refer to our SI. The main changes are summarized below:

- We have refined our causal inference terminology for precision, including clarifying the equivalence and theoretical exactness of adjustment sets under our SCM, replacing “approximate” with “compute” for backdoor adjustment, substituting all instances of “counterfactual” with “confounder-conditioned probability”, and incorporating optimal adjustment set references in the Discussion.
- We improved symbol and visualization consistency by standardizing variable notation, converting Fig. 3 to bar charts for clarity, and updating legend labels from “SMILES” to “drug”.
- We have extended our ablation study by incorporating two additional datasets—the Davis dataset and the BindingDB cross-domain dataset—and redesigned the experimental setup to include 7 ablation variants. This expansion allows a more comprehensive assessment of each module’s contribution. The ablation analysis section has been thoroughly rewritten to reflect these improvements.
- Results not related to ablation experiments have been moved to the SI, with appropriate citations added in the manuscript. Additionally, since hyperparameter tuning was conducted on the BioSNAP cross-domain dataset, in the cross-domain comparison section, all conclusions derived from this dataset have been removed. Only experimental results are retained for reference.
- We have comprehensively addressed misreferenced tables and figures, carefully standardized model nomenclature and symbolic notation, removed redundant variables, and ensured full adherence to academic writing conventions. Additionally, we have streamlined the description of the datasets in our manuscript, reducing the total number from six to four.
- We have thoroughly reviewed all citations of the SI to ensure accurate and complete referencing of every figure, table, and algorithm within the manuscript.
- We have released a streamlined version of our code on GitHub, allowing our results to be reproduced directly with a single line of Python command.

We have also enriched the code with additional comments to improve readability and facilitate understanding. To ensure accessibility for readers in regions where Hugging Face may be unavailable, we have also uploaded the ESM-2 and Molformer weights required to reproduce our results to OneDrive.

We sincerely appreciate the reviewers for giving us the opportunity to revise our manuscript. We have thoroughly revised the paper and hope that the meaningful findings and innovative aspects of our methodology are now more clearly highlighted. We believe that our revisions have addressed the comments accordingly and hope the manuscript now meets your expectations.

Best regards,

Gaoming Lin, Xin Zhang, Zhonghao Ren, Quan Zou, Prayag Tiwari, Changjun Zhou, Yijie Ding

Aug 23, 2025

Contents

1	Response to Reviewer 1	4
2	Response to Reviewer 2	17
3	Response to Reviewer 3	26

1 Response to Reviewer 1

Comment: (Remarks to the Author)

The authors have performed an extensive revision and have addressed all my comments in detail. I was pleased to read that they have made their method more rigorous from the causal inference perspective, and that this has led to an overall improved performance across their benchmarks. Only a few minor comments remain:

Response:

We sincerely appreciate the reviewer's exceptional expertise in causal inference and patience with our work. Your comments have significantly strengthened the methodological rigor and clarity of our manuscript. Following your guidance, we revised terminology and notation throughout the manuscript, and we have implemented the following key revisions:

1. Adjustment Set Terminology Precision: We corrected our manuscript to reflect the theoretical equivalence of adjustment sets under our SCM. We explicitly stated equivalence between adjusting for C and X_t , replaced "approximate" with "compute" for backdoor adjustment, and incorporated optimal adjustment set literature in the Discussion.

2. Counterfactual Terminology Correction: We replaced all instances of "counterfactual" with "confounder-conditioned probability", aligning with standard causal inference usage.

3. Symbol and Visualization Consistency: We standardized the notation of all variables across equations and figures, converted Fig. 3 to bar charts for clearer frequency representation, and updated legend labels from "SMILES" to "drug" for terminological consistency.

All modifications are highlighted. Again, we sincerely appreciate the reviewer's time, expert guidance, and rigorous evaluation of our causal methodology. We hope that these improvements align with the expectations

Comments:

Comment: 1.1 (* Response A.2)

While the authors were previously too optimistic in their causal inference language, they now err a little bit in the other direction. For instance, they write that their adjustment for C achieves a balance between theoretical rigor and computational cost. However, if the data is truly generated by the causal

graph in Fig. 9b, then adjusting for C or adjusting for the true confounder X_t is theoretically equivalent and both result in the correct causal effect identification. Of course, different adjustment sets may be theoretically equivalent but differ in their finite-sample efficiency, variance, etc. This is something that would be difficult to analyze in the present context, but the authors may well refer to their current backdoor formula as theoretically exact, while perhaps referring to some of the literature around optimal adjustment sets in their discussion.

Response 1.1.:

We sincerely appreciate the reviewer's professional comments and instructive guidance. In our previous revision, we mischaracterized our adjustment approach—as theoretically equivalent to adjusting for the true confounder X_t —as merely "balancing theoretical rigor with computational cost." Following your guidance, we have revised the relevant sections to explicitly clarify that adjusting for C is theoretically equivalent to adjusting for X_t and thus identifies the correct causal effect, and confirm that the current backdoor formula is theoretically exact. Furthermore, we have removed the inappropriate use of the term 'approximate', which may have led readers to mistakenly think that our method is not theoretically exact, and we have expanded the discussion to incorporate references to the literature on optimal adjustment sets, acknowledging potential differences in finite-sample efficiency while emphasizing the theoretical equivalence. All these revisions are comprehensively summarized in Revision 1.1. Additionally, as requested by Reviewer 3, variables have been bolded accordingly. Tensors representing features, such as \mathbf{D} and \mathbf{T} , have been bolded, whereas Y remains in regular font since it denotes a scalar.

Revision: 1.1.

Abstract

PAGE 1-2

Drug Target Interaction (DTI) prediction is vital for drug repurposing. Previous studies trained on BioSNAP and BindingDB datasets have incorrectly identified "drug bias" as the cause of biased predictions, while our work reveals "target prior bias" as the predominant issue. This bias stems from the "prior tendency," characterized by the imbalanced label distribution of targets in the training data. From causal lens, "~~target prior bias~~ target **"prior tendency"**" is a confounder, causing models trained with $P(Y|\mathbf{D}, \mathbf{T})$ ~~$P(\mathbf{Y}|\mathbf{D}, \mathbf{T})$~~ to learn spurious associations between targets and labels rather than genuine interaction mechanisms. In this study, we introduce alleviating **Target Prior Bias** in Drug-Target Interaction Prediction (TAPB), a novel debiasing frame-

work that employs amino acid randomization, confounder alignment module (CAM), and interventional training to approximate $P(Y|D, do(T))$ via backdoor adjustment, thereby addressing this bias. TAPB outperforms existing methods on both in-domain and cross-domain datasets, offering enhanced generalization and providing interpretable insights into DTIs. TAPB achieves superior performance over existing approaches, not only in generalization but also in providing interpretable insights for DTIs.

1 Introduction

PAGE 3-4

Figure 1: Overview of DTI Bias Analysis and our framework. (a) Dataset Construction of BioSNAP and BindingDB. (b) An example sketch of target biased training sets shows that certain targets (T_1 and T_2) exhibit positive and negative "prior tendencies", respectively, while the bias in drugs is less pronounced. (c) Outline of the biased training process, where models learn from datasets containing an inherent "prior tendency." (d) "Target bias" tests demonstrate that pairing a randomly generated feature R with target T_1 yields higher positive scores compared to negative ones, with the opposite observed for T_2 . (e) "Drug bias" tests show relatively balanced scores when a drug D is paired with a randomly generated feature R, indicating lesser influence on predictions. (f) Our TAPB interventional training, combining amino acid randomization, confounder alignment module (CAM), and multi-head cross-attention (MHCA) to approximate $P(Y|D, do(T))$ via backdoor adjustment, thereby adjusting confounders.

As shown in Fig. 1f, TAPB employs amino acids randomization and

confounder alignment module to approximate $P(Y|\mathbf{D}, do(\mathbf{T}))$ via **theoretically exact** backdoor adjustment, where $do(\cdot)$ denotes the intervention that sets the variable to a specific value, blocking all incoming paths to the variable. The backdoor adjustment computes $P(Y|\mathbf{D}, do(\mathbf{T}))$ via observational data without performing actual interventions.

2.3 TAPB framework

PAGE 11

A shared classifier $g_y(\cdot)$ then computes $P(Y|\mathbf{D}, \mathbf{T}, \mathbf{c}_i)$ for all $\mathbf{F}_{\mathbf{c}_i}$, **enabling the approximation of $P(Y|\mathbf{D}, do(\mathbf{T}))$. enabling the computation of $P(Y|\mathbf{D}, do(\mathbf{T}))$ via backdoor adjustment under our SCM.**

PAGE 12

We did not employ domain adaptation techniques, e.g. CDAN [1], and achieved better results on the cross-domain datasets, indicating the strong generalization of TAPB. Note that TAPB is a debiasing framework, and replacing encoders or aggregators can further enhance performance. The components of TAPB are generic, with the **approximate computation of** backdoor adjustment requiring the satisfaction of certain assumptions. The pseudocode of our method is provided in Supplementary Algorithm 1.

3 Discussion

PAGE 22

While our adjustment for \mathbf{C} satisfies the backdoor criterion and is theoretically exact for causal effect identification in the SCM of Figure 9b, where amino acid randomization effectively disrupts target patterns, different valid adjustment sets may vary significantly in their finite-sample performance. As demonstrated by [2], in SCM with hidden variables, multiple adjustment sets can be theoretically equivalent for causal identification but exhibit different asymptotic variances. For observable adjustment sets, there exist optimal minimal adjustment sets that yield the smallest asymptotic variance among all minimal valid sets [3]. Our choice of adjusting for \mathbf{C} balances statistical robustness and computational efficiency, acknowledging that while alternative valid adjustment sets might offer improved statistical efficiency in certain scenarios, they could come with higher computational costs or data requirements. Future work could explore optimal adjustment set selection specifically for DTI predictions.

4.1 Analysis DTI through causal inference

PAGE 24

Figure 9: SCM of conventional DTI models and TAPB. (a) SCM of the conventional DTI models exhibiting "target prior bias." (b) SCM of our stage 1 biased training without amino acids randomization, adding C into $X_t \rightarrow T$. (c) SCM after applying amino acid randomization, which weakens the confounding path $T \leftarrow C \leftarrow X_t \rightarrow Y$. Note that the dashed lines, which are not rigorous representations of causal relationships within SCM, indicate a weakened connection rather than a complete removal of this path. (d) SCM for $P(Y|D, do(T))$ where $do(T)$ blocks the path $X_t \rightarrow C \rightarrow T$. We compute $P(Y|D, do(T))$ via backdoor adjustment without actual intervention.

4.2.4 TAPB interventional training

PAGE 26-27

Therefore, to estimate $P(Y|D, T, X_t)$, we adopt approximations balancing causal inference rigor, deep learning feasibility, and training efficiency.

Since X_t is unobservable, direct estimation of $P(Y|D, T, X_t)$ is infeasible. However, under the causal assumptions of Fig. 9b, the confounder dictionary C serves as a valid adjustment set that is theoretically equivalent to adjusting for X_t .

PAGE 27

Unlike previous deep learning debiasing methods, e.g. VCRCNN and IB-MIL, that employ the Normalized Weighted Geometric Mean (NWGM) [4] to approximate the backdoor adjustment, we implement the **theoretically** exact backdoor adjustment formula, approximating $P(Y|D, T, c_i)$ to compute $P(Y|D, do(T))$. As shown in Fig. 9b, our SCM yields the backdoor adjustment: by estimating $P(Y|D, T, c_i)$ for all $c_i \in C$ to compute $P(Y|D, do(T))$, while maintaining computational efficiency in deep learning frameworks. Our SCM yields the backdoor adjustment:

$$P(Y|D, do(T)) = \sum_i P(Y|D, T, c_i)P(c_i) \quad (17)$$

PAGE 26-27

This enables simultaneous estimation of all $P(Y|\mathbf{D}, \mathbf{T}, c_i)$ in one forward. ~~balancing theoretical rigor with computational efficiency.~~ Note that H must equal the confounder dictionary size I and satisfy that D_m is divisible by H .

PAGE 28

then, we can parameterize Equation 17 via TAPB in the following form:

$$P(Y|\mathbf{D}, do(\mathbf{T})) = \sum_i P(c_i) \text{Softmax}(g_y(P(Y|\mathbf{D}, \mathbf{T}, c_i))) \quad (22)$$

Therefore, $P(Y|\mathbf{D}, do(\mathbf{T}))$ can be computed via Equation 22 and integrated into deep learning training to adjust for confounders, ~~achieving a practical balance between theoretical rigor and computational cost~~ **Under our SCM in Fig. 9b, this implementation provides a theoretically exact estimation of the causal effect, while maintaining computational efficiency in deep learning frameworks.** The binary classification loss \mathcal{L}_b for TAPB can be denoted as follows:

$$\mathcal{L}_b = - \sum_{i=1} y_i \log(\hat{y}_i) \quad (23)$$

Comment: 1.2 (* Response A.2)

The authors refer to the conditional distributions $P(Y|D, T, c_i)$ for a given value c_i of the clustering variable C as a "counterfactual". Unless I have misunderstood what is being estimated here, I don't think this is a correct use of the term counterfactual. A counterfactual in causal inference usually refers to a statement about a single unit (in this case drug-target interaction) and asks what would have been true under different circumstances. For instance, if the authors' method predicts an interaction between a specific drug and target, they could ask the counterfactual question if the interaction would still have been predicted if the drug or target would have taken a specific value different from its true value ("counter to the facts") for one of its features.

Response 1.2.:

We sincerely appreciate the reviewer's professional guidance and patient explanations. We acknowledge that we incorrectly used the term "counterfactual" in our previous revision. Our initial usage of "counterfactual" stemmed from the

observation that only one probability $P(Y|D, T, c_i)$ exists in reality; however, as you correctly pointed out, the genuine counterfactual in DTI prediction should involve binding pocket-related counterfactual scenarios. Following your instructive guidance, we have now replaced all instances of "counterfactual" probability with "confounder-conditioned" probability throughout the manuscript. This change better aligns with the causal inference framework we employ, where c_i represents confounder clusters rather than counterfactual interventions. We have carefully revised all occurrences of this term. All these revisions are comprehensively summarized in Revision 1.2.

Revision: 1.2.

2.3 TAPB framework

PAGE 11

Figure 5: Architecture of the TAPB framework. (a) **TAPB Interventional Training:** The drug encoder BERT $f_d(\cdot)$ generates drug features \mathbf{D} from SMILES. ESM-2 Pre-extracted target features \mathbf{E} undergo amino acid randomization and are then processed by the CAM g_t . all cluster centers $c_i \in \mathbf{C}$ act as keys/values in CAM g_t with \mathbf{E} . Fused features \mathbf{F} are partitioned into I segments \mathbf{F}_{c_i} , each globally pooled and fed to classifier g'_y for estimating counterfactual **confounder-conditioned probabilities** $P(Y|\mathbf{D}, \mathbf{T}, c_i)$. Finally, $P(Y|\mathbf{D}, \text{do}(\mathbf{T}))$ is computed via backdoor adjustment. (b) **Target Confounder Dictionary \mathbf{C} :** Obtained via K-means clustering on ESM-2 target features from training sets. (c) **Amino Acid Randomization:** 1. Random deletion of 70% residue features; 2. Mutation of remaining residues to random features from the amino acid dictionary. (d) **Confounder Alignment Module (CAM, $g_t(\cdot)$):** Attention-weighted summation fuses c_i with target features, followed by dimensionality reduction and residual connection. Maintains explicit path $X_t \rightarrow \mathbf{C} \rightarrow \mathbf{T}$ across training.

2.3 TAPB framework

PAGE 11

The confounder alignment module $g_t(\cdot)$, illustrated in Fig. 5d, operates during interventional training. It processes each confounder cluster center c_i to generate counterfactual **confounder-conditioned** representations \mathbf{T}_{c_i} , and partitioned fused features \mathbf{F}_{c_i} . A shared classifier $g_y(\cdot)$ then computes $P(Y|\mathbf{D}, \mathbf{T}, c_i)$ for all \mathbf{F}_{c_i} , enabling the approximation of $P(\mathbf{Y}|\mathbf{D}, \text{do}(\mathbf{T}))$. **enabling the computation of $P(Y|\mathbf{D}, \text{do}(\mathbf{T}))$ via backdoor adjustment under our SCM.**

4.2.1 Encoders

PAGE 25

Following TransformerCPI, TAPB adopts the same aggregator $\mathcal{F}(\cdot)$ with Multi-head Cross Attention (MHCA), which is essential for our estimation of counterfactual **confounder-conditioned probabilities** $P(Y|\mathbf{D}, \mathbf{T}, c_i)$. First, each fusion layer of the aggregator takes the output of the previous layer $\mathbf{F} \in \mathbb{R}^{B \times L_d \times D_m}$ into the self-attention layer, followed by residual connection and layer normalization:

$$\mathbf{F} = \text{LayerNorm}(\mathbf{F} + \text{Self Attention}(\mathbf{F}, \mathbf{F}, \mathbf{F})) \quad (11)$$

4.2.4 TAPB interventional training

PAGE 27-28

Since data augmentation weakens **amino acid randomization disrupts** the original confounding information **and pattern** within the target features, this enables counterfactual **confounder-conditioned** features to be integrated into \mathbf{T} , establishing the path $X \rightarrow \mathbf{C} \rightarrow \mathbf{T}$. Note that computational costs remain minimal

since $I \ll \text{length}(X_t)$.

To approximately estimate all counterfactual **confounder-conditioned probabilities** $P(Y|\mathbf{D}, \mathbf{T}, \mathbf{c}_i)$ within per forward, we leverage the independent interaction mechanism of MHCA in the aggregator.

Comment: 1.3 (* Response 1.3)

The authors should carefully review the different symbols used in their revised text. They now use z_i where before they wrote t_i , but their figure axis label still says t_i . Likewise they still use T before eq. (4) which should probably be a Z . In eq. (5), why is a new symbol g_i introduced while this should be equal/related to the z_i ? Fig. 3(a)-(d) - wouldn't a bar chart of the frequencies be more informative than a line chart; the legend label "SMILES" should be consistent with the caption which refers to "drugs".

Response 1.3.:

We sincerely appreciate the reviewer for the meticulous attention to detail and the highly constructive comments. We apologize for the inconsistencies in the previous version due to insufficient scrutiny. In this revision, we have ensured consistent usage of the variable z_i throughout the text (replacing t_i in figure axis labels and T before eq. 4 with Z where applicable), and in eq. 5, we have replaced the symbol g_i with z_i to maintain coherence. Additionally, as suggested, we have updated Fig. 3 to a bar chart for improved clarity and adjusted the legend label from "SMILES" to "drug" to align with the caption, thereby guaranteeing consistency between the text and figures for enhanced reader comprehension. Furthermore, we analyzed the two supplemented datasets, i.e. Davis and Human, and presented the results in Supplementary Section 3. All these revisions are comprehensively summarized in Revision 1.3.

Revision: 1.3.

2.2.2 Prior tendency causes biased predictions

PAGE 7-9

Figure 3: Statistical visualization of "prior tendency" for drugs and targets in the BindingDB and BioSNAP datasets. (a) "Prior tendency" frequency distribution of z_i and p-value for drugs and targets in the BindingDB in-domain training set. (b) "Prior tendency" frequency distribution of z_i and p-value for drugs and targets in the BindingDB cross-domain training set. (c) "Prior tendency" frequency distribution of z_i and p-value for drugs and targets in the BioSNAP in-domain training set. (d) Label frequency distribution of $t_i Z$ for drugs and targets in the BioSNAP cross-domain training set. "Prior tendency" frequency distribution of z_i and p-value for drugs and targets in the BioSNAP cross-domain training set. (e) Quantification of the overall "prior tendency" for labels associated with drugs and targets across different datasets.

We assume that the biased predictions are caused by "prior tendency", which

we formally define as systematic label distribution biases inherent to individual drugs or targets in the DTI sequence dataset. Specifically, this refers to statistically significant deviations in the positive/negative sample ratios observed across different drugs (drug-level prior) or targets (target-level prior), which create spurious correlations that models can exploit to minimize loss without learning true interaction mechanisms. To quantify "prior tendency" across different DTI datasets, we devised the following label test:

$$z_i = \frac{\sum_j y_{ij}}{n_i} \quad (2)$$

$$Z = \sum_i |z_i - 0.5| + 0.5 \quad (3)$$

where y_{ij} denotes the j -th label of the i -th sequence, n_i denotes the total number of elements in the i -th sequence, n_i denotes the occurrence count of the i -th sequence, z_i denotes each sequence's "prior tendency" which is rounded to one decimal place for better visualization, and Z denotes the overall "prior tendency" across all sequences in the dataset, ranging from 0.5 to 1.0.

Furthermore, beyond the heuristic score Z , we designed a rigorous permutation test grounded in a null model where interaction labels Y are independent of target-specific effects. Under this null hypothesis, each drug-target pair's label follows a Bernoulli distribution parameterized by the global positive interaction proportion:

$$g = \frac{\sum Y}{N} \quad (4)$$

where N denotes the total number of drug-target pairs, representing random label assignment without target-specific biases. To evaluate statistical significance, we employed a weighted sum of squared deviations as our test statistic:

$$T = \sum_{i=1}^M n_i (g_i - g)^2$$

$$T = \sum_{i=1}^M n_i (z_i - g)^2 \quad (5)$$

where M is the total number of unique sequences, i.e. total number of unique drugs or targets in the dataset. n_i is the occurrence count of target

i , and g_i is its observed positive interaction proportion. The n_i weighting ensures proportional contribution by sample size while maintaining sensitivity for sparse targets. Our permutation procedure preserves drug-protein pair structures while randomly reshuffling labels across all pairs for $B = 1000$ iterations, with p-value computed as:

$$\text{p-value} = \frac{1 + \sum_{b=1}^B \mathbf{1}(T_b \geq T_{\text{obs}})}{1 + B} \quad (6)$$

where T_b is the permuted statistic and T_{obs} the observed value. This non-parametric approach maintains DTI data structures through fixed pairings with permuted labels, ensures small-sample robustness by avoiding asymptotic assumptions, and naturally adapts to class imbalance.

Supplementary Section 3. Extend data statistics

PAGE 12

We quantified the "prior tendency" and total overall "prior tendency" in the Davis and human cold-split datasets. As illustrated in Supplementary Fig. 3a, b and c, the Davis dataset exhibits a certain "target prior bias", while a more pronounced "drug prior bias" is observed in the human cold-split data. **Statistical analysis confirmed significant deviations for both biases ($P_t < 0.001$, $P_d < 0.001$).**

Supplementary Figure 3: Statistics of Davis and Human cold-split. (a) "Prior tendency" frequency distribution of z_i for drugs and targets in the Davis training set. (b) "Prior tendency" frequency distribution of z_i for drugs and targets in the Human cold-split training set. (c) Overall "prior tendency" for labels associated with drugs and targets across Davis and Human cold-split datasets.

References

- [1] Mingsheng Long, Zhangjie Cao, Jianmin Wang, and Michael I Jordan. Conditional adversarial domain adaptation. *Advances in neural information processing systems*, 31, 2018.
- [2] Jakob Runge. Necessary and sufficient graphical conditions for optimal adjustment sets in causal graphical models with hidden variables. *Advances in Neural Information Processing Systems*, 34:15762–15773, 2021.
- [3] Ezequiel Smucler, Facundo Sapienza, and Andrea Rotnitzky. Efficient adjustment sets in causal graphical models with hidden variables. *Biometrika*, 109(1):49–65, 2022.
- [4] Kelvin Xu, Jimmy Lei Ba, Ryan Kiros, Kyunghyun Cho, Aaron Courville, Ruslan Salakhutdinov, Richard S Zemel, and Yoshua Bengio. Show, attend and tell: neural image caption generation with visual attention. In *Proceedings of the 32nd International Conference on International Conference on Machine Learning-Volume 37*, pages 2048–2057, 2015.

2 Response to Reviewer 2

Comment: (Remarks to the Author)

In this revised paper, the authors introduce a work of alleviating Target Prior Bias in Drug-Target Interaction Prediction (TAPB), a debiasing framework that employs amino acid randomization, confounder alignment module (CAM), and interventional training to approximate $P(Y|D, do(T))$ via backdoor adjustment, thereby addressing this bias. The experimental results show that TAPB achieves better performance, which is offering enhanced generalization and providing interpretable insights into drug-target interactions.

The authors employ several way to alleviate target prior bias in drug-target interaction prediction, including data level, model level, and training process level. The effectiveness seems to be sound.

Response:

We sincerely appreciate your thoughtful guidance and your understanding of our work. Your expert guidance has been invaluable in enhancing the methodological rigor of our work. Below, we detail the key revisions made in response to your constructive comments:

1. **Additional Ablation Experiments:** We have conducted comprehensive ablation studies on the Davis and BindingDB cross-domain datasets, expanding the total to three datasets (BioSNAP cross-domain, BindingDB cross-domain, and Davis). Each configuration was rigorously evaluated across five independent random seeds, with results reported as mean and standard deviation to ensure statistical robustness and reliability.

2. **Revision of Figures and Manuscript:** To improve clarity and accessibility, we have revised the figures presenting the ablation results, prioritizing the newly added experiments at the forefront as per your valuable guidance. This restructuring enables readers to directly compare the contribution of each module. Concurrently, the corresponding textual descriptions in the ablation section have been updated to align precisely with these visual enhancements and provide a cohesive narrative.

3. **Code and Source Availability:** We have updated the GitHub repository to include all code, accompanied by significantly enhanced documentation to facilitate reader comprehension. Furthermore, to address potential source accessibility constraints with Hugging Face, we have uploaded the ESM-2 and Molformer weights to the OneDrive cloud storage platform, enabling easier one-click downloads for reproducibility.

All revisions are clearly highlighted in the manuscript. We are deeply grateful for your constructive feedback, which has substantially strengthened this work. We

hope that these improvements align with the expectations.

Comments:

Comment: 2.1

In TAPB interventional training, the authors combine amino acid randomization, confounder alignment module (CAM), and multi-head cross-attention (MHCA) to approximate $P(Y|D, do(T))$. These three key components should be evaluated with more datasets (the authors conducted the ablation study only on the BioSNAP in-domain dataset).

Response 2.1.:

We sincerely appreciate your instructive comments regarding the addition of more comprehensive ablation studies for the key modules across more datasets. Your feedback is highly valuable, as expanded ablation analyses are essential for rigorously isolating the contribution of each component within our TAPB framework.

In direct response to your guidance, we have conducted additional ablation studies on the BindingDB cross-domain and Davis datasets, bringing the total to three evaluated datasets: BioSNAP cross-domain, BindingDB cross-domain, and Davis. The revised study encompasses seven clearly defined variants per dataset: (1) TAPB-CNN: replacing the ESM-2 encoder with an untrained CNN, (2) TAPB-Base (TAPB Plain in our last revision): Baseline dual-tower architecture with ESM-2 encoders, binary classification loss, and average pooling (no causal components), (3) TAPB-R: TAPB-Base augmented with amino acid randomization, (4) TAPB-RM: TAPB-R further enhanced with masked language modeling (MLM) loss, (5) TAPB-RM-BA: TAPB-RM with backdoor adjustment (without causal attention module, omitting $X_t \rightarrow C \rightarrow T$), (6) TAPB-RM-CAM: TAPB-RM with causal attention module (without backdoor adjustment), and (7) TAPB-Full: Complete model integrating all proposed components (ESM-2, randomization, MLM, CAM, and backdoor adjustment).

Comparative analysis of the first four variants confirms that each proposed module incrementally enhances performance, while the collective evaluation of the latter three variants demonstrates that accurate estimation of $P(Y|D, do(T))$ necessitates both causal inference components—CAM and backdoor adjustment—as neither alone achieves optimal results, thereby validating the theoretical foundation of our design.

You specifically requested ablation evaluations for the three core components—amino acid randomization, confounder alignment module (CAM), and multi-head cross-attention (MHCA)—across multiple datasets. We have fully addressed the first

two components through the expanded experiments described above. However, for MHCA, ablation is not feasible in our context, as it constitutes a standard and immutable configuration in widely adopted transformer-based aggregators, as evidenced in foundational Multi-modal work ALBEF [1] and DTI work Transformer-CPI [2], where it is typically treated as a fixed architectural element.

Acknowledging the significance of your comment, we have revised the manuscript to prominently feature the ablation results at the outset for immediate clarity on module contributions, while relocating hyperparameter tuning analyses and t-SNE visualizations to the SI, with appropriate cross-references in the manuscript. All modifications related to our expanded ablation studies are comprehensively documented in Revision 2.1. We are grateful for your insightful guidance, which has strengthened the rigor and transparency of our work.

Revision: 2.1.

2.4.4 Ablation study

PAGE 16-19

Figure 7: Ablation Studies results. (a) AUROC and AURPC of TAPB on BioSNAP cross-domain dataset under various target confounder dictionary sizes. (b) AUROC and AURPC of TAPB on the BioSNAP cross-domain dataset under various target random drop ratios. (c) AUROC and AURPC of TAPB on the BioSNAP cross-domain dataset under various mutation rates. (d) AUROC and AURPC of plain DrugBAN and DrugBAN w/ random deletion on the BioSNAP cross-domain dataset. (e) AUROC and AURPC of plain TransformerCPI and TransformerCPI w/ random deletion on the BioSNAP cross-domain dataset. (f) Ablation study of TAPB key components on the BioSNAP cross-domain dataset. (g) Ablation study of ESM-2 on the BioSNAP cross-domain dataset. (h) Target bias test on a prior-balanced dataset using ESM-2 features. (i) Drug bias test on a prior-balanced dataset using ESM-2 features. (a) Ablation study of TAPB key components on the Davis dataset. (b) Ablation study of TAPB key components on the BioSNAP cross-domain dataset. (c) Ablation study of TAPB key components on the BindingDB cross-domain dataset. (d) AUROC and AURPC of TransformerCPI, TransformerCPI w/ random deletion, DrugBAN, and DrugBAN w/ random deletion on the BioSNAP cross-domain dataset.

We conducted ablation studies on three datasets—DAVIS, BioSNAP cross-domain, and BindingDB cross-domain—to evaluate the impact of our key components, with a total of 7 TAPB variants, to ensure that our ablation study comprehensively captures the contribution of each component: (1) TAPB-CNN: Replacing the ESM-2 encoder with an untrained CNN, (2) TAPB-Base: Baseline dual-tower architecture with ESM-2 encoders, binary classification loss, and average pooling (w/o interventional training), (3) TAPB-R: TAPB-Base augmented with amino acid randomization, (4) TAPB-RM: TAPB-R enhanced with masked language modeling (MLM) loss, (5) TAPB-RM-BA: TAPB-RM with backdoor adjustment (without CAM, omitting $X_t \rightarrow C \rightarrow T$), (6) TAPB-RM-CAM: TAPB-RM with CAM (w/o backdoor adjustment), and (7) TAPB-Full: Complete TAPB model integrating all proposed components (ESM-2, randomization, MLM, CAM, and backdoor adjustment). Unless specified, all experiments of TAPB were conducted with identical hyperparameters to those in Supplementary Table 13. Each experiment included five independent runs with different random seeds. Comprehensive ablation results for Davis, BioSNAP cross-domain, BindingDB cross-domain, and residue random deletion generalizability are presented in Supplementary Tables 5–8.

ESM-2 encoder contribution: Given ESM-2’s strong representation capacity, we ablated its usage in the TAPB-Base architecture by replacing it with a randomly initialized CNN encoder (denoted TAPB-CNN). As shown in Fig. 7a,

b, and c, TAPB-Base significantly outperformed the TAPB-CNN, demonstrating the advantage of pretrained protein encoders. Meanwhile, to confirm that the ESM-2 features do not cause "target bias", we conducted "target bias" and "drug bias" tests on the "balanced" dataset introduced in our previous section, as detailed in Supplementary Section 1.3. As shown in Supplementary Fig. 2, neither test exhibited clustering similar to that in Fig. 2, indicating that incorporating the ESM-2 encoder enhances target representation and does not cause "target bias", which is primarily triggered by the data.

Amino acid randomization and MLM loss: Amino acid randomization significantly enhances model performance and serves as the most direct approach to prevent model from memorizing the target, thereby avoiding insufficient learning of interaction patterns. As shown in Fig. 7a, b, and c, TAPB-R consistently achieves higher AUROC and AUPRC scores than TAPB-Base across all three datasets, particularly on the Davis dataset, demonstrating the effectiveness of our randomization strategy and validating the rationale behind preventing target memorization. TAPB-RM marginally outperforms both TAPB-R and TAPB-Base on all three datasets. Although the performance improvement is less pronounced compared to amino acid randomization, the drug MLM loss effectively strengthens drug representation in target-biased datasets, thereby reducing the influence of the target.

Interventional training: According to the theory, TAPB requires both CAM and backdoor adjustment to compute $P(Y|\mathbf{D}, do(\mathbf{T}))$. Solely employing CAM violates the assumptions of our SCM, while the backdoor adjustment is specifically designed for our SCM and is theoretically invalid without CAM. To validate this, we designed ablation variants—TAPB-RM-BA, TAPB-RM-CAM, and TAPB-Full. As shown in Fig. 7a, b and c, when operating with only one module, TAPB-RM-BA and TAPB-RM-CAM exhibit comparable performance, while significant performance gains are exclusively observed in TAPB-Full. This pattern is particularly pronounced on the Davis dataset and consistently evident across the BioSNAP and BindingDB cross-domain datasets. The comparative analysis of these three variants empirically validates that our theoretically grounded design aligns with the expected theoretical outcomes.

Generalizability of residue random deletion: To test the generalizability of our approach, we integrated residue random deletion into DrugBAN (Non_DA) and TransformerCPI, using the hyperparameters specified in Supplementary Tables 16 and 9, respectively. Only residue random deletion was selected because our residue mutation and interventional training require both a pre-trained encoder and MHCA-based aggregation. Results on the BioSNAP cross-

domain dataset are as shown in Fig. 7d, TransformerCPI with random deletion exhibited substantial gains of nearly 10% in both AUROC and AUPRC, while DrugBAN with random deletion showed a 1% AUROC improvement over the baseline. This discrepancy may arise from TransformerCPI’s aggregator architecture being more suitable for modeling DTI under this modification. These results confirm our residual random deletion as a general, model-agnostic design.

We conducted ablation studies to evaluate the impact of key components and hyperparameters of the TAPB framework on DTI prediction performance. Unless specified otherwise, all experiments were conducted on the BioSNAP in-domain dataset using TAPB with identical hyperparameters to those in Supplementary Table 8. Each experiment included five independent runs with different random seeds.

Target confounder dictionary size: For that we compute $P(\mathbf{Y}|\mathbf{D}, do(\mathbf{T}))$ via backdoor adjustment, the target confounder dictionary plays a critical role. We evaluated the appropriate dictionary size by comparing TAPB performance on the BioSNAP cross-domain dataset across varying sizes. As shown in Figure 7a, TAPB achieved optimal performance with a dictionary size of 8, yielding an average AUROC of 0.79 and AUPRC of 0.80. This configuration demonstrated superior AUPRC compared to other sizes and was consequently selected as the default setting.

Target random drop ratio: This hyperparameter governs amino acid randomization by deleting features at fixed proportions. We evaluated TAPB performance on the BioSNAP cross-domain dataset across drop ratios ranging from 0 to 0.9. Figure 7b shows that performance was lowest without random dropping (ratio=0). Introducing any drop ratio (>0) increased AUROC by approximately 4% across configurations. Random target masking prevents spurious correlation memorization, suppresses target prior bias, and reduces computational costs through shortened target feature length P . We selected a ratio of 0.7 as the default due to its stable predictive performance.

Mutation rate: This hyperparameter governs amino acid mutations during randomization. We evaluated TAPB performance on the BioSNAP cross-domain dataset across mutation rates ranging from 0 to 0.8, with results presented in Figure 7c. Model performance demonstrates a unimodal trend: it initially improves then declines as mutation rate increases. Peak performance occurs at a mutation rate of 0.4, achieving 0.79 AUROC and 0.80 AUPRC. This indicates that higher mutation rates introduce excessive noise that destabilizes training. We selected a mutation rate of 0.2 as default since it maintains

competitive metrics approaching those at 0.4 while exhibiting the lowest variance across runs.

TAPB components contributions: We analyzed individual component contributions by evaluating four configurations on the BioSNAP cross-domain dataset: Plain TAPB, TAPB w amino acid randomization, TAPB w randomization & MLM, and full TAPB. As shown in Figure 7f: Plain TAPB yielded the lowest performance; adding amino acid randomization increased AUROC and AUPRC by 3% each; incorporating MLM loss provided marginal AUPRC improvement with little AUROC gain; finally, the full implementation (achieving backdoor adjustment via CAM and MHCA) boosted both metrics by 1% over preceding configurations. This ablation confirms the individual efficacy of each component.

ESM-2 Encoder Contribution: Given ESM-2's strong representation capacity, we isolated its contribution by comparing TAPB wo ESM-2 (CNN encoder) against plain TAPB on the BioSNAP in-domain dataset. Results in Figure 7g show plain TAPB outperformed the CNN variant by 2.5% AUROC and 1% AUPRC, demonstrating the significant advantage of pretrained protein encoders. Notably, while our debiasing framework requires a pretrained encoder, the component ablation in Figure 7f separately validates the efficacy of TAPB's novel modules beyond encoder selection.

Integration of residual random deletion into other methods: To test the generalizability of our approach, we integrated residual random deletion into DrugBAN and TransformerCPI. Only residual random deletion was selected because our residual mutation and backdoor adjustment approximation require both a pretrained encoder and MHCA-based aggregation. Results on the BioSNAP cross-domain dataset as shown in Figure 7d and e, DrugBAN with random deletion showed a 1% AUROC improvement over the baseline, while TransformerCPI with deletion exhibited substantial gains of nearly 10% in both AUROC and AUPRC. This discrepancy may arise from TransformerCPI's aggregator architecture being more suitable for modeling DTI under this modification. These results confirm our residual random deletion as a general, model-agnostic design.

Effect of ESM-2 features: In addition to proving that "prior tendency" is the cause of biased predictions in DTI sequence datasets, we also tested whether the stronger features extracted by ESM-2 could lead to biased predictions. For this purpose, we conducted bias tests (T, R) and (D, R) on the previously constructed unbiased dataset using the TAPB. As shown in Figure 7h and i, TAPB does not exhibit "target bias" when receiving random inputs R,

unlike in the BindingDB and BioSNAP datasets. This suggests that in DTI sequence datasets, labels and other priors have a more significant impact on DTI predictions, making them more susceptible to biased predictions.

Comment: 2.2.(Remarks on code availability):

It was difficult to load the sources.

Response 2.2

We sincerely appreciate your valuable feedback and important comments regarding the code issue. Our code is openly accessible on GitHub, and we have comprehensively documented all required libraries and detailed setup procedures in the README.md file, enabling readers to easily download the source code and configure the necessary environment using Anaconda.

The difficulty in loading the sources you mentioned likely pertains to the ESM-2 and Molformer weights; as explicitly noted in the README.md, the code requires accessing ESM-2 weights from Hugging Face for target feature generation and Molformer weights for drug tokenization. However, potential regional restrictions on Hugging Face access may have contributed to the loading challenges. To address this, we have uploaded both model weights to a dedicated OneDrive link, allowing readers to simply download, extract, and place them in the designated directory as specified in the README.md to ensure seamless reproducibility.

Furthermore, we have streamlined the code such that, after completing the environment setup, readers can reproduce our results with a single Python command. Additionally, recognizing that insufficient code comments previously hindered alignment with the paper's methodological details, we have enhanced the repository with extensive annotations to improve clarity and facilitate reader comprehension. We hope that these improvements will facilitate reproducibility for readers.

References

- [1] Junnan Li, Ramprasaath Selvaraju, Akhilesh Gotmare, Shafiq Joty, Caiming Xiong, and Steven Chu Hong Hoi. Align before fuse: Vision and language representation learning with momentum distillation. *Advances in neural information processing systems*, 34:9694–9705, 2021.
- [2] Lifan Chen, Xiaoqin Tan, Dingyan Wang, Feisheng Zhong, Xiaohong Liu, Tianbiao Yang, Xiaomin Luo, Kaixian Chen, Hualiang Jiang, and Mingyue Zheng. TransformerCPI: improving compound–protein interaction prediction by

sequence-based deep learning with self-attention mechanism and label reversal experiments. *Bioinformatics*, 36(16):4406–4414, 2020.

3 Response to Reviewer 3

Comment: (Remarks to the Author)

I'd like to thank the authors for their efforts in addressing the concerns raised in the first round of review. I can see that the manuscript has been improved; however, I also notice some newly identified problems and several unresolved issues remaining. I will write according to the index of my previous comments first and then add on other issues.

Response:

We sincerely appreciate the reviewer's professional comments and patience with our work. Your guidance has been instrumental in enhancing the rigor of our experiments and the readability of the manuscript. We offer our sincerest apologies for the imprecise writing and incomplete experiments described in our previous response. We highly value each of your meticulous suggestions and the opportunity you have provided. We have made every effort to provide point-by-point responses to your comments and have thoroughly revised every detail of the manuscript. Below are key revisions implemented:

1. We have revised the description of the datasets in our manuscript, reducing the number from six to four.
2. We have redesigned seven rigorous ablation experiments and additionally included two datasets, the Davis dataset and the BindingDB cross-domain dataset. The ablation study section has been completely rewritten to ensure each component is rigorously evaluated.
3. The conclusions drawn from the BioSNAP cross-domain dataset have been deleted from the cross-domain comparison section.
4. We have comprehensively corrected misreferenced tables and figures, carefully reviewed and standardized model nomenclature and symbols, eliminated redundant variables, and ensured compliance with writing conventions.
5. We have meticulously reviewed and revised the legends for Fig. 3 and Supplementary Fig. 3, replacing "smiles" with "drug" to align with the manuscript description and avoid reader confusion, and converted the line graphs to bar charts.
6. Results not pertaining to ablation experiments have been moved to the Supplementary Information (SI), with appropriate references included in the main manuscript.
7. We have carefully verified all SI references to ensure that every figure, table, and algorithm in the SI is correctly cited within the manuscript.
8. We have uploaded the latest simplified code to our GitHub repository, enabling direct reproduction of our results with a single line of Python code. Additionally, we have added more comments to facilitate reader comprehension.

Thanks to your guidance, the quality of our manuscript has been significantly improved, allowing our key findings—the "target prior bias," its underlying causes, and our novel interventional training approach—to be better understood by readers. We sincerely thank you for your rigorous guidance and the opportunity provided. We hope that our revisions meet your expectations.

Comments:

Comment: 3.1.

I do not agree that BioSNAP (BindingDB) in-domain and BioSNAP (BindingDB) cross-domain are considered as two datasets. They are the same dataset under two different settings (versions), an in-domain setting and a cross-domain setting. Even the authors' original manuscript considers BioSNAP and BindingDB as two datasets, rather than four so I am not sure why this revision calls it differently. Therefore, the claim of six datasets in the revised manuscript is misleading.

Response 3.1.:

We sincerely thank the reviewer's meticulous and insightful comments. We fully agree that BioSNAP (BindingDB) in-domain and BioSNAP (BindingDB) cross-domain should not be treated as distinct datasets, but rather as the same dataset under two different evaluation settings.

In our previous revision, we referred to them as separate datasets solely to streamline exposition for readers from diverse disciplinary backgrounds—particularly those in drug discovery who may not be familiar with the technical distinction between these settings. However, we recognize that this phrasing inadvertently led to the misleading characterization of "six datasets" in the revised manuscript.

Following your valuable suggestion, we have now consolidated the in-domain and cross-domain as a single dataset throughout the manuscript. All relevant sections have been updated accordingly, and these revisions are reflected in Revision 3.1.

Revision: 3.1.

1 Introduction

PAGE 4

Experiments conducted on ~~six~~ **four** publicly available datasets demonstrate that TAPB establishes a new benchmark in DTI prediction performance. The framework's adaptability offers potential improvements for other DTI models, provided our assumptions are satisfied.

2.4.1 Datasets and evaluation protocol.

PAGE 12

Following DrugBAN, we evaluated the in-domain and cross-domain classification performance of 6 publicly available datasets: BindingDB in-domain, BindingDB cross-domain, BioSNAP in-domain, BioSNAP cross-domain, Davis, Human cold-split. To ensure a rigorous and comprehensive assessment, we evaluated the model's classification performance across four publicly available datasets under six settings: BindingDB (in-domain and cross-domain), BioSNAP (in-domain and cross-domain), Davis, and Human cold-split. Supplementary Table 1 provides an overview of the datasets. Additionally, Supplementary Fig. 3 reveals "target prior bias" in the Davis dataset and "drug prior bias" in the Human cold-split dataset.

Comment: 3.2.

On page 65 of the response and page 12 of the revised manuscript, the authors say, "As shown in Table 1, TAPB achieves state-of-the-art performance ...". However, Table 1 provides statistics on the dataset instead of a performance comparison. Again, the authors need to proofread their writing before submission.

Response 3.2.:

We sincerely appreciate the reviewer's meticulous attention and patience with our work; your rigorous scrutiny is important to enhancing the quality of our manuscript. We acknowledge that Table 1 was incorrectly referred to. The correct reference should be to Supplementary Table 3, where we present detailed comparative results, including the mean and variance across all metrics (AUROC, AUPRC, ACC, Sensitivity, Specificity) for our method and all baselines under five different random seeds. Following your valuable comment, we have rewritten the in-domain comparison section related to Supplementary Table 3, ensuring that all citations in this section are accurate and that no references to Table 1 remain. We have also carefully reviewed all references to Supplementary Tables throughout the manuscript to ensure accurate citations. All corrections have been highlighted and summarized in Revision 3.2.

Revision: 3.2.

2.4.2 In-domain comparison

PAGE 13-15

We compared TAPB with five baseline models—MolTrans [1], TransformerCPI [2], DrugBAN [3], PSICHIC [4], and MlanDTI [5]—under

random splitting settings. Unlike these models, which rely on $P(Y|D, T)$ for predictions, TAPB $P(Y|D, do(T))$ via backdoor adjustment. The bias analysis for Davis and Human cold-split can be found in Supplementary Section 3. Davis shows "target prior bias" while Human cold-split shows "drug prior bias" as shown in Supplementary Fig. 3.

Figure 6: Radar chart comparisons of in-domain and cross-domain performance on BioSNAP and BindingDB, Davis, Human datasets and Human Cold-split datasets. (a) In-domain evaluation on BioSNAP. (b) Cross-domain evaluation on BioSNAP. In-domain evaluation on BindingDB. (c) In-domain evaluation on Davis. (d) In-domain evaluation on BindingDB. Cross-domain evaluation on BioSNAP. (e) Cross-domain evaluation on BindingDB. (f) Cold-split evaluation on Human.

TAPB exhibits comprehensive dominance over all baselines on the BioSNAP in-domain dataset, as evidenced by its larger polygon area across each evaluation metric in Fig. 6a. As shown in Supplementary Table 3, compared to the next best baseline PSICHIC, TAPB shows significant improvements: a 2.3% increase in AUROC; 2.2% in AUPRC; 2.7% in accuracy; 3.4% in sensitivity; and 1.9% in specificity. These statistically significant gains underscore TAPB’s effectiveness in alleviating target prior bias.

Fig. 6a demonstrates TAPB’s in-domain superiority through comprehensive

multi-metric analysis. The radar chart reveals TAPB's complete dominance over baselines in BioSNAP, forming a larger polygon area across all evaluation axes. As shown in Table 1, TAPB achieves state-of-the-art performance in BioSNAP with significant improvements: =2.3% higher AUROC, 2.2% higher AUPRC, 2.7% higher accuracy, 3.4% higher sensitivity, and 1.9% higher specificity than the next best baseline PSICHIC as shown in Supplementary Table 3. These statistically significant gains underscore TAPB's effectiveness in addressing target prior bias.

Despite the severe "target bias" in BindingDB, where models can achieve high performance by merely memorizing the target—leading to strong results across all baselines—we still conducted a fair comparison. As illustrated in Fig. 6b, TAPB maintains strong competitiveness, narrowly trailing the top-performing method DrugBAN by only 0.2% in AUROC and 0.3% in AUPRC as shown in Supplementary Table 3, demonstrating superior performance.

As shown in Fig. 6d, BindingDB exhibits pronounced target prior bias where models tend to overfit. Despite this challenge, TAPB maintains robust competitiveness, achieving near state-of-the-art performance with only 0.2% lower AUROC and 0.3% lower AUPRC than the top-performing DrugBAN as shown in Supplementary Table 3, while demonstrating superior generalization capability.

Notably, TAPB outperforms all baselines across every metric on the Davis dataset, as shown in Fig. 6c. Supplementary Table 3 confirms that it exceeds the strongest baseline, PSICHIC, by 2.1% in AUROC and 7.4% in AUPRC—the largest performance gap observed among all datasets. This underscores TAPB's exceptional ability to capture complex interaction patterns. In the challenging Human cold-split scenario, where "drug bias" is the dominant one, we also evaluated TAPB's performance under this opposite condition. As shown in Fig. 6f, TAPB maintains competitive performance, surpassing DrugBAN by 2.0% in AUROC and 2.8% in AUPRC, and outperforming three out of five baselines. This result demonstrates that TAPB, although designed to alleviate "target bias", also achieves strong performance on drug-biased datasets, highlighting its robustness and generalizability beyond its intended application context.

Remarkably, TAPB dominates the Davis dataset across all metrics, outperforming the strongest baseline PSICHIC by 2.1% in AUROC and 7.4% in AUPRC—the largest improvement observed among all datasets as shown in Supplementary Table 3. This highlights TAPB's exceptional ability to capture complex interaction patterns. In the challenging Human cold-split scenario,

as shown in Fig. 6f. TAPB maintains competitive performance, surpassing DrugBAN by 2.0% in AUROC and 2.8% in AUPRC while outperforming three of five baselines. This consistent multi-dataset superiority confirms TAPB's architectural advantages in mitigating bias while preserving predictive power.

Comment: 3.3.

Page 67 of the response, the figure at the top, subfigures a and b label legends as "Target" and "SMILES" and subfigure c labels the legends as "Drug" and "Target". Why such a discrepancy? Is "Drug" in c not represented as "SMILES"? If using representations as the legend, why not use the representation of "Target"? Basically, "Target" and "SMILES" are not parallel concepts. This problem persists in other similar figure legends.

Response 3.3.:

We sincerely thank the reviewer's meticulous comments, as your rigorous scrutiny has greatly improved the quality of our manuscript. In our previous revision, we indeed used "SMILES" instead of "Drug" in the legends of Fig. 3 and Supplementary Fig. 3. As correctly noted by the reviewer, "Target" and "SMILES" are not parallel concepts, which could cause confusion for readers. We sincerely apologize for this oversight during our revision process. We have now revised Fig. 3 and Supplementary Fig. 3 by replacing all instances of "SMILES" with "Drug," and have also updated the line charts to more informative bar charts (as requested by Reviewer 1), along with corresponding textual revisions. Additionally, we have included the p-values from our null model tests in Supplementary Fig. 3. All modifications have been highlighted and summarized in Revision 3.3.

Revision: 3.3.

2.2.2 Prior tendency causes biased predictions

PAGE 7-9

Figure 3: Statistical visualization of "prior tendency" for drugs and targets in the BindingDB and BioSNAP datasets. (a) "Prior tendency" frequency distribution of z_i and p-value for drugs and targets in the BindingDB in-domain training set. (b) "Prior tendency" frequency distribution of z_i and p-value for drugs and targets in the BindingDB cross-domain training set. (c) "Prior tendency" frequency distribution of z_i and p-value for drugs and targets in the BioSNAP in-domain training set. (d) Label frequency distribution of z_i for drugs and targets in the BioSNAP cross-domain training set. "Prior tendency" frequency distribution of z_i and p-value for drugs and targets in the BioSNAP cross-domain training set. (e) Quantification of the overall "prior tendency" for labels associated with drugs and targets across different datasets.

We assume that the biased predictions are caused by "prior tendency", which

we formally define as systematic label distribution biases inherent to individual drugs or targets in the DTI sequence dataset. Specifically, this refers to statistically significant deviations in the positive/negative sample ratios observed across different drugs (drug-level prior) or targets (target-level prior), which create spurious correlations that models can exploit to minimize loss without learning true interaction mechanisms. To quantify "prior tendency" across different DTI datasets, we devised the following label test:

$$z_i = \frac{\sum_j y_{ij}}{n_i} \quad (2)$$

$$Z = \sum_i |z_i - 0.5| + 0.5 \quad (3)$$

where y_{ij} denotes the j -th label of the i -th sequence, n_i denotes the total number of elements in the i -th sequence, n_i denotes the occurrence count of the i -th sequence, z_i denotes each sequence's "prior tendency" which is rounded to one decimal place for better visualization, and Z denotes the overall "prior tendency" across all sequences in the dataset, ranging from 0.5 to 1.0.

Furthermore, beyond the heuristic score Z , we designed a rigorous permutation test grounded in a null model where interaction labels Y are independent of target-specific effects. Under this null hypothesis, each drug-target pair's label follows a Bernoulli distribution parameterized by the global positive interaction proportion:

$$g = \frac{\sum Y}{N} \quad (4)$$

where N denotes the total number of drug-target pairs, representing random label assignment without target-specific biases. To evaluate statistical significance, we employed a weighted sum of squared deviations as our test statistic:

$$T = \sum_{i=1}^M n_i (g_i - g)^2$$

$$T = \sum_{i=1}^M n_i (z_i - g)^2 \quad (5)$$

where M is the total number of unique sequences, i.e. total number of unique drugs or targets in the dataset. n_i is the occurrence count of target

i , and g_i is its observed positive interaction proportion. The n_i weighting ensures proportional contribution by sample size while maintaining sensitivity for sparse targets. Our permutation procedure preserves drug-protein pair structures while randomly reshuffling labels across all pairs for $B = 1000$ iterations, with p-value computed as:

$$\text{p-value} = \frac{1 + \sum_{b=1}^B \mathbf{1}(T_b \geq T_{\text{obs}})}{1 + B} \quad (6)$$

where T_b is the permuted statistic and T_{obs} the observed value. This non-parametric approach maintains DTI data structures through fixed pairings with permuted labels, ensures small-sample robustness by avoiding asymptotic assumptions, and naturally adapts to class imbalance.

Supplementary Section 3. Extend data statistics

PAGE 12

We quantified the "prior tendency" and total overall "prior tendency" in the Davis and human cold-split datasets. As illustrated in Supplementary Fig. 3 a, b and c, the Davis dataset exhibits a certain "target prior bias", while a more pronounced "drug prior bias" is observed in the human cold-split data. **Statistical analysis confirmed significant deviations for both biases ($P_t < 0.001$, $P_d < 0.001$).**

Supplementary Figure 3: Statistics of Davis and Human cold-split. (a) "Prior tendency" frequency distribution of z_i for drugs and targets in the Davis training set. (b) "Prior tendency" frequency distribution of z_i for drugs and targets in the Human cold-split training set. (c) Overall "prior tendency" for labels associated with drugs and targets across Davis and Human cold-split datasets.

Comment: 3.4.

Comment 3.4.1.

The figure on page 78 (Fig. 7 in the revised manuscript). For the same label “Plain TAPB”, why is there a big difference in its performance in subfigures f and g? Are there more than one version of “Plain TAPB”? For subfigure f, why is the use of ESM-2 not included here for this ablation study in subfigure f?

Comment 3.4.2.

For this same figure, the caption states that it is about ablation studies; however, not all subfigures are ablation studies. For example, subfigures a, b, and c are not ablation studies, but rather hyperparameter sensitivity studies. Such inconsistencies need to be addressed.

Comment 3.4.3.

For this same figure, the captions of subfigures h and i are not clear enough to help readers understand what to observe from the figures.

Comment 3.4.4.

On page 79 of the response, it says, “all experiments were conducted on the BioSNAP in-domain dataset ...”. May I know which of the ablation studies were conducted on the BioSNAP in-domain dataset? It seems mostly the cross-domain dataset was used.

Comment 3.4.5.

Page 80 of the response, line 4, “Drug BAN”?

Response 3.4.:

We sincerely thank the reviewer for the meticulous review and highly detailed comments. You patiently identified numerous issues in our manuscript, and we deeply appreciate your rigor in both experimental design and writing. To facilitate a point-by-point review, we have numbered your comments accordingly:

Comment: 3.4.1.

The figure on page 78 (Fig. 7 in the revised manuscript). For the same label “Plain TAPB”, why is there a big difference in its performance in subfigures f and g? Are there more than one version of “Plain TAPB”? For subfigure f, why is the use of ESM-2 not included here for this ablation study in subfigure f?

Response 3.4.1.:

We sincerely apologize for the shortcomings in our previous manuscript regarding the presentation of the ablation study and the design of our experiments, which

failed to adequately demonstrate the innovation of our modules and the specific contribution of each component. Specifically, in Fig. 7 of our prior revision, a significant discrepancy was observed between subfigures f and g: subfigure f reported an AUROC of only 0.74 while subfigure g showed 0.92. This inconsistency stemmed from an error in the caption of subfigure g, which incorrectly attributed the results to the cross-domain dataset; in reality, subfigure g presents outcomes on the BioSNAP in-domain dataset, whereas subfigure f corresponds to the cross-domain ablation study. Consequently, the ablation of the ESM-2 encoder was omitted from subfigure f due to this misalignment.

Acknowledging the lack of rigor in our ablation study design—including the exclusion of ESM-2 ablation under identical dataset conditions and insufficient ablation for the causal inference components—we have conducted comprehensive experiments on two additional datasets. To ensure objectivity, each ablation variant was evaluated using five independent random seeds.

The revised ablation study encompasses seven clearly defined variants per dataset: (1) TAPB-CNN: replacing the ESM-2 encoder with an untrained CNN, (2) TAPB-Base (TAPB Plain in our previous revision): Baseline dual-tower architecture with ESM-2 encoders, binary classification loss, and average pooling (no causal components), (3) TAPB-R: TAPB-Base augmented with amino acid randomization, (4) TAPB-RM: TAPB-R further enhanced with masked language modeling (MLM) loss, (5) TAPB-RM-BA: TAPB-RM with backdoor adjustment (without causal attention module, omitting $X_t \rightarrow C \rightarrow T$), (6) TAPB-RM-CAM: TAPB-RM with causal attention module (without backdoor adjustment), and (7) TAPB-Full: Complete model integrating all proposed components (ESM-2, randomization, MLM, CAM, and backdoor adjustment).

Comparative analysis of the first four variants confirms that each proposed module incrementally enhances performance, while the collective evaluation of the latter three variants demonstrates that accurate estimation of $P(Y|\mathbf{D}, do(\mathbf{T}))$ necessitates both causal inference components—CAM and backdoor adjustment—as neither alone achieves optimal results, thereby validating the theoretical foundation of our design.

We have thoroughly rewritten the ablation study section and redrawn Fig. 7 to ensure methodological rigor and clarity in substantiating the role of each module, with all captions meticulously verified for accuracy. All changes are fully detailed in Revision 3.4.1, and we deeply appreciate the reviewer’s valuable feedback.

Revision: 3.4.1.

2.4.4 Ablation study

PAGE 16-19

Figure 7: Ablation Studies results. (a) AUROC and AUPRC of TAPB on BioSNAP cross-domain dataset under various target confounder dictionary sizes. (b) AUROC and AUPRC of TAPB on the BioSNAP cross-domain dataset under various target random drop ratios. (c) AUROC and AUPRC of TAPB on the BioSNAP cross-domain dataset under various mutation rates. (d) AUROC and AUPRC of plain DrugBAN and DrugBAN w/ random deletion on the BioSNAP cross-domain dataset. (e) AUROC and AUPRC of plain TransformerCPI and TransformerCPI w/ random deletion on the BioSNAP cross-domain dataset. (f) Ablation study of TAPB key components on the BioSNAP cross-domain dataset. (g) Ablation study of ESM-2 on the BioSNAP cross-domain dataset. (h) Target bias test on a prior-balanced dataset using ESM-2 features. (i) Drug bias test on a prior-balanced dataset using ESM-2 features. (a) Ablation study of TAPB key components on the Davis dataset. (b) Ablation study of TAPB key components on the BioSNAP cross-domain dataset. (c) Ablation study of TAPB key components on the BindingDB cross-domain dataset. (d) AUROC and AUPRC of TransformerCPI, TransformerCPI w/ random deletion, DrugBAN, and DrugBAN w/ random deletion on the BioSNAP cross-domain dataset.

We conducted ablation studies on three datasets—DAVIS, BioSNAP cross-domain, and BindingDB cross-domain—to evaluate the impact of our key

components, with a total of 7 TAPB variants, to ensure that our ablation study comprehensively captures the contribution of each component: (1) TAPB-CNN: Replacing the ESM-2 encoder with an untrained CNN, (2) TAPB-Base: Baseline dual-tower architecture with ESM-2 encoders, binary classification loss, and average pooling (w/o interventional training), (3) TAPB-R: TAPB-Base augmented with amino acid randomization, (4) TAPB-RM: TAPB-R enhanced with masked language modeling (MLM) loss, (5) TAPB-RM-BA: TAPB-RM with backdoor adjustment (without CAM, omitting $X_t \rightarrow C \rightarrow T$), (6) TAPB-RM-CAM: TAPB-RM with CAM (w/o backdoor adjustment), and (7) TAPB-Full: Complete TAPB model integrating all proposed components (ESM-2, randomization, MLM, CAM, and backdoor adjustment). Unless specified, all experiments of TAPB were conducted with identical hyperparameters to those in Supplementary Table 13. Each experiment included five independent runs with different random seeds. Comprehensive ablation results for Davis, BioSNAP cross-domain, BindingDB cross-domain, and residue random deletion generalizability are presented in Supplementary Tables 5–8.

ESM-2 encoder contribution: Given ESM-2’s strong representation capacity, we ablated its usage in the TAPB-Base architecture by replacing it with a randomly initialized CNN encoder (denoted TAPB-CNN). As shown in Fig. 7a, b, and c, TAPB-Base significantly outperformed the TAPB-CNN, demonstrating the advantage of pretrained protein encoders. Meanwhile, to confirm that the ESM-2 features do not cause "target bias", we conducted "target bias" and "drug bias" tests on the "balanced" dataset introduced in our previous section, as detailed in Supplementary Section 1.3. As shown in Supplementary Fig. 2, neither test exhibited clustering similar to that in Fig. 2, indicating that incorporating the ESM-2 encoder enhances target representation and does not cause "target bias", which is primarily triggered by the data.

Amino acid randomization and MLM loss: Amino acid randomization significantly enhances model performance and serves as the most direct approach to prevent model from memorizing the target, thereby avoiding insufficient learning of interaction patterns. As shown in Fig. 7a, b, and c, TAPB-R consistently achieves higher AUROC and AUPRC scores than TAPB-Base across all three datasets, particularly on the Davis dataset, demonstrating the effectiveness of our randomization strategy and validating the rationale behind preventing target memorization. TAPB-RM marginally outperforms both TAPB-R and TAPB-Base on all three datasets. Although the performance improvement is less pronounced compared to amino acid randomization, the drug MLM loss effectively strengthens drug representation in target-biased datasets, thereby

reducing the influence of the target.

Interventional training: According to the theory, TAPB requires both CAM and backdoor adjustment to compute $P(Y|\mathbf{D}, do(\mathbf{T}))$. Solely employing CAM violates the assumptions of our SCM, while the backdoor adjustment is specifically designed for our SCM and is theoretically invalid without CAM. To validate this, we designed ablation variants—TAPB-RM-BA, TAPB-RM-CAM, and TAPB-Full. As shown in Fig. 7a, b and c, when operating with only one module, TAPB-RM-BA and TAPB-RM-CAM exhibit comparable performance, while significant performance gains are exclusively observed in TAPB-Full. This pattern is particularly pronounced on the Davis dataset and consistently evident across the BioSNAP and BindingDB cross-domain datasets. The comparative analysis of these three variants empirically validates that our theoretically grounded design aligns with the expected theoretical outcomes.

Generalizability of residue random deletion: To test the generalizability of our approach, we integrated residue random deletion into DrugBAN (Non_DA) and TransformerCPI, using the hyperparameters specified in Supplementary Tables 16 and 9, respectively. Only residue random deletion was selected because our residue mutation and interventional training require both a pre-trained encoder and MHCA-based aggregation. Results on the BioSNAP cross-domain dataset are as shown in Fig. 7d, TransformerCPI with random deletion exhibited substantial gains of nearly 10% in both AUROC and AUPRC, while DrugBAN with random deletion showed a 1% AUROC improvement over the baseline. This discrepancy may arise from TransformerCPI’s aggregator architecture being more suitable for modeling DTI under this modification. These results confirm our residual random deletion as a general, model-agnostic design.

~~We conducted ablation studies to evaluate the impact of key components and hyperparameters of the TAPB framework on DTI prediction performance. Unless specified otherwise, all experiments were conducted on the BioSNAP in-domain dataset using TAPB with identical hyperparameters to those in Supplementary Table 8. Each experiment included five independent runs with different random seeds.~~

~~**Target confounder dictionary size:** For that we compute $P(Y|\mathbf{D}, do(\mathbf{T}))$ via backdoor adjustment, the target confounder dictionary plays a critical role. We evaluated the appropriate dictionary size by comparing TAPB performance on the BioSNAP cross-domain dataset across varying sizes. As shown in Figure 7a, TAPB achieved optimal performance with a dictionary size of 8, yielding an average AUROC of 0.79 and AUPRC of 0.80. This configuration demonstrated~~

superior AUPRC compared to other sizes and was consequently selected as the default setting.

Target random drop ratio: This hyperparameter governs amino acid randomization by deleting features at fixed proportions. We evaluated TAPB performance on the BioSNAP cross-domain dataset across drop ratios ranging from 0 to 0.9. Figure 7b shows that performance was lowest without random dropping (ratio=0). Introducing any drop ratio (>0) increased AUROC by approximately 4% across configurations. Random target masking prevents spurious correlation memorization, suppresses target prior bias, and reduces computational costs through shortened target feature length P . We selected a ratio of 0.7 as the default due to its stable predictive performance.

Mutation rate: This hyperparameter governs amino acid mutations during randomization. We evaluated TAPB performance on the BioSNAP cross-domain dataset across mutation rates ranging from 0 to 0.8, with results presented in Figure 7c. Model performance demonstrates a unimodal trend: it initially improves then declines as mutation rate increases. Peak performance occurs at a mutation rate of 0.4, achieving 0.79 AUROC and 0.80 AUPRC. This indicates that higher mutation rates introduce excessive noise that destabilizes training. We selected a mutation rate of 0.2 as default since it maintains competitive metrics approaching those at 0.4 while exhibiting the lowest variance across runs.

TAPB components contributions: We analyzed individual component contributions by evaluating four configurations on the BioSNAP cross-domain dataset: Plain TAPB, TAPB w amino acid randomization, TAPB w randomization & MLM, and full TAPB. As shown in Figure 7f: plain TAPB yielded the lowest performance; adding amino acid randomization increased AUROC and AUPRC by 3% each; incorporating MLM loss provided marginal AUPRC improvement with little AUROC gain; finally, the full implementation (achieving backdoor adjustment via CAM and MHCA) boosted both metrics by 1% over preceding configurations. This ablation confirms the individual efficacy of each component.

ESM-2 Encoder Contribution: Given ESM-2's strong representation capacity, we isolated its contribution by comparing TAPB wo ESM-2 (CNN encoder) against plain TAPB on the BioSNAP in-domain dataset. Results in Figure 7g show plain TAPB outperformed the CNN variant by 2.5% AUROC and 1% AUPRC, demonstrating the significant advantage of pretrained protein encoders. Notably, while our debiasing framework requires a pretrained encoder, the component ablation in Figure 7f separately validates the efficacy

of TAPB's novel modules beyond encoder selection.

Integration of residual random deletion into other methods: To test the generalizability of our approach, we integrated residual random deletion into DrugBAN and TransformerCPI. Only residual random deletion was selected because our residual mutation and backdoor adjustment approximation require both a pretrained encoder and MHCA-based aggregation. Results on the BioSNAP cross-domain dataset as shown in Figure 7d and e, DrugBAN with random deletion showed a 1% AUROC improvement over the baseline, while TransformerCPI with deletion exhibited substantial gains of nearly 10% in both AUROC and AUPRC. This discrepancy may arise from TransformerCPI's aggregator architecture being more suitable for modeling DTI under this modification. These results confirm our residual random deletion as a general, model-agnostic design.

Effect of ESM-2 features: In addition to proving that "prior tendency" is the cause of biased predictions in DTI sequence datasets, we also tested whether the stronger features extracted by ESM-2 could lead to biased predictions. For this purpose, we conducted bias tests (T, R) and (D, R) on the previously constructed unbiased dataset using the TAPB. As shown in Figure 7h and i, TAPB does not exhibit "target bias" when receiving random inputs R , unlike in the BindingDB and BioSNAP datasets. This suggests that in DTI sequence datasets, labels and other priors have a more significant impact on DTI predictions, making them more susceptible to biased predictions.

Comment: 3.4.2

For this same figure, the caption states that it is about ablation studies; however, not all subfigures are ablation studies. For example, subfigures a, b, and c are not ablation studies, but rather hyperparameter sensitivity studies. Such inconsistencies need to be addressed.

Response 3.4.2.:

We sincerely thank the reviewer for the insightful comments and constructive suggestions, which have greatly helped us improve the manuscript. In Fig. 7b and c, the initial values for the target random drop ratio and mutation rate were set to 0, indicating that these techniques were not applied; consequently, in our previous revision, we treated subfigures b and c as ablation studies to demonstrate the performance improvements achieved when incorporating target random drop and mutation rate. We acknowledge that subfigure a is not an ablation experiment but rather a dedicated parameter tuning study.

Following your valuable comment, we have relocated the parameter tuning experiments—specifically subfigures a, b, and c—to Supplementary Fig. 1 in Supplementary Section 1.2, and have ensured proper reference within the manuscript. All revisions are comprehensively summarized in Revision 3.4.2.

Revision: 3.4.2.

2.4.4 Ablation study

PAGE 18-19

~~Target confounder dictionary size: For that we compute $P(Y|D, do(T))$ via backdoor adjustment; the target confounder dictionary plays a critical role. We evaluated the appropriate dictionary size by comparing TAPB performance on the BioSNAP cross-domain dataset across varying sizes. As shown in Figure 7a, TAPB achieved optimal performance with a dictionary size of 8, yielding an average AUROC of 0.79 and AUPRC of 0.80. This configuration demonstrated superior AUPRC compared to other sizes and was consequently selected as the default setting.~~

~~Target random drop ratio: This hyperparameter governs amino acid randomization by deleting features at fixed proportions. We evaluated TAPB performance on the BioSNAP cross-domain dataset across drop ratios ranging from 0 to 0.9. Figure 7b shows that performance was lowest without random dropping (ratio=0). Introducing any drop ratio (≥ 0) increased AUROC by approximately 4% across configurations. Random target masking prevents spurious correlation memorization, suppresses target prior bias, and reduces computational costs through shortened target feature length. We selected a ratio of 0.7 as the default due to its stable predictive performance.~~

~~Mutation rate: This hyperparameter governs amino acid mutations during randomization. We evaluated TAPB performance on the BioSNAP cross-domain dataset across mutation rates ranging from 0 to 0.8, with results presented in Figure 7c. Model performance demonstrates a unimodal trend: it initially improves then declines as mutation rate increases. Peak performance occurs at a mutation rate of 0.4, achieving 0.79 AUROC and 0.80 AUPRC. This indicates that higher mutation rates introduce excessive noise that destabilizes training. We selected a mutation rate of 0.2 as default since it maintains competitive metrics approaching those at 0.4 while exhibiting the lowest variance across runs.~~

Supplementary Section 1.2 Hyperparameters setting and tuning

PAGE 3-4

Supplementary Figure 1: Hyperparameters tuning results. (a) AUROC and AURPC of TAPB on BioSNAP cross-domain dataset under various target confounder dictionary sizes. (b) AUROC and AURPC of TAPB on the BioSNAP cross-domain dataset under various target random drop ratios. (c) AUROC and AURPC of TAPB on the BioSNAP cross-domain dataset under various mutation rates.

We report the hyperparameter optimization results on BioSNAP cross-domain dataset for our three key parameters—Target Confounder Dict Size, Target Random Drop Ratio, and Mutation Rate—in Supplementary Table 2, presenting the mean values and variances obtained from five independent runs per configuration with different random seeds; the outcomes across seeds showed minimal variation, with variances generally remaining below 0.0001.

Supplementary Table 2: Detailed hyperparameter optimization results for Target Confounder Dict Size, Target Random Drop Ratio, and Mutation Rate. All values are rounded to five decimal places. For clarity, the highest results are indicated in bold.

Value	AUROC	AUPRC
	Target Confounder Dict Size	
2	0.77476±0.00003	0.79014±0.00003
4	0.78514±0.00003	0.80292±0.00007
8	0.79122±0.00004	0.79956±0.00002
16	0.78076±0.00008	0.78884±0.00008
	Target Random Drop Ratio	
0	0.73492±0.00010	0.75326±0.00007
0.3	0.77898±0.00010	0.78992±0.00007
0.5	0.78166±0.00011	0.79658±0.00021
0.7	0.79122±0.00004	0.79956±0.00002
0.9	0.78450±0.00004	0.79822±0.00009
	Mutation Rate	
0	0.78144±0.00011	0.78944±0.00014
0.2	0.79122±0.00004	0.79956±0.00002
0.4	0.79432±0.00011	0.80428±0.00007
0.6	0.78700±0.00004	0.79518±0.00002
0.8	0.75884±0.00015	0.77144±0.00003

Target confounder dictionary size: For that we compute $P(Y|\mathbf{D}, do(\mathbf{T}))$ via back-door adjustment; the target confounder dictionary plays a critical role. We evaluated the appropriate dictionary size by comparing TAPB performance on the BioSNAP cross-domain dataset across varying sizes. As shown in Supplementary Fig. 1a, TAPB achieved optimal performance with a dictionary size of 8, yielding an average AUROC of 0.79 and AUPRC of 0.80. This configuration demonstrated superior AUPRC compared to other sizes and was consequently selected as the default setting.

Target random deletion ratio: This hyperparameter governs amino acid randomization by deleting features at fixed proportions. We evaluated TAPB performance on the BioSNAP cross-domain dataset across drop ratios ranging from 0 to 0.9. Supplementary Fig. 1b shows that performance was lowest without random drop (ratio=0). Introducing any drop ratio (>0) increased AUROC by approximately 4% across configurations. Target random deletion prevents spurious correlation memorization, suppresses target prior bias, and reduces computational costs through shortened target feature length. We selected a ratio of 0.7 as the default due to its stable predictive performance.

Mutation rate: This hyperparameter governs amino acid mutations during randomization. We evaluated TAPB performance on the BioSNAP cross-domain dataset across mutation rates ranging from 0 to 0.8, with results presented in Supplementary Fig. 1c. Model performance demonstrates a unimodal trend: it initially improves, then declines as the mutation rate increases. Peak performance occurs at a mutation rate of 0.4, achieving 0.79 AUROC and 0.80 AUPRC. This indicates that higher mutation rates introduce excessive noise that destabilizes training. We selected a mutation rate

of 0.2 as default since it maintains competitive metrics approaching those at 0.4 while exhibiting the lowest variance across runs.

Comment: 3.4.3.

For this same figure, the captions of subfigures h and i are not clear enough to help readers understand what to observe from the figures.

Response 3.4.3.:

We sincerely appreciate the reviewer's valuable feedback. Given ESM-2's powerful encoding capabilities, we sought not only to investigate its impact on model performance but also to test whether the enhanced features extracted by ESM-2 might induce a stronger preference for target-based prediction. Since conventional datasets inherently exhibit target prior bias, evaluation within such datasets is precluded. Consequently, we conducted both "target bias" and "drug bias" tests on the "balanced" dataset designed in Section 2, with visualizations presented in subfigures g and h (Now Supplementary Fig. 2a and b). Both visualizations demonstrate uniform distribution patterns, indicating that ESM-2's enhanced features do not cause the model to disproportionately favor target-based predictions; instead, "target bias" is primarily attributable to the dataset.

We have relocated these subfigures to Supplementary Fig. 2 in Supplementary Section 1.3, titling this section "ESM-2 Features Do Not Drive DTI Prediction Bias" to enhance reader comprehension. The figure captions have been comprehensively revised to clarify the experimental methodology and provide detailed explanations of the visualizations. All related modifications are summarized in Revision 3.4.3.

Revision: 3.4.3.

Supplementary Section 1.3: ESM-2 features do not drive DTI prediction bias

PAGE 4-5

In addition to proving that "prior tendency" is the cause of biased predictions in DTI sequence datasets, we also tested whether the stronger features extracted by ESM-2 could lead to biased predictions. For this purpose, we conducted bias tests ((T, R) and (D, R)) on the previously constructed unbiased dataset using the TAPB-Base. As shown in Supplementary Fig. 2a and b, TAPB does not exhibit "target bias" when receiving random inputs R , unlike in the BindingDB and BioSNAP datasets. This suggests that in DTI sequence datasets, labels and other priors have a more significant impact on DTI predictions, making them

more susceptible to biased predictions.

Supplementary Figure 2: T-SNE Visualization of bias tests. (a) TAPB "target bias" test on a prior-balanced dataset using ESM-2 features. (b) TAPB "drug bias" test on a prior-balanced dataset using ESM-2 features.

Comment: 3.4.4.

On page 79 of the response, it says, "all experiments were conducted on the BioSNAP in-domain dataset ...". May I know which of the ablation studies were conducted on the BioSNAP in-domain dataset? It seems mostly the cross-domain dataset was used.

Response 3.4.4.:

We sincerely thank the reviewer's meticulous scrutiny. We acknowledge an error in our description of the ablation study datasets: while all ablation experiments were intended to be conducted on BioSNAP cross-domain, the ESM-2 ablation was performed on BioSNAP in-domain, which constituted an inappropriate experimental design. We recognize both the oversight in our reporting and the flaw in the experimental setup. To address this comprehensively, we have now incorporated two additional datasets—BioSNAP cross-domain and Davis—designing seven distinct experimental versions to rigorously evaluate each model component through detailed ablation analysis. The entire ablation study section has been thoroughly rewritten and consolidated in Revision 3.4.1.

Comment: 3.4.5.

Page 80 of the response, line 4, “Drug BAN”?

Response 3.4.5.:

We sincerely thank the reviewer for their meticulous review. We are deeply grateful for your rigor and patience in evaluating our work. We have revised the manuscript, thoroughly rewriting the ablation experiments section (where we wrote ‘Drug BAN’) to ensure that readers can better understand our ablation studies. Additionally, we have carefully checked and corrected all model abbreviations throughout the paper to eliminate any inappropriate spacing or spelling errors.

Revision: 3.4.5

2.4.4 Ablation study

PAGE 18-19

~~Integration of residual random deletion into other methods: To test the generalizability of our approach, we integrated residual random deletion into Drug BAN and TransformerCPI. Only residual random deletion was selected because our residual mutation and backdoor adjustment approximation require both a pretrained encoder and MHCA-based aggregation. Results on the BioSNAP cross-domain dataset as shown in Figure 7d and e, DrugBAN with random deletion showed a 1% AUROC improvement over the baseline, while TransformerCPI with deletion exhibited substantial gains of nearly 10% in both AUROC and AUPRC. This discrepancy may arise from TransformerCPI’s aggregator architecture being more suitable for modeling DTI under this modification. These results confirm our residual random deletion as a general, model-agnostic design.~~

Generalizability of residue random deletion: To test the generalizability of our approach, we integrated residue random deletion into DrugBAN (Non_DA) and TransformerCPI, using the hyperparameters specified in Supplementary Tables 16 and 9, respectively. Only residue random deletion was selected because our residue mutation and interventional training require both a pre-trained encoder and MHCA-based aggregation. Results on the BioSNAP cross-domain dataset are as shown in Figure 7d, TransformerCPI with random deletion exhibited substantial gains of nearly 10% in both AUROC and AUPRC, while DrugBAN with random deletion showed a 1% AUROC improvement over the baseline. This discrepancy may arise from TransformerCPI’s aggregator architecture being more suitable for modeling DTI under this modification. These results confirm our residual random deletion as a general, model-agnostic design.

Comment: 3.5.

The performance of TAPB has been tuned using the BioSNAP cross-domain dataset to find the best settings for the best results, so the performance on this dataset should not be used to draw conclusions. On the other hand, how many hyperparameters or settings of the competing methods have been tuned? On what datasets? It is important to clarify such details to ensure a fair comparison between them.

Also, the authors refer to such hyperparameter tuning as ablation studies, which is incorrect.

Response 3.5.:

We sincerely appreciate the reviewer's thorough examination of our experiments. Your feedback is crucial to the content of our experimental section and the rigor of our work.

We independently tuned 3 hyperparameters—target confounder dictionary size, target random deletion ratio, and mutation rate—on the BioSNAP cross-domain dataset. You pointed out that since hyperparameter tuning was performed on the BioSNAP cross-domain, this dataset should not be used to draw conclusions. We acknowledge that tuning parameters only for our method on this dataset—without adjusting those of other baselines—would lead to an unfair comparison. We have therefore removed all conclusive statements related to BioSNAP cross-domain in the "Cross-domain comparison" section, explicitly informed readers in the manuscript that parameter tuning was conducted on this dataset, and retained the results only for reference. We now base our conclusions solely on the BindingDB cross-domain dataset.

Regarding the ablation studies, we have supplemented the experiments with two additional datasets, Davis and BindingDB cross-domain, and redesigned 7 TAPB variants to thoroughly evaluate the contribution of each component. We have also expanded the original ablation experiments on the BioSNAP cross-domain. Since these experiments only involve comparisons among our own method's variants—using an incremental approach starting from TAPB-CNN and gradually adding modules—even if some modules were independently tuned, the comparisons remain fair. The comprehensively rewritten ablation study is detailed in Revision 3.4.1.

We did not tune the hyperparameters of other competing methods across any datasets for two reasons:

1. Competing methods such as DrugBAN, MlanDTI, and TransformerCPI have already been evaluated and reported in their papers using exactly the same datasets as in our study. In other words, these hyperparameters were already optimized by the original authors using a consistent set of values.

2. We used the same hyperparameter settings as provided in their official GitHub implementations. Our reproduced results are highly consistent with those reported in previous work, ensuring reliability.

Since our previous revision did not clearly state that these hyperparameters were kept fixed—which may have caused misunderstanding—we have now explicitly added a dedicated subsection at the beginning of the experimental section listing all hyperparameters of the competing methods. This ensures readers can fully understand our experimental setup. All related details are summarized in Revision 3.5.

Additionally, you mentioned in Comment 3.4.2 that the results of our hyperparameter tuning should not be placed in the ablation studies section. We agree with this point. As emphasized in your comment, we have moved the hyperparameter tuning experiments to Supplementary Section 1.2 (Supplementary Fig. 1), with the caption revised to "Hyperparameter tuning results". and cited them accordingly earlier in the manuscript. These changes are summarized in Revision 3.4.2.

Revision: 3.5.

Datasets and evaluation protocol.

PAGE 24-25

~~Following DrugBAN, we evaluated the in-domain and cross-domain classification performance of 6 publicly available datasets: BindingDB in-domain, BindingDB cross-domain, BioSNAP in-domain, BioSNAP cross-domain, Davis, Human cold-split.~~ **To ensure a rigorous and comprehensive assessment, we evaluated the model's classification performance across four publicly available datasets under six settings: BindingDB (in-domain and cross-domain), BioSNAP (in-domain and cross-domain), Davis, and Human cold-split. Supplementary Table 1 provides an overview of the datasets. Additionally, Supplementary Fig. 3 reveals "target prior bias" in the Davis dataset and "drug prior bias" in the Human cold-split dataset.**

~~The in-domain datasets were randomly divided into training, validation, and test sets in a 7:1:2 ratio. However, due to the sparsity of the data, nearly half of the drugs in the training set and test set of BindingDB in-domain do not overlap, while the targets exhibit a high degree of overlap. The cross-domain datasets, constructed by DrugBAN, consists of a source domain training set, a target domain training set, and a target domain testing set, with no overlap between the drugs and target in the source and target domain, i.e. CVS4.~~ **Notably, in these in-domain datasets, targets exhibit significantly higher overlap across training, validation, and test sets compared to drugs. In contrast, the cross-domain datasets—constructed by DrugBAN—consist of a source domain training set, a target domain training set, and a target domain test set,**

with no overlap between source domain drugs/targets and the target domain data (CVS4). We adopted the Youden Index to adjust the optimal threshold, which balances sensitivity and specificity more effectively. Davis dataset For in-domain datasets, Davis and Human, we conducted five independent runs using different seeds for each dataset, selecting the weight with the highest AUROC on the validation set, and then evaluating the model on the test set to report the performance metrics. For cross-domain datasets, following DrugBAN, we conducted five independent runs using different seeds on the source domain training set and reported metrics on the target domain test set directly, because we did not apply domain adaptation techniques.

For all datasets, we performed five independent runs with different random seeds and reported the area under the receiver operating characteristic curve (AUROC), the area under the precision-recall curve (AUPRC), accuracy, sensitivity, and specificity. The Youden Index was adopted to adjust the optimal threshold, offering a more effective balance between sensitivity and specificity. For in-domain datasets, we selected the model checkpoint with the highest validation AUROC and reported test set performance. Following DrugBAN’s protocol for cross-domain datasets, we trained models without domain adaptation techniques on the source domain and evaluated them directly on the target domain test set, reporting the resulting metrics. Table 1 provides detailed information on the datasets.

We compared TAPB with five baseline models—MolTrans [1], Transformer-CPI [2], DrugBAN [3], PSICHIC [4], and MlanDTI [5]. Unlike these models, which rely on $P(Y|D, T)$ for predictions, TAPB computes $P(Y|D, do(T))$ via backdoor adjustment for predictions. The hyperparameter settings for TransformerCPI, MolTrans, DrugBAN (on in-domain, Davis, Human), DrugBAN-da (on cross-domain), TAPB, PSICHIC, and MlanDTI are detailed in Supplementary Tables 9-15. For each model, hyperparameters remained consistent across all datasets unless otherwise specified. The key hyperparameters of TAPB—target confounder dictionary size, target random deletion ratio, and mutation rate—were tuned on the BioSNAP cross-domain dataset, as shown in Supplementary Fig. 1. Accordingly, comparative conclusions were not drawn from this dataset. Supplementary Table 2 provides the per-seed AUROC and AUPRC values across all optimizations.

Figure 6: Radar chart comparisons of in-domain and cross-domain performance on BioSNAP and BindingDB, Davis, Human datasets and Human Cold-split datasets. (a) In-domain evaluation on BioSNAP. (b) Cross-domain evaluation on BioSNAP. In-domain evaluation on BindingDB. (c) In-domain evaluation on Davis. (d) In-domain evaluation on BindingDB. Cross-domain evaluation on BioSNAP. (e) Cross-domain evaluation on BindingDB. (f) Cold-split evaluation on Human.

Cross-domain comparison.

PAGE 24-25

As shown in Fig. 6b and d, TAPB demonstrates superior cross-domain generalization capability. Supplementary Table 4 provides comprehensive performance comparisons. On the BioSNAP dataset, TAPB achieves state-of-the-art performance with 0.791 AUROC, 0.800 AUPRC, and 0.732 accuracy, outperforming the strongest baseline PSICHIC by 1.3% in AUROC, 1.4% in AUPRC, and 0.8% in accuracy. Notably, TAPB also attains the highest sensitivity among all methods at 0.676. On the BindingDB cross-domain dataset, TAPB maintains robust competitiveness, achieving leading performance in AUROC 0.676, accuracy 0.630, and specificity 0.565, while surpassing DrugBAN with CDAN (w/ CDAN) by 7.5% in AUROC, 5.8% in AUPRC, and 5.1% in accuracy.

As illustrated in Fig. 6e, TAPB exhibits outstanding cross-domain

generalization capabilities. Comprehensive performance comparisons are provided in Supplementary Table 4. On the BindingDB cross-domain dataset, TAPB maintains strong competitiveness, delivering notable results with an AUROC of 0.676, accuracy of 0.630, and specificity of 0.565, while surpassing DrugBAN-da (w/ CDAN) by 7.5% in AUROC, 5.8% in AUPRC, and 5.1% in accuracy. Our method, even without relying on domain adaptation, still surpasses DrugBAN-da and achieves better generalization performance, demonstrating the validity of our debiasing framework. The results on the BioSNAP cross-domain dataset, as shown in Fig. 6d, are included for completeness, for the interested reader.

TAPB's cross-domain superiority stems from its fundamental approach to alleviating target prior bias, which prevents reliance on spurious target-label correlations. Conventional models depending on $P(Y|D, T)$ exhibit severe performance degradation when encountering out-of-distribution targets, as evidenced in Fig. 6b and d, where baseline sensitivity collapses to near-zero levels and accuracy stagnates at approximately 0.5. In contrast, TAPB's interventional training paradigm effectively mitigates this bias, enabling consistent generalization beyond training data distributions. The observed multi-metric advancements across both BioSNAP and BindingDB datasets confirm TAPB's architectural advantage in disentangling authentic drug-target interaction patterns from dataset-specific biases.

TAPB's strong cross-domain generalizability stems from its core approach of mitigating target prior bias, thereby avoiding reliance on spurious target-label correlations. Conventional models relying on $P(Y|D, T)$ to predict drug-target interactions exhibit severe performance degradation when encountering out-of-distribution targets. In contrast, TAPB's interventional training paradigm, which incorporates amino acid randomization, disrupts these spurious correlations. Our method enables consistent generalization beyond training distributions, allowing TAPB to disentangle authentic DTI patterns from dataset-specific biases.

Comment: 3.6.

On page 83 of the response, the authors are talking about Figures 1g and 1h, but I still cannot find where they are either in the original manuscript or the revised one.

Response 3.6.:

We sincerely appreciate the reviewer's thorough and careful review of our manuscript. We apologize for the citation errors in our previous response. To clarify, Fig. 1g and h correspond respectively to the "target bias" tests and "drug bias" tests as illustrated in Fig. 1d and e.

Figure 1: Overview of DTI Bias Analysis and our framework. (a) **Dataset Construction of BioSNAP and BindingDB.** (b) An example sketch of target biased training sets shows that certain targets (T_1 and T_2) exhibit positive and negative "prior tendencies", respectively, while the bias in drugs is less pronounced. (c) Outline of the biased training process, where models learn from datasets containing an inherent "prior tendency." (d) "Target bias" tests demonstrate that pairing a randomly generated feature R with target T_1 yields higher positive scores compared to negative ones, with the opposite observed for T_2 . (e) "Drug bias" tests show relatively balanced scores when a drug D is paired with a randomly generated feature R , indicating lesser influence on predictions. (f) Our TAPB interventional training, combining amino acid randomization, confounder alignment module (CAM), and multi-head cross-attention (MHCA) to compute $P(Y|D, do(T))$ via backdoor adjustment, thereby adjusting confounders.

In our previous Revision 3.6, we removed the erroneous references to Fig. 1g and h (which had been mislabeled as Fig. 3g and h). To prevent the recurrence of such errors, we have rewritten and carefully reviewed the Introduction section, in addition to conducting a full verification of the entire manuscript to ensure the accuracy of all figure citations. In accordance with your suggestions, Revision 3.6 now more clearly highlights our distinctive findings and novel methodology, thereby improving the overall clarity for readers.

1 Introduction

PAGE 3-4

The cause of "target prior bias" is "prior tendency." Intuitively, "prior tendency" describes the imbalance of positive and negative interaction labels across individual drugs or targets in the DTI training set. in positive versus negative interaction labels associated with individual drugs or targets within the DTI sequence training set. Models can minimize loss by simply capturing the label tendencies of drugs or targets rather than the true interaction mechanisms. For example, in Fig. 1b, targets T_1 and T_2 in the training set have more positive and negative labels, respectively, while drugs' label distribution is relatively averaged. This imbalance can cause models to remember memorize the observed targets' label in the training set rather than genuine drug-target interactions, leading to biased predictions. Inspired by CF-VQA [6], we designed and performed our bias test for targets and drugs as shown in Fig. 1d and e, respectively. When inputs change from DTI pairs (D, T) to pairs with a randomly generated tensor R , i.e. (D, R) and (T, R) , the model receives (T, R) tends to predict based on training set target label tendencies, highlighting the influence of target "prior tendency." For the training set in Fig. 1b, when the input changes from DTI pairs (D, T) to pairs containing a randomly generated tensor R —i.e., (D, R) and (T, R) —the model tends to make predictions based on the observed target label tendencies from the training set when given (T, R) . In contrast, predictions from (D, R) remain close to average scores. This result underscores the significant influence of target "prior tendency." In Fig. 1c, we characterize this data-distorted training as biased training.

Comment: 3.7.

Are all supplementary tables referred to in the main text so that the readers are aware of the existence of such results?

Response 3.7.:

We sincerely appreciate the reviewer's rigorous and valuable reminders regarding our work. After careful verification, we hereby confirm that all supplementary figures, tables, and algorithms have been completely and correctly cited throughout the manuscript, ensuring that readers can fully access the detailed aspects of our study. The SI includes 3 figures, 16 tables, and 1 algorithm. Below, we specify the locations where each SI element is referenced in the main text:

- Supplementary Table 1 is cited in Section 2.4.1 "Datasets and evaluation proto-

col,” page 12.

- Supplementary Fig. 1 and Supplementary Table 2 are cited in Section 2.4.1 “Datasets and evaluation protocol,” page 13.
- Supplementary Fig. 2 is cited in Section 2.4.4 “Ablation study,” page 14.
- Supplementary Fig. 3 is cited in Section 2.4.2 “In-domain comparison,” page 13.
- Supplementary Table 4 is cited in Section 2.4.3 “Cross-domain comparison,” page 15.
- Supplementary Tables 5–8 are cited in Section 2.4.4 “Ablation study,” page 17.
- Supplementary Tables 9–15 are cited in Section 2.4.1 “Datasets and evaluation protocol,” page 13.
- Supplementary Table 16 is cited in Section 2.4.4 “Ablation study,” page 18.
- Supplementary Algorithm 1 is cited in Section 2.3 “TAPB framework,” page 12.

Your comments have significantly enhanced the readability and overall quality of our work. We thank you once again for your thorough and attentive review.

Comment: 3.8.

I did not find the response is addressing my 8th comment. The response reads largely identical to response 3.7 and hence it is addressing the 7th comment rather than the 8th one.

Response 3.8.:

We sincerely apologize that our previous response did not meet your expectations. Regarding Comment 8: “In Figure 6, sensitivity for many baselines approaches zero, but specificity approaches one. This is a well-known trade-off in binary classification. The authors should clarify whether performance is reported using a fixed threshold and whether adjusting the threshold could yield a better trade-off.”

Your observation is highly accurate. Our original Fig. 6-o (labeled here as 6-o for clarity) is shown below:

Figure 6-o: Radar chart comparisons of in-domain and cross-domain performance on BioSNAP and BindingDB datasets. (a) In-domain evaluation on BioSNAP. (b) In-domain evaluation on BindingDB. (c) Cross-domain evaluation on BioSNAP. (d) Cross-domain evaluation on BindingDB.

As correctly noted, the sensitivity approaches zero while specificity approaches one in the cross-domain evaluations. In the original manuscript, following DrugBAN [3] (published in NMI 2023), the threshold was selected to maximize the F1-score. However, this threshold does not necessarily lead to a desirable trade-off between sensitivity and specificity. Therefore, in the latest revision, we changed the threshold selection method to the Youden Index, which yields a more balanced performance. The Youden Index (J) is defined as follows:

$$J = \text{Sensitivity} + \text{Specificity} - 1 \quad (1)$$

Now, the optimal threshold is selected as the one that maximizes the value of J , thus providing a balanced trade-off between true positive and false positive rates. Our updated results are presented in Fig. 6:

Figure 6: Radar chart comparisons of in-domain and cross-domain performance on BioSNAP, BindingDB, Davis, and Human cold-split datasets. (a) In-domain evaluation on BioSNAP. (b) Cross-domain evaluation on BioSNAP. (c) In-domain evaluation on Davis. (d) In-domain evaluation on BindingDB. (e) Cross-domain evaluation on BindingDB. (f) Cold-split evaluation on Human.

A comparison between the current Fig. 6 and the original Fig. 6-o clearly shows that after applying the Youden Index, the sensitivity values of all models no longer cluster near zero on the more challenging datasets, indicating a better trade-off. All models in our experiments now use the Youden Index for threshold selection to ensure a fair and improved comparison. Furthermore, we provide detailed average thresholds in Supplementary Tables 3 and 4. We have also clearly stated in Section 2.4.1 of the manuscript that we used the Youden Index to adjust the threshold, as shown below:

The Youden Index was adopted to adjust the optimal threshold, offering a more effective balance between sensitivity and specificity. (Page 12)

Comment: 3.9. (Reviewer 3 labeled this as comment 10.)

The writing is still problematic. The writing use both Bert and BERT. Are they refer to the same BERT? The symbols are not standardised, the same entity occurs both in bold and nonbold versions, which is confusing. For example, the last two lines on page 88 of the response, the line above writes D and T and the line below use the bold versions. Are they the same? There are many other places of the similar issues.

Response 3.9.:

We sincerely thank the reviewers for their meticulous review of our manuscript. In our manuscript, "Bert" and "BERT" refer to the same model, and the bold and non-bold versions of "D" and "T" carry identical meanings. We sincerely apologize for the inconsistencies in our writing. Meanwhile, considering that Y is a scalar and should not be bolded, we have revised all instances from $P(\mathbf{Y}|\mathbf{D}, \mathbf{T})$ to $P(Y|\mathbf{D}, \mathbf{T})$, and $P(\mathbf{Y}|\mathbf{D}, do(\mathbf{T}))$ to $P(Y|\mathbf{D}, do(\mathbf{T}))$. We have now carefully reviewed the model naming conventions and standardized the symbols throughout the manuscript to ensure consistency and prevent reader confusion. The relevant modifications are summarized in Revision 3.9.

Revision: 3.9.

Abstract

PAGE 1-2

Drug Target Interaction (DTI) prediction is vital for drug repurposing. Previous studies trained on BioSNAP and BindingDB datasets have incorrectly identified "drug bias" as the cause of biased predictions, while our work reveals "target prior bias" as the predominant issue. This bias stems from the "prior tendency," characterized by the imbalanced label distribution of targets in the training data. From causal lens, "target prior bias" **target "prior tendency"** is a confounder, causing models trained with $P(\mathbf{Y}|\mathbf{D}, \mathbf{T})$ $P(\mathbf{Y}|\mathbf{D}, \mathbf{T})$ to learn spurious associations between targets and labels rather than genuine interaction mechanisms. In this study, we introduce alleviating **Target Prior Bias in Drug-Target Interaction Prediction (TAPB)**, a novel debiasing framework that employs amino acid randomization, confounder alignment module (CAM), and interventional training to approximate **compute $P(Y|\mathbf{D}, do(\mathbf{T}))$** $P(\mathbf{Y}|\mathbf{D}, do(\mathbf{T}))$ via backdoor adjustment, thereby addressing this bias. TAPB ~~outperforms existing methods on both in-domain and cross-domain datasets, offering enhanced generalization and providing interpretable insights into DTIs.~~ **TAPB achieves superior performance over existing approaches, not only in generalization but also in providing interpretable insights for DTIs.**

1. Introduction

PAGE 2

These inputs are encoded into embeddings and then aggregated for binary classification tasks, estimating the **interaction** probability $P(Y|D, T)$ $P(Y|\mathbf{D}, \mathbf{T})$ of an interaction between a drug D \mathbf{D} and a target T \mathbf{T} .

PAGE 4

When inputs change from DTI pairs (D, T) to pairs with a randomly generated tensor R , i.e. (D, R) and (T, R) , the model receives (\mathbf{T}, \mathbf{R}) tends to predict based on training set target label tendencies, highlighting the influence of target "prior tendency." For the training set in Fig. 1b, when the input changes from DTI pairs (\mathbf{D}, \mathbf{T}) to pairs containing a randomly generated tensor \mathbf{R} —i.e., (\mathbf{D}, \mathbf{R}) and (\mathbf{T}, \mathbf{R}) —the model tends to make predictions based on the observed target label tendencies from the training set when given (\mathbf{T}, \mathbf{R}) . In contrast, predictions from (\mathbf{D}, \mathbf{R}) remain close to average scores. This result underscores the significant influence of target "prior tendency." In Fig. 1c, we characterize this data-distorted training as biased training.

From causal lens, the "target prior bias" target "prior tendency" of DTI sequence datasets is a confounder that opens up backdoor paths for targets and predictions, making it difficult for DTI models to make unbiased predictions through $P(Y|\mathbf{D}, do(\mathbf{T}))$ $P(\mathbf{Y}|\mathbf{D}, do(\mathbf{T}))$. This issue has not been adequately addressed in previous studies. In this paper, we introduce an interventional debiasing framework for alleviating Target Prior Bias (TAPB) in drug-target interaction prediction. As shown in Fig. 1f, TAPB employs amino acids randomization and confounder alignment module to approximate compute $P(Y|\mathbf{D}, do(\mathbf{T}))$ $P(\mathbf{Y}|\mathbf{D}, do(\mathbf{T}))$ via theoretically exact backdoor adjustment, where $do(\cdot)$ denotes the intervention that sets the variable to a specific value, blocking all incoming paths to the variable. The backdoor adjustment computes $P(Y|\mathbf{D}, do(\mathbf{T}))$ $P(\mathbf{Y}|\mathbf{D}, do(\mathbf{T}))$ via observational data without performing actual interventions.

2.1 DTI prediction formulation on sequence datasets

PAGE 5

The general approach for DTI prediction involves three main steps: 1) Feature encoding: segment or convert the input SMILES and target sequences separately, and employ various respective encoders $f_d(\cdot)$ and $f_t(\cdot)$ to encode features, e.g. CNN [7], ResNet [8], GCN [9], LSTM [10] and Bert BERT [11], etc. Drug feature and target feature are denoted as D and T \mathbf{D} and \mathbf{T} , respectively.

2.3 TAPB framework

PAGE 10-11

The TAPB framework fundamentally differs from conventional DTI models

through its integration of interventional training, confounder alignment, and amino acid randomization to estimate $P(Y|\mathbf{D}, do(\mathbf{T}))$ via backdoor adjustment. As shown in Fig. 5a, the interventional training module computes $P(Y|\mathbf{D}, do(\mathbf{T}))$ via backdoor adjustment by incorporating all target confounder clusters $e_i \in \mathcal{E}$ $\mathbf{c}_i \in \mathbf{C}$. This requires the confounder dictionary \mathcal{E} \mathbf{C} and the confounder alignment module $g_t(\cdot)$ as prerequisites.

The confounder dictionary \mathbf{C} is constructed as shown in Fig. 5b. We perform K-means clustering on ESM-2 features extracted from all training targets in the training set, where the cluster centers form \mathbf{C} and the sample proportion per cluster \mathbf{c}_i defines the adjustment weight $P(\mathbf{c}_i)$. Since ESM-2 was pre-trained on datasets disjoint from DTI benchmarks, this eliminates label leakage risks.

The confounder alignment module $g_t(\cdot)$, illustrated in Fig. 5d, operates during interventional training. It processes each confounder cluster center \mathbf{c}_i to generate counterfactual **confounder-conditioned** representations $\mathbf{T}_{\mathbf{c}_i}$, and partitioned fused features $\mathbf{F}_{\mathbf{c}_i}$. A shared classifier $g_y(\cdot)$ then computes $P(Y|\mathbf{D}, \mathbf{T}, \mathbf{c}_i)$ for all $\mathbf{F}_{\mathbf{c}_i}$, **enabling the approximation of $P(\mathbf{Y}|\mathbf{D}, do(\mathbf{T}))$. enabling the computation of $P(Y|\mathbf{D}, do(\mathbf{T}))$ via backdoor adjustment under our SCM.**

Amino acid randomization in Fig. 5c regularizes input sequences. First, 70% of residues in ESM-2 features are randomly deleted to reduce computation and disrupt sequence patterns. Subsequently, each residue feature undergoes independent mutation with 20% probability by replacement via random sampling from the amino acid dictionary. This dual randomization prevents spurious correlation learning by disrupting label-specific motifs.

Figure 5: Architecture of the TAPB framework. (a) **TAPB Interventional Training:** The drug encoder BERT $f_d(\cdot)$ generates drug features D from SMILES. ESM-2 Pre-extracted target features E undergo amino acid randomization and are then processed by the CAM g_t . all cluster centers $c_i \in C$ act as keys/values in CAM g_t with E . Fused features F are partitioned into I segments F_{c_i} , each globally pooled and fed to classifier g'_y for estimating counterfactual **confounder-conditioned probabilities** $P(Y|D, T, c_i)$. Finally, $P(Y|D, do(T))$ is computed via backdoor adjustment. (b) **Target Confounder Dictionary C :** Obtained via K-means clustering on ESM-2 target features from training sets. (c) **Amino Acid Randomization:** 1. Random deletion of 70% residue features; 2. Mutation of remaining residues to random features from the amino acid dictionary. (d) **Confounder Alignment Module (CAM, $g_t(\cdot)$):** Attention-weighted summation fuses c_i with target features, followed by dimensionality reduction and residual connection. Maintains explicit path $X_t \rightarrow C \rightarrow T$ across training.

4.2.1 Encoders

PAGE 24-25

Next, TAPB stacks n layers ($n = 3$) of BERT layers with RoPE to construct more complex and abstract contextual representations. The output $D \in \mathbb{R}^{B \times L_d \times D_m}$ of the entire drug encoder is a context-sensitive depth representation of the input drug SMILES sequence:

$$D = f_d(E_d) = Bert(E_d)$$

$$\mathbf{D} = f_d(\mathbf{E}_d) = \text{BERT}(\mathbf{E}_d) \quad (8)$$

4.2.3 Amino acids randomization

PAGE 26

By randomly deleting and independently mutating residues, we create a scenario akin to a randomized experiment that helps to disrupt backdoor path $\mathbf{T} \leftarrow X_t \rightarrow \mathbf{Y}$, as shown in Fig. 9b. Intuitively, amino acid randomization can prevent models from memorizing the spurious correlations between targets and labels. Furthermore, the residue random deletion, reducing the sequence length, lowers computational costs, thereby accelerating training and allowing for the exploration of larger models. This is essential for deepening our understanding of the extensive and complex space of drug-target interactions.

4.2.4 TAPB interventional training

PAGE 28

Here, \mathbf{F}_{c_i} represents the feature segment associated with the i -th confounder cluster. Finally, after applying average pooling to each \mathbf{F}_{c_i} , a classification head $g_y(\cdot)$ is used to estimate all $P(\mathbf{Y}|\mathbf{D}, \mathbf{T}, \mathbf{c}_i)$:

$$P(\mathbf{Y}|\mathbf{D}, \mathbf{T}, \mathbf{c}_i) = \text{Softmax}(g_y(\mathbf{F}_{c_i})) \quad (21)$$

then, we can parameterize Equation 17 via TAPB in the following form:

$$P(\mathbf{Y}|\mathbf{D}, do(\mathbf{T})) = \sum_i P(\mathbf{c}_i) \text{Softmax}(g_y(P(\mathbf{Y}|\mathbf{D}, \mathbf{T}, \mathbf{c}_i))) \quad (22)$$

Additional comments:

Comment: A

This manuscript has nine figures and one table for the main text. To my understanding, this exceeded what is allowed by Nature Communications.

Response A.1.:

We are grateful to the reviewer for this reminder. We have now moved the dataset statistics table (Table 1) to Supplementary Table 1, and the main text now contains only 9 figures, which complies with the requirements of Nature Communications. All revisions are summarized in Revision A.

2.4.1 Datasets and evaluation protocol.

PAGE 12

Following DrugBAN, we evaluated the in-domain and cross-domain classification performance of 6 publicly available datasets: BindingDB in-domain, BindingDB cross-domain, BioSNAP in-domain, BioSNAP cross-domain, Davis, Human cold-split. To ensure a rigorous and comprehensive assessment, we evaluated the model's classification performance across four publicly available datasets under six settings: BindingDB (in-domain and cross-domain), BioSNAP (in-domain and cross-domain), Davis, and Human cold-split. Supplementary Table 1 provides an overview of the datasets. Additionally, Supplementary Fig. 3 reveals "target prior bias" in the Davis dataset and "drug prior bias" in the Human cold-split dataset.

Table 1 provides detailed information on the datasets.

Supplementary Section 1.1. Dataset statistics

PAGE 2

We summarize the key statistics of the public datasets used in this study in Supplementary Table 1, including the number of Drugs and Targets, the sequence lengths of Drugs and Targets (along with the maximum sequence length, which may consume excessive GPU memory if too large), and the total counts as well as the ratios of positive and negative samples for each dataset.

Supplementary Table 1: Statistics of BindingDB and BioSNAP in-domain and cross-domain datasets and Davis in-domain dataset and Human cold-split dataset. Length data is represented as (average length/maximum length). For positive and negative pairs, the values denote (number of pairs/percentage of total pairs)

Dataset	Drug	Target	Drug length	Target length	Pos pairs	Neg pairs
BindingDB in-domain	14643	2623	58.58/680	691.93/7073	20674/0.42	28525/0.58
BioSNAP in-domain	4505	2181	56.71/748	549.95/5183	13830/0.50	13634/0.5
BindingDB cross-domain	8969	1587	57.57/660	692.20/4128	6466/0.39	10241/0.61
BioSNAP cross-domain	3219	1751	52.53/409	542.04/5179	5456/0.51	5217/0.49
Davis in-domain	68	379	63.26/92	801.44/2549	1506/0.14	9597/0.86
Human cold-split	1813	1503	47.32/420	639.74/4655	1995/0.51	1924/0.49

Comment: B

There are many forward references where the figures are several (unknown at the place of references) pages ahead, making it difficult for the readers to follow.

Response B.:

We sincerely appreciate the reviewer’s insightful comment regarding the excessive forward references to figures, which indeed could disrupt the reading flow. In the previous revision, figures were inadvertently placed too far from their corresponding references due to LaTeX compilation issues. To address this, we have systematically revised the manuscript by relocating all figures to appear immediately after their first textual mention (or within the same section where contextually appropriate), thereby minimizing the page separation between each reference and its corresponding figure. These modifications have significantly enhanced the manuscript’s readability, and we thank the reviewer for highlighting this critical issue.

Comment: C. (Remarks on code availability):

I did not find meaningful changes in the code repo comparing to the version for the initial submission, except some figure/documentation update.

Response C.:

We sincerely appreciate your valuable feedback and insightful comments regarding the code reproducibility issue. We are deeply sorry for the incomplete code submission in our initial revision. We have now fully updated our repository with the latest version of TAPB’s implementation. Furthermore, to enhance reproducibility:

1. The code has been refactored for simplicity, enabling readers to reproduce all results with a single Python command after environment setup.
2. Comprehensive inline comments have been added throughout the code to explicitly align implementation details with the methodology, addressing prior gaps in documentation.

Additionally, recognizing that potential regional restrictions may hinder access to Hugging Face Hub, we have uploaded both model weights to a dedicated OneDrive link. This link is also shared in our ‘README.md’. Readers need only download the weights, extract them, and place the files in the specified directory per the ‘README.md’ instructions to ensure reliable reproduction.

These revisions comprehensively address the reproducibility concerns raised, and we hope that these improvements will facilitate reproducibility for readers.

References

- [1] Kexin Huang, Cao Xiao, Lucas M Glass, and Jimeng Sun. Moltrans: molecular interaction transformer for drug–target interaction prediction. *Bioinformatics*,

37(6):830–836, 2021.

- [2] Lifan Chen, Xiaoqin Tan, Dingyan Wang, Feisheng Zhong, Xiaohong Liu, Tianbiao Yang, Xiaomin Luo, Kaixian Chen, Hualiang Jiang, and Mingyue Zheng. Transformerpci: improving compound–protein interaction prediction by sequence-based deep learning with self-attention mechanism and label reversal experiments. *Bioinformatics*, 36(16):4406–4414, 2020.
- [3] Peizhen Bai, Filip Miljković, Bino John, and Haiping Lu. Interpretable bilinear attention network with domain adaptation improves drug–target prediction. *Nature Machine Intelligence*, 5(2):126–136, 2023.
- [4] Huan Yee Koh, Anh TN Nguyen, Shirui Pan, Lauren T May, and Geoffrey I Webb. Physicochemical graph neural network for learning protein–ligand interaction fingerprints from sequence data. *Nature Machine Intelligence*, 6(6):673–687, 2024.
- [5] Zhousan Xie, Shikui Tu, and Lei Xu. Multilevel attention network with semi-supervised domain adaptation for drug-target prediction. In *Proceedings of the AAAI Conference on Artificial Intelligence*, volume 38, pages 329–337, 2024.
- [6] Yulei Niu, Kaihua Tang, Hanwang Zhang, Zhiwu Lu, Xian-Sheng Hua, and Ji-Rong Wen. Counterfactual vqa: A cause-effect look at language bias. In *Proceedings of the IEEE/CVF conference on computer vision and pattern recognition*, pages 12700–12710, 2021.
- [7] Yann LeCun, Léon Bottou, Yoshua Bengio, and Patrick Haffner. Gradient-based learning applied to document recognition. *Proceedings of the IEEE*, 86(11):2278–2324, 1998.
- [8] Kaiming He, Xiangyu Zhang, Shaoqing Ren, and Jian Sun. Deep residual learning for image recognition. In *Proceedings of the IEEE conference on computer vision and pattern recognition*, pages 770–778, 2016.
- [9] Thomas N Kipf and Max Welling. Semi-supervised classification with graph convolutional networks. In *International Conference on Learning Representations*, 2022.
- [10] S Hochreiter. Long short-term memory. *Neural Computation MIT-Press*, 1997.
- [11] Jacob Devlin Ming-Wei Chang Kenton and Lee Kristina Toutanova. Bert: Pre-training of deep bidirectional transformers for language understanding. In *Proceedings of naacL-HLT*, volume 1, page 2. Minneapolis, Minnesota, 2019.

Response letter for “TAPB: An Interventional Debiasing Framework for Alleviating Target Prior Bias in Drug-Target Interaction Prediction”

Gaoming Lin ^{†1}, Xin Zhang ^{†2,3}, Zhonghao Ren^{4,5}, Quan Zou²,
Prayag Tiwari^{*6}, Changjun Zhou ^{*1}, and Yijie Ding ^{*2}

¹School of Computer Science and Technology, Zhejiang Normal
University, Jinhua, 321000, Zhejiang, China

²Yangtze Delta Region Institute (Quzhou), University of Electronic
Science and Technology of China, Quzhou, 324003, Zhejiang, China

³The Quzhou Affiliated Hospital of Wenzhou Medical University,
Quzhou People’s Hospital, Quzhou, 324000, Zhejiang, China

⁴State Key Laboratory of Chemo and Biosensing, College of
Computer Science and Electronic Engineering, Hunan University,
Changsha, 410082, Hunan, China

⁵The Ministry of Education Key Laboratory of Fusion Computing of
Supercomputing and Artificial Intelligence, Hunan University,
Changsha, 410082, Hunan, China

⁶School of Information Technology, Halmstad University, Halmstad,
301 18, Sweden

To reviewers

Dear reviewers:

Our sincere thanks go out to the reviewers who reviewed our manuscript and provided constructive comments that significantly improved it. We have made detailed revisions in a point-by-point response to comments made by reviewers.

We first quote the comments and then reply with how we have revised the manuscript to accommodate the changes. We use **black sans serif font** for our responses and **red box** for comments. Revisions in the manuscript are indicated within a **blue box**, with modified text shown in **red font** and deleted content marked by "~~strikethrough formatting~~." For further results, please refer to our SI.

We sincerely appreciate the reviewers for giving us the opportunity to revise our manuscript. We believe that our final revisions have addressed the comments accordingly and hope the manuscript now meets your expectations.

Best regards,

Gaoming Lin, Xin Zhang, Zhonghao Ren, Quan Zou, Prayag Tiwari, Changjun Zhou, Yijie Ding

Oct 19, 2025

Contents

1	Response to Reviewer 1	3
2	Response to Reviewer 2	4
3	Response to Reviewer 3	9

1 Response to Reviewer 1

Comment: (Remarks to the Author)

The authors have addressed my previous comments and I have no further comments.

Response:

We sincerely appreciate your expert comments and meticulous review, which have greatly benefited our work. We are also truly grateful for your patience with our research. In the future, we will continue to explore potential applications of causal inference theory in deep learning.

2 Response to Reviewer 2

Comment: (Remarks to the Author)

The authors propose a method for predicting drug-target interactions (DTI). They believe that when dealing with datasets such as BioSNAP and BindingDB, one must consider the issue of 'target prior bias.' This bias arises from some factors like imbalanced label distribution of targets in the training data. Their model demonstrates advantages over selected methods on both in-domain and cross-domain datasets.

Response:

We sincerely appreciate your expert guidance and your understanding of our work. Your feedback has been highly valuable and has significantly improved the quality of our manuscript. In this revision, we have cited and discussed UdanDTI as suggested, explicitly clarifying the differences between our method, UdanDTI, and previous methods. We have also added a discussion on related UDA techniques in the manuscript. All revisions have been clearly highlighted in the response letter. We are deeply grateful for your constructive feedback and hope that the revised manuscript now meets your expectations.

Comment: 2.1

This manuscript has been revised and supplemented in response to the reviewers' comments. Additionally, the results of the ablation experiments have been added. If possible, Table 3 and Table 4 in the supplements are the main predictive results of this paper, and it is recommended to describe them in the main text.

Response 2.1.:

We sincerely appreciate your constructive feedback regarding the detailed description of the experimental results from Supplementary Tables 3 and 4 in the main text. We have already cited and described these tables in Sections 2.6 and 2.7 to detail the model's performance. If your suggestion involves moving Supplementary Tables 3 and 4 directly into the main text, we agree that this would offer readers a more intuitive comparison of model performance. However, according to the guidelines of Nature Communications, the main text is limited to a total of 10 figures and tables. After our previous revisions, the main text already contains 9 figures and tables, which meet this limit. Therefore, to avoid exceeding the limit—which would occur if Tables 3 and 4 were included (bringing the total to 11)—we have placed

these tables in the Supplementary Information. In the main text, model performance is summarized using a radar chart, where TAPB’s performance is represented by the outermost radar plot, enabling an intuitive and clear comparison across models.

Comment: 2.2

Since DTI has been a relatively popular research topic in recent years, many research results on DTI have been published in relevant journals. For example, “Escaping the Drug-Bias Trap: Using Debiasing Design to Improve Interpretability and Generalization of Drug-Target Interaction Prediction, IEEE TRANSACTIONS ON COMPUTATIONAL BIOLOGY AND BIOINFORMATICS, VOL. 22, NO. 4, JULY/AUGUST 2025” (after acceptance, the paper should be in an OPEN ACCESS state), the paper has corresponding prediction results (both in-domain and cross-domain experiments) on the BioSNAP and BindingDB datasets. Although the method TAPB described in this manuscript shows certain advantages compared to the methods it is compared with on the above datasets, when compared to the results of the aforementioned papers, the superiority of this TAPB’s results does not exist. We hope the authors provide a better explanation and clarification.

Response 2.2.:

We sincerely appreciate your instructive comments regarding the recently published UdanDTI [1] in June 2025. We have carefully studied the UdanDTI method and are pleased to discuss some key points here.

UdanDTI also recognizes the issue of biased prediction in DTI and attributes it to the dominant role of "drug" features. Consequently, it employs an asymmetric structure and an attentive aggregation module to enhance target information. To improve cross-domain evaluation performance, UdanDTI further incorporates the Unsupervised Domain Adaptation (UDA) technique MCD, proposed by [2] in 2018. We have cited UdanDTI and incorporated a discussion on UDA techniques in the main text.

Our approach differs fundamentally from prior works, including TAPB and UdanDTI, in the following aspects:

1. **Bias Identification.** A key distinction of TAPB and UdanDTI compared to prior work lies in the understanding of dataset bias. Earlier studies generally assumed the presence of “drug bias” in datasets such as BioSNAP and BindingDB. In contrast, through systematic experiments—including t-SNE visualization, bias tests, statistical quantification, and dataset reversal—we conclusively identify “target prior bias” as the dominant factor influencing model predictions. This redefined bias perspective diverges from previously recognized forms such as “hidden pattern

bias.”

2. Methodological Approach. Building on this insight, we reformulate the DTI prediction problem from a causal perspective and introduce TAPB, a framework that employs backdoor adjustment and amino acid randomization to mitigate target prior bias. Specifically, randomization is applied through residue deletion and mutation to disrupt spurious correlations, while backdoor adjustment enables the computation of $P(Y|\mathbf{D}, do(\mathbf{T}))$ via a confounder dictionary and alignment module. This causal interventional approach offers a new direction for debiasing in DTI prediction.

We understand and acknowledge the value of UDA techniques in specific scenarios. UDA requires the use of both the source domain training set and the target domain training set, whereas our TAPB uses only the source domain training set. One of our core objectives is to explore a more universal and convenient zero-shot prediction paradigm. In practical DTI application scenarios, when users face novel drugs or targets, it is often challenging to pre-construct a representative target domain training set and retrain the model using UDA. This undoubtedly introduces significant computational burden and application complexity.

Therefore, the focus of our method is to achieve exceptional cross-domain generalization capability using only the source domain training set. Table 2 demonstrates a fair comparison, showing that TAPB exhibits comparable performance to UdanDTI, with a slight advantage on the BioSNAP in-domain split. Notably, the zero-shot performance of TAPB surpasses that of UdanDTI on cross-domain splits without requiring UDA techniques. Specifically, for cross-domain splits, TAPB exceeds UdanDTI by 3.5% in AUROC and 2.7% in AUPRC on BioSNAP, and by 3.5% in AUROC and 5.7% in AUPRC on BindingDB, demonstrating that TAPB inherently possesses generalization ability.

Table 1: Comparison of TAPB and UdanDTI performance on BioSNAP, BindingDB datasets. Results for UdanDTI are taken directly from its original paper, and results for our TAPB are from our Supporting Information. The best results are in bold.

Dataset	Method	AUROC	AUPRC
BioSNAP in-domain split	UdanDTI	0.941±0.003	0.942±0.004
	TAPB	0.943±0.003	0.944±0.003
BindingDB in-domain split	UdanDTI	0.965±0.001	0.955±0.001
	TAPB	0.961±0.001	0.950±0.002
BioSNAP cross-domain split	UdanDTI	0.756±0.013	0.773±0.021
	TAPB	0.791±0.007	0.800±0.005
BindingDB cross-domain split	UdanDTI	0.636±0.021	0.571±0.027
	TAPB	0.676±0.016	0.628±0.029

We extended the UdanDTI approach by leveraging additional data to fine-tune the TAPB_S model. As shown in Table 2, our model achieved a significant performance improvement: on the BioSNAP cross-domain split, TAPB_S increased AUROC and AUPRC by 2.1% and 2.5%, respectively; on BindingDB, the corresponding gains were 16.4% in AUROC and 18.3% in AUPRC. We have added the discussion on UdanDTI to the manuscript. The related results are provided in the SI. Thank you once again for your valuable comments, which have helped us improve this work.

Table 2: Comparison of TAPB_S and UdanDTI_MCD performance on BioSNAP, BindingDB datasets. We directly cite the mean and standard deviation of UdanDTI_MCD from the UdanDTI paper, with the best results highlighted in bold.

Dataset	Method	AUROC	AUPRC
BioSNAP cross-domain split	UdanDTI_MCD	0.805±0.011	0.825±0.008
	TAPB_S	0.812±0.009	0.825±0.015
BindingDB cross-domain split	UdanDTI_MCD	0.713±0.017	0.671±0.019
	TAPB_S	0.840±0.014	0.811±0.017

Revison

1. Introduction

PAGE 2

Previous research, e.g. TransformerCPI [3], DrugBAN [4] and UdanDTI [1], has attributed this issue to "hidden pattern bias", i.e. "drug bias".

PAGE 2

UdanDTI leverages an asymmetrical architecture and attentive aggregation module to strengthen the target branch and downplay the drug branch, thus mitigating "drug bias."

3. Discussion

PAGE 19

UDA techniques, e.g. CDAN [5] used in DrugBAN [4] and MCD [2] used in UdanDTI [1], require access to both source and target domain data for model adaptation, which typically leads to improved cross-domain generalization performance. In contrast, we aim to explore a more universal and convenient zero-shot prediction paradigm, where our method TAPB utilizes only the source domain training set. This approach avoids the computational burden and application complexity associated with repeatedly constructing target domain sets and retraining for novel drugs or targets. Our comparisons with UdanDTI are provided in Supplementary Tables 17 and 18.

The focus of our method is to achieve exceptional cross-domain generalization capability using only the source domain training set. Supplementary Table 17 demonstrates a fair comparison, showing that TAPB exhibits comparable performance to UdanDTI, with a slight advantage on the BioSNAP in-domain split. Notably, the zero-shot performance of TAPB surpasses that of UdanDTI on cross-domain splits without requiring UDA techniques. Specifically, for cross-domain splits, TAPB exceeds UdanDTI by 3.5% in AUROC and 2.7% in AUPRC on BioSNAP, and by 3.5% in AUROC and 5.7% in AUPRC on BindingDB, demonstrating that TAPB inherently possesses generalization ability.

Supplementary Table 17: Comparison of TAPB and UdanDTI performance on BioSNAP, BindingDB datasets. Results for UdanDTI are taken directly from its original paper. The best results are in bold.

Dataset	Method	AUROC	AUPRC
BioSNAP in-domain split	UdanDTI	0.941±0.003	0.942±0.004
	TAPB	0.943±0.003	0.944±0.003
BindingDB in-domain split	UdanDTI	0.965±0.001	0.955±0.001
	TAPB	0.961±0.001	0.950±0.002
BioSNAP cross-domain split	UdanDTI	0.756±0.013	0.773±0.021
	TAPB	0.791±0.007	0.800±0.005
BindingDB cross-domain split	UdanDTI	0.636±0.021	0.571±0.027
	TAPB	0.676±0.016	0.628±0.029

We adapted and enhanced the core data-centric idea of UdanDTI by fine-tuning the TAPB_S model on fewer target domain data. As shown in Supplementary Table 18, our model achieved a significant performance improvement: on the BioSNAP cross-domain split, TAPB_S increased AUROC and AUPRC by 2.1% and 2.5%, respectively; on BindingDB, the corresponding gains were 16.4% in AUROC and 18.3% in AUPRC.

Supplementary Table 18: Comparison of TAPB_S and UdanDTI_MCD performance on BioSNAP, BindingDB datasets. We directly cite the mean and standard deviation of UdanDTI_MCD from the UdanDTI paper, with the best results highlighted in bold.

Dataset	Method	AUROC	AUPRC
BioSNAP cross-domain split	UdanDTI_MCD	0.805±0.011	0.825±0.008
	TAPB_S	0.812±0.009	0.825±0.015
BindingDB cross-domain split	UdanDTI_MCD	0.713±0.017	0.671±0.019
	TAPB_S	0.840±0.014	0.811±0.017

3 Response to Reviewer 3

Comment: (Remarks to the Author)

The authors have substantially revised the manuscript to address all the detailed comments raised in the previous round of review. I found the revisions largely satisfactory, except for one issue detailed below and some minor writing problems that should be fixed during the editorial process.

Response:

We sincerely appreciate the reviewer’s thoughtful guidance and thorough assessment of our work. Thank you for the valuable opportunity and for your detailed comments. In this revision, we have meticulously revised the dataset naming conventions in accordance with your suggestions. All revisions are clearly highlighted in the response letter. We hope that these improvements align with your expectations.

Comment: 3.1

Remaining issue: The authors refer to the Human dataset with a cold split strategy as the “Human Cold-split” dataset. The naming style of DATASET_NAME+SPLIT is inappropriate and inconsistent with the convention and literature, as well as the naming of other datasets in this paper.

Response 3.1.:

We sincerely appreciate the reviewer’s insightful comment regarding the dataset naming convention. We fully agree that incorporating the split strategy into the dataset name was inappropriate and inconsistent with common practice in the literature. Following your suggestion, we have revised the manuscript throughout to refer to the dataset simply as the "Human" dataset, and explicitly describe the "cold-split" as the data partitioning strategy applied to it. Corresponding adjustments have also been made to the other datasets. All revisions are summarized in Revision 3.1.

Revision: 3.1.

1.Introduction

PAGE 2

~~Recently, these datasets have been redivided into in-domain and cross-domain datasets in DrugBAN [4], enabling a more rigorous evaluation of model performance under diverse conditions.~~ Recently, DrugBAN [4] has divided these datasets into in-domain and cross-domain splits, enabling a more system-

atic and rigorous evaluation of model performance under diverse conditions. However, our findings suggest that "target prior bias", i.e. "target bias", which indicates that models rely predominantly on target-specific features for prediction, plays a more significant role in biased predictions across both in-domain and cross-domain datasets from BioSNAP and BindingDB.

However, our findings indicate that "target prior bias", i.e. "target bias", plays a more substantial role in biased predictions across both in-domain and cross-domain splits of BioSNAP and BindingDB. This bias reflects the tendency of models to rely primarily on target-specific features when making predictions.

PAGE 4

In this study, we re-evaluate the BioSNAP and BindingDB datasets, both in-domain and cross-domain splits, identifying what we term "target prior bias" as a significant cause of biased prediction in these datasets, diverging from the previously recognized "hidden pattern bias."

2.2 Drugs bias vs target bias: Which is more severe?

PAGE 5

For in-domain datasets splits, the training set was chosen as the visualization data, whereas for cross-domain datasets splits, the source training set was selected for visualization.

PAGE 6

Figure 2: The t-SNE visualization of classification features (D, R) and (T, R) of DrugBAN and TransformerCPI on the BindingDB and BioSNAP in-domain, and cross-domain datasets. (a) DrugBAN trained on the in-domain split of the BindingDB dataset with inputs (T, R). (b) DrugBAN trained on the in-domain split of the BioSNAP dataset with inputs (T, R). (c) DrugBAN trained on the cross-domain split of the BindingDB dataset with inputs (T, R). (d) DrugBAN trained on the cross-domain split of the BioSNAP dataset with inputs (T, R). (e) DrugBAN trained on the in-domain split of the BindingDB dataset with inputs (D, R). (f) DrugBAN trained on the in-domain split of the BioSNAP dataset with inputs (D, R). (g) DrugBAN trained on the cross-domain split of the BindingDB dataset with inputs (D, R). (h) DrugBAN trained on the cross-domain split of the BioSNAP dataset with inputs (D, R). (i) TransformerCPI trained on the in-domain split of the BindingDB dataset with inputs (T, R). (j) TransformerCPI trained on the in-domain split of the BioSNAP dataset with inputs (T, R). (k) TransformerCPI trained on the cross-domain split of the BindingDB dataset with inputs (T, R). (l) TransformerCPI trained on cross-domain split of the BioSNAP dataset with inputs (T, R). (m) TransformerCPI trained on the in-domain split of the BindingDB dataset with inputs (D, R). (n) TransformerCPI trained on the in-domain split of the BioSNAP dataset with inputs (D, R). (o) TransformerCPI trained on the cross-domain split of the BindingDB dataset with inputs (D, R). (p) TransformerCPI trained on the cross-domain split of the BioSNAP dataset with inputs (D, R).

PAGE 7

However, the visualization results contradict these expectations. As shown in Fig. 2e, g, m, and o, under the BindingDB in-domain and cross-domain datasets settings, no matter what drug encoder $f_d(\cdot)$ or target encoder $f_t(\cdot)$ or aggregators $\mathcal{F}(\cdot)$ were used, the positive classification features of (D, R) are nearly randomly distributed, while t-SNE visualizations of (T, R) in Fig. 2a, c, i, and k exhibits significant positive class clustering. Similarly, under the BioSNAP in-domain and cross-domain datasets settings, Fig. 2b, d, j, and l shows more pronounced positive class clustering for (T, R) compared to Fig. 2f, h, n, and p. These observations suggest that models trained on these datasets exhibit stronger "target bias" than "drug bias", prompting the question: why do models have inner patterns that rely more heavily on targets?

2.5 Datasets and evaluation protocol

PAGE 12

To ensure a rigorous and comprehensive assessment, we evaluated the model's classification performance across four publicly available datasets under six settings: BindingDB (in-domain and cross-domain), BioSNAP (in-domain and

cross-domain), ~~Davis, and Human cold-split.~~ **in-domain and cross-domain splits of BindingDB and BioSNAP, in-domain split of Davis, and cold-split of Human.** Supplementary Table 1 provides an overview of the datasets. Additionally, Supplementary Fig. 3 reveals "target prior bias" in the Davis dataset and "drug prior bias" in the Human cold-split dataset.

~~The in-domain~~ **For the in-domain splits,** datasets were randomly divided into training, validation, and test sets in a 7:1:2 ratio. Notably, in these in-domain datasets **scenarios,** targets exhibit significantly higher overlap across training, validation, and test sets compared to drugs. In contrast, the cross-domain **splits** datasets—constructed by DrugBAN—consist of a source domain training set, a target domain training set, and a target domain test set, with no overlap between source domain drugs/targets and the target domain data (CVS4).

For all datasets, we performed five independent runs with different random seeds and reported the area under the receiver operating characteristic curve (AUROC), the area under the precision-recall curve (AUPRC), accuracy, sensitivity, and specificity. The Youden Index was adopted to adjust the optimal threshold, offering a more effective balance between sensitivity and specificity. ~~For in-domain datasets~~ **For the in-domain splits,** we selected the model checkpoint with the highest validation AUROC and reported test set performance. Following DrugBAN’s protocol for cross-domain datasets, we trained models without domain adaptation techniques on the source domain and evaluated them directly on the target domain test set, reporting the resulting metrics.

We compared TAPB with five baseline models—MolTrans [6], TransformerCPI [3], DrugBAN [4], PSICHIC [7], and MlanDTI [8]. Unlike these models, which rely on $P(Y|D, T)$ for predictions, TAPB computes $P(Y|D, do(T))$ via backdoor adjustment for predictions. The hyperparameter settings for TransformerCPI, MolTrans, ~~DrugBAN (on in-domain, Davis, Human), DrugBAN-da (on cross-domain)~~ **DrugBAN (on the in-domain splits of BindingDB, BioSNAP, and Davis and cold-split of Human), DrugBAN-da (on the cross-domain splits of BindingDB and BioSNAP),** TAPB, PSICHIC, and MlanDTI are detailed in Supplementary Tables 9-15. For each model, hyperparameters remained consistent across all datasets unless otherwise specified. The key hyperparameters of TAPB—target confounder dictionary size, target random deletion ratio, and mutation rate—were tuned on the **cross-domain split of the** BioSNAP ~~cross-domain~~ dataset, as shown in Supplementary Fig. 1. Accordingly, comparative conclusions were not drawn from this dataset. ~~Supplementary Table 2 provides the per-seed AUROC and AUPRC values across all optimizations.~~ **A summary of the per-seed AUROC and AUPRC**

values across all hyperparameter tuning experiments is provided in Supplementary Table 2.

2.6 In-domain comparison

PAGE 12-13

In the challenging Human cold-split scenario of the Human dataset, where "drug bias" is the dominant one, we also evaluated TAPB's performance under this opposite condition.

2.7 Cross-domain comparison

PAGE 13

On the BindingDB cross-domain dataset cross-domain split of the BindingDB dataset, TAPB maintains strong competitiveness, delivering notable results with an AUROC of 0.676, accuracy of 0.630, and specificity of 0.565, while surpassing DrugBAN-da (w/ CDAN) by 7.5% in AUROC, 5.8% in AUPRC, and 5.1% in accuracy.

2.8 Ablation study

PAGE 14

We conducted ablation studies on three splits—DAVIS, BioSNAP cross-domain, and BindingDB cross-domain—to evaluate the impact of our key components

We conducted ablation studies on three datasets: the in-domain split of the Davis dataset, and the cross-domain splits of the BioSNAP and BindingDB datasets. Using a total of seven TAPB variants, these studies were designed to comprehensively evaluate the impact of our key components.

Comprehensive ablation results for Davis, BioSNAP cross-domain, BindingDB cross-domain and residue random deletion generalizability are presented in Supplementary Tables 5–8.

SI: 3. Extend data statistics

PAGE 14

We quantified the "prior tendency" and overall "prior tendency" in the in-domain split of Davis and Human cold-split cold-split of Human. As illustrated in Supplementary Fig. 3a, b and c, the Davis dataset exhibits a certain "target prior bias", while a more pronounced "drug prior bias" is observed in the Human dataset. Statistical analysis confirmed significant deviations for both biases ($P_t = 0.000$, $P_d = 0.000$).

Supplementary Figure 3: Statistics of the Davis and Human datasets. (a) "Prior tendency" frequency distributions of z_i for drugs and targets in the Davis training set. (b) "Prior tendency" frequency distribution of z_i for drugs and targets in the Human cold-split training set. (c) Overall "prior tendency" for labels associated with drugs and targets across Davis and Human cold-split datasets. P-values were derived from a one-sided permutation test with 1000 iterations (P_d for drugs, P_t for targets), with no adjustments for multiple comparisons. Source data are provided as a Source Data File.

Comment: 3.2

I see substantial efforts to improve the code; for example, comments in Chinese have been converted to English, along with other improvements detailed in the response to my comments.

Response 3.2.:

We sincerely thank your thorough review of our manuscript and code. We hope that the optimized code will facilitate readers' understanding and reproduction of our results.

References

- [1] Peidong Zhang, Jianzhu Ma, and Ting Chen. Escaping the drug-bias trap: Using debiasing design to improve interpretability and generalization of drug-target interaction prediction. *IEEE Transactions on Computational Biology and Bioinformatics*, 22(4):1902–1911, 2025.
- [2] Kuniaki Saito, Kohei Watanabe, Yoshitaka Ushiku, and Tatsuya Harada. Maximum classifier discrepancy for unsupervised domain adaptation. In *Proceedings of the IEEE conference on computer vision and pattern recognition*, pages 3723–3732, 2018.

- [3] Lifan Chen, Xiaoqin Tan, Dingyan Wang, Feisheng Zhong, Xiaohong Liu, Tianbiao Yang, Xiaomin Luo, Kaixian Chen, Hualiang Jiang, and Mingyue Zheng. Transformerpci: improving compound–protein interaction prediction by sequence-based deep learning with self-attention mechanism and label reversal experiments. *Bioinformatics*, 36(16):4406–4414, 2020.
- [4] Peizhen Bai, Filip Miljković, Bino John, and Haiping Lu. Interpretable bilinear attention network with domain adaptation improves drug–target prediction. *Nature Machine Intelligence*, 5(2):126–136, 2023.
- [5] Mingsheng Long, Zhangjie Cao, Jianmin Wang, and Michael I Jordan. Conditional adversarial domain adaptation. *Advances in neural information processing systems*, 31, 2018.
- [6] Kexin Huang, Cao Xiao, Lucas M Glass, and Jimeng Sun. Moltrans: molecular interaction transformer for drug–target interaction prediction. *Bioinformatics*, 37(6):830–836, 2021.
- [7] Huan Yee Koh, Anh TN Nguyen, Shirui Pan, Lauren T May, and Geoffrey I Webb. Physicochemical graph neural network for learning protein–ligand interaction fingerprints from sequence data. *Nature Machine Intelligence*, 6(6):673–687, 2024.
- [8] Zhousan Xie, Shikui Tu, and Lei Xu. Multilevel attention network with semi-supervised domain adaptation for drug-target prediction. In *Proceedings of the AAAI Conference on Artificial Intelligence*, volume 38, pages 329–337, 2024.